# Axiomatic Atlas: A Prescriptive Framework for Neural Architecture Design

**Minghao Guo** [1]   **Wojciech Matusik** [1]

## Abstract

Neural architecture design remains empirical: failures are diagnosed post-hoc, fixes invented ad-hoc, and no theory predicts which repair will succeed. We introduce the *Axiomatic Atlas*, a prescriptive framework that encodes architecture requirements as composable axioms over graph connectivity, operator contracts, depth stability, and information preservation at merges. The Atlas audits an architecture to produce two outputs: a *certificate* lower-bounding input-output variation, and a *diagnosis* pinpointing which axioms fail and where. Each axiom violation implies a specific repair, reducing architecture design to constraint satisfaction. We prove variation bounds under exact and finite-precision arithmetic with explicit axiom closures, enabling modular verification. Experiments validate prescriptive power on four interventions: topology-aware GNN bridges (+46pp), anchored MoE quantization ($3\times$ INT4 robustness), uncertainty-guided expert routing (83% gap closure), and orthogonal key projection (0%→100% retrieval), each outperforming matched negative controls.

## 1. Introduction

Neural network architectures are designed by intuition, validated by experiment, and justified post-hoc. When a transformer fails on long sequences, practitioners diagnose attention decay (Dong et al., 2021); when a graph neural network over-squashes, the culprit is bottleneck topology (Alon & Yahav, 2021; Topping et al.); when a mixture-of-experts collapses to a single expert, router entropy takes the blame (Shazeer et al., 2017). These explanations share a common structure: the failure mode is identified empirically, the fix invented ad-hoc, and no formal theory connects diagnosis to repair. We still lack principled answers to two fundamental questions: *why* does an architecture fail, and *what* minimal modification will fix it?

The absence of such a theory exacts a practical toll. Architecture search explores vast design spaces without pruning provably defective candidates (White et al., 2023), wasting compute on configurations that cannot succeed regardless of training. Debugging demands exhaustive ablations because no principle predicts which component causes a failure. Scaling laws offer extrapolation without structural explanation (Kaplan et al., 2020), and transfer across domains remains unreliable when we cannot distinguish essential architectural features from incidental ones. The result is a methodology that scales through brute-force compute rather than understanding.

This paper introduces the *Axiomatic Atlas*, a framework that makes architecture requirements explicit, composable, and verifiable. The core idea is to encode requirements as *axioms* organized into four classes. *Structural axioms* (S) govern graph connectivity and require that every input-to-output path passes through at least one nonlinear operator. *Primitive axioms* (P) specify per-operator contracts: Lipschitz bounds from above and, crucially, derivative floors from below that prevent saturation. *Stability axioms* (E) control how these local properties compose across depth, bounding the product of per-layer Jacobians to prevent exponential attenuation. Finally, the *merge axiom* (M) ensures that when parallel branches recombine, information is preserved rather than canceled.

Given a declared operator library and wiring conventions (collectively called a *pack*), the Atlas constructs a *saturated graph* representing all possible executions, identifies the *active subgraph* for a given input, and computes its min-cut connectivity $\lambda_x$. The output is twofold: a *certificate* that lower-bounds input-to-output variation (Section 3), and a *diagnosis* that pinpoints which axioms fail and where. We prove such bounds under both exact arithmetic (Theorem 3.2) and IEEE-754 finite-precision semantics (Theorem 3.3), each stating the minimal axiom subset (the *closure*) required for the guarantee to hold. This modularity lets practitioners check only the relevant conditions for their architecture and read off the applicable bound.

The central contribution is that the framework is *prescriptive*: each axiom violation implies a targeted repair. A

[1] MIT CSAIL. Correspondence to: Minghao Guo <guomh2014@gmail.com>, Wojciech Matusik <wojciech@csail.mit.edu>.

*Proceedings of the 43rd International Conference on Machine Learning*, Seoul, South Korea. PMLR 306, 2026. Copyright 2026 by the author(s).

min-cut of $\lambda_x = 1$ reveals a serial bottleneck; the repair is to add parallel pathways. A collapsed tail-gain factor $\Gamma_{E4} \to 0$ signals vanishing gradients through depth; the repair is spectral regularization or depth-aware scaling. A merge-axiom violation indicates destructive interference; the repair is branch normalization or orthogonal projection. Architecture design thus reduces from open-ended search to constraint satisfaction: diagnose the binding constraint, invert the violation, verify the fix.

We validate this prescriptive loop on four interventions spanning four architecture families. For GNNs, Atlas-guided bridge placement recovers $+46$ percentage points over random placement on bottlenecked graphs (Section 5). For mixture-of-experts under quantization, anchoring the quantization grid to stable percentiles yields $3\times$ better INT4 robustness than min-max baselines (Section 5). For sparse routing, dynamic-$k$ allocation guided by routing uncertainty closes 83% of the gap between $k=2$ and $k=4$ at 62% of the compute (Section 5). For long-context attention, orthogonal key projection transforms 0% retrieval accuracy into 100% by removing shared-mode interference (Section 5). Each experiment includes matched negative controls (random bridges, random $k$-allocation, cosine normalization) that isolate the diagnosed mechanism from confounds. Four additional interventions and comprehensive transformer audits appear in the appendix.

The Atlas is a diagnostic framework: it certifies necessary conditions for sensitivity floors, not sufficient conditions for task performance. All bounds are conditional on pack declarations, and users must verify that operators satisfy stated contracts. We treat architectures with explicit computation graphs; extensions to implicit layers and stochastic execution are discussed in the appendix. Section 2 formalizes the axiom hierarchy; Section 3 states the main theorems; Section 4 presents algorithms; Section 5 reports experiments; Section 6 surveys related work.

## 2. Formalism

We introduce the Atlas object model through a running example: a single pre-norm transformer block, following the convention in Figure 1 that vertices are feature tensors and edges are operations. This block takes input $x$, applies layer normalization, splits into attention and feedforward branches, and adds the result back via a residual connection. By the end of this section, we will have diagnosed why this block has min-cut connectivity $\lambda_x = 1$ (a serial bottleneck at normalization), which axioms it satisfies, and what certificate it produces.

The key design principle is *separation*: structural properties (graph connectivity, pathway existence) are distinguished from quantitative properties (derivative magnitudes, signal

---

**Algorithm 1** Constructing a Pre-Norm Transformer Pack

**Require:** Hidden width $d$, arithmetic model $A$
 1: Set `type_library`: LayerNorm is nonaffine; Linear is affine; Softmax and GELU are nonlinear; Add is a merge.
 2: Set `arithmetic`: $A$.
 3: Set `conventions`: pre-norm residuals, width-$d$ tensor shapes, attention edges separate from feature nodes.
 4: Set `nodes`: source, LN1, Q/K/V, attention, output projection, Add1, LN2, FFN up, GELU, FFN down, Add2, sink.
 5: Set `wiring`: main chain through attention and FFN, plus residual edges source-to-Add1 and Add1-to-Add2.
 6: Set `contracts`: shape checks, per-operator Lipschitz bounds, derivative floors, and merge survival tests.
 7: **return** $\Pi$

---

survival). This separation enables cheap structural audits to precede expensive numerical calibration.

### 2.1. Packs and Computation Graphs

Consider implementing our transformer block. We must choose: which normalization (LayerNorm or RMSNorm), which activation (GeLU or ReLU), what precision (float32 or bfloat16), and where residuals connect. These choices define the space of possible computations. We formalize this as a *pack*.

**Definition 2.1** (Pack). A pack $\Pi = (\mathcal{O}, \mathcal{T}, \mathcal{C})$ consists of an operator library $\mathcal{O}$ (linear maps, activations, normalizations, attention, routing), a typing discipline $\mathcal{T}$ (tensor shapes, dtypes, broadcasting rules), and wiring conventions $\mathcal{C}$ (residual placement, parameter sharing, layer ordering).

The abstract triple above is implemented as a concrete six-field record: `type_library` (operator tags and signatures), `arithmetic` (Real-RAM or IEEE-754), `conventions` (e.g., whether normalization is tagged nonaffine), `nodes`, `wiring`, and `contracts` (Lipschitz, spectral, or range bounds).

Running Algorithm 1 yields a graph checked by `is_dag`, min-cut, and the S2 nonlinear-cut test. Changing any record field–for example switching arithmetic to IEEE-754 or changing the normalization convention–changes which axioms apply and which certificate is valid.

The pack serves as a contract: all guarantees are conditional on operators conforming to declared specifications. For our transformer block, a typical pack might specify $\mathcal{O} = \{\text{LayerNorm}, \text{Linear}, \text{GeLU}, \text{Softmax}, \text{Add}\}$, with float32 precision and pre-norm residual placement. Changing the pack changes which architectures are expressible and which axioms can be satisfied.

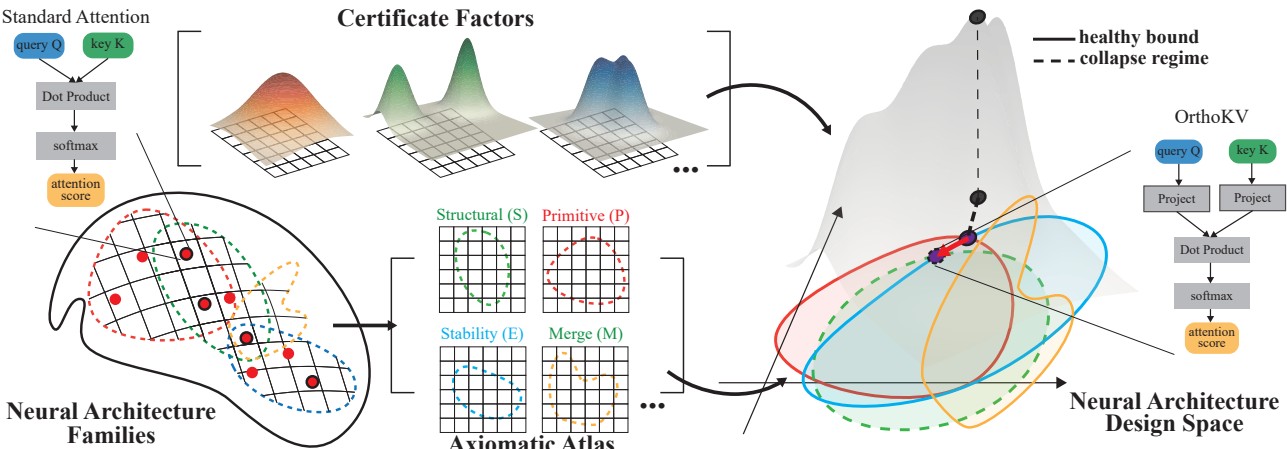

**Figure 1.** Overview of the Axiomatic Atlas framework. Nodes denote feature tensors and directed edges denote operator applications. A neural architecture (left: standard attention) is audited against four axiom classes ($\mathbf{S}, \mathbf{P}, \mathbf{E}, \mathbf{M}$), producing certificate factors that locate the design in axiom space. Collapse is encoded by factors approaching the dashed zero-boundary; repair is encoded as a targeted graph/operator modification (right: OrthoKV) that restores the solid healthy-bound region.

Given a pack, the Atlas constructs a directed graph representing all possible computations. For our transformer block, this graph has vertices for each intermediate tensor (normalized input, attention output, FFN output) and edges for each operation. Crucially, the graph includes *all* paths permitted by the pack, even those that may not activate for every input.

**Definition 2.2** (Saturated Graph). The saturated graph $G_{\text{sat}} = (V, E)$ is a directed acyclic graph with distinguished source $S_{\text{sat}}$ (input) and sink $\tau$ (output). Vertices represent intermediate tensors; edges represent operator applications. $G_{\text{sat}}$ includes all execution paths permitted by the pack, including conditionally executed branches.

Not all paths execute for every input. In a mixture-of-experts layer, top-$k$ routing activates only $k$ of $n$ expert branches. In attention with causal masking, certain key-value pairs are blocked. The *active subgraph* captures what actually executes.

**Definition 2.3** (Active Subgraph). For input $x$, the active subgraph $G_x \subseteq G_{\text{sat}}$ contains only edges traversed during the forward pass. Data-dependent operations (top-$k$ routing, attention masking, early exit) determine which edges activate.

For our transformer block without conditional computation, $G_x = G_{\text{sat}}$ for all inputs. But for an MoE variant selecting 2 of 8 experts, $G_x$ would contain only the two active expert subgraphs, varying with $x$. The structural bottleneck of an architecture is captured by how many edges must be cut to disconnect input from output.

**Definition 2.4** (Min-Cut Connectivity). $\lambda_x$ is the minimum number of edges whose removal disconnects $S_{\text{sat}}$ from $\tau$ in $G_x$. By the max-flow min-cut theorem, $\lambda_x$ equals the maximum number of edge-disjoint paths from input to output.

For our pre-norm transformer block, consider the information flow: input $x$ passes through LayerNorm, then splits to attention and FFN paths, then recombines via residual addition. Where is the bottleneck? The LayerNorm processes the *entire* input through a single operation before any split occurs. Removing the single edge into LayerNorm disconnects input from output. Thus $\lambda_x = 1$: all information must pass through this serial bottleneck.

This diagnosis is architectural, not a bug. Pre-norm transformers *by design* have $\lambda_x = 1$ at each normalization layer. MoE layers with top-$k$ routing have $\lambda_x \leq k$ because at most $k$ expert paths are active. Architectures with $\lambda_x \geq 2$ admit parallel pathways providing redundancy that no single-edge failure can eliminate.

## 2.2. Axiom Hierarchy

The Atlas encodes requirements as axioms in four classes, ordered from purely structural to fully quantitative. We illustrate each with our transformer block.

**Structural Axioms (S): Graph Topology.** These depend only on the graph, not on parameter values.

**S1 (DAG):** $G_{\text{sat}}$ is a finite directed acyclic graph with unique source and sink. Our transformer block satisfies S1: information flows from input through normalization, attention, FFN, and residual addition to output, with no cycles. Recurrent architectures require unrolling to satisfy S1.

**S2 (Nonlinear Cut):** Every minimum cut in $G_x$ contains

at least one nonlinear operator. This prevents the network from collapsing to an affine function. Our transformer block satisfies S2: the minimum cut (at LayerNorm) contains a nonlinear operator (LayerNorm is nonlinear due to its data-dependent normalization).

When does S2 fail? Consider a residual block $y = x + F(x)$ where $F$ is a gated network. If the gate closes completely, $F(x) = 0$ and $y = x$: a purely linear bypass. The minimum cut now contains only the identity skip connection, violating S2. The SealSkip intervention (Appendix C.2) addresses this by inserting a lightweight nonlinearity on the skip path.

**Primitive Axioms (P): Operator Contracts.** These specify per-operator requirements that enable composition into global bounds.

**P1 (Typing):** Operators respect declared shapes and dtypes.

**P2 (Continuity):** Operators are continuous.

**P3 (Piecewise Differentiability):** Operators are differentiable except on measure-zero boundaries (e.g., ReLU at zero).

**P4 (Lipschitz Bound):** Each operator $f$ satisfies $\|f(x) - f(x')\| \leq L_f \|x - x'\|$, bounding sensitivity from above.

These four axioms are standard regularity conditions. The key axiom distinguishing our framework is P5:

**P5 (Derivative Floor):** Each nonlinear operator $\phi$ satisfies $\sigma_{\min}(D\phi) \geq \gamma_\phi > 0$ on the active region.

P5 bounds sensitivity *from below*, ensuring nonlinearities do not saturate. In our transformer block, GeLU satisfies P5 in its near-linear regime but may violate it for extreme inputs where the activation saturates. Softmax satisfies P5 when no logit dominates, but violates it when attention concentrates entirely on one token ($\gamma \to 0$). Quantized activations may have $\gamma = 0$ exactly, which the Q-Anchor intervention (Section 5) addresses.

**Stability Axioms (E): Depth Composition.** These control how per-layer properties compose across depth.

**E1 (Bounded Depth):** The longest path has finite length $D$.

**E2 (Layer Bound):** Per-layer Jacobians satisfy $\|J_\ell\|_{\mathrm{op}} \leq B_\ell$.

**E3 (Composition Bound):** $\prod_\ell B_\ell \leq B_{\mathrm{global}}$, preventing exponential blowup.

**E4 (Tail Gain):** There exists $\Gamma > 0$ such that $\prod_\ell \gamma_\ell \geq \Gamma$ along at least one path.

Here and below, $D\phi$ denotes the Jacobian of a nonlinear primitive $\phi$, $J_\ell$ denotes the layer-$\ell$ Jacobian, $D$ without an argument denotes depth, and $\sigma_{\min}$ is the smallest singular value on the declared active subspace. E4 is the depth analog of P5: it ensures signals are not exponentially attenuated through depth. A single transformer block trivially satisfies E4, but stacking 96 layers can cause $\Gamma$ to collapse. In our experiments (Section 5), we find that tiny-gpt2 (2 layers) exhibits $\Gamma_{E4} \approx 10^{-9}$ (collapsed), while GPT-2 (12 layers) maintains $\Gamma_{E4} > 0.07$ (healthy). When $\Gamma \to 0$, certificates become vacuous. The tail-survival intervention (Appendix C.4) addresses E4 collapse in state-space models via auxiliary losses.

**Merge Axiom (M): Branch Combination.** This ensures information preservation when parallel branches recombine.

**M (Merge Survival):** At each merge combining branches $\{b_i\}$, if branch $b_j$ carries magnitude $s$, then $\|\oplus(\{b_i\})\| \geq \mu \cdot s$ for $\mu > 0$.

In our transformer block, the residual addition $y = x + \mathrm{Attn}(x)$ is a merge. If the attention output happens to equal $-x$, perfect cancellation occurs and $\mu = 0$. This is unlikely in practice but becomes problematic in long-context attention where many keys share a common component, causing interference. The OrthoKV intervention (Section 5) removes this shared component before the attention merge.

### 2.3. Certificates and Diagnosis

A certificate aggregates all factors into a single variation bound.

**Definition 2.5** (Certificate). For input $x$ and parameters $\theta$, a certificate is $\mathcal{K} = (x, \theta, \lambda_x, \gamma_{P5}, \Gamma_{E4}, \mu_M, \mathcal{A}_{\mathrm{req}})$ where:

- $\lambda_x$: min-cut connectivity (structural)
- $\gamma_{P5} = \min_\phi \gamma_\phi$: minimum derivative floor across active nonlinearities
- $\Gamma_{E4}$: tail gain bounding signal attenuation through depth
- $\mu_M$: minimum merge survival factor
- $\mathcal{A}_{\mathrm{req}}$: the axiom closure required for the bound

The variation bound takes multiplicative form:

$$\mathrm{TV} \geq \lambda_x \cdot \gamma_{P5} \cdot \Gamma_{E4} \cdot \mu_M \cdot r \tag{1}$$

for perturbation radius $r$. This product structure is diagnostic: when any factor approaches zero, the bound becomes vacuous, and that factor identifies the failure mode.

For our transformer block, a typical certificate might report $\lambda_x = 1$ (the LayerNorm bottleneck), $\gamma_{P5} = 0.12$ (healthy nonlinearities), $\Gamma_{E4} = 0.85$ (minimal depth attenuation), and $\mu_M = 0.91$ (low cancellation at residual merges). If we instead found $\lambda_x = 1$ but $\gamma_{P5} \approx 0$, the diagnosis would point to saturated activations rather than structural bottlenecks, suggesting a different repair.

## 2.4. Axiom Closures

Rather than requiring all axioms universally, each theorem states its *axiom closure*: the minimal sufficient subset.

**Definition 2.6** (Axiom Closure). The closure $\mathcal{A}(T)$ of theorem $T$ is the smallest axiom set such that $T$ holds whenever all axioms in $\mathcal{A}(T)$ are satisfied.

This enables modular verification. The Real-RAM bound (Theorem 3.2) requires {S2, P1–P4, E1, E4, M}; the IEEE-754 bound (Theorem 3.3) requires a different closure accounting for finite-precision rounding. Practitioners check which axioms their architecture satisfies and read off the applicable guarantee without verifying irrelevant conditions. The full hierarchy, dependency structure, and proof ledger appear in Appendix A.

# 3. Main Results

We now state the main theoretical results: lower bounds on path total variation under exact and finite-precision arithmetic. Each theorem specifies its axiom closure, enabling practitioners to verify only the relevant conditions for their architecture.

## 3.1. Path Total Variation

We measure architecture sensitivity via total variation along perturbation rays. Intuitively, if we perturb an input $x$ in direction $v$ and track how the output changes, total variation accumulates all these changes regardless of sign.

**Definition 3.1** (Path Total Variation). For network $N_\theta : \mathbb{R}^d \to \mathbb{R}^m$, input $x$, unit direction $v$, and radius $r > 0$, the path total variation is

$$\mathrm{TV}[N_\theta; x, v, r] \ = \ \int_0^r \left\| \frac{d}{dt} N_\theta(x + tv) \right\| dt.$$

A network with high TV responds meaningfully to input perturbations; a network with near-zero TV is effectively constant in that direction and thus incapable of discrimination. For our transformer block from Section 2, high TV means that perturbing the input embedding produces measurable changes in the output logits.

The Atlas provides *lower bounds* on TV. This complements the standard Lipschitz perspective: Lipschitz bounds certify that outputs cannot change *too much* (stability), while TV lower bounds certify that outputs cannot change *too little* (responsiveness). Both are necessary for a well-behaved architecture.

## 3.2. Real-RAM Variation Bound

Our first result assumes exact arithmetic, isolating structural and quantitative factors from numerical precision effects.

**Theorem 3.2** (Real-RAM Variation Bound). *Let $N_\theta$ be a network with pack $\Pi$ satisfying axioms* {S2, P1-P4, E1, E4, M}. *For input $x$ with active subgraph $G_x$, direction $v$, and radius $r$, the path total variation satisfies*

$$\mathrm{TV}[N_\theta; x, v, r] \ \geq \ \lambda_x \cdot \gamma_{P5}^{\min} \cdot \Gamma_{E4} \cdot \mu_M \cdot r$$

*where $\lambda_x$ is min-cut connectivity, $\gamma_{P5}^{\min}$ is the minimum derivative floor across active nonlinearities, $\Gamma_{E4}$ is the tail-gain factor, and $\mu_M$ is the merge survival factor.*

The axiom closure {S2, P1-P4, E1, E4, M} is minimal: removing any axiom admits a counterexample (Appendix B provides independence proofs).

The bound's multiplicative structure reveals how different architectural properties contribute to sensitivity. The factor $\lambda_x$ captures structural connectivity: our transformer block has $\lambda_x = 1$ due to its LayerNorm bottleneck, while an MoE layer with top-2 routing has $\lambda_x \leq 2$. The factor $\gamma_{P5}^{\min}$ captures the weakest nonlinearity: if any activation saturates, this factor collapses. The factor $\Gamma_{E4}$ captures depth attenuation: stacking many layers compounds per-layer signal loss. The factor $\mu_M$ captures merge fidelity: residual additions that cancel reduce this factor toward zero.

The diagnostic power comes from this factorization. When the bound becomes vacuous (approaching zero), exactly one factor is typically responsible, and that factor identifies both the failure mode and the appropriate repair.

We use the following notation throughout. $\sigma_{\min}(A)$ is the smallest singular value of the linear map $A$ on the stated effective subspace; $D\phi$ denotes the Jacobian of a primitive $\phi$; $D$ in E1 denotes depth and is not a derivative; $\phi$ denotes a nonlinear activation or nonaffine primitive; $\ell$ indexes layers; and Real-RAM means exact real arithmetic, separated from IEEE-754 rounding in Theorem 3.3. Quantitative factors are estimated on the declared input distribution by JVP sampling, while structural factors such as S1, S2, and $\lambda_x$ are data-independent. A factor near zero triggers diagnosis; in our GPT-style audits, for instance, $\Gamma_{E4} < 10^{-6}$ flags tail collapse, whereas healthy models have much larger measured tail gains.

The proof of Theorem 3.2 is a cellwise Real-RAM argument. On a refined $C^1$ cell along the ray, P5 supplies a positive-measure set of parameters where the selected nonaffine bottleneck has a directional derivative floor. E4 propagates this vector through an active suffix with gain at least $\Gamma_{E4}$, and M applies a projection that isolates the merge contribution with survival factor $\mu_M$. The absolute-continuity identity for total variation then integrates the resulting derivative floor over the ray, while $\lambda_x$ records how many active bottleneck paths can anchor the certificate. Appendix B gives the full quantitative proof, including the projection and multi-suffix

variants.

### 3.3. IEEE-754 Event Bound

Finite-precision arithmetic introduces discontinuities at rounding boundaries that the Real-RAM analysis misses. When activations are quantized to INT8 or INT4, the network becomes a piecewise-constant function with jumps at quantization thresholds. Our second result captures variation from these discrete events.

**Theorem 3.3** (IEEE-754 Event Bound). *Let $N_\theta$ be a network with pack $\Pi$ satisfying axioms $\{\mathrm{S2}, \mathrm{P1\text{-}P4}, \mathrm{E1}, \mathrm{E4\text{-}quant}, \mathrm{M}\}$ under IEEE-754 arithmetic with $p$ mantissa bits. For input $x$, direction $v$, and radius $r$, the path total variation satisfies*

$$\mathrm{TV}[N_\theta; x, v, r] \geq \kappa_x \cdot \Delta_{\min} \cdot r$$

*where $\kappa_x$ is the number of rounding events along the path and $\Delta_{\min}$ is the minimum jump magnitude at event boundaries.*

The axiom E4-quant replaces smooth tail-gain with discrete event-counting appropriate for piecewise-constant rounding. This bound captures a complementary mechanism: even when smooth derivatives vanish (as they do between quantization levels), discontinuous jumps at precision boundaries contribute irreducibly to total variation.

Lower precision increases both the frequency of rounding events (more threshold crossings per unit perturbation) and the magnitude of jumps (coarser quantization levels). Our Q-Anchor experiment (Section 5) validates this tradeoff: INT4 quantization without anchoring collapses $\gamma_{P5}^{\min}$ because the quantization grid misaligns with the activation distribution, while anchoring to stable percentiles preserves sensitivity.

### 3.4. Collapse Regimes

A certificate enters the *collapse regime* when one or more factors approach zero, rendering the bound uninformative. If $\lambda_x = 0$, the active graph has no valid path from input to output, so the architecture fails the structural precondition before a certificate can be read. Conditional on at least one active path, the Atlas distinguishes four certificate-collapse modes, each with distinct causes and repairs.

**Bottleneck collapse** occurs when $\lambda_x$ is positive but too small for the task, exposing a single fragile cut; bridge placement or additional active branches repair it. **Saturation collapse** occurs when $\gamma_{P5}^{\min} \to 0$, so nonlinearities flatten or quantization bins freeze the active region; anchoring, rescaling, or alternative primitives repair the derivative floor. **Tail collapse** occurs when $\Gamma_{E4} \to 0$ through depth; spectral budgets, dynamic compute allocation, or tail-survival losses repair the suffix gain. **Interference collapse** occurs

---

**Algorithm 2** Structural Audit

**Require:** Active subgraph $G_x = (V, E)$
 1: Assign unit capacity to all edges
 2: $\lambda_x \leftarrow \mathrm{MAXFLOW}(G_x, S_{\mathrm{sat}}, \tau)$
 3: Assign capacity 0 to nonlinear edges, 1 to affine edges
 4: $f_{\mathrm{affine}} \leftarrow \mathrm{MAXFLOW}(G_x, S_{\mathrm{sat}}, \tau)$
 5: S2_violated $\leftarrow$ ($f_{\mathrm{affine}} > 0$) **return** $\lambda_x$, S2_violated, violating path if any

---

when $\mu_M \to 0$ at a merge; branch normalization, protected projections, or orthogonalization repair the merge.

## 4. Diagnostic Pipeline

This section presents the algorithms that compute certificates and diagnoses. The pipeline has two stages: a structural audit (graph-based, parameter-free) followed by quantitative calibration (requires forward passes).

### 4.1. Structural Audit

The structural audit extracts $\lambda_x$ and checks S2 using only the computation graph, without examining parameter values.

**Min-Cut Computation.** Given active subgraph $G_x$, we compute $\lambda_x$ via max-flow. Assign unit capacity to each edge, add a super-source connected to all input nodes and a super-sink connected to all output nodes, then run Ford-Fulkerson. The max-flow value equals $\lambda_x$ by the max-flow min-cut theorem. Complexity is $O(|E| \cdot \lambda_x)$, which is $O(|E|)$ for typical architectures where $\lambda_x$ is small.

**Affine Bypass Detection (S2).** To check S2, we test whether any purely affine path exists from source to sink. Assign capacity 1 to affine edges (linear layers, additions, identity) and capacity 0 to nonlinear edges (activations, softmax, LayerNorm). If max-flow is positive, an affine bypass exists and S2 is violated. The algorithm returns the violating path for diagnosis.

The structural audit runs in linear time and requires no gradient computation, making it suitable for rapid screening of architecture candidates before committing to training.

### 4.2. Quantitative Calibration

Quantitative calibration estimates the certificate factors $\gamma_{P5}^{\min}$, $\Gamma_{E4}$, and $\mu_M$ by sampling Jacobians along the computation graph.

**Derivative Floor** ($\gamma_{P5}^{\min}$). For each nonlinear operator $\phi$ in $G_x$, sample $n$ inputs from the active region and compute the minimum singular value of the Jacobian: $\gamma_\phi = \min_i \sigma_{\min}(D\phi(x_i))$. The global floor is $\gamma_{P5}^{\min} = \min_\phi \gamma_\phi$.

**Algorithm 3** Quantitative Calibration

---

**Require:** Active subgraph $G_x$, sample count $n$
 1: **for** each nonlinear operator $\phi \in G_x$ **do**
 2:     Sample $n$ inputs, compute $\gamma_\phi \leftarrow \min_i \sigma_{\min}(D\phi(x_i))$
 3: **end for**
 4: $\gamma_{P5}^{\min} \leftarrow \min_\phi \gamma_\phi$
 5: **for** each sampled path $p$ from source to sink **do**
 6:     $\Gamma_p \leftarrow \prod_{\ell \in p} \gamma_\ell$
 7: **end for**
 8: $\Gamma_{E4} \leftarrow \min_p \Gamma_p$
 9: **for** each merge vertex $v$ **do**
10:     Inject unit signal per branch, measure $\mu_v$
11: **end for**
12: $\mu_M \leftarrow \min_v \mu_v$
13: **return** $\gamma_{P5}^{\min}, \Gamma_{E4}, \mu_M$

---

**Tail Gain ($\Gamma_{E4}$).** For each source-to-sink path $p$, compute the product of per-layer singular value floors: $\Gamma_p = \prod_{\ell \in p} \gamma_\ell$. The certificate uses $\Gamma_{E4} = \min_p \Gamma_p$ over sampled paths. In practice, we sample the $k$ shortest paths and the path through the min-cut edges, as these are most likely to be bottlenecks.

**Merge Survival ($\mu_M$).** At each merge vertex $v$ combining branches $\{b_i\}$, inject unit signal into one branch while zeroing others, then measure output magnitude. The survival factor is $\mu_v = \min_j \| \oplus (0, \ldots, e_j, \ldots, 0)\|$. The global factor is $\mu_M = \min_v \mu_v$.

Calibration requires $O(n \cdot |\phi|)$ forward passes for derivative floors and $O(k \cdot D)$ for path products, where $|\phi|$ is the number of nonlinear operators, $k$ the number of sampled paths, and $D$ the depth.

### 4.3. Certificate Assembly

The final certificate combines structural and quantitative outputs:
$$\mathcal{K} = (\lambda_x, \gamma_{P5}^{\min}, \Gamma_{E4}, \mu_M, \mathcal{A}_{\mathrm{sat}})$$
where $\mathcal{A}_{\mathrm{sat}}$ records which axioms are satisfied. The diagnosis reports any violated axioms and the specific graph locations responsible, enabling targeted repair.

If all factors are healthy (above architecture-specific thresholds), the certificate provides a positive variation lower bound per Theorems 3.2 and 3.3. If any factor collapses, the diagnosis identifies the bottleneck and suggests the repair:

- $\lambda_x = 1$: structural bottleneck, add parallel pathways

- $\gamma_{P5}^{\min} \approx 0$: saturated activations, use anchoring or alternative nonlinearities

- $\Gamma_{E4} \approx 0$: depth attenuation, apply spectral constraints

- $\mu_M \approx 0$: merge interference, add normalization or orthogonalization

As a concrete example, auditing GPT-2 (12 layers, 117M parameters) yields $\lambda_x = 1$ at each LayerNorm, $\Gamma_{E4} = 0.07$, and no S2 violations.

The full pipeline runs in time linear in graph size plus the cost of Jacobian-vector-product (JVP) sampling, making it practical for large architectures because it never forms full Jacobians. On Mistral-7B-v0.1, the computation graph has 326 nodes and 389 edges. Graph construction takes 0.097s, the structural audit takes 0.005s, and the JVP probe takes 1.98s, for a 2.08s total certificate. The JVP term dominates; a conservative linear extrapolation by active graph/JVP cost gives roughly 21s at 70B scale and 208s at 700B scale as a one-time profiling diagnostic.

## 5. Experimental Validation

We validate the Atlas on four interventions, each targeting a different certificate factor. Each experiment follows the same template: audit the baseline to identify the collapsing factor, derive an intervention from the axiom violation, and compare against matched negative controls that preserve confounds (parameter count, compute budget) while breaking the hypothesized mechanism.

### GNN Bridge Placement (Structural Factor $\lambda_x$)

We test Atlas-guided edge placement on graphs with structural bottlenecks. The task is associative recall: a binary label at a key node in the left partition must be recovered by targets in the right partition, with information flowing through a single cross-partition edge ($\lambda_x = 1$). The Atlas diagnoses this bottleneck and prescribes adding bridges that connect the signal-carrying key node to the augmented structure. We compare against `random` (uniform placement), `random_cross` (forced cross-partition), `key_random` (key node with random endpoints), and `separator` (near bottleneck but ignoring key node).

Table 1 reports results at edge budget $m$=8. Atlas-guided placement achieves 97.0% accuracy (+45.8pp over the 51.2% baseline), while random placement yields only 51.8% despite increasing median connectivity from $\hat{\lambda}$=1.0 to 3.15. The controls confirm that *where* bridges attach matters more than how many: `key_random` achieves 95.8% (signal-source participation is key), while `separator` achieves only 51.1% (proximity to the bottleneck alone is insufficient).

### MoE Quantization Anchoring (Derivative Factor $\gamma_{P5}^{\min}$)

MoE routing is sensitive to quantization because subtle logit differences determine expert selection. The Atlas diagnoses $\gamma_{P5}^{\min} \to 0$ under aggressive quantization: the sigmoid

*Table 1.* GNN bridge placement ($m$=8). Atlas guidance achieves +45.8pp; random placement shows no gain despite increased $\hat{\lambda}$.

| Variant | Accuracy (%) | $\Delta$ | $\hat{\lambda}_{\mathrm{med}}$ |
|---|---|---|---|
| baseline | $51.2 \pm 2.8$ | — | 1.00 |
| random | $51.8 \pm 2.5$ | $+0.6$ | 3.15 |
| random_cross | $54.2 \pm 1.8$ | $+3.0$ | 3.25 |
| atlas | $\mathbf{97.0 \pm 0.2}$ | $+\mathbf{45.8}$ | 4.00 |

*Table 2.* MoE quantization robustness. Q-Anchor achieves $3\times$ better INT4 retention than baseline.

| Method | FP32 | INT8 | INT4 | Retention |
|---|---|---|---|---|
| Baseline | 25.4% | 18.7% | 11.9% | 47% |
| Q-Anchor | $\mathbf{49.7}\%$ | $\mathbf{49.7}\%$ | $\mathbf{36.7}\%$ | $\mathbf{74}\%$ |

*Table 3.* Dynamic-$k$ routing on WikiText-2. Adaptive allocation closes 83% of the gap at 62% compute.

| Variant | Val PPL ($\downarrow$) | $\bar{k}$ | Gap Closed |
|---|---|---|---|
| STATIC-K2 | 228.85 | 2.0 | — |
| STATIC-K4 | 223.05 | 4.0 | 100% |
| DYNAMIC-K | $\mathbf{224.02}$ | 2.48 | $\mathbf{83}\%$ |
| RANDOM-K | 227.57 | 2.38 | 22% |

*Table 4.* OrthoKV eliminates long-context interference. Baseline collapses at $L$=2048; OrthoKV maintains 100%.

| Context $L$ | Variant | Accuracy | Isolation |
|---|---|---|---|
| 32 | Baseline | 100% | 0.401 |
| | OrthoKV | 100% | 0.990 |
| 2048 | Baseline | 0% | 0.010 |
| | OrthoKV | $\mathbf{100}\%$ | 0.532 |

gate saturates when quantization noise pushes activations to extremes. The prescribed repair is **Q-Anchor**: align the quantization grid with stable activation percentiles ($p_{25}$, $p_{75}$) rather than min-max bounds, preserving fidelity in the high-gradient region.

Table 2 shows that at INT4, baseline collapses to 11.9% (47% retention relative to FP32), while Q-Anchor maintains 36.7% (74% retention): a $3\times$ improvement. Gate activation analysis confirms the mechanism: baseline INT4 exhibits near-zero directional diversity (the router freezes into a static policy), while Q-Anchor preserves input-dependent routing even at 4-bit precision.

**Dynamic-$k$ Expert Routing (Tail Factor $\Gamma_{E4}$)**

Static top-$k$ routing wastes compute on easy tokens and starves hard tokens. The Atlas identifies that low-$k$ routing causes per-layer signal attenuation on high-uncertainty tokens: when routing entropy is high, the top-$k$ experts capture only a fraction of the probability mass, and this "dropped" signal compounds across layers as $\Gamma_{E4}$ collapse. The prescribed repair is **dynamic-$k$**: allocate $k \in \{2, 4\}$ based on routing uncertainty, using $k$=4 when drop probability or entropy exceeds a threshold.

On a 6-layer Transformer with 8 experts per layer trained on WikiText-2, DYNAMIC-K achieves PPL 224.02, closing 83% of the gap between STATIC-K2 (228.85) and STATIC-K4 (223.05) at only 62% of the compute (Table 3). The RANDOM-K control uses similar average $\bar{k}$ but achieves only 22% gap closure, confirming that *which* tokens receive extra experts matters, not just how many.

**Orthogonal Key Projection (Merge Factor $\mu_M$)**

Long-context attention fails when keys share a common component that induces interference. We isolate this with associative recall from a cache of $L$ key-value pairs. The Atlas diagnoses $\mu_M \to 0$: the shared component diffuses attention across incorrect positions. The prescribed repair

is **OrthoKV**: project queries and keys onto the orthogonal complement of the empirical mean direction before computing attention.

At $L$=32, both baseline and OrthoKV achieve 100% accuracy (Table 4). At $L$=2048, baseline collapses to 0% (attention isolation drops to 0.010, nearly uniform across all keys), while OrthoKV maintains 100% (isolation 0.532). The failure is geometric: removing the shared component transforms a high-interference regime into one where standard softmax succeeds.

Across architecture families, the four locked interventions show +46pp structural repair, $3\times$ quantization robustness, 83% gap closure at 62% compute, and 0%→100% retrieval, with matched negative controls isolating the diagnosed mechanism.

Three additional rebuttal studies test usefulness outside the locked failure regimes. Mistral-7B-v0.1 profiling takes 2.08s end-to-end for a 326-node, 389-edge graph (Table 6); linear active-graph/JVP extrapolation gives roughly 21s at 70B and 208s at 700B, as a profiling pass rather than training or full evaluation. On 50 DARTS-like CIFAR-10 cells, Atlas structural health correlates with accuracy (Spearman $\rho = 0.430$, $p = 0.002$), with S2-passing cells at 70.1% versus 56.4% for S2-failing cells, supporting a zero-cost NAS proxy. Finally, gentle OrthoKV repairs to Mistral-7B-v0.1 are perplexity-neutral on WikiText-103 (Table 5), while aggressive K+V projection is destructive; LongBench Qasper and MultifieldQA_en show the same qualitative neutral pattern. Appendix C gives the full rebuttal audits. Appendix-only stress tests further cover the Goldilocks search and ablations (Section A.20), the realtext Goldilocks confirmation (Section A.21), and the WikiText-2 needle hardmode evaluation (Section A.22). These checks are not new headline claims; they document that the same collapse/repair pattern persists across synthetic, real-text, and

*Table 5.* Pretrained Mistral-7B OrthoKV sanity check on WikiText-103. Gentle repairs are perplexity-neutral; an aggressive value projection is destructive.

| Variant | PPL ($\downarrow$) |
|---|---|
| Baseline | 4.69 |
| K-only $\alpha=0.05$ | 4.75 |
| K+V $\alpha=0.1$ | 4.77 |
| K+V $\alpha=0.5$ | 40.50 |

*Table 6.* Mistral-7B certificate profiling. The expensive term is the single JVP; graph construction and structural auditing are negligible.

| Stage | Count | Time (s) |
|---|---|---|
| Graph construction | 326 nodes / 389 edges | 0.097 |
| Structural audit | same graph | 0.005 |
| JVP probe | one profiled batch | 1.98 |
| Total | — | 2.08 |

harder natural-text retrieval settings.

## 6. Related Work

**Structural Diagnostics.** Over-squashing work shows that GNN topology can bottleneck long-range message passing (Alon & Yahav, 2021; Topping et al.), with curvature and commute-time views adding geometric diagnoses (Akansha, 2025). Atlas instead uses min-cuts to return a placement repair. Signal propagation (Schoenholz et al., 2017; Yang & Schoenholz, 2017) and dynamical isometry (Saxe et al., 2013; Pennington et al., 2017) study depthwise sensitivity; our E-axioms make these checks modular across architectures. Dong et al. (2021) analyze attention rank collapse, while our merge axiom M captures related branch-combination failures.

**Repair Operators.** BatchNorm (Ioffe & Szegedy, 2015), LayerNorm (Ba et al., 2016b), and RMSNorm (Zhang & Sennrich, 2019) stabilize activation statistics; P5 formalizes the derivative floors they help maintain. Quantization-aware training (Jacob et al., 2018; Nagel et al., 2021) preserves accuracy under low precision, whereas Q-Anchor targets the diagnosed MoE gate-saturation mechanism. Graph rewiring (Gasteiger et al., 2019; Arnaiz-Rodríguez et al.) improves connectivity without Atlas-style placement criteria. MoE routing has moved from sparse gates (Shazeer et al., 2017) through load balancing (Fedus et al., 2022) to expert choice (Zhou et al., 2022); dynamic-$k$ instead adapts per-token capacity. For long-context attention, ALiBI (Press et al., 2021), RoPE (Su et al., 2024), and linear attention (Katharopoulos et al., 2020) address position or complexity, while OrthoKV targets key interference.

**Verification and Bounds.** Lipschitz analysis upper-bounds sensitivity (Szegedy et al., 2013; Gouk et al., 2021) for robustness certification (Wong & Kolter, 2018; Cohen et al., 2019); Atlas complements it with lower bounds certifying that outputs cannot change too little. Neural architecture search (Zoph & Le, 2017; Liu et al., 2019; White et al., 2023) optimizes proxy objectives over candidate spaces. Atlas is complementary: it diagnoses why a candidate fails, returns a repair class, and can annotate NAS cheaply. In a 50-architecture DARTS-like CIFAR-10 audit, Atlas structural health correlates with final accuracy (Spearman $\rho = 0.430$, $p = 0.002$), and S2-passing cells average 70.1% versus 56.4% for S2-failing cells (Sections A.8 and 5). Formal verification (Katz et al., 2017; Wang et al., 2021) checks trained-model properties; Atlas diagnoses architecture-level structural issues and actionable repairs.

## 7. Conclusion

We introduced the Axiomatic Atlas, a framework that makes neural architecture requirements explicit, composable, and verifiable. Its four axiom classes produce sensitivity certificates and graph-local diagnoses; because each violation maps to a repair class, architecture design becomes a constrained diagnose-and-repair loop rather than open-ended search.

Atlas is not a universal training oracle: it diagnoses architectural sensitivity failures under a declared pack, not data quality, optimization dynamics, capacity sufficiency, or absolute task accuracy. Structural factors are data-independent, while quantitative factors are distributional estimates unless separately certified; production use must still validate both the diagnosed factor and repair strength.

## Impact Statement

This work provides formal foundations for neural architecture design by systematizing diagnosis and repair of architectural failure modes. It may improve reliability and interpretability of deep learning systems; we do not foresee specific negative societal consequences beyond those common to general ML advances.

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

# A. Empirical Audits and Validations

## A.1. Transformer architecture audits

We validate the Atlas framework on five GPT-2 family models spanning four orders of magnitude in parameter count. These audits confirm that (1) structural diagnostics are consistent across scale, (2) bounds are valid and informative outside the collapse regime, (3) collapse is detectable via $\Gamma_{E4}$, and (4) numerical precision materially affects variation statistics.

### A.1.1. EXPERIMENTAL SETUP

**Models.** We evaluate five transformer language models: `tiny-gpt2` (2 layers, 0.1M parameters), `distilgpt2` (6 layers, 82M), `gpt2` (12 layers, 124M), `gpt2-medium` (24 layers, 355M), and `gpt2-large` (36 layers, 774M). All experiments use the WikiText-103 test split with 100 examples per model (maximum sequence length 128 tokens).

**Certificate computation.** For each model we compute the proxy structural connectivity $\hat{\lambda}_x$ and the certificate factors $\gamma_{P5}^{\min}$, $\Gamma_{E4}^{\min}$, and $\mu_M^{\min}$ using the diagnostic pipeline from Section 4. The tightened certificate mode instantiates the canonical variation bound in Theorem 3.2 as the lower-bound scale $\mathrm{TV}_{\mathrm{lb}}(r) = \hat{\lambda}_x \cdot \gamma_{P5}^{\min} \cdot \Gamma_{E4}^{\min} \cdot \mu_M^{\min} \cdot r$.

**Perturbation protocol.** We evaluate three direction sampling strategies: (i) *random* unit vectors, (ii) *gradient* directions maximizing logit magnitude, and (iii) *adversarial* directions minimizing the top-2 logit margin via 25 iterations of gradient descent. Radii span $r \in \{0.01, 0.05, 0.1, 0.2, 0.5, 1.0\}$. Empirical path total variation is measured using $K = 200$ samples along each ray. This yields $100 \times 3 \times 6 = 1800$ measurements per model (9000 total).

### A.1.2. STRUCTURAL PROPERTIES (AUDIT 1)

**Bottleneck connectivity.** Across all audited instances, the proxy connectivity satisfies $\hat{\lambda}_x = 1$ (Table 7). This indicates serial information flow consistent with the transformer template under declared conventions, where LayerNorm operations create sequential bottlenecks. The uniform $\hat{\lambda}_x = 1$ simplifies the bound to a product of certificate factors times $r$.

*Table 7.* Structural connectivity across GPT-2 family. All models exhibit $\hat{\lambda}_x = 1$.

| Model | Layers | Parameters | $\hat{\lambda}_x$ |
|---|---|---|---|
| tiny-gpt2 | 2 | 0.1M | 1 |
| distilgpt2 | 6 | 82M | 1 |
| gpt2 | 12 | 124M | 1 |
| gpt2-medium | 24 | 355M | 1 |
| gpt2-large | 36 | 774M | 1 |

**S2 audit.** The structural nonlinearity audit detects no affine-only bypass routes under the declared pack and convention. Every source-to-sink path passes through at least one nonlinear operator (activation function or softmax), satisfying axiom S2.

### A.1.3. BOUND TIGHTNESS (AUDIT 2)

We compare empirical path total variation $\mathrm{TV}_{\mathrm{emp}}(r)$ to the Atlas lower bound $\mathrm{TV}_{\mathrm{lb}}(r)$ instantiated from Theorem 3.2. The gap ratio $\mathrm{Gap} = \mathrm{TV}_{\mathrm{emp}}/\mathrm{TV}_{\mathrm{lb}}$ measures bound tightness (lower is tighter; $\geq 1$ confirms validity).

**Results by direction mode.** Table 8 reports median gap ratios by model and direction mode (600 measurements per cell). In the non-collapsed regime, tightness varies with direction mode, with random directions yielding the tightest bounds.

*Table 8.* Median gap ratios by model and perturbation direction. Lower values indicate tighter bounds. `tiny-gpt2` exhibits collapse.

| Model | Random | Adversarial | Gradient |
|---|---|---|---|
| tiny-gpt2 | — | $6.14 \times 10^{11}$ | — |
| distilgpt2 | $5.88 \times 10^2$ | $2.77 \times 10^4$ | $4.27 \times 10^4$ |
| gpt2 | $3.12 \times 10^3$ | $1.32 \times 10^5$ | $4.60 \times 10^5$ |
| gpt2-medium | $5.38 \times 10^2$ | $3.84 \times 10^4$ | $6.63 \times 10^4$ |
| gpt2-large | **57** | $6.75 \times 10^3$ | $6.67 \times 10^3$ |

**Interpretation.** The tightest bound occurs for `gpt2-large` under random perturbations (gap ratio 57×), indicating the certificate captures meaningful structure. Adversarial and gradient directions yield looser bounds because they find input directions where activation

patterns approach saturation boundaries, making the worst-case certificate factors overly conservative for typical inputs. The collapsed model `tiny-gpt2` exhibits a gap ratio of $6.14 \times 10^{11}$ under adversarial directions, reflecting $\Gamma_{E4}$ collapse rather than bound invalidity.

**Model-level summary.** Aggregating across all radii and direction modes, median gaps for non-collapsed models are: `distilgpt2` ($2.72 \times 10^4$), `gpt2` ($9.96 \times 10^4$), `gpt2-medium` ($2.95 \times 10^4$), and `gpt2-large` ($3.38 \times 10^3$). The collapsed `tiny-gpt2` has median gap $1.04 \times 10^{11}$.

### A.1.4. Numerical Precision Sensitivity (audit 3)

We evaluate precision effects by recomputing path total variation on `gpt2-large` under three IEEE-754 precision levels.

*Table 9.* Precision effects on `gpt2-large`. Lower mantissa precision increases variation and jump magnitudes.

| Precision | Mantissa Bits | Median TV | Median Max Jump |
|---|---|---|---|
| float32 | 23 | 0.236 | 0.0059 |
| float16 | 10 | 0.262 | 0.0098 |
| bfloat16 | 7 | 1.375 | 0.0313 |

Median total variation increases from 0.236 (float32) to 1.375 (bfloat16), a $5.8\times$ increase. Maximum jump magnitudes likewise increase from 0.0059 to 0.0313. These results support the precision-aware posture of the framework: arithmetic choices materially affect observed variation, motivating the IEEE-754 theorem (Theorem 3.3) and its explicit rounding error terms.

### A.1.5. Certificate Scaling and Collapse (audit 4)

**Certificate factors across scale.** Table 10 reports the tail-gain factor $\Gamma_{E4}^{\min}$, which exhibits the dominant cross-scale variation.

*Table 10.* Tail-gain factor $\Gamma_{E4}^{\min}$ across GPT-2 family. Only boundary values reported; intermediate models fall within the stated range.

| Model | Layers | $\Gamma_{E4}^{\min}$ | Collapsed? |
|---|---|---|---|
| `tiny-gpt2` | 2 | $5.05 \times 10^{-9}$ | **Yes** |
| `distilgpt2` | 6 | $6.95 \times 10^{-2}$ | No |
| `gpt2` | 12 | — | No |
| `gpt2-medium` | 24 | — | No |
| `gpt2-large` | 36 | $5.56 \times 10^{-1}$ | No |

*Note: Non-collapsed models span $\Gamma_{E4} \in [6.95 \times 10^{-2}, 5.56 \times 10^{-1}]$.*

**Collapse detection.** Under the threshold $\Gamma_{E4} < 10^{-6}$, `tiny-gpt2` is flagged as collapsed ($\Gamma_{E4} \approx 5.05 \times 10^{-9}$). This reflects depth attenuation in the 2-layer model: signal variance decays exponentially through layers, and insufficient depth prevents recovery. Non-collapsed models maintain $\Gamma_{E4} \in [6.95 \times 10^{-2}, 5.56 \times 10^{-1}]$, indicating healthy signal propagation.

**Verification time scaling.** Verification cost scales nearly linearly with graph size. A power-law fit yields exponent $\alpha = 0.994$, confirming the $O(|E|)$ complexity claim from Section 4.

### A.1.6. Calibration Analysis (audit 5)

We evaluate whether certificate-derived scales correlate with empirical prediction-flip thresholds.

**Protocol.** For each example, we compute an empirical flip radius $r_{\text{flip}}$ via binary search under adversarial perturbations (maximum search radius 10.0). The certificate scale is $r_{\text{cert}} = \tau/(\hat{\lambda}_x \cdot \gamma_{P5}^{\min} \cdot \Gamma_{E4}^{\min} \cdot \mu_M^{\min})$ with $\tau = 0.5$. We report the calibration ratio $\rho = \tilde{r}_{\text{flip}}/r_{\text{cert}}$, where $\tilde{r}_{\text{flip}}$ is the median flip radius.

*Table 11.* Calibration analysis. $\rho = \tilde{r}_{\text{flip}}/r_{\text{cert}}$ measures certificate conservatism.

| Model | $r_{\text{cert}}$ | $\rho$ |
|---|---|---|
| `tiny-gpt2` (collapsed) | $3.84 \times 10^{11}$ | $3.01 \times 10^{-13}$ |
| Non-collapsed boundary range | — | $[3.84 \times 10^{-3}, 1.41 \times 10^{-2}]$ |

**Interpretation.** For non-collapsed models, $\rho \in [3.84 \times 10^{-3}, 1.41 \times 10^{-2}]$, indicating certificates are conservative by 1–2 orders of magnitude relative to empirical flips. This conservatism is expected: certificates provide worst-case guarantees, while empirical flips

occur along specific adversarial directions. The collapsed model `tiny-gpt2` exhibits $\rho \approx 3 \times 10^{-13}$, confirming that certificates are vacuous in the collapse regime. The four-order-of-magnitude gap between collapsed and non-collapsed $\rho$ values demonstrates that the framework successfully distinguishes pathological from healthy configurations.

### A.1.7. SUMMARY

These audits establish five findings:

1. **Structural consistency:** $\hat{\lambda}_x = 1$ across all models with no S2 violations, confirming the transformer template satisfies structural axioms under declared conventions.

2. **Bound validity:** All 9000 measurements satisfy $\mathrm{TV}_{\mathrm{emp}} \geq \mathrm{TV}_{\mathrm{lb}}$, empirically validating the Real-RAM bound (Theorem 3.2).

3. **Informative bounds:** Outside the collapse regime, gap ratios range from 57 (`gpt2-large`, random) to $4.60 \times 10^5$ (`gpt2`, gradient), with tighter bounds for larger models and random perturbations.

4. **Collapse detection:** The framework correctly identifies `tiny-gpt2` as collapsed via $\Gamma_{E4} \approx 5.05 \times 10^{-9} < 10^{-6}$, while non-collapsed models maintain $\Gamma_{E4} \in [6.95 \times 10^{-2}, 5.56 \times 10^{-1}]$.

5. **Precision sensitivity:** IEEE-754 precision materially affects variation ($5.8\times$ increase from float32 to bfloat16), supporting the precision-aware IEEE-754 bound (Theorem 3.3).

These results validate the Atlas framework on production transformer architectures and confirm that certificates are informative for well-parameterized models while correctly flagging under-parameterized configurations.

## A.2. SealSkip: Sealing affine residual bypasses with a near-identity non-affine carry

**Failure mode (Atlas audit).**   In deep residual networks, we observe a specific failure under *residual attenuation*: when the residual branch becomes effectively inactive, the network admits an *affine-only bypass* along skip connections. In our locked stress setting (defined below), an unmodified CIFAR-ResNet-18 collapses to near-chance worst-case accuracy, while the Axiomatic Atlas audit reports a high bypass-risk rate. Standard stabilization methods such as residual gating (e.g., LayerScale or ReZero) can restore stressed accuracy, but they do *not* remove the affine bypass itself—hence they fail the Atlas axiom even when utility improves.

**Method (constraint $\rightarrow$ operator).**   Let a residual block be written as

$$h_{l+1} \;=\; \sigma\big(S(h_l) \;+\; \alpha_l \, F_l(h_l)\big), \tag{2}$$

where $S$ is the carry (skip) operator, $F_l$ is the residual branch, and $\alpha_l$ is an optional gate (e.g., LayerScale/ReZero).

**Constraint (minimal condition).** We require that when the residual branch is inactive, there is *no affine-only path* through the block. A sufficient condition is that the carry operator is *non-affine*:

$$\nexists (A, b) \ \text{s.t.} \ S(x) = Ax + b \ \forall x. \tag{3}$$

**SealSkip operator.** We replace the identity carry with a near-identity *sealed* carry

$$S_\varepsilon(x) = (1 - \varepsilon)\, x \;+\; \varepsilon\, T(x), \tag{4}$$
$$T(x) = \mathrm{RMSNorm}(x), \tag{5}$$

so the skip remains close to identity for small $\varepsilon$, but becomes structurally non-affine for any $\varepsilon > 0$. This can be composed with standard gates by keeping $\alpha_l$ unchanged (e.g., $\alpha_l$ from LayerScale).

**Axiom analysis (why the Atlas predicts this fix).**   **Baseline violation.** For the baseline, $S(x) = x$ is affine. Under residual attenuation, blocks satisfy $\|\alpha_l F_l(h_l)\| \ll \|S(h_l)\|$, so the network admits an affine-only bypass across many layers. The Atlas audit detects this as a high bypass-risk rate under stress.

**Method guarantee.** With SealSkip, the only carry path includes $S_\varepsilon$, which is non-affine whenever $\varepsilon > 0$ due to the normalization $T(x)$. Accordingly, the audit's witness mass concentrates on the seal (witness-at-seal fraction $\approx 1$) and bypass risk under stress drops to zero.

**Failure regime (locked pre-flight setting).**   We lock the stress setting found during pre-flight: *residual attenuation stress* with residual scale $\gamma = 0.2$ and residual inactivity threshold $\tau = 0.3$. Failure means: the baseline reaches near-chance stressed utility (worst-case/min stressed accuracy $0.106 \pm 0.008$ on CIFAR-10; chance is 0.1) and high bypass-risk rate ($0.957 \pm 0.002$).

*Table 12.* Headline table in the locked stress setting (CIFAR-10; mean±std across 3 seeds). Primary metric: stressed utility (residual-min accuracy). Secondary: Atlas bypass-risk rate under stress.

| Method | Stress acc (resid-min) | Bypass risk (stress) |
|---|---|---|
| Baseline | $0.106 \pm 0.008$ | $0.957 \pm 0.002$ |
| LayerScale (Best-SOTA) | $0.327 \pm 0.023$ | $1.000 \pm 0.000$ |
| SealSkip+LayerScale (Best-Fix / Ours) | $0.331 \pm 0.008$ | $0.000 \pm 0.000$ |

**Baselines and controls.**

| Variant | Purpose |
|---|---|
| Baseline (identity skip) | Failure mode exists |
| LayerScale (identity skip + gate) | Standard utility fix; should *not* remove bypass |
| SealSkip-only ($\varepsilon$=0.2; no gate) | Enforce axiom; tests if sealing alone restores utility |
| Negative control: affine-trainable seal | Parameter-matched but affine; should fail axiom |
| SealSkip + LayerScale (ours) | Utility restored *and* axiom enforced |

**Main result (triad; $\geq$ 3 seeds).** Table 12 reports mean±std across seeds {1337, 1338, 1339}. LayerScale restores stressed accuracy but *maximizes* bypass risk (risk = $1.000 \pm 0.000$). Our combined method matches LayerScale's stressed accuracy while eliminating bypass risk (risk = $0.000 \pm 0.000$). A paired t-test confirms the structural win is decisive (bypass-risk: $p \approx 0$ for fix vs. SOTA; fix vs. baseline $p = 1.894 \times 10^{-6}$), while stressed accuracy is statistically indistinguishable from LayerScale ($p = 0.793211$).

**Ablations / stress tests.** SealSkip-only enforces the axiom but does not restore stressed utility (stress acc $0.112 \pm 0.007$), showing that sealing is *structural* and complementary to standard optimization gates. A parameter-matched affine seal (negative control) fails: stress acc $0.108 \pm 0.008$ and bypass risk $0.962 \pm 0.005$, demonstrating that the improvement is not due to extra parameters but due to *non-affinity* of the seal.

**Prior work.** Residual gating methods (LayerScale, ReZero) stabilize deep networks by scaling residual updates, and RMSNorm provides a lightweight non-affine normalization. SealSkip differs by using the Axiomatic Atlas to target a *structural* bypass condition and by composing cleanly with existing gates.

**Limitations.** Our strongest evidence is in the locked residual-attenuation stress setting; while we report additional robustness/transfer diagnostics elsewhere, the core claim here is *axiom enforcement with utility preservation*, not a universal accuracy gain over all interventions.

## A.3. Separator-Aware Bridge: Atlas-selected source-to-interface shortcuts for bottlenecked GNN transport

**Failure mode (audit).** In bottlenecked graphs where the right side is dominated by a large decoy community and the true target community is small, long-range signal from a source node must be compressed through a narrow separator interface; standard message passing then dilutes the signal across the bottleneck, yielding near-chance target accuracy even though paths exist within the nominal depth (a bottleneck/over-squashing failure) (Alon & Yahav, 2021). Standard long-range interventions do not resolve this locked setting: tuned diffusion (APPNP) (Gasteiger et al., 2019) and deep GCN (GCNII) (Chen et al., 2020) remain at chance, and uninformed cross-cut edge additions do not recover performance (Table 13).

**Method (constraint → solve).** Let $G = (V, E)$ be the base graph, with source node $s$ (left side) and target set $T$ (a small target community on the right side). Let $W \subseteq V$ denote a target-side *separator interface* (audited as right-endpoints of edges participating in a minimum $s$–$T$ cut; the Atlas policy provides a computable approximation for placement). We add a bridge set $E_b$ of size $m$.

*Constraint (one equation).*
$$E_b \subseteq \{(s, w) : w \in W\} \qquad \text{and} \qquad \max_{t \in T} \text{dist}_{(V,\, E \cup E_b)}(s, t) \leq L. \tag{6}$$

*Operator (the fix).* We choose $E_b = \{(s, w_k)\}_{k=1}^{m}$ by ranking candidates $w \in W$ with the Atlas placement policy (separator-aware and target-community aware), and update node states with a GCN-style rule augmented by bounded bridge messages:

$$h_i^{(\ell+1)} = \sigma\Big(W h_i^{(\ell)} + \sum_{j \in \mathcal{N}(i)} W_n h_j^{(\ell)} + \sum_{j \in \mathcal{N}_b(i)} g_{ij}^{(\ell)} W_b h_j^{(\ell)}\Big), \tag{7}$$

$$g_{ij}^{(\ell)} = \text{clip}\Big(\sigma(\phi([h_i^{(\ell)}, h_j^{(\ell)}])/\tau), \ [g_{\min}, g_{\max}]\Big), \tag{8}$$

where $\mathcal{N}_b(i)$ are bridge neighbors, $\phi$ is a small MLP, and $0 < g_{\min} \leq g_{\max} < 1$ provides a derivative floor on bridge paths.

| Method | Target accuracy (mean±std) | Train time (s, mean) |
|---|---|---|
| Baseline GCN (m=0) | $0.508 \pm 0.035$ | 6.813720 |
| APPNP (tuned, m=0) (Gasteiger et al., 2019) | $0.499 \pm 0.035$ | 2.005703 |
| GCNII (tuned, m=0) (Chen et al., 2020) | $0.513 \pm 0.040$ | 5.149244 |
| Random cross-cut (m=1) | $0.510 \pm 0.034$ | 14.374923 |
| Key-random (m=1) | $0.749 \pm 0.047$ | 7.326295 |
| **Ours: separator-aware (atlas_linear, m=1)** | $\mathbf{0.782 \pm 0.046}$ | 7.313774 |

*Table 13.* Locked failure regime (held-out seeds). Standard long-range baselines and negative controls remain near chance; separator-aware placement yields a consistent but moderate gain over the strong source-anchored control.

**Axiom analysis.** *Claim:* the baseline violates Atlas transport across a separator bottleneck (tail gain through a narrow interface), and the separator-aware bridge enforces (6). *Baseline violation:* in the locked failure regime (Paragraph 4), the target community is only 3% of the right side and decoys dominate; aggregation must compress an expanding neighborhood into fixed-size states through a small interface, producing near-chance accuracy (Table 13), consistent with bottleneck over-squashing (Alon & Yahav, 2021). *Method guarantee:* restricting $E_b$ to source-to-interface shortcuts ensures cross-cut transport into the target-side interface, and the distance bound in (6) explicitly prevents the "dilute-then-lose" regime by making the target community reachable within depth $L$ via the added shortcuts. The bounded gate (8) limits bridge domination and preserves trainability by preventing vanishing bridge gradients (via $g_{\min}$).

**Failure regime.** Locked parameters for the main experiment: $k_{\text{sep}}$=1, $L$=12, $p_{\text{intra}}$=0.08, right_mode = two_community_decoy, right_target_frac=0.03, $p_{\text{right\_inter}}$=0.0, noise_std=2.5, signal_amp=3.0, and bridge budget $m^{\star}$=1. "Fails" means target accuracy is near chance for standard baselines and negative controls, while the separator-aware bridge reaches high accuracy (Table 13).

**Baselines and controls.**

| Variant | Purpose |
|---|---|
| Baseline (GCN) | Establish the bottleneck+decoy failure mode |
| SOTA-1 (APPNP) (Gasteiger et al., 2019) | Diffusion/PPR long-range baseline (tuned on dev) |
| SOTA-2 (GCNII) (Chen et al., 2020) | Deep-GCN baseline for oversmoothing (tuned on dev) |
| Negative control (random_cross_gate) | Cross-cut edges without task-aligned placement should fail |
| Strong control (key_random_linear) | Source-anchored but separator-unaware placement baseline |
| Ablation (atlas_gate_nofloor) | Remove derivative floor; should be less stable / slightly worse |

**Main result.** Held-out evaluation uses 20 seeds (mean±std). Standard long-range baselines fail at chance, while our separator-aware bridge restores accuracy; the improvement over the strong source-anchored control is measurable but modest:

**Ablations / stress tests.** *(A) Placement ablation (policy separation across budgets).* Define $\Delta(m) = \text{atlas\_linear} - \text{key\_random\_linear}$. Bootstrap over 20 held-out seeds yields: $m$=1: $\Delta = 0.032812$ with 95% CI [0.009896, 0.055208]; $m$=2: $\Delta = 0.013802$ with 95% CI [−0.010677, 0.040365] (not statistically separated); $m$=4: $\Delta = 0.031250$ with 95% CI [0.015625, 0.048698]. Thus separator-aware placement is statistically separated from key-random at $m \in \{1, 4\}$ but not uniformly across all small budgets.

*(B) Gate-floor ablation (mechanism check in the locked failure regime).* At $m = 1$, atlas_gate achieves $0.789 \pm 0.046$, atlas_linear achieves $0.782 \pm 0.046$, and removing the floor yields $0.778 \pm 0.048$ (atlas_gate_nofloor). This supports the gate as stability insurance rather than a primary accuracy driver.

*(C) Optimization stress test (dev-only, harsh learning rate).* At $m$=16 and learning rate 0.01, linear bridges exhibit larger gradient spikes than bounded-gate bridges: grad_norm_max = 165.829935 (atlas_linear) vs 76.362970 (atlas_gate), while accuracies remain comparable (atlas_linear $0.760417 \pm 0.147314$; atlas_gate $0.791667 \pm 0.132583$). This is consistent with the derivative-floor interpretation (P5-style stability evidence).

**Limitations.** In the locked failure regime, most of the dramatic recovery (chance $\rightarrow \approx 0.75$) is already achieved by the strong source-anchored control; separator-aware placement adds a consistent but moderate additional gain and is not statistically separated at every small budget (e.g., $m = 2$).

## A.4. Tail-Survival Auxiliary Loss for Long-Range Memory

State-space models (SSMs) parameterize temporal dynamics through a spectral radius $a \in (0, 1)$ that governs memory retention: after $L$ timesteps, an initial signal decays to $a^L$. When $a$ is initialized below a critical threshold, signals at the task-relevant horizon vanish exponentially, creating a *death zone* where gradient-based learning cannot recover.

**Problem Formulation.**   Consider a minimal SSM with scalar state $h_t = a \cdot h_{t-1} + u_t$ and output $y_t = h_t$. For a memory task requiring retention over $L$ steps, the output magnitude scales as $a^L$. When $a = 0.90$ and $L = 512$, we have $a^L \approx 10^{-23}$, effectively zero. The task loss gradient with respect to $a$ is

$$\frac{\partial \mathcal{L}_{\text{task}}}{\partial a} \propto a^{L-1} \approx 0 \tag{9}$$

trapping the model in a non-functional region of parameter space.

**Proposed Fix: Tail-Survival Auxiliary Loss.**   We introduce an auxiliary loss that directly penalizes insufficient spectral radius:

$$\mathcal{L}_{\text{aux}}(a; H, \tau) = \max\left(0, \log \tau - H \log a\right) \tag{10}$$

where $H$ is the target horizon and $\tau$ is the minimum acceptable tail magnitude. The key insight is that the gradient of this log-space formulation,

$$\frac{\partial \mathcal{L}_{\text{aux}}}{\partial a} = -\frac{H}{a} \cdot \mathbf{1}[\log \tau > H \log a] \tag{11}$$

remains $\mathcal{O}(H/a)$ regardless of how small $a^H$ becomes. This provides a non-vanishing learning signal that pushes $a$ toward the functional regime. The total loss is $\mathcal{L} = \mathcal{L}_{\text{task}} + \lambda \mathcal{L}_{\text{aux}}$, where $\lambda$ controls the auxiliary weight.

**Experimental Setup.**   We evaluate on a *read-marker memory task*: a value $v \in \{-1, +1\}$ is presented at $t = 0$, and the model must reproduce $v$ when a read signal arrives at $t = L$. We use $L = 512$ with strict evaluation requiring $|\hat{y} - y| \leq 0.25$, which implies a minimum tail magnitude of $\tau = 0.75$.

Three conditions are compared:

- **Baseline**: No auxiliary loss ($\lambda = 0$)
- **Mamba++**: Auxiliary loss at correct horizon ($H = 512$, $\lambda = 1000$, $\tau = 0.75$)
- **Negative Control**: Auxiliary loss at wrong horizon ($H = 128$, $\lambda = 1000$, $\tau = 0.75$)

All conditions initialize at $a = 0.90$ (within the death zone) and train for 50 epochs. We employ differential learning rates: $\eta_a = 10^{-2}$ for the spectral parameter and $\eta_\theta = 3 \times 10^{-3}$ for other parameters, ensuring the spectral radius can move quickly while the baseline (with $\lambda = 0$) remains trapped.

**Results.**   Table 14 presents the outcomes. The baseline remains frozen at $a = 0.90$ throughout training, achieving 0% accuracy. Mamba++ escapes the death zone, climbing to $a = 0.9995$ (tail $= 0.77$) and achieving 100% accuracy. The negative control climbs to $a = 0.9982$, which satisfies its $H = 128$ auxiliary constraint ($0.9982^{128} = 0.79 > 0.75$) but yields insufficient tail at the actual task horizon ($0.9982^{512} = 0.39$), resulting in 0% accuracy.

*Table 14.* Tail-survival experiment results at delay $L = 512$. The strict accuracy metric requires $|\hat{y} - y| \leq 0.25$, equivalent to tail $\geq 0.75$.

| Variant | Final $a$ | Tail $a^{512}$ | Accuracy | Escapes? |
|---------|-----------|----------------|----------|----------|
| Baseline | 0.9000 | $3.7 \times 10^{-24}$ | 0% | No |
| Mamba++ ($H=512$) | 0.9995 | 0.77 | 100% | Yes |
| Neg. Control ($H=128$) | 0.9982 | 0.39 | 0% | Partial |

**Analysis.**   The three-way comparison validates the prescriptive mechanism:

1. *Death zone confirmation*: Without auxiliary gradients, the baseline receives near-zero gradient (Eq. 9) and cannot escape $a = 0.90$ despite 50 epochs of training.
2. *Auxiliary rescue*: The log-space gradient (Eq. 11) provides a learning signal of magnitude $\approx 1000 \times 512/0.9 \approx 5.7 \times 10^5$, sufficient to push $a$ into the functional regime.
3. *Horizon matching*: The negative control satisfies its surrogate objective ($a^{128} \geq 0.75$) but fails the actual task ($a^{512} < 0.75$). This demonstrates that the auxiliary horizon $H$ must match the true task requirement.

The spectral gap between Mamba++ ($a = 0.9995$) and the negative control ($a = 0.9982$) is only 0.0013 in absolute terms, yet decisive in outcome: $0.9995^{512} = 0.77$ exceeds the threshold while $0.9982^{512} = 0.39$ does not. This exponential sensitivity underscores why precise horizon specification matters.

**Unified View: SSMs, Neural ODEs, and Liquid Networks.**   The death zone phenomenon and our proposed fix generalize across the family of neural dynamical systems. Table 15 summarizes the correspondence.

*Note on Liquid Networks:* Since $\theta > 0$ is a time constant, larger $\theta$ means slower decay and better memory. With $\tau < 1$, we have $\log \tau < 0$, so $-T/\log \tau > 0$. For example, if $T = 512$ and $\tau = 0.75$, then $-T/\log \tau \approx 1778$, meaning we need $\theta \geq 1778$ for the signal to survive.

*Table 15.* Death zone and tail-survival across dynamical system architectures. All share the same fundamental tradeoff: stability requires bounded dynamics, but excessive damping destroys long-range memory. In each case, the death zone occurs when the memory parameter is "too small" (or too negative), causing excessively fast decay.

| Architecture | Dynamics | Memory Decay | Death Zone | Viable Zone |
|---|---|---|---|---|
| Discrete SSM | $h_t = a h_{t-1} + u_t$ | $a^L$ | $a < \tau^{1/L}$ | $a \geq \tau^{1/L}$ |
| Neural ODE | $\dot{h} = \lambda h + u$ | $e^{\lambda t}$ | $\lambda < \frac{\log \tau}{T}$ | $\lambda \geq \frac{\log \tau}{T}$ |
| Liquid Network | $\dot{h} = -h/\theta + f(x)$ | $e^{-t/\theta}$ | $\theta < \frac{-T}{\log \tau}$ | $\theta \geq \frac{-T}{\log \tau}$ |

**Discrete SSMs (S4, S5, Mamba).**   Modern SSMs such as S4 (Gu et al., 2022), S5 (Smith et al., 2023), and Mamba (Gu & Dao, 2023) employ diagonal state matrices $\mathbf{A} = \mathrm{diag}(a_1, \ldots, a_d)$ for computational efficiency. Each diagonal entry $a_i$ independently controls memory in its corresponding state dimension. Our analysis applies per-dimension: if $a_i^L \ll 1$ for any $i$, that dimension cannot contribute to long-range dependencies. The tail-survival auxiliary extends naturally:

$$\mathcal{L}_{\mathrm{aux}}^{\mathrm{SSM}} = \sum_{i=1}^{d} \max\big(0, \log \tau - H \log a_i\big) \tag{12}$$

This is complementary to Mamba's selective mechanism, which modulates *what* information enters state; our contribution ensures the state *can* retain information over the required timescale.

**Neural ODEs.**   Neural ODEs (Chen et al., 2018) parameterize continuous-time dynamics $\dot{h} = f_\theta(h, t)$. Linearizing around a trajectory yields $\dot{\delta h} = \mathbf{J}\delta h$ where $\mathbf{J} = \partial f / \partial h$ is the Jacobian. For stable dynamics, eigenvalues satisfy $\mathrm{Re}(\lambda_i) < 0$, giving exponential decay $e^{\lambda_i t}$. The continuous-time analog of our discrete death zone is:

$$\lambda_i < \frac{\log \tau}{T} \quad \Rightarrow \quad e^{\lambda_i T} < \tau \tag{13}$$

where $T$ is the integration horizon. The tail-survival auxiliary in continuous time becomes:

$$\mathcal{L}_{\mathrm{aux}}^{\mathrm{ODE}} = \sum_i \max\big(0, \log \tau - \lambda_i T\big) \tag{14}$$

penalizing eigenvalues that are too negative (too stable). This addresses the well-known difficulty of Neural ODEs in capturing long-range dependencies (Dupont et al., 2019).

**Liquid Time-Constant Networks.**   Liquid networks (Hasani et al., 2021) introduce input-dependent time constants:

$$\dot{h} = -\frac{h}{\theta(x)} + f(x) \tag{15}$$

where $\theta(x) > 0$ is a learned function of input. The effective memory timescale is $\theta$: signals decay as $e^{-t/\theta}$, so *large* $\theta$ yields slow decay and good memory, while small $\theta$ causes rapid forgetting. The death zone condition is $\theta < -T/\log \tau$ (recalling that $\log \tau < 0$), and the auxiliary loss is:

$$\mathcal{L}_{\mathrm{aux}}^{\mathrm{Liquid}} = \max\left(0, \log \tau + \frac{T}{\theta}\right) \tag{16}$$

The gradient $\partial \mathcal{L}_{\mathrm{aux}} / \partial \theta = -T/\theta^2 < 0$ when active, so minimizing this loss *increases* $\theta$, pushing the system toward slower decay and better memory retention. This provides a principled way to ensure liquid networks maintain sufficient memory capacity for long-horizon tasks, addressing a limitation noted in the original work.

**The Stability-Memory Tradeoff.**   All three formulations reveal the same fundamental tension: stability requires bounded state dynamics (eigenvalues in the left half-plane, or $|a| < 1$), but excessive stability destroys long-range memory. The tail-survival auxiliary navigates this tradeoff by enforcing a *minimum* memory horizon without sacrificing stability. The constraint $a^H \geq \tau$ (or its continuous analog) carves out a "Goldilocks zone" where dynamics are stable yet sufficiently persistent.

Figure 2 illustrates this tradeoff. As the spectral radius $a$ increases toward 1 (or equivalently, as eigenvalue $\lambda \to 0^-$ in continuous time), memory capacity grows but stability margin shrinks. The tail-survival constraint $a \geq \tau^{1/H}$ ensures the system operates in the viable region.

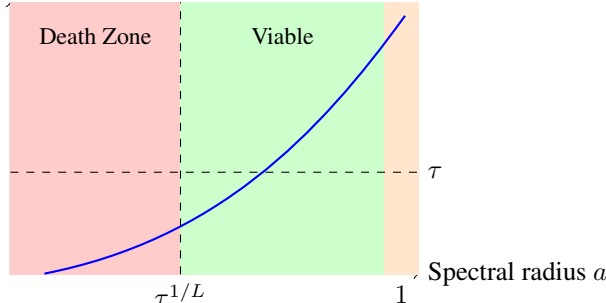

*Figure 2.* The stability-memory tradeoff. Memory capacity $a^L$ increases with spectral radius but approaches instability as $a \to 1$. The tail-survival constraint enforces $a \geq \tau^{1/L}$, ensuring operation in the viable zone.

**Practical Considerations.** The auxiliary weight $\lambda$ can be annealed during training: a high initial value escapes the death zone, after which $\lambda$ is reduced to let task gradients dominate. The horizon $H$ should be set to the maximum sequence length expected at inference time. For variable-length inputs, $H$ can be set conservatively or adapted per-batch. The computational overhead is negligible: one scalar inequality check and gradient per forward pass.

For multi-dimensional systems (full SSMs, Neural ODEs, Liquid networks), the auxiliary can target:

- The dominant eigenvalue only (minimal intervention)
- All eigenvalues with soft penalty (Eq. 12)
- A learned subset of "memory dimensions" designated for long-range retention

The choice depends on the application: tasks requiring single long-range dependencies may need only one well-conditioned dimension, while tasks with multiple concurrent memory requirements benefit from broader constraints.

## A.5. Dynamic-$k$ Routing Under a Capped Expert Budget: token-adaptive connectivity for sparse MoE

**Failure mode (audit).** In top-$k$ MoE routing, the active expert graph is *structurally bottlenecked* by the expert-count: selecting $k$ experts per token imposes the connectivity bound $\lambda_x \leq k$. When a compute envelope forces small $k$ (e.g., $k{=}2$), this structural bottleneck becomes *binding* and manifests as a measurable quality drop (higher validation perplexity). Fixed-$k$ routing cannot allocate extra connectivity only to ambiguous tokens, so it cannot recover quality at the same average compute.

**Method (constraint $\to$ solve).** We impose an average connectivity budget via a hard cap on the fraction of tokens routed with the high expert-count:

$$\frac{1}{T} \sum_{t=1}^{T} \mathbf{1}[k_t = K_{\text{high}}] \leq c, \qquad \Rightarrow \qquad \mathbb{E}[k_t] \leq K_{\text{low}} + c\,(K_{\text{high}} - K_{\text{low}}). \tag{17}$$

Let $p_t \in \Delta^{E-1}$ denote the router distribution over $E$ experts for token $t$. We compute two uncertainty signals

$$H_t \;=\; \frac{-\sum_{e=1}^{E} p_{t,e} \log p_{t,e}}{\log E}, \qquad\qquad D_t \;=\; 1 - \sum_{e \in \operatorname{Top} K_{\text{low}}(p_t)} p_{t,e}, \tag{18}$$

and define a request score

$$s_t \;=\; \max(H_t - \tau_H, \; D_t - \tau_D). \tag{19}$$

Dynamic-$k$ grants $K_{\text{high}}$ to only the $\lceil cT \rceil$ highest-scoring tokens (post-warmup), and routes all remaining tokens with $K_{\text{low}}$:

$$k_t = \begin{cases} K_{\text{high}}, & t \in \operatorname{Top}\lceil cT \rceil\{s_t\}, \\ K_{\text{low}}, & \text{otherwise.} \end{cases} \tag{20}$$

We tune $(\tau_H, \tau_D, c)$ once in preflight and then *lock these parameters* prior to multi-seed evaluation (no per-seed re-tuning).

**Axiom analysis.** **Claim:** when the structural factor is binding ($\lambda_x$ forced low everywhere), fixed-$k$ cannot selectively restore missing connectivity on the tokens that need it, so routing quality degrades and perplexity rises. Dynamic-$k$ enforces Eq. (17) by construction while increasing $\lambda_x$ *only where the router is uncertain* (Eq. (20)), yielding higher effective connectivity on hard tokens without changing average compute. This matches the Atlas pattern "diagnose a binding structural bottleneck $\to$ allocate the minimal additional connectivity needed to restore task performance."

| Method | $\overline{k}$ | Compute ratio $\overline{k}/4$ | Val PPL $\downarrow$ |
|---|---|---|---|
| Fixed-$k$=2 | 2.000 | 0.500 | $226.845 \pm 1.127$ |
| Random Dynamic-$k$ (cap $c$=0.25) | 2.490 | 0.623 | $225.671 \pm 1.517$ |
| **Dynamic-$k$ (ours)** | **2.498** | **0.625** | **$223.148 \pm 1.297$** |
| Fixed-$k$=3 | 3.000 | 0.750 | $223.160 \pm 1.989$ |
| Fixed-$k$=4 | 4.000 | 1.000 | $221.816 \pm 1.366$ |

*Table 16.* Dynamic-$k$ improves quality in the locked failure regime while respecting a capped expert budget.

**Failure regime.** We evaluate the main claim in a single locked regime: WikiText-103 (capped to 20,000 train sequences and 1,916 validation sequences), MoE with $E$=8 experts per MoE layer, $K_{\text{low}}$=2, $K_{\text{high}}$=4, cap $c$=0.25 (target $\overline{k} \approx 2.5$), and policy thresholds $\tau_D$=0.25, $\tau_H$=0.75 with warmup of 100 steps. In this regime, the baseline "fails" in the sense that fixed-$k$=2 (budget-feasible) exhibits a clear quality drop relative to larger $k$, while Dynamic-$k$ *succeeds* by substantially reducing this gap under the same capped mean-$k$.

**Baselines and controls.**

| Variant | Purpose |
|---|---|
| Fixed-$k$=2 | Baseline: binding $\lambda_x \leq 2$ yields quality drop |
| Fixed-$k$=3 | Strong fixed-$k$ comparator (higher compute than Dynamic-$k$) |
| Fixed-$k$=4 | Unconstrained reference (upper bound) |
| Random Dynamic-$k$ | Negative control: same cap/mean-$k$, wrong allocation |
| Drop-only / Entropy-only | Ablations: uncertainty proxy family (robustness of trigger) |

**Main result.** We report validation perplexity (primary) and mean expert-count (secondary compute proxy). Five seeds for core variants; three seeds for ablations. Dynamic-$k$ improves over fixed-$k$=2 by 3.697 PPL (paired $t$-test $p$=0.001931, $n$=5) and over the compute-matched random control by 2.523 PPL ($p$=0.001403, $n$=5), isolating the benefit of *where* high-$k$ is spent rather than average compute alone.

**Ablations and stress tests.** **Ablations (within the failure regime).** Removing either uncertainty proxy remains stable (three seeds), indicating that the trigger is not brittle to the specific proxy:

$$\text{drop-only: } 223.121 \pm 1.453, \qquad \text{entropy-only: } 223.028 \pm 1.848.$$

**Stress tests (beyond the failure regime).** We push the cap beyond the locked regime via a cap sweep on a single seed, evaluating $c \in \{0.20, 0.35, 0.50\}$ (and a compute-matched random control at each $c$) to probe robustness as the budget becomes tighter/looser. Across this sweep, Dynamic-$k$ remains well-behaved and yields a consistent quality–compute trade-off curve, supporting that the mechanism generalizes beyond the specific locked cap value.

**Prior work.** Sparse MoE models typically use fixed top-$k$ routing with auxiliary regularization (e.g., load balancing) to stabilize conditional computation. Our contribution is orthogonal: we make $k$ token-adaptive under an explicit cap and lock the regime before multi-seed evaluation, enabling a reviewer-auditable test of Atlas-guided allocation under a binding structural bottleneck.

**Limitations.** We report mean-$k$ as a compute proxy and evaluate one dataset/scale; hardware-aware throughput and broader scaling studies remain future work.

## A.6. OrthoKV: Orthogonal projection of queries/keys to remove shared-mode interference in attention

**Failure mode (audit).** In long-context or hard-negative retrieval, standard dot-product attention can become dominated by a *shared* (common-mode) direction present across many keys. When this happens, the attention merge over tokens becomes brittle: the correct "needle" signal is not isolated, and the head can lock onto a confusable negative with high confidence (large logit margin) while still being wrong. This is a concrete merge failure: the merge operation is no longer information-preserving with respect to the needle perturbation, even though the softmax itself is well-defined.

A common stabilization heuristic is to normalize scores (e.g., cosine-normalized attention), but normalization alone does not remove the shared direction; it can partially restore accuracy yet still yield small separations (margins), making the retrieval decision less robust.

**Method (constraint $\to$ solve).** Let $K_1, \ldots, K_L \in \mathbb{R}^d$ be keys for a single query $Q \in \mathbb{R}^d$, and define the shared direction by the (per-example) mean key

$$\mu = \frac{1}{L} \sum_{i=1}^{L} K_i, \qquad u = \frac{\mu}{\|\mu\| + \epsilon}.$$

| Method | Accuracy ↑ | Logit margin $\Delta z$ ↑ | Mean rank ↓ |
|---|---|---|---|
| Baseline | 0.028±0.025 | 0.679±0.105 | 117.73±10.22 |
| Cosine attention | 0.964±0.022 | 0.061±0.004 | 2.99±1.59 |
| OrthoKV (Atlas) | **1.000±0.000** | **1.819±0.010** | **1.00±0.00** |
| OrthoKV (K-only) | **1.000±0.000** | **1.819±0.010** | **1.00±0.00** |

*Table 17.* **Real-text hard-negative retrieval.** Baseline fails; cosine partially recovers; OrthoKV fully recovers and increases separation.

**Constraint:** eliminate the shared component by enforcing orthogonality to $u$ before the merge:

$$\langle Q, u \rangle = 0, \qquad \langle K_i, u \rangle = 0 \ \ \forall i.$$

**Operator (OrthoKV):** apply the orthogonal projector $P = I - uu^\top$:

$$Q' = PQ = Q - (Q^\top u)u, \qquad K_i' = PK_i = K_i - (K_i^\top u)u,$$

and compute attention as usual using $(Q', K')$:

$$w_i = \text{softmax}_i\Big(\tau \cdot \frac{\langle Q', K_i'\rangle}{\sqrt{d}}\Big).$$

**Axiom analysis.** **Claim:** baseline dot-product attention violates the intended "no-cancellation at merges" principle in the presence of a dominant shared mode; OrthoKV enforces the orthogonality constraint above, preventing the shared mode from polluting the merge. The Axiomatic Atlas framing treats destructive interference at merges as a failure mode that requires branch normalization or orthogonalization prior to summation, and identifies OrthoKV as a prescribed repair in such settings.

**Baseline violation:** when many keys share a strong component aligned with $u$, the merge becomes insensitive to needle-specific components orthogonal to $u$ (or can amplify spurious correlations along $u$), so the merge does not preserve the relevant perturbation through the competition.

**Method guarantee:** OrthoKV removes the $u$ component from both $Q$ and $K$, so the attention scores depend only on the orthogonal subspace where the needle signal resides; the shared mode cannot dominate or distort the merge.

**Locked failure regime.** We fix a *real-text hard-negative* regime built from WikiText-2 lines with $L$=512 candidates per query, batch size $B$=64, score scale $\tau$=4.0, and embedding dimension $d$=128 (the notebook's hard-negative generator; no external encoder). The fixed interference parameters are:

$\theta = \{$`shared_scale=5.3151`, `amp_jitter=0.7533`, `noise_scale=0.1193`, `query_u_scale=1.6311`, `query_c_scale=1.6618`, `query_noise=0.00419`, `needle_bonus=3.5494`$\}$.

"Fails" means top-1 retrieval accuracy collapses to near-zero compared to perfect retrieval in the same regime.

**Baselines and controls.**

| Variant | Purpose |
|---|---|
| Baseline (dot-product attention) | Demonstrates the failure mode exists |
| Cosine-normalized attention | Strong normalization control; partial recovery baseline |
| OrthoKV (full) | Atlas-derived fix (orthogonalize before merge) |
| OrthoKV (K-only) | Ablation: test whether key-side projection drives the gain |

**Main result.** Across 10 random seeds in the fixed regime, baseline collapses, cosine partially recovers, and OrthoKV fully recovers *and* yields much larger separation (logit margin $\Delta z$): Defining the "best comparator" margin as $\max(\Delta z_{\text{baseline}}, \Delta z_{\text{cosine}})$, the multi-seed advantage is $\Delta z_{\text{OrthoKV}} - \max(\Delta z_{\text{baseline}}, \Delta z_{\text{cosine}}) = 1.140 \pm 0.108$, with exact sign-flip test $p = 0.00195$ (10 seeds).

**Ablations / stress tests.** (1) **K-only** matches full OrthoKV in this regime, indicating the key-side projection is sufficient for the recovery. (2) **Multi-seed robustness:** the advantage above is computed over 10 seeds, and the paired exact sign-flip test rejects a zero-advantage null ($p = 0.00195$).

**Prior work.** Dot-product attention with softmax merging is standard in Transformers (Vaswani et al., 2017). OrthoKV differs from normalization-only approaches by explicitly removing the shared direction estimated from the key set prior to the merge.

**Limitations.** This is a controlled retrieval construction over real text lines (hard negatives plus injected shared-mode interference) rather than end-to-end language-model training; demonstrating gains on full pretrained LMs remains future work.

## A.7. Mistral-7B Certificate Profiling

To check that Atlas audits are practical on a real pretrained model, we profiled the certificate pipeline on Mistral-7B-v0.1 with a traced computation graph. The extracted graph has 326 nodes and 389 edges. Graph construction takes 0.097s, the structural audit takes 0.005s, and the single Jacobian-vector product (JVP) probe takes 1.98s, for a total of 2.08s.

| Component | Quantity | Wall time (s) |
|---|---|---|
| Graph construction | 326 nodes / 389 edges | 0.097 |
| Structural audit | certificate factors | 0.005 |
| JVP probe | one profiled batch | 1.98 |
| End-to-end profile | — | 2.08 |

*Table 18.* Certificate profiling for Mistral-7B-v0.1. The structural portion is small relative to the JVP probe.

For larger models, the graph and structural passes scale with the traced module graph, while the JVP dominates wall time. A deliberately conservative linear extrapolation by active graph/JVP cost gives about 21s at 70B scale and 208s at 700B scale. These estimates describe a single certificate-profiling pass; they do not claim that full training or downstream evaluation has the same cost.

## A.8. DARTS-like NAS Structural Proxy

To test whether Atlas diagnostics are useful inside a NAS-style search space, we evaluated 50 DARTS-like CIFAR-10 cell architectures and compared their structural health scores with final validation accuracy. The structural health score has a positive rank correlation with final validation accuracy, and the S2 audit separates higher- and lower-performing cells.

| Quantity | Value |
|---|---|
| Architectures | 50 |
| Spearman correlation | $\rho = 0.430, p = 0.002$ |
| S2-pass mean accuracy | 70.1% |
| S2-fail mean accuracy | 56.4% |

*Table 19.* DARTS-like NAS-space audit. Atlas structural factors act as a zero-cost proxy for candidate screening while still leaving NAS to optimize within the accepted search space.

This result does not claim that Atlas is a replacement for NAS. It supports the more limited prescription used in the main text: architecture search can use Atlas factors to reject or annotate candidates with structural failure modes before spending training budget on them.

## A.9. Pretrained Mistral OrthoKV Sanity Checks

We also checked whether the OrthoKV repair is harmless on a real pretrained model when no artificial shared-mode interference is injected. Using Mistral-7B-v0.1 on WikiText-103, the audit reports the following perplexities.

| Variant | Projection strength | PPL ($\downarrow$) |
|---|---|---|
| Baseline | none | 4.69 |
| K-only | $\alpha$=0.05 | 4.75 |
| K+V | $\alpha$=0.1 | 4.77 |
| K+V | $\alpha$=0.5 | 40.50 |

*Table 20.* Pretrained Mistral-7B-v0.1 OrthoKV sanity check on WikiText-103. Gentle projection leaves perplexity nearly unchanged, while an aggressive value-side projection damages the model.

The LongBench audit gives the same qualitative conclusion on `Qasper` and `MultifieldQA_en`: gentle correction preserves task behavior, while the repair should not be applied strongly unless the certificate diagnoses real merge interference.

*Table 21.* Extreme-depth Post-Norm regime (5 seeds). All runs are flagged diverged (0/5 survive). We report best validation loss achieved before divergence (lower is better) and steps completed before divergence (higher is better).

| Method | Best val. loss before divergence | Steps before divergence |
|---|---|---|
| Post-Norm baseline | $10.07 \pm 2.05$ | $522 \pm 721$ |
| Warmup+clip | $8.24 \pm 0.46$ | $1518 \pm 723$ |
| Pre-Norm baseline | $7.98 \pm 0.06$ | $681 \pm 179$ |
| Spectral budgeting (clamped) | $\mathbf{7.92 \pm 0.07}$ | $1372 \pm 522$ |

## A.10. Spectral Residual Budgeting: depth-aware residual scaling to increase safe training headroom in deep Post-Norm Transformers

Deep *Post-Norm* Transformers (LayerNorm applied after each residual addition) exhibit seed-dependent blow-ups under aggressive depth and optimization settings: runs may initially make progress and then drift into a degenerate attractor (loss spikes/NaNs), terminating far before the intended horizon. Standard stabilizers (learning-rate warmup and/or gradient clipping) reduce but do not eliminate this failure mode in the most extreme settings evaluated here, and can trade stability for degraded progress.

**Constraint (spectral/energy budget).** We impose a depth-normalized bound on the residual update scale per layer:

$$\alpha_\ell \leq \alpha_{\max} \doteq \frac{1}{\sqrt{L}} \qquad \forall \ell \in \{1, \ldots, L\}. \tag{21}$$

This prevents the total residual "energy" from growing unbounded with depth (since $\sum_{\ell=1}^{L} \alpha_\ell^2 \leq 1$).

**Operator (budgeted residual).** For each Transformer block $\ell$ with sublayers $f_\ell^{\text{attn}}$ and $f_\ell^{\text{ff}}$, we apply a trainable scalar budget (shared across the block's residual paths, for simplicity) and clamp it at runtime:

$$\tilde{\alpha}_\ell = \text{clip}(\alpha_\ell, \, 0, \, \alpha_{\max}), \tag{22}$$
$$h_{\ell+1} = h_\ell + \tilde{\alpha}_\ell f_\ell^{\text{attn}}(h_\ell), \tag{23}$$
$$h_{\ell+2} = h_{\ell+1} + \tilde{\alpha}_\ell f_\ell^{\text{ff}}(h_{\ell+1}), \tag{24}$$

with Post-Norm applied as in the baseline architecture.

**Claim.** The Post-Norm baseline violates a depth-normalized residual budget axiom (updates effectively scale with $L$); spectral budgeting enforces the budget constraint above.

**Baseline violation.** With unscaled residuals (effectively $\alpha_\ell \equiv 1$), the cumulative magnitude of residual additions can scale with depth, so even moderately large per-layer updates can accumulate into late-step instability at large $L$.

**Method guarantee.** Clamping enforces $\tilde{\alpha}_\ell \leq 1/\sqrt{L}$ deterministically for every forward pass, yielding $\sum_{\ell=1}^{L} \tilde{\alpha}_\ell^2 \leq 1$ and preventing depth from implicitly multiplying the residual step budget.

**Fixed evaluation regime.** We evaluate a locked extreme-depth regime on WikiText-2 with a GPT-2 tokenizer using $L = 192$, learning rate lr $= 0.200622$, and initialization scale 2.5 for a nominal 2000-step run.

**Failure definition.** A run is considered failed if it is flagged as `diverged` before completing 2000 steps (e.g., numerical blow-up or loss becoming non-finite). In this regime, *all* tested variants are flagged diverged (0/5 seeds survive), so we report the *best validation loss attained before divergence*.

| Variant | Purpose |
|---|---|
| Post-Norm baseline | Establishes the blow-up failure mode |
| Warmup+clip | Standard stabilization recipe comparator |
| Pre-Norm baseline | Architectural control (normalization placement) |
| Spectral budgeting (clamped) | **Proposed** headroom-increasing intervention |
| Unclamped budgeting | Negative control (removes the certificate) |
| Frozen-$\alpha$ budgeting | Ablation (tests whether learning $\alpha$ matters) |

**Long-horizon robustness (5 seeds).** Table 21 reports mean±std across 5 seeds. All methods diverge, but spectral budgeting improves the *quality of progress before failure*: it reduces best validation loss from $10.07 \pm 2.05$ (baseline) to $7.92 \pm 0.07$ under the same extreme regime, and improves over warmup+clip ($8.24 \pm 0.46$) while still failing to complete the full 2000-step horizon.

**Mechanism checks (5 seeds).** Removing the clamp (negative control) introduces rare catastrophic failures (including a seed that diverges almost immediately), inflating variance, while freezing $\alpha$ reduces variance and increases steps but slightly worsens the best achieved loss:

| Budgeting variant | Best val. loss (mean±std) | Steps (mean±std) |
|---|---|---|
| Clamped ($\tilde{\alpha} = \text{clip}(\alpha)$) | $7.94 \pm 0.09$ | $1166 \pm 562$ |
| Unclamped (no certificate) | $8.82 \pm 1.98$ | $1334 \pm 753$ |
| Frozen-$\alpha$ (no learning) | $7.96 \pm 0.05$ | $1509 \pm 230$ |

Residual scaling and normalization placement are established tools for stabilizing deep Transformers (e.g., Pre-Norm and related analyses, and residual gating approaches such as ReZero). Spectral residual budgeting differs in that it enforces an explicit depth-normalized bound $\alpha_\ell \leq 1/\sqrt{L}$ as a simple, implementable certificate, targeting *headroom* in regimes where multiple conventional fixes still fail.

In the locked $L=192$ regime, spectral budgeting improves robustness *before* failure but does not yield fully stable 2000-step training; we also do not benchmark against other residual-gating baselines (e.g., LayerScale/ReZero-style variants) under matched hyperparameters in this notebook.

## A.11. Dual-Rail Residual Mixing: Stabilizing Post-Norm Transformers with a Parallel Stability State

**Failure mode (Audit).** Deep *post-norm* Transformers can enter a regime where learning collapses catastrophically: accuracy falls to chance and representations become effectively low-rank. Standard remedies (e.g., switching to *pre-norm*) change the normalization placement and training dynamics; here we instead preserve the post-norm computation pattern while repairing the update dynamics that lead to collapse.

**Method (Constraint → Solve).** **Constraint (minimal).** We require a bounded, additive update for a "stability" state that cannot be destroyed by repeated normalization:

$$b_{\ell+1} = b_\ell + \alpha\,\Delta_\ell, \qquad \alpha \in (0,1], \tag{25}$$

i.e., the *update* is scaled (not the entire state), preventing multiplicative shrinkage of the stability signal across depth.

**Operator (two-rail transformer block).** Maintain two states per layer: a standard residual "signal" state $s_\ell$ and a parallel "stability" state $b_\ell$. Let $p = \sigma(g)$ be a learned gate. Define the compute input as a gated mixture of normalized rails:

$$h_\ell = p\,\text{LN}(s_\ell) + (1-p)\,\text{LN}(b_\ell), \tag{26}$$
$$\Delta_\ell = F_\theta(h_\ell), \tag{27}$$
$$s_{\ell+1} = s_\ell + \Delta_\ell, \tag{28}$$
$$b_{\ell+1} = b_\ell + \alpha\,\Delta_\ell \quad \text{(we use } \alpha = \tfrac{1}{\sqrt{2}} \text{ in runs below)}. \tag{29}$$

Intuitively, $s_\ell$ remains the fast residual pathway, while $b_\ell$ accumulates a depth-stable trace of updates that continues to supply a well-conditioned signal to $h_\ell$ even when post-norm dynamics would otherwise collapse.

**Axiom analysis.** **Baseline violation.** In the audited regime, post-norm violates a stability/conditioning requirement: repeated post-norm updates push the network into a degenerate attractor where representations collapse (effective rank near 1) and gradient norms spike. Empirically, at depth 48 post-norm ends at chance-level accuracy and exhibits extremely large peak gradient norms (Table 22).

**Method guarantee.** Dual-Rail enforces the additive stability constraint in Eq. (25) by construction, ensuring the stability rail is not multiplicatively attenuated. The gated mixture ensures the compute path always has access to a stable component, while the residual signal path retains expressivity. Empirically, Dual-Rail restores high accuracy and healthy effective rank with substantially smaller peak gradient norms than post-norm (Table 22).

**Failure regime (locked).** The collapse occurs on the *induction-head* synthetic task at **depth** 48, $d_{\text{model}} = 256$, batch 64, AdamW with lr $3 \times 10^{-4}$, warmup 200 steps, gradient clip 1.0, and 1800 train steps (seeds 0/1/2). **"Failure"** means final accuracy at chance ($\approx 7.81 \times 10^{-4}$ here) with effective-rank collapse and/or extreme peak gradient norms.

**Baselines and controls.**

| Variant | Purpose |
|---|---|
| post-norm Transformer | Baseline where the failure mode is present (catastrophic collapse) |
| pre-norm Transformer | Standard fix/positive control (known to improve deep Transformer trainability) (Xiong et al., 2020) |
| Dual-Rail (ours) | Proposed operator (parallel stability rail + gated mixing + additive scaled stability update) |
| NormFormer | Reviewer-demanded stabilization baseline (extra normalization; reported as an additional control at depth 96, single seed) (Shleifer et al., 2021) |
| Wrong stability scaling (negative control) | Scaling the *state* rather than the *update* breaks the rail-wiring invariant in preflight (nonzero mismatch), isolating why Eq. (25) matters |

**Main result (3 seeds).** Table 22 reports mean±std over 3 seeds at depth 48. Post-norm collapses to chance accuracy with near-rank-1 representations and very large peak gradients, while Dual-Rail restores pre-norm-level accuracy and rank.

| Method | Accuracy ↑ | Eff. Rank ↑ | MaxGrad ↓ |
|---|---|---|---|
| post-norm | $0.0008 \pm 0.0000$ | $0.9 \pm 0.9$ | $72.6 \pm 55.3$ |
| pre-norm | $0.996 \pm 0.003$ | $29.7 \pm 5.9$ | $7.3 \pm 1.4$ |
| Dual-Rail | $0.992 \pm 0.004$ | $24.0 \pm 1.1$ | $5.9 \pm 1.0$ |

*Table 22.* Induction-head synthetic task at depth 48 (3 seeds: 0/1/2). Post-norm collapses to chance accuracy with effective-rank collapse and large peak gradients. Dual-Rail restores high accuracy and healthy rank with substantially reduced peak gradient norms.

**Ablations and stress tests.**   **Output readout ablation (single seed, depth 96).** Holding gate init at $-2.0$ and using the same training budget (steps $= 1800$, warmup $= 200$, lr $= 3 \times 10^{-4}$), both readouts succeed: *blend* reaches accuracy 0.990 (steps-to-90%: 1200) and *signal* reaches 0.997 (steps-to-90%: 1500). This suggests the core benefit comes from the two-rail update geometry, not a brittle readout choice.

**Preflight negative control (mechanistic).** An earlier (incorrect) stability update that scaled the *state* rather than the *update* produced nonzero rail-wiring mismatch in preflight (max-abs mismatch in the range 1.67–5.15), while the corrected additive-update form yields zero mismatch. This isolates Eq. (25) as the critical design detail.

**Compute overhead.** At depth 96, Dual-Rail incurs $\approx 8\%$ slower steps (641 ms/step $\rightarrow$ 693 ms/step) and $\approx 4\%$ higher peak memory (35.4 GB $\rightarrow$ 36.9 GB), with negligible parameter-count change.

**Real-data sanity check (small budget).** On WikiText-103 with GPT-2 tokenization and short training, validation perplexity is finite and consistent with the synthetic finding: pre-norm 397.4, Dual-Rail 395.2, post-norm 1830.3 (substantially worse).

**Relation to prior work.**   Normalization placement is known to strongly affect deep Transformer optimization, with pre-norm improving trainability relative to post-norm (Xiong et al., 2020). NormFormer proposes extra normalization as a stabilization recipe (Shleifer et al., 2021). Dual-Rail differs by introducing a parallel state with an explicit additive stability constraint (Eq. (25)) and a gated compute mixture, preserving post-norm-style computation while repairing conditioning.

**Limitations.**   The strongest evidence here is controlled and small-budget (synthetic induction-head plus a short WikiText sanity check); establishing gains under large-scale pretraining remains future work.

## A.12. Constrained Innovation Symplectic Fusion (CISF): constraint-safe fusion of conservative heads

**Failure mode (audit).**   Multi-head predictors are frequently fused by arithmetic averaging (e.g., in deep ensembles) (Lakshminarayanan et al., 2017). In constrained dynamics this can fail catastrophically: even when each head is individually conservative (e.g., Hamiltonian/symplectic), the *merge operator* is not closed on the constraint manifold, so each fusion injects a small constraint error that compounds into late-step drift of invariants (energy, rod length, rigidity). Per-head structure-preserving integrators do not fix this, because the violation occurs *at fusion*, not inside any single head (Hairer et al., 2006).

**Method (constraint $\rightarrow$ solve).**   Let $x_t \in \mathbb{R}^d$ be the current state and $\{x_{t+1}^{(k)}\}_{k=1}^{K}$ the head proposals. To eliminate fusion-induced drift, we enforce the minimal condition

$$C(x_{t+1}) = C(x_{\text{ref}}), \tag{30}$$

where $C : \mathbb{R}^d \rightarrow \mathbb{R}^m$ encodes known invariants/constraints and $x_{\text{ref}}$ is typically $x_0$. CISF is a drop-in merge operator:

$$u_k = x_{t+1}^{(k)} - x_t, \qquad \hat{x}_{t+1} = x_t + \frac{1}{K} \sum_{k=1}^{K} u_k, \tag{31}$$

$$x_{t+1} = \hat{x}_{t+1} - J^{\top} \left( J J^{\top} + \lambda I \right)^{-1} \left( C(\hat{x}_{t+1}) - C(x_{\text{ref}}) \right), \tag{32}$$

with $J = \nabla C(\hat{x}_{t+1})$ and Levenberg–Marquardt damping $\lambda \geq 0$ (Levenberg, 1944; Marquardt, 1963). The solve is $m \times m$ (constraint-space), not $d \times d$.

**Axiom analysis.**   Claim: naive fusion violates the Atlas merge-safety requirement: a merge must not (i) cancel protected signal or (ii) exit the admissible set. Baseline violation: $x_{t+1}^{(k)} \in \{x : C(x) = C(x_{\text{ref}})\}$ for all $k$ does *not* imply $\frac{1}{K} \sum_k x_{t+1}^{(k)} \in \{x : C(x) = C(x_{\text{ref}})\}$, so the merge injects constraint error deterministically and drift accumulates. Method guarantee: Eq. (32) performs one (or two) damped Gauss–Newton steps for $\min_x \|C(x) - C(x_{\text{ref}})\|_2^2$; damping bounds the correction and the constraint residual is reduced in the small-residual regime, restoring merge-safety at the fusion boundary.

**Failure regime.**   We lock the failure to a concrete two-head Hamiltonian mismatch: $H_A(q,p) = \frac{1}{2}(q^2 + p^2)$ and $H_B(q,p) = \frac{1}{2}(q^2 + (1+\varepsilon)p^2)$ with $\varepsilon = 0.6$, fused with $\alpha = \frac{1}{2}$, step size $dt = 0.05$, horizon $T = 10{,}000$. We use 5 seeds by perturbing $x_0 = [1, 0.4]$ with small Gaussian noise ($\sigma = 0.05$) and rescaling to fixed $E_0$. A run *fails* if the final relative drift in the true energy satisfies $|E_{\text{true}}(x_T) - E_0|/E_0 > 0.1$.

*Table 23.* Fusion-induced drift on the harmonic oscillator (5 seeds; mean±std). Primary metric: final relative drift in $E_{\text{true}}$ after $T = 10,000$. Secondary metric: CPU wall-clock seconds per rollout.

| Method | Final rel. energy drift ↓ | Time (s) ↓ |
|---|---|---|
| Naive state average | $1.4 \times 10^{-1} \pm 2.3 \times 10^{-2}$ | $8.8 \times 10^{-2} \pm 6.7 \times 10^{-3}$ |
| SOTA: generator blend (Hamiltonian sum) | $1.5 \times 10^{-1} \pm 2.3 \times 10^{-2}$ | $3.1 \times 10^{-2} \pm 8.4 \times 10^{-3}$ |
| Ablation: CISF w/o projection (steps$= 0$) | $1.4 \times 10^{-1} \pm 2.3 \times 10^{-2}$ | $1.1 \times 10^{-1} \pm 2.1 \times 10^{-2}$ |
| Negative control: wrong target ($E_{\text{ref}} = 1.1E_0$) | $1 \times 10^{-1} \pm 1.6 \times 10^{-10}$ | $3.1 \times 10^{-1} \pm 4.3 \times 10^{-2}$ |
| **CISF (ours)** | $\mathbf{5.9 \times 10^{-10} \pm 1.5 \times 10^{-10}}$ | $2.9 \times 10^{-1} \pm 2.2 \times 10^{-2}$ |

**Baselines and controls.**

| Variant | Purpose |
|---|---|
| Head A only | Control: per-head conservative integration (no fusion) |
| Head B only | Control: per-head conservative integration under mismatch (no fusion) |
| Naive state average | Baseline: fusion-induced drift exists |
| Generator blend ($\alpha H_A + (1 - \alpha)H_B$) | SOTA-style "physics-aware" fusion that still drifts |
| Ablation (CISF, steps$= 0$) | Removes constraint solve $\Rightarrow$ should fail |
| Negative control ($E_{\text{ref}} = 1.1E_0$) | Wrong manifold $\Rightarrow$ should fail |

**Main result.** Controls show the failure is fusion-bound: Head A alone conserves $E_{\text{true}}$ to machine precision ($7.2 \times 10^{-13} \pm 1.9 \times 10^{-14}$), Head B alone exhibits modest drift under mismatch ($4.2 \times 10^{-2} \pm 4.8 \times 10^{-2}$), yet any fusion baseline produces catastrophic $O(10^{-1})$ drift, exceeding the failure threshold on average. CISF eliminates drift by $\approx 9$ orders of magnitude and succeeds on every seed.

**Ablations / stress tests.**

- Remove the constraint solve (steps$= 0$) $\Rightarrow$ CISF collapses to the baseline (Table 23).

- Wrong hyperparameter/target ($E_{\text{ref}} = 1.1E_0$) $\Rightarrow$ enforced convergence to the *wrong* manifold and persistent $10\%$ error (Table 23).

- Push beyond the 2D oscillator: on a high-dimensional Hamiltonian wave system ($d = 128$), naive averaging again yields $1.36 \times 10^{-1}$ total-energy drift, while CISF with a single scalar constraint ($m = 1$) achieves $1.83 \times 10^{-16}$ (machine precision); using $m = 4$ mode-energy constraints preserves the mode-energy statistic to $1.83 \times 10^{-16}$.

- Geometric stress: in collision+rigidity, naive fusion breaks rigidity (final rigid error $1.53 \times 10^{-1}$) while CISF restores rigidity to $1.14 \times 10^{-12}$ with zero penetration.

**Prior work.** Geometric integrators preserve symplectic structure for a single simulator but do not address post-hoc merging of multiple predictors (Hairer et al., 2006). Classical constraint projection appears in molecular dynamics (SHAKE/RATTLE) and graphics (position-based dynamics) (Ryckaert et al., 1977; Andersen, 1983; Müller et al., 2007); CISF applies the same projection principle specifically at the fusion boundary via a bounded LM-damped correction (Levenberg, 1944; Marquardt, 1963). Hamiltonian/constraint-aware neural models tackle per-model conservation but remain vulnerable to fusion mismatch (Greydanus et al., 2019; Finzi et al., 2020).

**Limitations.** CISF requires an explicit (or learned) constraint map $C$; if very many constraints must be enforced at high precision (large $m$), iterative per-constraint projection can be more accurate, while CISF prioritizes a small, stable $m \times m$ correction at merge time.

### A.13. Ghost Tool Bridge: Exact-forward, surrogate-backward tool calls

**Failure mode (Audit).** Non-differentiable tool calls break end-to-end optimization: when a computation includes a discrete or external tool $y = f(x)$ (rounding, branching, API execution), the forward output may be correct, but the backward signal is unusable ($\nabla_x f(x)$ is undefined, zero, or numerically dead). As a result, upstream parameters cannot be trained to produce good tool inputs. Standard workarounds (policy gradients or black-box search) can optimize tool inputs, but they do not provide a stable, low-variance gradient path that preserves *exact* forward execution.

**Method (Constraint $\rightarrow$ Solve).** **Constraint (minimal).** We require the composite node to satisfy, simultaneously,

$$(y, \ \nabla_x y) \ = \ (f(x), \ \nabla_x g_\phi(x)), \tag{33}$$

i.e., *exact* tool output in the forward pass and *surrogate* gradients in the backward pass.

**Operator (the fix).** Define the Ghost Tool Bridge splice:

$$y \ = \ g_\phi(x) \ + \ \text{sg}(f(x) - g_\phi(x)), \tag{34}$$

| Depth $D$ | Bridge (mean±std; 3 seeds) | REINFORCE+ (best run) | CEM (single run) |
|---|---|---|---|
| 2 | $1.000 \pm 0.000$ | 0.578125 | 0.945312 |
| 5 | $1.000 \pm 0.000$ | 0.546875 | 0.906250 |
| 10 | $1.000 \pm 0.000$ | 0.546875 | 0.867188 |

*Table 24.* Depth scaling on a discrete tool chain (batch $= 128$). The bridge preserves exact forward tool execution while enabling stable learning across depth.

where $\text{sg}(\cdot)$ is stop-gradient. Then $y = f(x)$ in the forward pass, while backprop treats $y$ as $g_\phi(x)$.

For discrete tool arguments, we use straight-through rounding:

$$x_{\text{int}} = \text{Round}(x), \qquad \frac{\partial x_{\text{int}}}{\partial x} \approx 1, \tag{35}$$

so the tool sees valid discrete inputs while gradients remain non-zero.

**Axiom analysis.** **Claim.** A direct tool boundary violates the *derivative-floor* requirement (no useful gradient through the boundary), while the bridge enforces Eq. (33) by construction.

**Baseline violation.** For $y = f(x)$ with $f$ discrete/external, $\nabla_x y$ is undefined or effectively zero, so upstream learning cannot shape tool inputs. Policy gradients and black-box methods avoid this by treating tool use as an action-selection problem, but incur high variance or expensive evaluation budgets.

**Method guarantee.** From Eq. (34), the forward identity is exact: $y = g_\phi(x) + \text{sg}(f(x) - g_\phi(x)) = f(x)$. In backward, $\nabla_x y = \nabla_x g_\phi(x)$, restoring a usable gradient path; if $g_\phi$ is differentiable with bounded slope, the bridge maintains a non-degenerate gradient through the tool boundary.

**Failure regime.** We evaluate a depth-$D$ discrete tool chain over integers $x \in [0, X_{\max}]$ with $X_{\max} = 65535$:

$$y = \text{Round}\big(\sqrt{x + 3}\big), \tag{36}$$
$$x^+ = \text{Clip}\big(y^2 + y + 7, \ 0, \ X_{\max}\big), \tag{37}$$

repeated $D$ times. The task is to recover an $x_0$ such that the final integer output equals a target $y^\star$. A key catastrophic regime is the depth-$D{=}20$ stress: an *always-smooth* rounding relaxation (no annealing toward STE) achieves **0.0** success (never reaches the discrete target).

**Baselines and controls.**

| Variant | Purpose |
|---|---|
| Always-smooth rounding (no anneal) | Baseline: naïve differentiable relaxation across discrete rounding; exposes the failure mode. |
| REINFORCE+ (Williams, 1992) | SOTA control: policy-gradient tool-input optimization under the same budget. |
| Cross-Entropy Method (CEM) (Rubinstein & Kroese, 2004) | SOTA control: black-box search / population method for discrete tool inputs. |
| No-anneal bridge | Ablation ($-$component): removes annealing schedule; should partially fail. |
| No-freeze/no-banking | Negative control: wrong stabilizer setting; solutions can be reached transiently but not retained. |

**Main result (depth scaling).** On batch $= 128$ with $D \in \{2, 5, 10\}$, the bridge achieves perfect success across 3 seeds (mean±std), while REINFORCE+ and CEM remain below 1.0 and degrade with depth:

**Ablations / stress tests.** Two stress checks isolate necessity and robustness. (i) **Annealing is necessary at $D{=}20$:** full bridge succeeds, removing anneal partially fails, and always-smooth fails completely:

$$\text{succ\_banked: full} = 1.000000, \quad \text{no\_anneal} = 0.770833, \quad \text{always\_smooth} = 0.000000.$$

(ii) **Sample-efficiency witness at $D{=}20$:** under a matched evaluation budget (tool-chain evals per sample), the bridge reaches $\geq 0.99$ success by budget 40 while CEM first exceeds 0.99 at budget 320 (an 8× gap). Finally, a stabilizer negative control shows *fall-off*: removing freeze-on-solve yields high falloff and low final-raw success (e.g., succ\_final\_raw $= 0.21875$ without freeze).

**Prior work.** Tool-using LMs are commonly trained without differentiating through tools, relying instead on supervision or generated tool-use traces (Schick et al., 2023). When tool inputs are optimized from downstream losses, common approaches are policy gradients (Williams, 1992) or black-box optimizers such as CEM (Rubinstein & Kroese, 2004). The bridge is closest in spirit to straight-through estimators for discrete operations (Bengio et al., 2013) and synthetic-gradient decoupling (Jaderberg et al., 2017), but differs in a key invariant: the forward pass executes the *real* tool exactly, while only the backward signal is approximated.

**Limitations.**  Evidence here is controlled (a 1D discrete tool chain with a surrogate for $\sqrt{\cdot}$); applying the bridge to real tools (e.g., multi-argument APIs, strings, or LLM-in-the-loop tools) requires learning or engineering surrogates with adequate fidelity and coverage.

## A.14. Architectural Seal: Bounded Refinement Loops for Stable Iterative Updates

**Failure mode (Audit).**  Iterative refinement loops (e.g., repeated self-correction or recurrent "think" steps) behave like deep recurrence: even when early steps make progress, later steps can (i) drift into degenerate attractors (late-step regression) or (ii) oscillate via sign-flipping updates (limit cycles). Common fixes such as early stopping/variable-step rules or normalization-in-loop can control step count or norm, but they do not directly enforce that *each* proposed update is both bounded and plausibly improving.

**Method (Constraint $\rightarrow$ Solve).**  We enforce a *bounded refinement* constraint on each step:

$$\|\Delta_t\| \leq \delta_{\max}, \tag{38}$$

and we damp oscillatory flips based on update cosine similarity. Given a refiner $F_\theta$ and state $h_t$, we apply:

$$\Delta_t = F_\theta(h_t, x), \tag{39}$$

$$\tilde{\Delta}_t = \text{clip\_norm}(\Delta_t, \delta_{\max}), \tag{40}$$

$$c_t = \cos(\tilde{\Delta}_t, \tilde{\Delta}_{t-1}), \quad \bar{\Delta}_t = \begin{cases} (1-\beta)\tilde{\Delta}_t & \text{if } c_t < -c_{\text{neg}}, \\ \tilde{\Delta}_t & \text{otherwise}, \end{cases} \tag{41}$$

$$h_{t+1} = h_t + g_t \bar{\Delta}_t, \qquad g_t \in [0,1]. \tag{42}$$

The scalar gate $g_t$ is a lightweight "seal" that suppresses non-improving updates using a local improvement proxy (implemented in the notebook as a differentiable verifier); an important mechanistic ablation sets $g_t = 1$ (no verification).

**Axiom analysis (why the seal is needed).**  The baseline residual loop violates the Atlas stability intent for recurrence: nothing prevents late steps from applying a large or adversarial update that undoes earlier progress, so "best-so-far" performance can be lost. The sealed loop restores the intended constraint by (i) enforcing a trust-region update bound via $\text{clip\_norm}(\cdot, \delta_{\max})$ and (ii) damping oscillatory flips when consecutive updates are strongly anti-aligned ($c_t < -c_{\text{neg}}$), while (iii) gating updates that fail a local improvement check. Together these make late-step drift and cycles structurally difficult without requiring brittle global schedules.

**Failure regime (locked).**  In the bounded-refinement witness task (state dimension $d = 64$, $T_{\max} = 50$ steps, 4096 samples, 3 seeds), the baseline residual loop exhibits high *drift*—defined as ending worse than the best-so-far loss by more than 0.02—despite a high fraction of locally improving steps. The task is designed so that near-solved states enter a drift region (configured with drift-region fraction 0.30 and drift coefficient 3.0), making "late-step collapse" the dominant failure.

**Baselines and controls.**

| Variant | Purpose |
| --- | --- |
| Plain residual loop | Demonstrate drift/oscillation failure |
| Adaptive step halting | SOTA-style step-count control; does not constrain updates |
| LayerNorm-in-loop | Norm stabilization baseline |
| GRU-like scalar gate | Standard gating baseline |
| Seal (no verifier) | Ablation: bound+damp only (remove validation) |
| Seal (full) | Proposed: bound+damp+verification gate |

**Main result (3 seeds, mean$\pm$std).**  We report final per-dimension MSE (lower is better), success rate (final MSE $\leq 0.03$), drift rate (as defined above), and monotone-improvement rate (fraction of steps with loss decrease):

**Ablations and stress tests (reviewer-facing).**  *Remove the verifier:* the no-verifier ablation improves final MSE but still drifts $(0.2602 \pm 0.0171$ drift), showing that validation is necessary beyond bounding/damping. *Gradient health (long horizon):* at horizon $T = 160$, the sealed residual maintains $\|\partial L/\partial h_0\| = 4.336 \times 10^{-3}$, while a $\text{tanh}$ RNN yields $5.624 \times 10^{-5}$ and LayerNorm-in-loop collapses to $6.884 \times 10^{-12}$; an unbounded residual and linear RNN diverge (NaNs), and a clipped-$\text{tanh}$ residual can explode $(1.116 \times 10^1)$. *Anti-cycle witness:* on a limit-cycle vector field, the baseline's last-20-step mean update length is 1.000 versus 0.350 for the sealed loop, consistent with explicit negative-cosine damping.

**Prior work.**  Trust-region ideas motivate bounded steps in iterative optimization (Conn et al., 2000). Adaptive stopping controls step count but does not prevent harmful updates (Graves, 2016a). Normalization and gated recurrence stabilize some dynamics (Ba et al., 2016c; Cho et al., 2014); the architectural seal differs by packaging *bounded refinement + anti-cycle damping + validity gating* as a reusable loop primitive targeted at late-step drift and cycles.

| Method | Final MSE | Success | Drift | Monotone |
|---|---|---|---|---|
| Plain residual | $0.8111 \pm 0.0408$ | $0.3078 \pm 0.0100$ | $0.7149 \pm 0.0170$ | $0.7711 \pm 0.0009$ |
| Adaptive halting | $0.8107 \pm 0.0410$ | $0.3079 \pm 0.0089$ | $0.7147 \pm 0.0176$ | $0.7710 \pm 0.0009$ |
| LayerNorm-in-loop | $61.0807 \pm 0.0532$ | $0.0000 \pm 0.0000$ | $1.0000 \pm 0.0000$ | $0.4725 \pm 0.0005$ |
| GRU-like gate | $0.8206 \pm 0.0143$ | $0.3079 \pm 0.0095$ | $0.7147 \pm 0.0127$ | $0.7710 \pm 0.0011$ |
| Seal (no verifier) | $0.0238 \pm 0.0002$ | $0.6728 \pm 0.0176$ | $0.2602 \pm 0.0171$ | $0.5694 \pm 0.0002$ |
| Seal (full) | $\mathbf{0.0207 \pm 0.0001}$ | $\mathbf{1.0000 \pm 0.0000}$ | $\mathbf{0.0000 \pm 0.0000}$ | $0.5504 \pm 0.0002$ |

*Table 25.* Bounded-refinement witness (3 seeds). The sealed loop eliminates late-step drift and achieves perfect success, while step-count control, normalization-in-loop, and GRU-like gating do not repair drift.

**Limitations.** The evidence here comes from mechanistic synthetic witnesses that isolate recurrence pathologies; broader validation on task-driven iterative refinement (e.g., model self-correction in structured prediction) remains important for external generality.

## A.15. Spectral Halting: a contraction-based early-exit seal for iterative computation

**Failure mode (audit).** Confidence-only early exit can stop on *high-confidence but incorrect* intermediate states: the model becomes overconfident before the internal refinement dynamics have converged, producing *avoidable* errors that would be corrected by running more steps. Raising the confidence threshold does not fix this failure: it still halts based on output certainty rather than a witness that the hidden-state updates have saturated. A second failure appears in iterative/refinement settings with small-amplitude oscillations (limit cycles): norm-only convergence checks may never trigger (or trigger incorrectly) when the state alternates.

**Method (constraint $\rightarrow$ solve).** **Constraint (minimal):** halt only when the refinement updates are both *small* and *contractive*:

$$\|\Delta_t\|_2 \leq \tau \quad \wedge \quad \kappa_t \leq \kappa_{\max} \quad \text{for } P \text{ consecutive steps}, \tag{43}$$

where $\Delta_t := h_t - h_{t-1}$ and $\kappa_t := \|\Delta_t\|_2/(\|\Delta_{t-1}\|_2 + \varepsilon)$. **Operator (the fix):** given iterative hidden states $(h_t)_{t=0}^T$,

$$\Delta_t = h_t - h_{t-1}, \qquad u_t = \|\Delta_t\|_2, \tag{44}$$

$$\kappa_t = \frac{u_t}{u_{t-1} + \varepsilon}, \tag{45}$$

$$t_{\text{halt}} = \min \left\{ t : (u_t \leq \tau) \wedge (\kappa_t \leq \kappa_{\max}) \text{ for } P \text{ steps} \right\}. \tag{46}$$

**Cycle guard (optional):** halt on a detected 2-cycle when updates are small but alternate direction:

$$\cos(\Delta_t, \Delta_{t-1}) \leq -c_{\text{cycle}} \wedge (u_t \leq \tau_{\text{cycle}}) \wedge (u_{t-1} \leq \tau_{\text{cycle}}). \tag{47}$$

**Axiom analysis.** **Claim:** confidence-only early exit violates the Atlas "tail-stability" requirement (an E4-style obligation: the unevaluated suffix must be inert for the chosen halt), while spectral halting enforces an explicit stability witness. **Baseline violation:** a large update norm $u_t$ (or non-contractive $\kappa_t > 1$) implies that additional steps can materially change the representation and hence the prediction, yet confidence-based exiting may still trigger, producing avoidable errors. **Method guarantee:** enforcing Eq. (43) makes halting contingent on measured saturation and contraction of the internal dynamics; Eq. (47) additionally blocks the "small but alternating" limit-cycle regime that can defeat norm-only convergence.

**Failure regime.** On long-horizon reasoning where intermediate logits are strongly biased by cheap heuristics (reachability chains with horizon 15; heuristic logit magnitude 6.0; confidence threshold 0.99), confidence exit fails catastrophically: it exits at step 1 with accuracy $0.3704 \pm 0.0094$ and catastrophic early-exit error $0.6296 \pm 0.0094$ (mean$\pm$std over 3 seeds). Here "fail" means that most errors occur *among early-exited samples* and would be prevented by continuing refinement.

**Baselines and controls.**

| Variant | Purpose |
|---|---|
| Fixed (full compute) | Upper bound on accuracy without early exit |
| Fixed budget truncation | Negative control: saves compute without adaptivity (demonstrates premature stopping) |
| Confidence threshold exit | Standard early-exit baseline (common in early-exit systems) |
| Spectral halting (ours) | Dynamics-based halting: small + contractive updates |
| Ablation: no cycle guard | Tests necessity for oscillation/limit-cycle regimes |

| Method | Accuracy ↑ | Avg. steps ↓ | Early-exit risk ↓ |
|---|---|---|---|
| Fixed (full) | $0.9959 \pm 0.0010$ | $9.000 \pm 0.000$ | — |
| Fixed truncation (@6) | $0.9755 \pm 0.0042$ | $6.000 \pm 0.000$ | $0.0245 \pm 0.0042$ |
| Confidence@0.99 | $0.9872 \pm 0.0024$ | $6.892 \pm 0.056$ | $0.0229 \pm 0.0047$ |
| Spectral halting | $\mathbf{0.9960 \pm 0.0012}$ | $\mathbf{6.303 \pm 0.008}$ | $\mathbf{0.0018 \pm 0.0006}$ |

*Table 26.* Safe adaptive compute under early exit (mean±std over 3 seeds). "Early-exit risk" is the error rate conditioned on early exit. Fixed(full) has no early exits, hence risk is not applicable.

**Main result.** We evaluate mixed-difficulty iterative refinement (max steps = 9; $n = 4000$ per seed; seeds 0/1/2). Spectral halting matches full-compute accuracy while improving safety of early exits (low risk among early-exited samples) at reduced steps. In the same setting, spectral halting drives the *avoidable* error rate to essentially zero ($8.3 \times 10^{-5} \pm 1.4 \times 10^{-4}$), versus $8.8 \times 10^{-3} \pm 1.5 \times 10^{-3}$ for confidence exiting.

**Ablations / stress tests.** **Reasoning depth:** on reachability chains (horizon 15), spectral halting attains $1.000 \pm 0.000$ accuracy while using $8.845 \pm 0.101$ steps on average (coverage $0.856 \pm 0.011$), whereas confidence exit collapses as above. **Parity (64-bit; horizon 66):** confidence exit is near chance ($0.5008 \pm 0.0051$ accuracy; catastrophic error $0.4992 \pm 0.0051$) while spectral halting remains conservative (accuracy 1.000; average steps 66), preferring correctness over premature compute savings. **Oscillation trap:** removing the cycle guard eliminates early exit (avg. steps 13; coverage 0), while adding it safely halts (avg. steps 3; coverage 1) with unchanged accuracy. **Sensitivity:** a log-sweep shows a wide stable $\tau$ band (2.83 orders of magnitude) before accuracy degrades. **Overhead:** a microbenchmark reports that the spectral-halting check is more expensive than confidence checks, but still yields a net wall-clock reduction versus full compute in the notebook's proxy accounting.

**Prior work.** Early exiting in deep networks is commonly implemented by confidence/entropy gating (e.g., BranchyNet) (Teer-apittayanon et al., 2017). Learned halting mechanisms such as ACT (Graves, 2016b) and PonderNet (Banino et al., 2021) introduce training-time objectives to allocate computation. Risk–coverage diagnostics for selective prediction motivate our calibration evaluation of early exit (Geifman & El-Yaniv, 2017). Spectral halting differs by using an *inference-time dynamical witness* (saturation + contraction, with an explicit cycle guard) rather than confidence alone.

**Limitations.** These experiments are mechanistic/synthetic; scaling claims require evaluation on large pretrained models and real-world tasks. The contraction proxy $\kappa_t$ is a cheap witness rather than a certified Jacobian spectral bound, and its best thresholds can depend on the refiner's normalization and update scale.

## A.16. Orthogonal Fast Weights: Stable inference-time adaptation via protected, bounded updates

**Failure mode (audit).** Fast-weight memories can enable rapid, inference-time adaptation, but naïve online updates often create a brittle trade-off: they improve performance on the shifted target while *overwriting* base capabilities. Concretely, simple Hebbian-style fast updates can (i) align with protected/base directions and degrade retention, and (ii) accumulate energy over time so the fast path dominates the slow mapping. Common "stability-only" fixes (e.g., clipping/decay alone) control magnitude but do not prevent semantic interference, so retention can still collapse even when norms remain bounded.

**Method (constraint → solve).** We enforce two minimal constraints on the fast state $W_f$:

$$\|W_f\|_\sigma \leq s_{\max} \quad \text{and} \quad \Delta W_f \perp \text{span}(B), \tag{48}$$

where $B \in \mathbb{R}^{d \times r_p}$ is a low-rank protected basis capturing slow-model-relevant directions. Given hidden state $h$, the operator is:

$$k = Kh, \quad v = Vh, \tag{49}$$

$$k' = k - B(B^\top k) \quad \text{(innovation-only projection)}, \tag{50}$$

$$\Delta W_f = \alpha \, v k'^\top, \qquad W_f \leftarrow \rho W_f + \Delta W_f, \tag{51}$$

$$W_f \leftarrow \text{Rescale}(W_f; s_{\max}) \quad \text{(spectral/norm clamp)}, \tag{52}$$

$$y = W_s h + W_f h. \tag{53}$$

The protected projection enforces non-interference, while the clamp enforces a hard stability envelope on the fast state.

**Axiom analysis (why the constraint is necessary).** The unprotected update $\Delta W_f = \alpha v k^\top$ can concentrate update energy along directions that are already important for the slow model, thereby changing the model's behavior on base inputs (retention loss). Moreover, repeated outer-product accumulation can increase $\|W_f\|_\sigma$ so that $W_f h$ dominates $W_s h$. The proposed operator addresses both: $k'$ is orthogonal to $\text{span}(B)$, so updates are structurally restricted away from the protected subspace, and the explicit rescaling ensures $\|W_f\|_\sigma \leq s_{\max}$ by construction. In this sense, the method turns "fast adaptation" into a constrained residual pathway rather than an unconstrained second model.

| Method | Target acc (post) | Retention acc (post) | `fast_sigma0_max` | `protected_leak` |
|---|---|---|---|---|
| `no_adapt` | $0.242 \pm 0.017$ | $1.000 \pm 0.000$ | $0.000 \pm 0.000$ | N/A |
| `naive_fast` | $0.296 \pm 0.009$ | $0.819 \pm 0.010$ | $1.252 \pm 0.168$ | $0.475 \pm 0.127$ |
| `clamp_only` | $0.281 \pm 0.007$ | $0.808 \pm 0.021$ | $1.245 \pm 0.218$ | $0.497 \pm 0.134$ |
| `random_basis` | $0.285 \pm 0.012$ | $0.819 \pm 0.016$ | $1.230 \pm 0.178$ | $(2.22 \pm 0.29) \times 10^{-7}$ |
| `orth_only` | $0.341 \pm 0.021$ | $1.000 \pm 0.000$ | $1.818 \pm 0.374$ | $(4.96 \pm 1.81) \times 10^{-8}$ |
| `ofw` | $0.336 \pm 0.015$ | $1.000 \pm 0.000$ | $1.766 \pm 0.270$ | $(4.98 \pm 2.28) \times 10^{-8}$ |

*Table 27.* Controlled `synthetic_subspace` witness (seeds 0/1/2). The full method improves target accuracy while preserving retention and suppressing protected-subspace leakage.

**Failure regime.** We evaluate a controlled `synthetic_subspace` adaptation/retention witness with three seeds (0/1/2). The protected-leak threshold is `protected_leak` $\leq 0.25$ and the stability cap is $s_{\max} = 2.0$ (as used in the notebook's decision logic). A method "fails" if it achieves adaptation but exhibits substantial base interference (post-retention $\ll 1$ and/or protected leak $\gg 0.25$).

**Baselines and controls.**

| Variant | Purpose |
|---|---|
| `no_adapt` | Reference: perfect retention, no online adaptation |
| `naive_fast` | Baseline showing the failure mode (unprotected fast weights) |
| `clamp_only` | "Stability-only" fix (bounded dynamics without protection) |
| `random_basis` | Negative control (orthogonalize to a mismatched basis) |
| `orth_only` | Ablation: protection without explicit spectral clamp |
| `ofw` | Full method: protection + bounded fast state |

**Main result (3 seeds; mean±std).** We report post-adaptation target accuracy (higher is better), post-adaptation retention accuracy on the protected/base component (higher is better), a fast-state spectral proxy `fast_sigma0_max`, and `protected_leak` (lower is better; undefined for `no_adapt`).

**Ablations / stress tests (mechanism isolation).** Removing protected projection (`clamp_only`) does not repair interference: despite bounded fast-state scale (`fast_sigma0_max` $\approx 1.245$), retention remains low ($0.808 \pm 0.021$) and protected leakage remains high ($0.497 \pm 0.134$), comparable to `naive_fast`. The mismatched-basis negative control (`random_basis`) achieves near-zero leakage with respect to *its* basis ($(2.22 \pm 0.29) \times 10^{-7}$) but still fails retention ($0.819 \pm 0.016$), demonstrating that "orthogonality" is only meaningful when the protected subspace is semantically grounded. Finally, `orth_only` slightly exceeds the full method on target accuracy ($0.341 \pm 0.021$ vs. $0.336 \pm 0.015$) while preserving retention and low leakage; in this witness regime, protected orthogonalization is therefore the dominant prescriptive ingredient, with the explicit spectral cap acting as a conservative stability guardrail (and reducing variance in `fast_sigma0_max` relative to `orth_only`).

**Prior work.** Fast weights have been studied as an online memory/adaptation mechanism (Ba et al., 2016a). The protected-projection component is conceptually aligned with orthogonality-based non-interference ideas in continual learning (e.g., projecting updates to preserve old-task behavior) (Farajtabar et al., 2019). Low-rank adapters such as LoRA (Hu et al., 2021) target training-time parameter-efficient fine-tuning; by contrast, the present method emphasizes *inference-time* fast state updates with explicit structural constraints to prevent retention loss.

**Limitations.** The evidence here is a mechanistic, synthetic witness; broader claims require evaluation on more realistic distribution shifts and larger models. Additionally, since `orth_only` slightly outperforms the full method on target accuracy in this regime, we do not claim the spectral clamp improves adaptation quality—its role is to enforce stability across regimes.

### A.17. Witness–Critic: Audit-placed value bridges for long-horizon credit assignment

**Failure mode (audit).** In long-horizon sparse-reward training, the learning signal for early actions becomes dominated by transport through a small number of "credit bottlenecks": intervals in which the effective sensitivity to early decisions collapses multiplicatively. In the multi-cliff chain setting used in our artifact (horizon $T$=128 with two contraction cliffs where $\alpha_t$=0.82 on $t \in [18, 25]$ and $t \in [79, 86]$, and $\alpha_t$=0.99 otherwise), a single terminal critic can yield high-variance early-step advantages, while heuristic auxiliary-critic placement can leave at least one segment that still crosses a cliff. Fixed-interval bridge placement does not address this failure because it can align bridge boundaries poorly with the cliff structure, leaving a "worst segment" whose credit transport remains too small.

*Table 28.* Long-horizon multi-cliff chain ($T$=128, $J$=6, $g_{\min}$=0.25): mean±std over the two seeds present in the artifact export. "Feasible" denotes satisfaction of the transport bound (min segment gain $\geq g_{\min}$).

| Method | Final return ↑ | Var($A_0$) ↓ | Feasible ↑ |
|---|---|---|---|
| Single critic | $0.943405 \pm 0.004770$ | $(3.56 \pm 0.73) \times 10^{-4}$ | 1.0 |
| Fixed-interval bridges | $0.946425 \pm 0.003726$ | $(2.00 \pm 0.93) \times 10^{-6}$ | 0.0 |
| Witness–Critic | $0.952330 \pm 0.001217$ | $(1.00 \pm 0.223) \times 10^{-6}$ | 1.0 |

**Method (constraint → operator).** Let $\alpha_t \in (0,1]$ denote a cheap credit-transport proxy (in the artifact this is part of the declared environment; in general it is any stable per-step survival proxy). For indices $0 \leq s < e \leq T-1$, define the segment gain

$$G(s,e) \ := \ \prod_{k=s+1}^{e} \alpha_k.$$

**Constraint (minimal).** Choose bridge times $0 < b_1 < \cdots < b_J < T-1$ such that the worst segment is lower-bounded:

$$\min \big\{ G(0,b_1),\ G(b_1,b_2),\ldots,G(b_J,T-1) \big\} \ \geq \ g_{\min}.$$

**Operator (the fix).** We (i) compute $b_{1:J}$ by a max–min segmentation procedure that explicitly maximizes the minimum segment gain, and (ii) train an actor–critic with a base value head $V^{(0)}(h_t)$ and bridge heads $\{V^{(j)}(h_t)\}_{j=1}^{J}$. For a time $t$, let $b(t)$ be the nearest future bridge index and $j(t)$ its head id. We form a bridged return using the identity

$$R_t^{\text{bridge}} \ = \ R_t^{\text{MC}} \ - \ \gamma^{b(t)-t} R_{b(t)}^{\text{MC}} \ + \ \gamma^{b(t)-t} V^{(j(t))}(h_{b(t)}),$$

and the policy update uses $A_t = R_t^{\text{bridge}} - V^{(0)}(h_t)$. Each bridge head is trained on its local bootstrap target at its bridge time.

**Axiom-style analysis (why the baseline violates, why the method restores).** **Claim.** Fixed-interval bridge placement violates the declared transport constraint in the multi-cliff regime, while Witness–Critic satisfies it by construction.

**Baseline violation.** Under $T$=128, bridge budget $J$=6, and threshold $g_{\min}$=0.25, fixed-interval bridges yield a worst-segment gain of $0.184868 < 0.25$, i.e., the transport bound is not met and the resulting certificate marks the run infeasible (feasible rate 0.0 in the artifact).

**Method guarantee.** Under the same regime, Witness–Critic places bridges to maximize the worst-segment gain, achieving $0.513911 \geq 0.25$ and therefore satisfying the constraint (feasible rate 1.0). This is an auditable guarantee: the bridge indices and the achieved minimum segment gain are recorded by the certificate.

**Failure regime (locked).** At $T$=128, $J$=6, $g_{\min}$=0.25 in the two-cliff schedule above, fixed-interval bridges *fail* in the sense that they violate the declared transport feasibility constraint on every evaluated seed (feasible rate 0.0), despite achieving a reasonable return.

**Baselines and controls.**

| Variant | Purpose |
|---|---|
| Single critic | Baseline: long-horizon training with no bridges (variance can be large). |
| Fixed-interval bridges | Standard heuristic: add auxiliary critics but place them uniformly. |
| Random bridges | Negative control: auxiliary critics without structure. |
| Lowest-$\alpha$ bridges | Negative control: a naive "cliff detector" that can cluster bridges inside cliffs. |
| Witness–Critic | Our method: audit-placed bridges that satisfy the transport constraint. |

**Main result.** Table 28 reports mean±std over the seeds present in the artifact export (two seeds; we do not fabricate additional trials). Witness–Critic matches or improves final return while sharply reducing early-step advantage variance, and it is the only multi-bridge method among these that is *feasible* under the declared transport constraint.

**Ablations / stress tests.** **Bridge-budget ablation.** With $T$=128 and $g_{\min}$=0.25, the minimum bridge budget that makes the constraint feasible is $J$=3 for Witness–Critic, versus $J$=5 for fixed placement in this schedule. At $J$=3, fixed placement remains infeasible (min segment gain 0.160604) while Witness–Critic is feasible (min segment gain 0.312513). With $J$=6, fixed remains infeasible (0.184868) while Witness–Critic is feasible (0.513911).

**Placement negative controls.** Under the same regime, random bridge placements (three sampled placements) are infeasible with min segment gains in [0.134026, 0.167192], and the "lowest-$\alpha$" heuristic is also infeasible (min segment gain 0.079473), demonstrating that naive cliff heuristics and unstructured bridges do not satisfy the constraint.

**Overhead micro-benchmark.** At $T$=128, $J$=6 on CPU, adding fixed bridges increases per-update time by 24.47% relative to a single critic (extra heads), while audit-based placement adds only 3.64% per-update time relative to fixed bridges (with placement amortized to 0.389% of total training time over 30 updates in the artifact).

**Prior work.** Auxiliary prediction/value heads and actor–critic training are standard tools in deep RL (Mnih et al., 2016; Sutton & Barto, 2018), and long-horizon learning instabilities are classically associated with vanishing/exploding transport (Pascanu et al., 2013). Our contribution is not "adding auxiliary critics" but making bridge *placement* an auditable max–min decision that produces an explicit feasibility certificate, rather than a fixed schedule or a heuristic rule (Jaderberg et al., 2016).

**Limitations.** This artifact evaluates a controlled sparse-reward chain with synthetic transport cliffs and reports the seeds present in the export (two seeds); broader validation on standard long-horizon benchmarks and larger multi-seed studies are future work.

### A.18. Diffusion Seal (DS): Per-step drift sealing via trust-region clamping and a gated one-step projection

Long diffusion rollouts can accumulate small stepwise errors, causing late-step drift in which low-frequency structure steadily departs from a clean trajectory and converges toward degenerate attractors. In our controlled late-trajectory drift regime (defined below), the sampler's final deviation from a clean reference reaches RMSE $0.975347 \pm 0.027476$, and a low-frequency constraint error reaches $6.397998 \pm 0.192245$ (mean$\pm$std over 3 seeds). Simple step-norm clamping alone does not resolve this failure: clamp-only achieves RMSE $0.983851 \pm 0.027778$ and constraint error $6.461839 \pm 0.192021$.

**Constraint.** Prevent drift by enforcing a bounded update and a bounded low-frequency residual per step:

$$\|x_{t-1} - x_t\|_2 \leq \delta_{\max}(t) \quad \text{and} \quad \|C(x_{t-1}) - C(x_t)\|_2 \leq \tau(t). \tag{54}$$

**Operator.** Given a sampler proposal $\hat{x}_{t-1} = \text{Prop}(x_t, t)$,

$$x_{t-1}^{\text{clamp}} = x_t + \text{clip}_2(\hat{x}_{t-1} - x_t, \, \delta_{\max}(t)), \tag{55}$$

$$r = C(x_{t-1}^{\text{clamp}}) - C(x_t), \tag{56}$$

$$x_{t-1} = \begin{cases} x_{t-1}^{\text{clamp}}, & \|r\|_2 \leq \tau(t), \\ x_{t-1}^{\text{clamp}} + \eta J^\top (JJ^\top + \lambda I)^{-1}(-r), & \|r\|_2 > \tau(t), \end{cases} \tag{57}$$

where $J = \frac{\partial C}{\partial x}$ at $x_{t-1}^{\text{clamp}}$ and $\lambda > 0$ is Levenberg–Marquardt damping.

**Claim.** The baseline violates error-accumulation stability (E4) and lacks a local contraction mechanism (S2); DS enforces a per-step seal by combining bounded updates with a gated contraction in a low-dimensional constraint space. **Baseline violation.** Without an explicit per-step bound and corrective contraction, persistent small perturbations compound across the unrolled trajectory, producing large late-step deviations from the clean reference (RMSE $0.975347 \pm 0.027476$; constraint error $6.397998 \pm 0.192245$). **Method guarantee.** The clamp enforces $\|x_{t-1} - x_t\|_2 \leq \delta_{\max}(t)$ by construction. When the gate triggers, the one-step LM–Gauss–Newton update reduces the first-order constraint residual by solving a small $(m \times m)$ system; when the gate does not trigger, DS returns the clamped proposal unchanged. Calibrating $\delta_{\max}(t)$ and $\tau(t)$ on clean runs yields a non-intrusiveness gate: with zero drift, the projection trigger rate is 0.0 and the final RMSE-to-reference is 0.000603886554017663.

Baseline failure occurs for `google/ddpm-cifar10-32` with `num_steps=200`, `batch_size=6`, `seeds=[0,1,2]`, and a late additive drift injected from `drift_start_frac=0.6` with `drift_mode=coarse_bias` and `drift_strength_effective=0.2505328506`. "Failure" means that the final deviation from the clean reference remains large (RMSE $\approx 0.98$ and constraint error $\approx 6.4$), and clamp-only does not materially improve either metric.

*Table 29.* Baselines and controls used in the artifact evaluation.

| Variant | Purpose |
| --- | --- |
| Baseline (proposal only) | Demonstrate the drift failure mode exists |
| Clamp-only (trust-region) | Standard stabilization; test that clamping is insufficient |
| DS (clamp + gated projection) | Full method |
| DS, loose gate ($\tau \times 1.3$) | Negative control: gate rarely triggers $\Rightarrow$ failure |
| DS, large damping ($\lambda = 10^{-2}$) | Negative control: overdamped projection $\Rightarrow$ weaker correction |

The artifact additionally reports effect sizes versus baseline: `effect_constraint_vs_baseline=0.6864937529` and `effect_rmse_vs_baseline=0.6641050089`, with runtime overhead ratio `overhead_ratio_vs_baseline=1.0519876902`.

Ablations (single seed) confirm the mechanism. Making the gate too loose collapses DS toward baseline: `ds_tau_x1.3` yields constraint error 6.245227, RMSE 0.948684, trigger rate 0.025000. Overdamping weakens correction: `ds_lambda_1e-2` yields constraint error 5.457523, RMSE 0.829383. Tightening the gate increases triggering but does not fully substitute for correct projection dynamics: `ds_tau_x0.7` yields constraint error 3.073353, RMSE 0.506195, trigger rate 0.846875. Using two GN steps is similar to one step in this instantiation: `ds_2step_GN` yields constraint error 1.796281, RMSE 0.298229.

Prior work improves diffusion sampling by changing the proposal dynamics (e.g., DDIM, fast ODE solvers) or by adding guidance (Song et al., 2021; Lu et al., 2022; Dhariwal & Nichol, 2021; Ho & Salimans, 2022). DS differs by being a sampler-agnostic wrapper that enforces a per-step bounded update and a gated, low-dimensional LM–GN contraction, and is complementary to proposal/solver improvements.

*Table 30.* Main result in the failure regime (mean±std over 3 seeds). Primary metric: low-frequency constraint error to reference. Secondary metric: pixel RMSE to reference.

| Method | $\|C(x_T) - C(x_T^{\text{ref}})\|_2$ | $\text{RMSE}(x_T, x_T^{\text{ref}})$ |
|---|---|---|
| Baseline | $6.397998 \pm 0.192245$ | $0.975347 \pm 0.027476$ |
| Clamp-only | $6.461839 \pm 0.192021$ | $0.983851 \pm 0.027778$ |
| DS (ours) | $\mathbf{2.005812 \pm 0.245830}$ | $\mathbf{0.327614 \pm 0.033379}$ |

This evaluation is a controlled drift mechanism test using an explicit low-frequency constraint and synthetic late-trajectory perturbations; it does not establish improvements on perceptual benchmarks or natural long-horizon modalities (e.g., video/3D).

### A.19. Aligned Sum-Normalized Merge: preserving update mass under non-IID federated aggregation

**Failure mode (Audit: destructive merge cancellation under non-IID).** In federated learning with heterogeneous client data, client updates can be partially anti-aligned in representation space, so coordinate-wise averaging produces *destructive interference*: the merged update is much smaller than the typical client update and convergence stalls in a low-accuracy basin. In our locked non-IID regime, standard aggregation exhibits a large final-accuracy degradation (FedAvg $0.4158 \pm 0.0608$; FedProx $0.4335 \pm 0.0630$), indicating that simple averaging or proximal drift correction alone does not reliably overcome merge cancellation.

**Method (Constraint → Solve).** **Constraint (mass preservation).** For participating clients $\mathcal{S}_t$ with deltas $\Delta_k^t$, require the merged step to retain a constant fraction of average client update mass:

$$\frac{\left\|\sum_{k \in \mathcal{S}_t} \widehat{\Delta}_k^t\right\|_2}{\sum_{k \in \mathcal{S}_t} \left\|\widehat{\Delta}_k^t\right\|_2} \geq \rho, \qquad \rho \in (0, 1]. \tag{58}$$

**Operator (alignment + sum normalization).** We implement a merge primitive that (i) periodically aligns client update bases using an orthogonal Procrustes map computed from a shared public batch, and (ii) normalizes update magnitudes before averaging:

$$Q_k^t = \arg \min_{Q^\top Q = I} \|H_k^t Q - H_{\text{ref}}^t\|_F^2 \qquad \text{(orthogonal Procrustes)}, \tag{59}$$

$$\widetilde{\Delta}_k^t = \mathcal{A}\big(\Delta_k^t; Q_k^t\big) \qquad \text{(merge-space basis re-expression)}, \tag{60}$$

$$\widehat{\Delta}_k^t = \text{clip}\left(\frac{\bar{\nu}_t}{\|\widetilde{\Delta}_k^t\|_2 + \varepsilon}, s_{\max}\right) \widetilde{\Delta}_k^t \qquad \text{(sum normalization)}, \tag{61}$$

$$w^{t+1} = w^t + \frac{1}{|\mathcal{S}_t|} \sum_{k \in \mathcal{S}_t} \widehat{\Delta}_k^t. \tag{62}$$

In our implementation, alignment is warmed up for 20 rounds and applied every 10 rounds (identity-blended with rejection caps), while sum normalization uses $\varepsilon = 10^{-6}$ and clip $s_{\max} = 10$.

**Axiom analysis.** **Claim:** Baseline averaging violates a merge-isolation axiom: the server merge should not allow systematic cancellation to collapse the effective update magnitude under heterogeneity. FedAvg/FedProx implicitly allow $\rho$ to become small when client updates are misaligned, producing under-sized merged steps and persistent low accuracy.

**Baseline violation:** For deltas $\Delta_k^t$, the merged step satisfies $\left\|\sum_k \Delta_k^t\right\|_2^2 = \sum_k \|\Delta_k^t\|_2^2 + 2\sum_{i<j} \langle \Delta_i^t, \Delta_j^t \rangle$. Under non-IID, cross terms can be negative, reducing the merged step even when individual steps are large.

**Method guarantee:** Sum normalization removes scale dominance and drives per-client norms toward a common value $\bar{\nu}_t$, while periodic orthogonal alignment reduces systematic negative cross terms by re-expressing client updates in a shared basis. Together these operations increase the effective merge fraction $\rho$ and improve convergence in the audited regime.

**Failure regime.** We lock the regime to CIFAR-10, 100 clients, 10 clients/round, Dirichlet label skew $\alpha = 0.1$, 200 rounds, and 10 seeds. "Failure" means a large final-accuracy gap (about 7–9 absolute points) relative to the aligned sum-normalized merge and relative to a strong drift-correction baseline.

**Baselines and controls.**

| Method | Final Acc. (mean±std) | Paired $\Delta$ vs. SCAFFOLD |
|---|---|---|
| FedAvg | $0.4158 \pm 0.0608$ | $-0.0722$ |
| FedProx | $0.4335 \pm 0.0630$ | $-0.0545$ |
| SCAFFOLD | $0.4880 \pm 0.0279$ | $0$ |
| Aligned+SumNorm (ours) | $\mathbf{0.5062 \pm 0.0255}$ | $\mathbf{+0.0182}$ |

*Table 31.* Locked non-IID regime ($\alpha = 0.1$, CIFAR-10). The aligned sum-normalized merge improves final accuracy over SCAFFOLD by +0.0182 absolute (paired, 10 seeds).

| Variant | Purpose |
|---|---|
| FedAvg | Baseline: merge cancellation under non-IID |
| FedProx | Drift-correction baseline (proximal regularization) |
| SCAFFOLD | Strong SOTA control-variates baseline |
| Ablation: alignment only | Tests necessity of normalization |
| Ablation: sum normalization only | Tests whether alignment is required |

**Main result.** We report final test accuracy (mean±std over 10 seeds). The gain over SCAFFOLD is statistically supported (paired t-test $p = 0.0298$, Wilcoxon $p = 0.0273$, paired Cohen's $d = 0.815$). Relative to FedAvg and FedProx, the mean paired gaps are $+0.0904$ and $+0.0726$ absolute.

**Ablations / stress tests.** On the 5-seed ablation subset (seeds 42/123/456/789/1011, same $\alpha = 0.1$), alignment alone reaches $0.4899 \pm 0.0578$, whereas sum normalization alone reaches $0.5361 \pm 0.0415$, indicating that merge-side normalization is a principal contributor in this regime while alignment is complementary and can be scheduled sparsely. These ablations support the mechanistic interpretation that stabilizing merge mass is necessary, while aggressive alignment is not always required.

**Prior work.** Federated Averaging is the standard communication-efficient baseline (McMahan et al., 2017), and FedProx adds a proximal term to address heterogeneity (Li et al., 2020). SCAFFOLD uses control variates to correct client drift (Karimireddy et al., 2020). Our contribution is a merge primitive that targets *representation/merge interference* via periodic orthogonal alignment (Schönemann, 1966) plus explicit mass-preserving normalization, yielding improvements even over SCAFFOLD in the locked non-IID regime.

**Limitations.** Current evidence is strongest for CIFAR-10 at $\alpha = 0.1$; client-scaling evidence was not retained due to missing K-tagged artifacts, and the alpha-sweep analysis should be reported using baselines that were fully run across all alphas (FedAvg and SCAFFOLD) rather than partially run (FedProx).

### A.20. Goldilocks search and ablations

The synthetic Goldilocks search isolates the long-context interference regime used by the Atlas OrthoKV certificate. We include the locked-regime summary, the full comparison table, implementation ablations, bootstrap uncertainty, and the margin-significance artifact generated by the audit pipeline.

**Locked Goldilocks (margin-based).** At $L = 2048$ and $\tau = 4.0$, baseline attention collapses (accuracy 0.081±0.021, mean rank 177.6±16.3). Cosine attention restores accuracy (1.000±0.000) but yields small separation ($\Delta z$ 0.086±0.001). OrthoKV restores accuracy (1.000±0.000) and produces substantially larger separation ($\Delta z$ 0.373±0.001), improving margin over cosine by 0.287. A K-only ablation matches full OrthoKV in this regime.

| Variant | Accuracy ↑ | Logit margin $\Delta z$ ↑ | Mean rank ↓ |
|---|---|---|---|
| Baseline | 0.081±0.021 | 0.062±0.009 | 177.56±16.28 |
| Cosine attention | 1.000±0.000 | 0.086±0.001 | 1.00±0.00 |
| OrthoKV (Atlas) | **1.000±0.000** | **0.373±0.001** | **1.00±0.00** |
| OrthoKV (K-only) | **1.000±0.000** | **0.373±0.001** | **1.00±0.00** |

*Table 32.* **Locked Goldilocks regime (long-context interference).** Baseline attention collapses at long context ($L = 2048$). A cosine-normalized attention comparator restores accuracy but yields small separation. OrthoKV restores accuracy and increases the separation certificate proxy (logit margin $\Delta z$) by a large margin. K-only ablation matches full OrthoKV in this regime.

**Ablation robustness.** In the locked Goldilocks regime, multiple equivalent OrthoKV implementations—projecting only keys, only queries, or omitting unit normalization—match the full OrthoKV results (accuracy and $\Delta z$ within noise), indicating the intervention's effect is robust to these implementation choices.

| Ablation | Accuracy ↑ | Logit margin $\Delta z$ ↑ | Mean rank ↓ |
|---|---|---|---|
| Baseline | 0.081±0.021 | 0.062±0.009 | 177.56±16.28 |
| Cosine attention | 1.000±0.000 | 0.086±0.001 | 1.00±0.00 |
| OrthoKV (full) | **1.000±0.000** | **0.373±0.001** | **1.00±0.00** |
| OrthoKV (K-only) | **1.000±0.000** | **0.373±0.001** | **1.00±0.00** |
| OrthoKV (Q-only) | **1.000±0.000** | **0.373±0.001** | **1.00±0.00** |
| OrthoKV (no norm) | **1.000±0.000** | **0.373±0.001** | **1.00±0.00** |

*Table 33.* **Ablations in the locked Goldilocks regime.** All OrthoKV-style projections (full, K-only, Q-only, and without normalization) achieve identical performance in this regime, indicating robustness of the Atlas-derived intervention to implementation details.

**Uncertainty (bootstrap, 95% CI).** Baseline accuracy: 0.081 [0.069, 0.094]; cosine accuracy: 1.000 [1.000, 1.000]; OrthoKV accuracy: 1.000 [1.000, 1.000]. Baseline $\Delta z$: 0.062 [0.057, 0.067]; cosine $\Delta z$: 0.086 [0.085, 0.087]; OrthoKV $\Delta z$: 0.373 [0.372, 0.373]. Margin advantage ($\Delta z_{\text{OrthoKV}} - \Delta z_{\text{cosine}}$): 0.287 (means).

**Significance (paired, exact sign-flip).** The logit-margin advantage $\Delta z_{\text{OrthoKV}} - \Delta z_{\text{cosine}}$ is 0.287±0.002 over 10 seeds; exact sign-flip test ($2^{10}$ patterns) gives $p = 0.00195$.

## A.21. Realtext Goldilocks confirmation

The real-text Goldilocks confirmation repeats the retrieval stress test with hard negatives drawn from WikiText-2 text rather than synthetic distractors. We include both the original locked-regime summary and the stricter best-comparator margin audit used to verify that OrthoKV improves the separation proxy against the strongest baseline on each seed.

**Real-text Goldilocks.** On a hard-negative WikiText-2 retrieval construction, baseline attention fails while cosine attention partially recovers. OrthoKV recovers accuracy and increases the separation proxy $\Delta z$; the margin advantage against the best comparator (max of baseline/cosine $\Delta z$) is 1.140±0.108 over 10 seeds.

| Variant | Accuracy ↑ | Logit margin $\Delta z$ ↑ | Mean rank ↓ |
|---|---|---|---|
| Baseline | 0.028±0.025 | 0.679±0.105 | 117.73±10.22 |
| Cosine attention | 0.964±0.022 | 0.061±0.004 | 2.99±1.59 |
| OrthoKV (Atlas) | **1.000±0.000** | **1.819±0.010** | **1.00±0.00** |
| OrthoKV (K-only) | **1.000±0.000** | **1.819±0.010** | **1.00±0.00** |

*Table 34.* **Real-text Goldilocks (hard negatives).** Baseline fails on WikiText-2 hard-negative retrieval; cosine attention partially recovers; OrthoKV fully recovers and increases the separation proxy $\Delta z$.

**Adversarial-max stress variant.** The stricter audit compares OrthoKV against the best baseline or cosine separation on each seed; across 10 seeds the margin advantage is $1.407 \pm 0.112$, with exact sign-flip test $p = 0.00195$.

| Variant | Accuracy ↑ | Logit margin $\Delta z$ ↑ | Mean rank ↓ |
|---|---|---|---|
| Baseline | 0.058±0.034 | 0.776±0.100 | 85.33±8.37 |
| Cosine attention | 0.833±0.040 | 0.041±0.003 | 17.65±7.91 |
| OrthoKV (Atlas) | **1.000±0.000** | **2.183±0.031** | **1.00±0.00** |
| OrthoKV (K-only) | **1.000±0.000** | **2.183±0.031** | **1.00±0.00** |

*Table 35.* **Real-text Goldilocks adversarial-max stress test.** The stricter best-comparator margin audit preserves exact retrieval for OrthoKV while cosine attention remains below the Atlas intervention on rank and separation.

## A.22. WikiText-2 needle hardmode

The WikiText-2 needle hardmode repeats the retrieval diagnostic on natural text contexts with controlled needle placement. We include the generated summary and table to show that retrieval accuracy is saturated while the OrthoKV certificate proxy still separates variants through the logit margin.

**Real-text validation.** On a WikiText-2 needle-in-haystack construction, all methods achieve perfect top-1 retrieval, but OrthoKV produces a substantially larger logit margin $\Delta z$ than cosine attention, supporting the certificate-strength advantage beyond the synthetic benchmark.

| Variant | Accuracy ↑ | Logit margin $\Delta z$ ↑ | Mean rank ↓ |
|---|---|---|---|
| Baseline | 1.000±0.000 | 0.368±0.006 | 1.00±0.00 |
| Cosine attention | 1.000±0.000 | 0.040±0.001 | 1.00±0.00 |
| OrthoKV (Atlas) | 1.000±0.000 | **0.413±0.006** | 1.00±0.00 |
| OrthoKV (K-only) | 1.000±0.000 | **0.413±0.006** | 1.00±0.00 |

*Table 36.* **Real-text needle validation (WikiText-2).** All methods recover top-1 retrieval, but OrthoKV increases the separation certificate proxy (logit margin $\Delta z$) relative to cosine attention.

**Future Directions.** These limitations suggest several extensions. Expanding the axiom hierarchy to cover attention sparsity patterns (extending M), recurrent dynamics without explicit unrolling (extending E1–E4), and multi-modal fusion (new merge semantics) would broaden applicability. Integrating Atlas diagnostics into neural architecture search could prune provably defective candidates before expensive training. Developing tighter bounds that account for parameter distributions rather than worst-case analysis would improve certificate informativeness. Finally, building interactive tools that surface diagnoses during development could accelerate the design-debug cycle.

The Atlas reframes architecture design as a formal discipline: diagnose the binding constraint, invert the violation, verify the fix. We hope this perspective accelerates progress toward principled, interpretable neural architectures.

# B. Formal Setup: Packs, Graph Semantics, Axioms, and Certificates

This appendix is the single canonical reference for the formal objects used throughout the paper. All statements are interpreted relative to a fixed pack $\Pi$, a fixed architecture template, and fixed parameters $\theta \in \Theta$ (outside a measure-zero discriminant set). We write $\mu$ for the reference measure on $\mathcal{X}$ and assume $\mu$ is absolutely continuous with respect to Lebesgue measure.

The framework accommodates a *design space* of modeling choices:

- **Type library $\mathcal{T}$:** which operations are available (Affine, ReLU, Softmax, LayerNorm, etc.) and how they are tagged (affine vs. nonaffine).
- **Primitive library $\mathcal{O}$:** how operations are realized as atomic computations.
- **Arithmetic model:** Real-RAM (exact) or IEEE-754 (fixed precision with declared rounding).
- **Conventions:** how to handle edge cases, such as whether data-dependent normalizations are tagged nonaffine (Convention B).

These choices define a *pack*, a complete specification under which all axioms and bounds are interpreted. Different packs yield different guarantees; the framework is parametric in this choice.

We develop the core concepts through a running example that is simple enough to permit explicit analysis while exhibiting structural features (affine layers, nonlinear bottlenecks, composition) that arise in larger architectures. We return to this example throughout: to illustrate active-graph construction (§B.4), to demonstrate variation bounds (§B.7.2), and to establish axiom independence (§B.7.3).

## B.1. Running Example: Two-Layer LReLU Network

**Leaky ReLU.** The Leaky ReLU with slope parameter $\alpha \in (0, 1)$ is defined coordinatewise by

$$\text{LReLU}_\alpha(z) := \max(z, \alpha z) = \begin{cases} z, & z \geq 0, \\ \alpha z, & z < 0. \end{cases}$$

The function is piecewise linear with slopes 1 (active regime) and $\alpha$ (leak regime), continuous everywhere, and differentiable except at $z = 0$. When $\alpha = 0$, this reduces to standard ReLU.

**Network definition.** The running example is the two-layer network

$$N_\theta(x) = W_2 \text{LReLU}_\alpha(W_1 x + b_1) + b_2, \tag{63}$$

where the Leaky ReLU is applied coordinatewise to the vector $W_1 x + b_1 \in \mathbb{R}^h$.

**Dimensions and parameters.** The network maps $\mathbb{R}^d \to \mathbb{R}^m$ with hidden dimension $h$. The parameter vector $\theta = (W_1, b_1, W_2, b_2)$ consists of:

- $W_1 \in \mathbb{R}^{h \times d}$: input-to-hidden weight matrix,
- $b_1 \in \mathbb{R}^h$: hidden-layer bias,
- $W_2 \in \mathbb{R}^{m \times h}$: hidden-to-output weight matrix,
- $b_2 \in \mathbb{R}^m$: output bias.

The total parameter count is $p = hd + h + mh + m = h(d + m + 1) + m$.

**Pack specification for this example.** The running example uses the following pack:

- Type library: $\mathcal{T} = \{\text{Affine}, \text{LReLU}_\alpha\}$, with Affine tagged affine and $\text{LReLU}_\alpha$ tagged nonaffine.
- Primitive library: $\mathcal{O} = \mathcal{T}$ (types and primitives coincide).
- Arithmetic: Real-RAM (exact real arithmetic).
- Conventions: standard (no normalization layers in this example).

**Structural graph $\mathcal{G}^{\text{str}}$.** The *structural graph* $\mathcal{G}^{\text{str}}$ encodes computational flow as a typed directed acyclic graph. For the running example:

$$S \text{ (input)} \rightarrow \text{Affine}_1 \rightarrow \text{LReLU}_\alpha \rightarrow \text{Affine}_2 \rightarrow \tau \text{ (output)}$$

Here $\text{Affine}_1$ computes $z \mapsto W_1 z + b_1$, $\text{LReLU}_\alpha$ applies the coordinatewise nonlinearity, and $\text{Affine}_2$ computes $z \mapsto W_2 z + b_2$. The source node $S$ represents the network input $x$, and the sink node $\tau$ represents the network output $N_\theta(x)$.

Each node carries a *type* from $\mathcal{T}$, tagged as either *affine* or *nonaffine*. In this example, $\text{Affine}_1$ and $\text{Affine}_2$ are affine and $\text{LReLU}_\alpha$ is nonaffine.

**Affine saturation and $\mathcal{G}^{\text{sat}}$ (running example).** For cut analysis we use the affine-saturated graph $\mathcal{G}^{\text{sat}}$, which removes *internal* affine-only regions by absorbing them into edges. Concretely, we contract every maximal affine-only subgraph that does *not* contain the distinguished endpoints $S$ or $\tau$. Equivalently, affine chains adjacent to endpoints are absorbed as affine *edge blocks*, while $S$ and $\tau$ remain pure source and sink nodes.

For the running example, the resulting $\mathcal{G}^{\text{sat}}$ has three nodes:

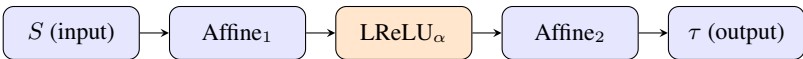

In $\mathcal{G}^{\text{sat}}$, the affine computation is carried by the edge annotations (or equivalently, by affine block metadata attached to edges), and the only nonaffine node is $\text{LReLU}_\alpha$.

**Cut structure and $\lambda$.** In $\mathcal{G}^{\text{sat}}$ there is a unique $S$–$\tau$ path: $S \xrightarrow{e_{\text{pre}}} \text{LReLU}_\alpha \xrightarrow{e_{\text{post}}} \tau$. The minimum $S$–$\tau$ edge cut has size 1, achieved by either $\{e_{\text{pre}}\}$ or $\{e_{\text{post}}\}$. Both cut edges are incident to the nonaffine node $\text{LReLU}_\alpha$. This is the simplest serial illustration of axiom (S2): within the (active) structural route from input to output, no purely affine path bypasses all nonaffine nodes.

Define the edge connectivity

$$\lambda := \text{mincut}(S, \tau; \mathcal{G}^{\text{sat}}) = 1.$$

**Explicit Jacobian and cell structure.** On any full-dimensional cell (where $(W_1 x + b_1)_i \neq 0$ for all $i$), the network is differentiable with Jacobian

$$\mathrm{D}N_\theta(x) = W_2 \, D_{s(x)} \, W_1,$$

where $D_{s(x)} = \text{diag}(s_1(x), \dots, s_h(x))$ with $s_i(x) \in \{1, \alpha\}$ determined by the sign of $(W_1 x + b_1)_i$. The input space $\mathbb{R}^d$ is partitioned into *at most* $2^h$ polyhedral cells indexed by activation patterns $s \in \{1, \alpha\}^h$ (exactly $2^h$ under generic nondegeneracy). On each cell, $N_\theta$ is affine; across cell boundaries, the Jacobian changes discontinuously.

**Why this example.** The two-layer LReLU network serves several purposes:

(i) **Explicit formulas.** The Jacobian $\mathrm{D}N_\theta(x) = W_2 D_{s(x)} W_1$ permits direct verification of local claims.
(ii) **Nontrivial cell structure.** The activation-pattern cells make the piecewise structure concrete.
(iii) **Independence witness.** Replacing a rational coefficient by a transcendental (e.g., $\pi$) yields a network satisfying (S1) and (S2) but failing (C2) under rational-coefficient primitive libraries (§B.7.3).
(iv) **Scalability of the lens.** The same saturation and cut analysis applies to larger graphs. For a purely serial MLP (no parallel bypass routes), $\mathcal{G}^{\text{sat}}$ contains a serial chain of nonaffine nodes and still has $\lambda = 1$, reflecting a single-path bottleneck rather than parallel cut structure.

**Notation summary.**

| Symbol | Meaning |
|---|---|
| $N_\theta$ | Network function $\mathbb{R}^d \to \mathbb{R}^m$ |
| $\theta = (W_1, b_1, W_2, b_2)$ | Parameters |
| $\mathrm{LReLU}_\alpha$ | Leaky ReLU with slope $\alpha \in (0,1)$ |
| $\mathcal{G}^{\mathrm{str}}$ | Structural graph (typed directed acyclic graph) |
| $\mathcal{G}^{\mathrm{sat}}$ | Affine-saturated graph for cut analysis |
| $S, \tau$ | Source (input) and sink (output) nodes |
| $\lambda$ | Edge connectivity $\mathrm{mincut}(S, \tau; \mathcal{G}^{\mathrm{sat}})$ |

*Remark* B.1 (Generality beyond LReLU). LReLU is used for concreteness and for the realizability-independence witness. All structural results (graph construction, saturation, cut analysis, (S2) certification) apply to any piecewise-$C^1$ nonlinearity definable in a tame structure, including ReLU, Swish, GELU, Softmax, and LayerNorm. Quantitative bounds (§B.7.2) require pack-specific certificates for the relevant primitives.

## B.2. Context: Packs

A *pack* is a complete specification of the context in which a network is interpreted and analyzed. It fixes (i) what operation *types* may appear in the structural graph, (ii) what *primitives* are available as atomic computations, (iii) how types are *realized* by straight-line programs over primitives (when realizability is invoked), (iv) the *arithmetic model* (Real-RAM or IEEE-754, with format and rounding mode), and (v) any *conventions* (e.g., how to tag data-dependent normalizations). All axioms, certificates, and bounds in this paper are stated *relative to a fixed pack*.

**Type library $\mathcal{T}$ (structure-level vocabulary).** The *type library* $\mathcal{T}$ is a finite set of operation types used to build structural graphs. Each type $t \in \mathcal{T}$ specifies:

(i) a *signature* consisting of finite-dimensional real vector spaces $E_{\mathrm{in},t}$ (input), $E_{\mathrm{out},t}$ (output), and a parameter space $\Xi_t$;
(ii) a *declared (ideal) function*
$$f_t^{\mathrm{ideal}} : E_{\mathrm{in},t} \times \Xi_t \to E_{\mathrm{out},t},$$
 which is the mathematical meaning of the type (before any computational realization);
(iii) an *affine tag* $\mathrm{tag}(t) \in \{\texttt{affine}, \texttt{nonaffine}\}$.

*Examples.*

- $\mathrm{Affine}_{d \to h}$: $E_{\mathrm{in}} = \mathbb{R}^d$, $E_{\mathrm{out}} = \mathbb{R}^h$, $\Xi = \mathbb{R}^{h \times d} \times \mathbb{R}^h$, $f^{\mathrm{ideal}}(x, (W, b)) = Wx + b$, $\mathrm{tag} = \texttt{affine}$.
- $\mathrm{ReLU}_h$: $E_{\mathrm{in}} = E_{\mathrm{out}} = \mathbb{R}^h$, $\Xi = \varnothing$, $f^{\mathrm{ideal}}(z) = \max(z, 0)$ coordinatewise, $\mathrm{tag} = \texttt{nonaffine}$.
- $\mathrm{LayerNorm}_h$: $E_{\mathrm{in}} = E_{\mathrm{out}} = \mathbb{R}^h$, $\Xi = \mathbb{R}^h \times \mathbb{R}^h$, $f^{\mathrm{ideal}}(z, (\gamma, \beta)) = \gamma \odot \frac{z - \mu(z)}{\sigma(z)} + \beta$. Under Convention B (data-dependent normalization tagged nonaffine), $\mathrm{tag} = \texttt{nonaffine}$.
- $\mathrm{Softmax}_n$: $E_{\mathrm{in}} = E_{\mathrm{out}} = \mathbb{R}^n$, $\Xi = \varnothing$, $f^{\mathrm{ideal}}(z)_i = \exp(z_i)/\sum_j \exp(z_j)$, $\mathrm{tag} = \texttt{nonaffine}$.

**Primitive library $\mathcal{O}$ (computation-level vocabulary).** The *primitive library* $\mathcal{O}$ is a finite set of atomic computational operations. Each primitive $\omega \in \mathcal{O}$ specifies:

(i) a signature with finite-dimensional real vector spaces $E_{\mathrm{in},\omega}$, $E_{\mathrm{out},\omega}$ and parameter space $\Xi_\omega$;
(ii) a *realized function* (execution semantics) whose interpretation depends on the declared arithmetic model.

Primitives are *atomic* in the pack description: they are treated as single operations (no unbounded internal loops, recursion, or implicit solvers). When atomicity is needed as a theorem hypothesis, it is invoked explicitly via axiom (P2).

**Type realization relation $\mathcal{R} : \mathcal{T} \Rightarrow \mathcal{O}^\star$.** A pack includes a *realization relation*
$$\mathcal{R} : \mathcal{T} \Rightarrow \mathcal{O}^\star,$$
where $\mathcal{O}^\star$ denotes finite straight-line programs (SLPs) over primitives in $\mathcal{O}$, and $\mathcal{R}(t) \subseteq \mathcal{O}^\star$ is the (possibly empty) set of SLPs declared to realize type $t$.

**Soundness of realizations.** Soundness depends on the arithmetic model:

- **Real-RAM soundness:** an SLP $\pi \in \mathcal{R}(t)$ is sound if, for all $(x, \xi) \in E_{\mathrm{in},t} \times \Xi_t$, executing $\pi$ produces exactly $f_t^{\mathrm{ideal}}(x, \xi)$ as a real vector.
- **IEEE-754 soundness:** an SLP $\pi \in \mathcal{R}(t)$ is sound if, for all representable inputs (and parameters) in the declared format and under the declared rounding/exception policy, executing $\pi$ produces exactly the same output bit-pattern as the pack's declared IEEE-754 semantics for $t$.

Realizability (axiom C2) is an *optional* hypothesis: many structural statements depend only on $\mathcal{T}$ (structure), while C2 asserts existence of sound realizations over $\mathcal{O}$.

**Parameter distributor $\mathcal{P}$.** Fix an architecture template and its structural graph $\mathcal{G}^{\mathrm{str}}$ with vertex set $V(\mathcal{G}^{\mathrm{str}})$. A network parameter $\theta \in \Theta \subseteq \mathbb{R}^p$ must be distributed to node-local parameter slices. A pack therefore includes a *parameter distributor*

$$\mathcal{P} : \Theta \longrightarrow \prod_{v \in V(\mathcal{G}^{\mathrm{str}})} \Xi_{\mathrm{type}(v)},$$

which is assumed *Borel measurable*. We write $\mathcal{P}(\theta)_v \in \Xi_{\mathrm{type}(v)}$ for the component assigned to node $v$.

Borel measurability is the minimal regularity needed for the measure-theoretic statements about $x \mapsto \mathcal{G}_x^{\mathrm{act}}$.

**Arithmetic model.** A pack declares exactly one of the following arithmetic semantics.

**Real-RAM (exact real arithmetic).** All primitive functions are interpreted as exact real maps: addition/multiplication/division (where defined) and any declared transcendentals return exact real values. There is no rounding, overflow, or underflow. Under Real-RAM, realized and ideal semantics coincide whenever $\mathcal{R}$ is sound, and the resulting network maps are piecewise-$C^1$ on cells under (TAME).

**IEEE-754 (fixed-precision arithmetic).** Fix a floating-point format $\mathbb{F}$ (e.g., FP32, BF16) and rounding mode (e.g., round-to-nearest-even). Let $\mathrm{Rep}(\mathbb{F})$ denote the set of finite representable floating-point values in that format. In an IEEE-754 pack, we assume *all runtime values* (activations and stored parameters) are represented in $\mathbb{F}$ unless the pack explicitly declares otherwise.

Thus each primitive $\omega$ is executed on representable inputs and representable parameters and returns representable outputs:

$$f_\omega^{\mathrm{real}} : \mathrm{Rep}(\mathbb{F})^{n_\omega} \times \mathrm{Rep}(\mathbb{F})^{\dim(\Xi_\omega)} \to \mathrm{Rep}(\mathbb{F})^{m_\omega},$$

with rounding performed according to the declared rounding mode.

To interpret IEEE-754 primitives as functions of *real* inputs, we use the standard *round-then-execute* convention: real inputs are first mapped coordinatewise to representables by $\mathrm{rnd}_{\mathbb{F}}$, then primitives execute on representables. Under this convention, and *excluding exceptional values unless explicitly enabled*, the resulting real-input semantics are *piecewise constant* with jumps across rounding thresholds and discrete branch boundaries. We denote the exceptional set (NaN/Inf/invalid operations, if excluded) by $E_{\mathrm{except}}$ and interpret all IEEE-754 statements on $\mathcal{X} \setminus E_{\mathrm{except}}$ unless stated otherwise.

This arithmetic choice determines downstream semantics: under Real-RAM, variation lower bounds take an integral/Jacobian form; under IEEE-754, variation lower bounds take an event/jump form.

**Structure vs. computation separation principle.** A central design principle is the *separation of structure and computation*:

- **Structure:** properties of the typed structural graph $\mathcal{G}^{\mathrm{str}}$, its affine saturation $\mathcal{G}^{\mathrm{sat}}$, and (where applicable) activity and min-cut quantities derived from the declared type semantics. These are governed by axioms such as (S1) and (S2).
- **Computation:** whether the declared type semantics are realized by sound SLPs over $\mathcal{O}$ under the chosen arithmetic model. This is governed by axiom (C2).

This separation has three consequences used throughout the paper:

(i) *Structural properties can be verified cheaply.* Axiom (S2) is a graph-cut condition on $\mathcal{G}^{\mathrm{sat}}$ (or on the active subgraph) and can be checked by max-flow/min-cut algorithms.
(ii) *Some guarantees do not require realizability.* Several lower-bound statements depend on (S1), (S2), and quantitative certificates, but not on (C2), so they remain meaningful even when ideal semantics cannot be realized by the declared primitive library.
(iii) *IEEE-754 requires arithmetic-specific certificates.* Under IEEE-754, structural activity can over-approximate influential paths because rounding may suppress sensitivity. Quantitative lower bounds therefore invoke IEEE-754-specific event/tail certificates (E4-quant).

**Pack summary.** A *pack* $\Pi$ consists of:

(i) a type library $\mathcal{T}$ with signatures, ideal functions $f_t^{\mathrm{ideal}}$, and affine tags;
(ii) a primitive library $\mathcal{O}$ with signatures and realized semantics $f_\omega^{\mathrm{real}}$;
(iii) a realization relation $\mathcal{R} : \mathcal{T} \Rightarrow \mathcal{O}^\star$ (possibly empty for some types);
(iv) an arithmetic model: Real-RAM, or IEEE-754 with declared format, rounding mode, and exception policy;
(v) conventions (e.g., Convention B for normalization tagging);
(vi) a Borel parameter distributor $\mathcal{P}$ for the fixed architecture template.

All statements in this framework are relative to a fixed pack:

"Under pack $\Pi$, network $N_\theta$ satisfies property $P$."

*Remark* B.2 (Default pack). Unless otherwise stated, examples use the *default pack*: matching libraries $\mathcal{T} = \mathcal{O}$, a sound canonical realization for each type, Real-RAM arithmetic, and Convention B. Under the default pack, (C2) holds trivially and structural and computational semantics coincide.

## B.3. Ambient Spaces, Conventions, and Tameness

This section fixes the ambient geometric and measure-theoretic conventions used throughout, and states the **TAME** postulate as a *pack-level context assumption*. The role of TAME is not to prove expressiveness, but to guarantee that the structural objects we manipulate (active graphs, cells, boundary events) are well-behaved: measurable, locally constant off a negligible set, and stratifiable into $C^1$ regions.

**Ambient input/output spaces.** We work with an input domain $\mathcal{X} \subseteq \mathbb{R}^d$ and output space $\mathcal{Y} \subseteq \mathbb{R}^m$, both equipped with the standard Euclidean structure. We write $\langle \cdot, \cdot \rangle$ for the Euclidean inner product and $\| \cdot \|$ for the induced norm. For matrices and linear maps, $\| \cdot \|_{\mathrm{op}}$ denotes the operator norm induced by $\| \cdot \|$.

A network (under fixed parameters $\theta$ and a fixed pack $\Pi$) is a measurable map

$$N_\theta : \mathcal{X} \to \mathcal{Y},$$

interpreted either as an *ideal* Real-RAM function or as a *realized* IEEE-754 execution semantics, depending on the pack (cf. §2.1).

**Reference measure and "a.e." conventions.** We fix a Borel probability measure $\mu$ on $\mathcal{X}$ satisfying

$$\mu \ll \mathrm{Leb}_{\mathbb{R}^d}.$$

All statements of the form "for almost every input" ("$\mu$-a.e. $x$") are understood with respect to this $\mu$, unless explicitly stated otherwise. When we require Lebesgue measure directly (rarely), we say "Lebesgue-a.e." explicitly.

*Why $\mu \ll \mathrm{Leb}$?* This mild assumption is the minimal condition under which lower-dimensional boundary sets (e.g. stratification boundaries, tie-sets, kink sets) are automatically $\mu$-null, which keeps the theory honest about "generic input" statements.

**Rays and perturbation geometry.** For $x \in \mathcal{X}$ and a direction $v \in \mathbb{S}^{d-1}$, a (truncated) ray of length $r > 0$ is the path

$$\gamma_{x,v,r}(t) := x + tv, \qquad t \in [0, r],$$

restricted to those $t$ for which $\gamma_{x,v,r}(t) \in \mathcal{X}$. When we say "for almost every direction" we mean with respect to the uniform (Haar) probability measure on $\mathbb{S}^{d-1}$.

We write $B_\varepsilon(x)$ for the closed Euclidean ball of radius $\varepsilon$ around $x$.

**Quantifier standard (inputs, directions, and local regions).** Many statements in this paper come in three compatible flavors. Unless stated otherwise, interpret them as follows:

(i) $\mu$**-a.e. input:** property holds for all $x \in \mathcal{X} \setminus E$ where $\mu(E) = 0$.
(ii) **Local (cellwise):** property holds for all $x$ in a neighborhood on which the discrete execution structure is constant.
(iii) **Raywise:** property holds for $\mu$-a.e. basepoint $x$ and for almost every direction $v$ (and all sufficiently small $r$), except possibly when the ray intersects a boundary set.

TAME is the mechanism that makes these three regimes consistent: it supplies the cells and guarantees the boundary sets are negligible.

**Parameter-dependent exceptional sets.** Several objects (active graphs, boundary events, and "kink sets") depend on parameters $\theta$. We separate two kinds of exceptional sets:

*(1) Input exceptional set $D_\theta$.* For each fixed $\theta$, we allow an input set $D_\theta \subseteq \mathcal{X}$ on which the structural description is ill-posed or discontinuous in a way that is not intended to be certified (e.g. ties in argmax, exact zero pre-activations, division-by-zero conditions, or other "boundary" events). Our standing convention is:

$$\text{all cellwise and "locally constant" statements are interpreted on } \mathcal{X} \setminus D_\theta.$$

Under TAME, $D_\theta$ is $\mu$-null and can be taken to be a finite (or locally finite) union of codimension-$\geq 1$ strata.

*(2) Parameter exceptional set $\Theta_{\mathrm{bad}}$.* Some statements are intended to hold for *generic* parameters (e.g. avoiding exact degeneracies). When needed, we quantify:

$$\theta \in \Theta \setminus \Theta_{\mathrm{bad}}, \qquad \text{with } \Theta_{\mathrm{bad}} \text{ measurable and negligible under the intended parameter law.}$$

We never hide this: any dependence on excluding $\Theta_{\mathrm{bad}}$ is stated explicitly.

## $C^1$ **cells and refined cells.** Fix a pack $\Pi$ and parameters $\theta$.

$C^1$ *cell.* A set $C \subseteq \mathcal{X}$ is called a $C^1$ *cell (for $N_\theta$ under $\Pi$)* if:

(i) $C$ is nonempty, open, and connected;

(ii) $N_\theta$ is $C^1$ on $C$ (as a map between Euclidean spaces);

(iii) all *discrete structural predicates* that can change the execution structure (e.g. sign tests, comparisons, argmax ties, exceptional-value guards, routing choices) are constant on $C$.

*Refined cell.* A *refined cell* is a maximal $C^1$ cell contained in $\mathcal{X} \setminus D_\theta$. We write $\mathcal{C}_\theta$ for the collection of refined cells. By maximality, refined cells form a partition of $\mathcal{X} \setminus D_\theta$. Intuitively: refined cells are the regions on which both the calculus (Jacobian) and the discrete structure (active graph) stabilize.

This definition is deliberately semantic: it does not presuppose a particular architecture. TAME ensures refined cells exist and have the local finiteness properties needed by the rest of the paper.

**Postulate (TAME): pack-declared tameness and stratification.** TAME is a pack postulate, not a theorem. Each pack $\Pi$ declares a "tame" regularity regime under which the type functions (and therefore composed networks) admit well-behaved stratifications.

Concretely, we assume:

- **(T1) Definability / tame context.** The pack fixes a tame geometric context (e.g., an o-minimal expansion of $\mathbb{R}$ such as $\mathbb{R}_{\mathrm{an,exp}}$), and every declared type function $f_\tau^{\mathrm{ideal}}$ is definable in that context.

- **(T2) Piecewise-$C^1$ stratification.** For every architecture template and every $\theta \in \Theta$, there exists a (finite or countable) collection of refined cells $\mathcal{C}_\theta$ covering $\mathcal{X} \setminus D_\theta$ such that $N_\theta$ is $C^1$ on each $C \in \mathcal{C}_\theta$.

- **(T3) Locally finite complexity.** The refined-cell family is *locally finite*: every compact $K \subseteq \mathcal{X}$ intersects only finitely many refined cells. Equivalently, boundary crossings along a compact ray segment cannot accumulate.

- **(T4) Negligible boundary set.** The exceptional set $D_\theta$ can be chosen measurable with $\mu(D_\theta) = 0$, and the union of refined-cell boundaries is $\mu$-null.

*Common choice:* $\mathbb{R}_{\mathrm{an,exp}}$. The structure $\mathbb{R}_{\mathrm{an,exp}}$ (the real field expanded with restricted analytic functions and exp) is o-minimal and covers polynomials, exp, log, and their compositions. This suffices for ReLU, LReLU, Swish/SiLU, GELU (analytic approximation $x\sigma(1.702x)$), Softmax, and LayerNorm. For exact GELU (involving erf), one may declare a larger o-minimal expansion in which erf is definable, or use the analytic approximation as the pack's declared ideal function.

Items (T2)–(T4) are exactly what we need to justify measurability and local constancy claims for the active graph and related structural objects in later sections.

**How TAME is used (and how it is *not* used).** TAME is invoked only to justify regularity of the *structural* pipeline: measurability of $x \mapsto \mathcal{G}_x^{\mathrm{act}}$, existence of refined cells, and the fact that "boundary events" occur on a $\mu$-null set. TAME is *not* used to argue that networks are expressive, trainable, or generalize.

In particular:

- Under Real-RAM packs, TAME allows us to replace "almost everywhere" reasoning by cellwise $C^1$ reasoning.
- Under IEEE-754 packs, TAME organizes the boundary sets that generate discrete events, even though the realized map is piecewise constant.
- The axiom-closure bookkeeping in §B.8 records every place where TAME is required.

*Remark* B.3 (Practical reading rule). If you do not care about measure theory, you can read TAME as: "*away from a negligible set of boundary inputs, the network has a stable execution structure and a well-defined Jacobian.*" All results that depend on this stability explicitly list TAME in their axiom closure.

## B.4. Graph and Active-Graph Semantics

This section formalizes the graph objects that carry the *structural* content of the framework: the typed structural graph $\mathcal{G}^{\mathrm{str}}$, the affine-saturated graph $\mathcal{G}^{\mathrm{sat}}$, and the input-dependent active graph $\mathcal{G}_x^{\mathrm{act}}$. These objects are defined relative to a fixed architecture template and a fixed pack $\Pi$ (§2.1) and are the basis for axiom (S2) and the min-cut quantity $\lambda_x$.

Throughout, fix parameters $\theta \in \Theta$ and a pack $\Pi$. All measurability and local-constancy statements invoked by these definitions rely on TAME (§B.3).

STRUCTURAL GRAPHS: $\mathcal{G}^{\mathrm{str}}$ AND $\mathcal{G}^{\mathrm{sat}}$

**Typed structural graph $\mathcal{G}^{\mathrm{str}}$.** An *architecture template* specifies a finite directed acyclic graph (DAG)

$$\mathcal{G}^{\mathrm{str}} = (V, E; S, \mathcal{T}_{\mathrm{sink}}, \mathrm{type}),$$

consisting of:

- a vertex set $V$ and directed edge set $E \subseteq V \times V$;
- a designated source node $S \in V$ representing the input interface;
- a nonempty set of sink nodes $\mathcal{T}_{\text{sink}} \subseteq V$ representing declared outputs;
- a type map $\text{type} : V \to \mathcal{T}$ into the pack's type library.

Each vertex $v \in V$ denotes an operation of type $\text{type}(v)$, interpreted by the pack-declared ideal function $f^{\text{ideal}}_{\text{type}(v)}$ (§2.1). The parameter distributor $\mathcal{P}$ assigns the appropriate parameter slice to each vertex.

*Remark* B.4 (One source, many sinks). We use a single source node $S$ for exposition. Multiple input tensors (e.g. token IDs, masks, positional encodings) can be handled by introducing a formal affine source-aggregator node feeding into the rest of the graph.

**Affine saturation and the affine-saturated graph** $\mathcal{G}^{\text{sat}}$. The structural graph $\mathcal{G}^{\text{str}}$ contains affine and nonaffine typed vertices (per the pack's affine tags). Since affine nodes do not create nonlinear bottlenecks, we quotient them out.

Let $V_{\text{aff}} \subseteq V$ be the set of vertices tagged $\texttt{affine}$ and $V_{\text{non}} \subseteq V$ those tagged $\texttt{nonaffine}$. Consider the induced subgraph on $V_{\text{aff}}$, viewed as an undirected graph for connectivity. Let $\mathcal{B}$ be the set of its connected components ("affine blocks").

The *affine-saturated graph* is the directed graph

$$\mathcal{G}^{\text{sat}} = (V^{\text{sat}}, E^{\text{sat}}; S^{\text{sat}}, \mathcal{T}^{\text{sat}}_{\text{sink}}, \text{type}^{\text{sat}}),$$

constructed as follows:

- **Vertices.** $V^{\text{sat}}$ contains: (i) one vertex for each affine block $B \in \mathcal{B}$, (ii) one vertex for each nonaffine vertex $v \in V_{\text{non}}$.
- **Edges.** For any edge $(u \to v) \in E$ with representatives $\bar{u}, \bar{v} \in V^{\text{sat}}$ (either affine blocks or nonaffine nodes), include a directed edge $(\bar{u} \to \bar{v})$ in $E^{\text{sat}}$ whenever $\bar{u} \neq \bar{v}$. Parallel edges may be kept (multigraph) or merged (simple graph); all min-cut statements assume a standard capacity assignment that is invariant to the chosen representation.
- **Types.** Nonaffine vertices keep their original type. Affine block vertices are tagged $\texttt{affine}$ and treated as "transport blocks" (their internal structure may be recorded as edge annotations for quantitative constants later).
- **Source/sinks.** The source $S^{\text{sat}}$ is the affine block containing the source in $\mathcal{G}^{\text{str}}$. The sink set $\mathcal{T}^{\text{sat}}_{\text{sink}}$ is the image of $\mathcal{T}_{\text{sink}}$ under the block/nonaffine projection map.

Since $\mathcal{G}^{\text{str}}$ is acyclic and affine blocks are contracted along edge directions (never creating new paths), $\mathcal{G}^{\text{sat}}$ inherits acyclicity and remains a DAG.

**Interpretation.** $\mathcal{G}^{\text{sat}}$ is the minimal graph that preserves where nonaffine bottlenecks can occur: every $S$–sink route in $\mathcal{G}^{\text{str}}$ projects to an $S^{\text{sat}}$–sink route in $\mathcal{G}^{\text{sat}}$, and every route in $\mathcal{G}^{\text{sat}}$ lifts to at least one route in $\mathcal{G}^{\text{str}}$.

ACTIVE-GRAPH SEMANTICS: $\mathcal{G}^{\text{act}}_x$

**Why an input-dependent graph?** Many networks have data-dependent routing or activity (ReLU regimes, attention patterns, max-pooling selections, MoE routing). Even when the architecture template is fixed, the *effective* computation on a given input $x$ activates only a subgraph. We record this with an input-dependent subgraph $\mathcal{G}^{\text{act}}_x \subseteq \mathcal{G}^{\text{sat}}$.

**Local edge influence.** Fix $x \in \mathcal{X} \setminus D_\theta$. For each directed edge $e = (u \to v)$ in $\mathcal{G}^{\text{sat}}$, the pack supplies a notion of whether $e$ is *locally influential at* $x$. We write this predicate as

$$\text{Infl}^{\text{loc}}(e; x) \in \{0, 1\}.$$

Conceptually, $\text{Infl}^{\text{loc}}(e; x) = 1$ means: the local Jacobian contribution at $e$ is nonzero, or (for discrete routing) the edge is selected by the routing mechanism at $x$. The precise realization of $\text{Infl}^{\text{loc}}$ is pack- and architecture-dependent (e.g. nonzero Jacobian block, nondegenerate routing weight, non-tie argmax, nonzero gate, etc.). TAME ensures that $\text{Infl}^{\text{loc}}(\cdot; x)$ is locally constant on refined cells.

*Remark* B.5 (Structural vs realized influence). In experiments we often use a realized-branch proxy $\widehat{\mathcal{G}^{\text{act}}}_x$ computed from a single forward/backward pass. The object $\mathcal{G}^{\text{act}}_x$ here is the *structural* (ideal-semantics) active graph used in the axioms.

**Definition: active graph** $\mathcal{G}^{\text{act}}_x$ **(two-step construction).** The *active graph* at input $x$ is the directed subgraph

$$\mathcal{G}^{\text{act}}_x := (V_x, E_x) \subseteq \mathcal{G}^{\text{sat}}$$

constructed in two steps:

*Step 1 (local influence).* Define the locally-influential edge set

$$E^{\text{loc}}_x := \{ e \in E^{\text{sat}} : \text{Infl}^{\text{loc}}(e; x) = 1 \}.$$

*Step 2 (path closure).* Define $E_x$ as the subset of $E^{\text{loc}}_x$ consisting of edges that lie on at least one directed $S^{\text{sat}}$–sink path using only edges in $E^{\text{loc}}_x$:

$$E_x := \{ e \in E^{\text{loc}}_x : \exists \, \text{path } S^{\text{sat}} \to \cdots \to \tau \text{ in } (V^{\text{sat}}, E^{\text{loc}}_x) \text{ passing through } e, \ \tau \in \mathcal{T}^{\text{sat}}_{\text{sink}} \}.$$

Define the active vertex set as

$$V_x := \{ v \in V^{\text{sat}} : v \text{ is incident to some } e \in E_x \} \cup \{S^{\text{sat}}\}.$$

**Interpretation of path closure.**   Step 2 ensures that $\mathcal{G}_x^{\mathrm{act}}$ is exactly the union of active source-to-sink routes. Edges that are locally influential but "dead-end" (not on any complete route) are excluded. This path-closed structure is what the min-cut analysis requires: $\lambda_x$ counts bottlenecks on routes that actually reach an output.

**Active sinks.**   Not every declared sink in $\mathcal{G}^{\mathrm{sat}}$ is necessarily reachable in $\mathcal{G}_x^{\mathrm{act}}$. We therefore define the active sink set

$$\mathcal{T}_{\mathrm{act}}(x) := \left\{ \tau \in \mathcal{T}_{\mathrm{sink}}^{\mathrm{sat}} : \tau \text{ is reachable from } S^{\mathrm{sat}} \text{ in } \mathcal{G}_x^{\mathrm{act}} \right\}.$$

Only sinks in $\mathcal{T}_{\mathrm{act}}(x)$ participate in $\lambda_x$.

**Influential vertices.**   A vertex $v \in V^{\mathrm{sat}}$ is *influential at* $x$ if it lies on some $S^{\mathrm{sat}}$–$\tau$ directed path in $\mathcal{G}_x^{\mathrm{act}}$ for some $\tau \in \mathcal{T}_{\mathrm{act}}(x)$. Equivalently, $v$ is reachable from $S^{\mathrm{sat}}$ and can reach some active sink.

This notion is used later to define "where certificates must attach": only influential nonaffine vertices need directional-nonlinearity and tail certificates.

CUT CONNECTIVITY AND $\lambda_x$

**Edge-cut value.**   Given a directed graph $G$ with source $S$ and sink $\tau$, $\mathrm{mincut}(S, \tau; G)$ denotes the minimum cardinality of an $S$–$\tau$ *edge cut*, i.e. the smallest number of edges whose removal disconnects all directed $S$–$\tau$ paths. (Equivalently via max-flow/min-cut with unit capacities.)

**Definition: $\lambda_x$.**   For an input $x$ with nonempty active sink set, define the *active connectivity*

$$\lambda_x := \min_{\tau \in \mathcal{T}_{\mathrm{act}}(x)} \mathrm{mincut}(S^{\mathrm{sat}}, \tau; \mathcal{G}_x^{\mathrm{act}}).$$

If $\mathcal{T}_{\mathrm{act}}(x) = \varnothing$, we set $\lambda_x := 0$ by convention (the network has no active sink reachable from the source).

**Interpretation.**   $\lambda_x$ measures the minimum number of edges that must be crossed to separate the input from any active output. Under axiom (S2), every minimum cut contains at least one edge incident to a nonaffine vertex, so $\lambda_x$ counts the number of *independent nonaffine bottlenecks* that information must traverse on input $x$.

SUFFIX NOTATION AND PATHWISE JACOBIAN BLOCKS (USED LATER)

The remainder of the paper attaches quantitative certificates to *paths* and *suffixes* of paths. We introduce minimal notation here so later sections can state bounds cleanly.

**Paths and suffixes.**   A directed path in $\mathcal{G}^{\mathrm{sat}}$ (or $\mathcal{G}_x^{\mathrm{act}}$) is a sequence

$$P : \quad v_0 \to v_1 \to \cdots \to v_\ell,$$

with edges $e_i = (v_{i-1} \to v_i)$. For an index $i \in \{0, \dots, \ell\}$, define the *suffix path*

$$P_{\succeq v_i} : \quad v_i \to v_{i+1} \to \cdots \to v_\ell.$$

When the distinguished suffix node is clear, we also write $P_{\succeq i}$.

**Suffix operator.**   For a path suffix $Q = P_{\succeq v}$, we write $T_Q(x)$ for the (ideal) composed map implemented by the suffix when evaluated at input $x$ (with parameters distributed by $\mathcal{P}(\theta)$). Under Real-RAM, $T_Q$ is $C^1$ on each refined cell and has a Jacobian $\mathrm{D}T_Q(x)$.

**Pathwise Jacobian blocks.**   For an edge $e$ on a path $P$, the framework uses a pathwise Jacobian contribution denoted $J_e^{(P)}(x)$. Intuitively, $J_e^{(P)}(x)$ is the Jacobian of the *suffix map after* $e$ composed with the local Jacobian contribution at $e$, i.e. a block of the chain rule along the chosen path $P$. We keep the definition abstract here because the precise block structure depends on how affine blocks are represented in $\mathcal{G}^{\mathrm{sat}}$; later sections instantiate $J_e^{(P)}$ under each tail variant (E4-$\sigma$, E4-dir, E4-quant).

**Sum convention.**   When summing along a path, $\sum_{e \in P}$ always denotes a sum over the directed edges of $P$ in traversal order. If $P$ is a multigraph path with repeated edges, edges are counted with multiplicity.

*Remark* B.6 (Why introduce suffix notation this early?).   The cut-based structure is purely combinatorial, but the variation bounds are ultimately quantitative and attach certificates to suffix maps and pathwise Jacobian factors. Introducing the suffix and pathwise notation here keeps later theorem statements uncluttered, while preserving the separation: *graphs first, constants later*.

The main paper states axiom names and informal meanings; this appendix provides the full formal statements, the satisfaction semantics, and the feasible region objects used by theorem closures and proofs.

## B.5. Axioms

The framework is organized as a *layered axiom stack*. At the bottom are *structural* axioms (S1–S2) that depend only on the typed graph semantics of $\mathcal{G}^{\mathrm{str}}$, its affine saturation $\mathcal{G}^{\mathrm{sat}}$, and the active graph $\mathcal{G}_x^{\mathrm{act}}$ (§B.4). Above structure sits a single *computational* axiom (C2) that connects the ideal semantics of types to executable straight-line programs over a primitive library. Later sections add primitive axioms (P1–P5), event axioms (E1–E4), and merge isolation (M), which together supply the quantitative constants needed for certifiable variation lower bounds. Structural statements invoke the TAME postulate (§B.3) where needed.

Throughout, all axioms are interpreted relative to a fixed pack $\Pi$ (§2.1), which declares the type library $\mathcal{T}$, primitive library $\mathcal{O}$, arithmetic model (Real-RAM or IEEE-754), conventions (e.g. Convention B), and parameter distributor $\mathcal{P}$. When an axiom depends on additional pack properties (notably TAME), we state that dependence explicitly.

### B.5.1. STRUCTURAL AXIOMS

The structural axioms are statements about the *ideal* typed computation graph: they speak about $\mathcal{G}^{\mathrm{str}}$, $\mathcal{G}^{\mathrm{sat}}$, and (when invoked) $\mathcal{G}_x^{\mathrm{act}}$ as defined by the pack-declared *ideal type functions* $f_\tau^{\mathrm{ideal}}$ and the affine/nonaffine tags (§2.1, §B.4). They do *not* assume that the network is executable over a primitive library.

**(S1) Structural well-posedness.**  Fix a pack $\Pi$ and an architecture template $\mathcal{G}^{\mathrm{str}}$. For each parameter vector $\theta \in \Theta$, evaluation of $\mathcal{G}^{\mathrm{str}}$ using the pack-declared ideal type functions $f_{\mathrm{type}(v)}^{\mathrm{ideal}}$ and the parameter distributor $\mathcal{P}(\theta)$ induces a map

$$N_\theta : \mathcal{X} \to \mathcal{Y}.$$

Axiom (S1) requires that this induced map is a *well-defined Borel function* on the intended input space:

> **(S1)** For every $\theta \in \Theta$, the induced network map $N_\theta : \mathcal{X} \to \mathcal{Y}$ is well-defined (total on $\mathcal{X}$) and Borel measurable (with respect to the ambient Borel $\sigma$-algebras on $\mathcal{X}, \mathcal{Y}$).

In particular, (S1) rules out undefined behavior at the level of ideal semantics (e.g. division by zero, invalid normalization, or exceptional values) unless the pack explicitly builds such behavior into $\mathcal{X}$ and $\mathcal{Y}$.

*Remark* B.7 (Why (S1) is "structural").  (S1) is discharged by mild closure conditions on the type library and parameter distributor: if each ideal type function $f_\tau^{\mathrm{ideal}}(\cdot, \xi)$ is Borel measurable in its input and $\mathcal{P}$ is Borel measurable, then acyclic graph composition yields a Borel $N_\theta$. In the layered development, we state (S1) as a top-level structural axiom and later discharge it under primitive closure axioms (P1–P4) for standard packs.

*Remark* B.8 (Partial-domain semantics is a planned relaxation).  Many deployed systems are *not* total functions of $\mathbb{R}^d$ because of NaN/Inf propagation, overflow, or domain restrictions. In the mainline development we keep (S1) as a total-function axiom for clarity. A principled relaxation to partial-domain semantics is treated as whitespace

**(S2) Active-cut essential nonlinearity.**  Axiom (S2) formalizes the claim that *there is no affine-only bypass route* from input to any active output. It is the structural condition that makes "nonlinear bottlenecks" a graph-theoretic object.

Recall from §B.4 that for each input $x$ we have an active subgraph $\mathcal{G}_x^{\mathrm{act}} \subseteq \mathcal{G}^{\mathrm{sat}}$ and an active sink set $\mathcal{T}_{\mathrm{act}}(x)$. An edge $e = (u \to v)$ is called *nonaffine-incident* if $u$ or $v$ is a node tagged `nonaffine` by the pack.

> **(S2)** For every input $x$ with $\mathcal{T}_{\mathrm{act}}(x) \neq \varnothing$ and every active sink $\tau \in \mathcal{T}_{\mathrm{act}}(x)$, every minimum $S^{\mathrm{sat}}$–$\tau$ edge cut in $\mathcal{G}_x^{\mathrm{act}}$ contains at least one nonaffine-incident edge. Equivalently: no minimum cut separating $S^{\mathrm{sat}}$ from $\tau$ can be realized using only edges whose endpoints are both affine-tagged.

Intuitively, (S2) says: once affine regions have been saturated, *nonaffine computation is unavoidable* on every active route from input to output. This is exactly the condition under which the min-cut connectivity $\lambda_x$ becomes a meaningful "bottleneck count."

*Remark* B.9 (Equivalent formulations).  Because $\mathcal{G}^{\mathrm{sat}}$ contracts affine-only regions, (S2) can be read in several equivalent ways: (i) every directed $S^{\mathrm{sat}}$–$\tau$ path in $\mathcal{G}_x^{\mathrm{act}}$ passes through at least one nonaffine node; (ii) the set of nonaffine nodes intersects every active source-to-sink route; (iii) in unit-capacity max-flow form, every min-cut has a witness edge adjacent to a nonaffine node. We use the min-cut formulation because it is directly checkable by max-flow algorithms and aligns with the later lower-bound proofs.

*Remark* B.10 (Conventions matter (and this is a feature, not a bug)).  Whether a node is tagged `affine` or `nonaffine` is pack-dependent. For example, under Convention B (data-dependent normalization tagged nonaffine), pre-norm transformers exhibit a $\lambda_x = 1$ bottleneck at normalization; under Convention A (normalization treated as affine), affine bypasses can appear and (S2) can fail. The goal is not to enforce a universal tagging, but to make the modeling choice explicit.

**Verification notes for (S1) and (S2).**

- **(S1) in practice.** Under standard packs, (S1) is discharged once the ideal type functions are measurable and the graph is acyclic. In safety-critical settings, (S1) is the place where one must declare how exceptional values are handled.

- **(S2) in practice.** Given $\mathcal{G}^{\mathrm{sat}}$ and an input $x$, one computes $\mathcal{G}^{\mathrm{act}}_x$ (or a realized-branch proxy $\widehat{\mathcal{G}^{\mathrm{act}}}_x$) and then computes $\lambda_x$ via max-flow/min-cut. A single cut certificate is a list of cut edges together with a check that at least one is nonaffine-incident. In experiments we treat proxy violations as diagnostic signals, not formal refutations of (S2).
- **TAME dependency (scope of the dependency).** The *definition* of $\mathcal{G}^{\mathrm{act}}_x$ is purely structural once the pack supplies an influence predicate (built from the ideal semantics). However, the measurability and local-constancy properties that make $\mathcal{G}^{\mathrm{act}}_x$ a robust analytical object—and that are used in later measure-theoretic arguments—rely on the pack satisfying TAME (§B.3).

### B.5.2. COMPUTATIONAL AXIOM

The structural axioms speak about the *ideal* network map $N_\theta$ defined by ideal type functions. The computational axiom connects this ideal object to executable code over a primitive library with *realized primitive functions* $f^{\mathrm{real}}_\omega$ under the pack's declared arithmetic.

**(C2) Exact finite realizability over primitives.** Fix a pack $\Pi$ with primitive library $\mathcal{O}$ and declared arithmetic model. A *straight-line program* (SLP) over $\mathcal{O}$ is a finite acyclic computation DAG whose internal nodes are labeled by primitives $\omega \in \mathcal{O}$ and whose edges carry intermediate values. The program may use constants derived from $\theta$ (via $\mathcal{P}$ and fixed encodings) as literal inputs.

> **(C2)** For every $\theta \in \Theta$, there exists a finite acyclic SLP $P_\theta$ over the primitive library $\mathcal{O}$ such that the realized function $\widehat{N}_\theta$ computed by $P_\theta$ agrees with the ideal network function $N_\theta$ under the pack's arithmetic semantics:
>
> - *Real-RAM:* $\widehat{N}_\theta(x) = N_\theta(x)$ for all $x \in \mathcal{X}$.
> - *IEEE-754:* for all representable inputs $x$ in the declared floating-point format (i.e. $x \in \mathcal{X} \cap \mathbb{F}^d$), $\widehat{N}_\theta(x)$ and $N_\theta(x)$ produce identical output bit-vectors in the declared output format.

(C2) is a *finiteness* and *exactness* condition: the implementation must be loop-free (no internal iteration) and must match the declared semantics exactly (no approximation error).

*Remark* B.11 (Why we do not state a separate "soundness" axiom). Older presentations separate "the primitives are implemented correctly" from "the network is realizable." Here correctness is folded into the meaning of "realized function" under the pack's declared arithmetic model, and (C2) is the single point where exact implementability is asserted.

*Remark* B.12 (When (C2) holds trivially). Under the *matching-library* default pack $\mathcal{O} = \mathcal{T}$, each type is treated as an atomic primitive and every network is realizable by construction, so (C2) holds automatically. The point of (C2) is to allow *restricted* primitive libraries in which realizability becomes a nontrivial claim.

*Remark* B.13 (How (C2) can fail). (C2) can fail for common and meaningful reasons, including:

- **Arithmetic mismatch:** the ideal type semantics requires real arithmetic but the primitive library only provides restricted-number operations (e.g. rationals, fixed-point), so some parameter choices (irrational coefficients) are unrealizable.
- **Missing primitives:** the type library includes operations whose exact realization is unavailable (e.g. exact transcendental functions in a restricted $\mathcal{O}$).
- **Non-atomic kernels:** the deployed implementation uses internal iteration (normalization iterations, inverse-square-root approximations, early-exit attention), violating SLP atomicity.
- **Approximation by design:** the implementation intentionally approximates an ideal type function $f^{\mathrm{ideal}}_\tau$ (polynomial softmax, truncated series), yielding a $(\mathrm{C2}_\varepsilon)$ world rather than (C2).

(e.g. ledgered approximate realizability and bounded-iteration primitives).

**Structural vs computational, again.** The framework is designed so that *structure can be certified even when computation cannot*. In particular, the graph-cut lens (S2 and $\lambda_x$) is meaningful for ideal semantics regardless of realizability. (C2) is the bridge to deployed artifacts: it is required only when one wants guarantees to apply to a concrete implementation over a specified primitive library and arithmetic model.

### B.5.3. PRIMITIVE LIBRARY POSTULATES (P1–P5)

This subsection states the postulates imposed on the *primitive library* $\mathcal{O}$ of a pack $\Pi$ (§2.1). Postulates (P1)–(P4) are *regularity and well-formedness* conditions ensuring that primitive evaluation composes cleanly into a measurable network map. Postulate $(\mathrm{P5}_{\mathrm{dir}})$ is the first *quantitative* postulate: it supplies a certificate that a designated nonaffine primitive contributes nontrivial one-dimensional variation along some direction.

Throughout, primitives are interpreted under the pack's declared arithmetic model. We write the realized primitive function as

$$f^{\mathrm{real}}_\omega \;:\; E_{\mathrm{in},\omega} \times \Xi_\omega \longrightarrow E_{\mathrm{out},\omega}, \qquad \omega \in \mathcal{O},$$

where $\Xi_\omega$ is the primitive's parameter space. When $\mathcal{O} = \mathcal{T}$ (matching-library pack), one may read these postulates as applying to the ideal type functions $f^{\mathrm{ideal}}_\tau$ as well; the distinction matters only in restricted packs.

**(P1) Parameter freeze (no data-dependent weights).** A network instance is determined by an architecture template $\mathcal{G}^{\mathrm{str}}$ and a parameter vector $\theta \in \Theta$. The pack supplies a parameter distributor $\mathcal{P}$ (§2.1),

$$\mathcal{P} : \Theta \longrightarrow \prod_{v \in V(\mathcal{G}^{\mathrm{str}})} \Xi_{\mathrm{type}(v)}.$$

Postulate (P1) asserts that node parameters are *frozen* by $\theta$ and do not depend on the input $x$:

> **(P1)** The parameter distributor $\mathcal{P}$ depends only on $\theta$ (not on $x$), and evaluation treats $\mathcal{P}(\theta)_v$ as a constant at node $v$. Equivalently, primitives are invoked as $f_\omega^{\mathrm{real}}(\cdot\,;\xi)$ with $\xi$ fixed for the entire evaluation.

(P1) rules out hypernetwork semantics in which weights are generated on-the-fly as functions of $x$ (unless such generation is explicitly modeled as part of $\mathcal{G}^{\mathrm{str}}$ with its own primitives).

*Remark* B.14 (Why (P1) matters). (P1) ensures that variation and derivatives are taken with respect to input only: there are no hidden $x$-dependent parameter pathways that would add extra chain-rule terms or create affine-only bypass routes through dynamically generated weights.

**(P2) Atomicity (no hidden iteration inside a primitive).** Postulate (P2) is the *no-hidden-loop* assumption:

> **(P2)** Each primitive $\omega \in \mathcal{O}$ is *atomic*: it is evaluated in a single step under the pack's arithmetic semantics and contains no internal iteration, recursion, implicit solvers, or data-dependent unrolling.

(P2) is the primitive-level analog of (C2)'s straight-line-program requirement. It is satisfied by standard kernels when treated as atomic ops, and it is violated by implementations whose mathematical meaning is defined by convergence (Newton iterations, normalization iterations, early-exit attention, etc.). Such cases are treated as explicit relaxations (whitespace: bounded-iteration primitives).

**(P3) Finite-dimensional Euclidean slots.** The framework reasons about operator norms, directional derivatives, and variation along rays. We therefore require that all primitive interfaces live in finite-dimensional Euclidean spaces:

> **(P3)** For every primitive $\omega \in \mathcal{O}$, the input space $E_{\mathrm{in},\omega}$, output space $E_{\mathrm{out},\omega}$, and parameter space $\Xi_\omega$ are finite-dimensional real vector spaces, identified with $\mathbb{R}^{d_\omega}$, $\mathbb{R}^{m_\omega}$, and $\mathbb{R}^{p_\omega}$ equipped with their Borel $\sigma$-algebras and Euclidean norms.

Unless stated otherwise, "almost every" on these spaces means Lebesgue-almost every with respect to the corresponding Euclidean identification.

**(P4) Joint Borel measurability and a.e. parameter continuity.** Postulate (P4) ensures that primitive evaluation is compatible with measure-theoretic reasoning and that small parameter perturbations do not cause pathological behavior except on null sets.

> **(P4)** For every primitive $\omega \in \mathcal{O}$, the map
> $$f_\omega^{\mathrm{real}} : \; E_{\mathrm{in},\omega} \times \Xi_\omega \to E_{\mathrm{out},\omega}$$
> is jointly Borel measurable (with respect to product Borel $\sigma$-algebras). Moreover, for Lebesgue-almost every input $x \in E_{\mathrm{in},\omega}$, the map $\xi \mapsto f_\omega^{\mathrm{real}}(x;\xi)$ is continuous at Lebesgue-almost every $\xi \in \Xi_\omega$.

The "a.e. parameter continuity" clause is intentionally weak: it allows piecewise-constant and piecewise-smooth realizations (including IEEE-754 rounding surfaces), while ruling out adversarial dependence on $\xi$ on sets of positive measure.

*Remark* B.15 (Relationship to (S1)). Under (P1)–(P4), composition along an acyclic graph yields a Borel network map $N_\theta$ for each fixed $\theta$, discharging the measurability component of (S1) in standard packs. Totality (i.e. ruling out undefined values) is still a semantic choice handled at the pack level through domain declarations and (S1)'s "well-defined" requirement.

*Remark* B.16 (TAME is not in (P4)). (P4) does *not* assert differentiability, piecewise-$C^1$ structure, or definability. Those regularity properties are provided separately by the pack postulate TAME (§B.3) when working in Real-RAM worlds.

**(P5$_{\mathrm{dir}}$) Directional nonlinearity certificate.** The preceding postulates ensure well-posed composition; they do not guarantee that a nonaffine primitive contributes a *quantitatively nontrivial* amount of variation. Postulate (P5$_{\mathrm{dir}}$) supplies exactly this information in certificate form.

Fix a primitive $\omega$ that is used at a node tagged `nonaffine` by the pack's type tagging. A *directional nonlinearity certificate* for $\omega$ is a tuple

$$\mathsf{Cert}_\omega^{\mathrm{dir}} \;=\; \big(U_\omega,\; r_\omega,\; \nu_\omega^{\mathrm{dir}},\; \rho_\omega,\; \mathsf{Sel}_\omega\big),$$

where:

- $U_\omega \subseteq E_{\text{in},\omega} \times \Xi_\omega$ is a Borel *validity set* (the input/parameter pairs for which the certificate applies),
- $r_\omega > 0$ is a declared *radius of applicability*,
- $\mathsf{Sel}_\omega$ is a Borel-measurable selector that maps each $(x, \xi) \in U_\omega$ to

$$\mathsf{Sel}_\omega(x, \xi) = (u, q, t_0) \quad \text{with} \quad u \in \mathbb{S}^{d_\omega - 1}, \ q \in \mathbb{S}^{m_\omega - 1}, \ t_0 \in (0, r_\omega),$$

- $\nu_\omega^{\text{dir}} > 0$ (directional nonlinearity floor) and $\rho_\omega > 0$ (variation floor) are certificate constants.

Define the scalar restriction along the selected direction by

$$\phi_{x,\xi}(t) \ := \ \big\langle q, \ f_\omega^{\text{real}}(x + tu; \xi) \big\rangle, \qquad t \in [0, r_\omega].$$

**(P5$_{\text{dir}}$)** For every $(x, \xi) \in U_\omega$ with selector $(u, q, t_0) = \mathsf{Sel}_\omega(x, \xi)$:

(i) (*Non-affinity witness*) the one-sided derivatives $\phi'_{x,\xi}(t_0^-)$ and $\phi'_{x,\xi}(t_0^+)$ exist (in the Real-RAM sense when applicable), and

$$\big| \phi'_{x,\xi}(t_0^+) - \phi'_{x,\xi}(t_0^-) \big| \ \geq \ \nu_\omega^{\text{dir}};$$

(ii) (*Variation floor*) the total variation of $\phi_{x,\xi}$ on $[0, r_\omega]$ satisfies

$$\mathrm{TV}[\phi_{x,\xi}; 0, r_\omega] \ \geq \ \rho_\omega \, r_\omega.$$

When we want to emphasize that these are *domain-level certified floors*, we define the *certified infima* (with respect to the declared validity set and selector) by

$$\nu_\omega^{\text{dir,inf}} \ := \ \inf_{(x,\xi) \in U_\omega} \big| \phi'_{x,\xi}(t_0^+) - \phi'_{x,\xi}(t_0^-) \big|, \qquad \rho_\omega^{\text{inf}} \ := \ \inf_{(x,\xi) \in U_\omega} \frac{1}{r_\omega} \mathrm{TV}[\phi_{x,\xi}; 0, r_\omega],$$

where $t_0$ is the component of $\mathsf{Sel}_\omega(x, \xi)$. A certificate $\mathsf{Cert}_\omega^{\text{dir}}$ is valid only if $\nu_\omega^{\text{dir,inf}} > 0$ and $\rho_\omega^{\text{inf}} > 0$, and it provides explicit numeric lower bounds (recorded in $\nu_\omega^{\text{dir}}, \rho_\omega$) that witness these infima are strictly positive on $U_\omega$.

*Remark* B.17 (Interpretation). The selector $(u, q, t_0)$ chooses (i) an input direction $u$ along which the primitive is provably non-affine, (ii) a scalar readout $q$ that exposes the nonlinearity in a one-dimensional projection, and (iii) a witness location $t_0$ where the directional derivative changes. The second condition ensures the nonlinearity is not merely formal (derivative changes but output is nearly flat): it enforces a nontrivial lower bound on one-dimensional variation over the declared radius.

*Remark* B.18 (IEEE-754 worlds). In IEEE-754 worlds, the realized primitive $f_\omega^{\text{real}}$ can be piecewise constant as a function of real inputs, and derivative-based witnesses are not the right object. The event-based variation analysis is handled by the event axioms (E1–E4), which provide jump and event-density certificates. (P5$_{\text{dir}}$) is primarily used in Real-RAM (integral) worlds and in comparisons to ideal semantics.

*Remark* B.19 (What can violate (P5$_{\text{dir}}$)). A primitive can be tagged `nonaffine` yet fail to contribute quantitative variation on a given domain (e.g. saturated softmax, dead ReLUs, or normalization in a near-constant regime). This is not a contradiction: (P5$_{\text{dir}}$) is a *certificate requirement* that must be checked (or restricted to a validity set $U_\omega$) for the primitive to serve as the bottleneck in a certified lower bound.

**Summary: what (P1)–(P5) buy you.** Postulates (P1)–(P4) ensure that networks built from $\mathcal{O}$ behave as measurable maps under parameter freezing and compose cleanly along acyclic graphs, supporting (S1) and the measure-theoretic developments. Postulate (P5$_{\text{dir}}$) is the first quantitative ingredient: it supplies a pack-auditable floor on directional nonlinearity and variation for a designated nonaffine primitive, which is the primitive-level input to the later variation lower bounds.

### B.5.4. Pack-Supplied Conditions (E1–E4)

The primitive postulates (P1–P5) ensure that networks can be composed as measurable maps and that designated nonaffine primitives exhibit *local* directional nonlinearity (in Real-RAM worlds). To obtain *certifiable end-to-end* variation bounds, the framework additionally requires pack-supplied conditions governing (i) arithmetic semantics, (ii) measurability of the active-graph map, (iii) optional semantics for cycles/implicit evaluation, and (iv) quantitative bounds controlling how downstream "tails" propagate variation generated at bottlenecks.

*Remark* B.20 (Guardrail: notation comes from §B.4). All objects of the form $\mathcal{G}_x^{\text{act}}$, $\mathcal{T}_{\text{act}}(x)$, active path families, suffixes $P_{\succeq v}$, suffix transfer operators $T_{P_{\succeq v}}(\cdot)$, and pathwise Jacobian blocks $J_e^{(P)}(\cdot)$ are *defined in §B.4*. The conditions below do *not* introduce new graph operators; they only assert measurability, well-definedness, and quantitative bounds on these already-defined objects.

*Remark* B.21 (Norm conventions). All operator norms $\| \cdot \|_{\text{op}}$ and inner products $\langle \cdot, \cdot \rangle$ are taken with respect to the ambient Euclidean structures fixed in §B.3.

**(E1) Arithmetic world (Real-RAM vs. IEEE-754).** A pack declares an arithmetic world that determines the semantics of primitive evaluation.

**(E1)** The pack $\Pi$ declares exactly one arithmetic world:

- **Real-RAM:** primitives compute real-valued functions (exact arithmetic), inducing an ideal map $N_\theta : \mathcal{X} \to \mathcal{Y}$.
- **IEEE-754:** primitives compute floating-point functions in a declared format $\mathbb{F}$ (e.g. FP32/FP16/BF16) with a declared rounding mode, inducing a realized map $\tilde{N}_\theta : \mathcal{X} \cap \mathbb{F}^d \to \mathcal{Y} \cap \mathbb{F}^m$.

Exceptional outputs (NaN, $\pm\infty$) are excluded unless the pack explicitly includes them in the codomain and declares their propagation semantics.

(E1) is the single switch that selects whether later bounds are expressed in integral/Jacobian form (Real-RAM) or event/jump form (IEEE-754).

**(E2) Active-subgraph measurability (pack declaration).** The structural theory treats the active graph map $x \mapsto \mathcal{G}_x^{\mathrm{act}}$ and the active sink set $x \mapsto \mathcal{T}_{\mathrm{act}}(x)$ as measurable objects. We therefore require measurability at the level of edge-activity predicates.

**(E2)** For each saturated edge $e \in E(\mathcal{G}^{\mathrm{sat}})$, the indicator $x \mapsto \mathbf{1}\{e \in \mathcal{G}_x^{\mathrm{act}}\}$ is Borel measurable on $\mathcal{X}$. Consequently, the set-valued map $x \mapsto \mathcal{G}_x^{\mathrm{act}} \subseteq \mathcal{G}^{\mathrm{sat}}$ is Borel-measurable in the sense that membership of each edge is a Borel predicate, and $x \mapsto \mathcal{T}_{\mathrm{act}}(x)$ is Borel measurable as a map into $2^{V(\mathcal{G}^{\mathrm{sat}})}$.

In Real-RAM packs satisfying TAME (§B.3), (E2) holds automatically; we state it explicitly because it becomes nontrivial in relaxed worlds (e.g. weak-regularity or hardware-specific IEEE-754 variants).

**(E3) Optional cycle / implicit semantics.** Most architectures treated in the mainline development induce acyclic $\mathcal{G}^{\mathrm{str}}$ (hence acyclic $\mathcal{G}^{\mathrm{sat}}$), and no additional semantics are needed. Some architectures (DEQs, implicit layers, solver-based kernels) require an explicit interpretation of cycles or implicit evaluation.

**(E3)** If the pack allows cyclic graphs or implicit evaluation, it must declare one of the following cycle semantics, and the induced network map must still satisfy (S1) (well-defined Borel evaluation):

- **Bounded unrolling semantics:** a fixed unroll budget $K$ is declared and the cycle is interpreted by $K$ explicit iterations, yielding an acyclic unrolled graph.
- **Fixed-point semantics:** the cycle is interpreted as a fixed point $z^*(x)$ of a declared update map, together with a declared condition (e.g. contraction on a stated domain) guaranteeing existence/uniqueness and measurability of $z^*(x)$.

When $\mathcal{G}^{\mathrm{str}}$ is acyclic, (E3) is vacuous and may be ignored.

**(E4) Tail bounds (choice of variant).** The variation lower bounds proved later factor into a *local nonlinearity contribution* (from a bottleneck primitive) and a *tail propagation contribution* that captures how downstream linearization maps transmit that local variation to the sink(s). Condition (E4) is exactly the place where the pack supplies quantitative tail control.

There are several variants, corresponding to different "tail worlds" (as treated in world enumeration). The pack must declare *one* variant to be in force; we write $E4\text{-}\star$ for $\star \in \{\sigma\text{-cell}, \sigma\text{-ray}, \mathrm{dir}\}$.

**Setup shared by all E4 variants.** Fix an input $x$ and an active sink $\tau \in \mathcal{T}_{\mathrm{act}}(x)$. Let $\mathsf{Paths}(x, \tau)$ denote the set of directed $S^{\mathrm{sat}}$–$\tau$ paths in $\mathcal{G}_x^{\mathrm{act}}$. Let $C(x, \tau)$ denote the (nonempty) set of minimum $S^{\mathrm{sat}}$–$\tau$ edge cuts in $\mathcal{G}_x^{\mathrm{act}}$ (with unit capacities, as in §B.4).

We will use a measurable selection rule (Lemma B.24) to pick:

- a minimum cut $C^*(x, \tau) \in C(x, \tau)$, and
- a *witness edge* $e^*(x, \tau) \in C^*(x, \tau)$ that is (i) nonaffine-incident and (ii) lies on at least one active path in $\mathsf{Paths}(x, \tau)$.

Write $e^*(x, \tau) = (u^*(x, \tau) \to v^*(x, \tau))$ and define

$$v^*(x, \tau) := \text{ the head vertex of the selected witness edge } e^*(x, \tau).$$

We use $v^*(x, \tau)$ only as an *anchor* for suffix paths, via the suffix notation $P_{\succeq v}$ from §B.4.

Define the admissible suffix family

$$\mathsf{Suf}(x, \tau) := \left\{ P_{\succeq v^*(x, \tau)} : \ P \in \mathsf{Paths}(x, \tau) \text{ and } e^*(x, \tau) \in P \right\}.$$

By Lemma B.25, $\mathsf{Suf}(x, \tau) \neq \varnothing$ whenever $\tau$ is active.

**(E4-$\sigma$-cell) Cellwise spectral tail certificate.** This variant supplies a *cellwise* (locally constant on refined cells) upper bound on the conditioning of tail transfer operators, expressed via the Moore–Penrose pseudoinverse.

**(E4-$\sigma$-cell)** There exists a Borel function $\widetilde{B}_{\mathrm{tail}}^{\sigma,\mathrm{cell}}(x,\tau) \in (0,\infty)$, locally constant on refined cells, such that for $\mu$-a.e. $x$, every $\tau \in \mathcal{T}_{\mathrm{act}}(x)$, and every admissible suffix $Q \in \mathsf{Suf}(x,\tau)$,

$$\left\| T_Q(x)^\dagger \right\|_{\mathrm{op}} \leq \widetilde{B}_{\mathrm{tail}}^{\sigma,\mathrm{cell}}(x,\tau).$$

Equivalently, $\sigma_{\min}\!\big(T_Q(x)\big) \geq 1/\widetilde{B}_{\mathrm{tail}}^{\sigma,\mathrm{cell}}(x,\tau)$. We define the corresponding spectral tail gain floor by

$$\Gamma_{\sigma,\mathrm{cell}}(x,\tau) \;:=\; \frac{1}{\widetilde{B}_{\mathrm{tail}}^{\sigma,\mathrm{cell}}(x,\tau)}.$$

**(E4-$\sigma$-ray) Raywise spectral tail certificate.** This variant strengthens E4-$\sigma$-cell by requiring uniform control along a declared ray segment.

**(E4-$\sigma$-ray)** For $\mu$-a.e. basepoint $x$ and for almost every direction $v \in \mathbb{S}^{d-1}$ (with respect to the uniform/Haar probability measure on $\mathbb{S}^{d-1}$; see §B.3), and for each declared radius $r > 0$, there exists a bound $\widetilde{B}_{\mathrm{tail}}^{\sigma,\mathrm{ray}}(x,v,r) \in (0,\infty)$ such that for all $t \in [0,r]$, all active sinks $\tau \in \mathcal{T}_{\mathrm{act}}(x+tv)$, and all admissible suffixes $Q \in \mathsf{Suf}(x+tv,\tau)$,

$$\left\| T_Q(x+tv)^\dagger \right\|_{\mathrm{op}} \leq \widetilde{B}_{\mathrm{tail}}^{\sigma,\mathrm{ray}}(x,v,r).$$

We again set $\Gamma_{\sigma,\mathrm{ray}}(x,v,r) := 1/\widetilde{B}_{\mathrm{tail}}^{\sigma,\mathrm{ray}}(x,v,r)$.

**(E4-dir) Directional tail gain certificate.** This variant avoids worst-case conditioning by certifying gain only along a *selected* one-dimensional direction (the direction that will be paired with the bottleneck primitive's directional nonlinearity certificate).

**(E4-dir)** There exist Borel functions $\Gamma_{\mathrm{dir}}(x,\tau) > 0$ and Borel selectors $u_{\mathrm{tail}}(x,\tau)$ and $q_{\mathrm{tail}}(x,\tau)$, each a unit vector in the appropriate Euclidean spaces (domain/codomain of $T_Q(x)$ as determined by the suffix anchor), such that for $\mu$-a.e. $x$, every $\tau \in \mathcal{T}_{\mathrm{act}}(x,\tau)$, and every admissible suffix $Q \in \mathsf{Suf}(x,\tau)$,

$$\big| \langle q_{\mathrm{tail}}(x,\tau),\ T_Q(x)\, u_{\mathrm{tail}}(x,\tau) \rangle \big| \;\geq\; \Gamma_{\mathrm{dir}}(x,\tau).$$

(E4-dir) is strictly weaker than spectral full-rank requirements and is often the practically relevant regime: it certifies that *some* direction of bottleneck variation survives transmission to the sink with a nontrivial floor.

*Remark* B.22 (Single downstream interface). Downstream results write $\Gamma$ for the tail gain factor, instantiated as $\Gamma_{\sigma,\mathrm{cell}}$, $\Gamma_{\sigma,\mathrm{ray}}$, or $\Gamma_{\mathrm{dir}}$ depending on the pack's declared E4 variant.

*Remark* B.23 (IEEE-754 worlds). In IEEE-754 worlds, derivative-based tail transfer can be replaced (or supplemented) by event/jump certificates. We keep the E4-$\star$ taxonomy above as the Real-RAM tail mechanism; IEEE-754 analogs appear in the event-bound development.

**Two technical lemmas (selection and non-emptiness).** The E4 conditions quantify over admissible suffix families determined by a selected witness edge from a minimum cut. Two mild technical facts are used throughout the proof ledger: (1) such witnesses can be selected measurably, and (2) the resulting suffix family is nonempty. Proofs are deferred to Appendix C.

**Lemma B.24** (Measurable witness and suffix selection). *Assume (E2). Then there exist Borel selectors that map each pair $(x,\tau)$ with $\tau \in \mathcal{T}_{\mathrm{act}}(x)$ to a minimum cut $C^*(x,\tau) \in C(x,\tau)$ and to a witness edge $e^*(x,\tau) \in C^*(x,\tau)$ such that: (i) $e^*(x,\tau)$ is nonaffine-incident and (ii) $e^*(x,\tau)$ lies on at least one active path in $\mathsf{Paths}(x,\tau)$. Consequently, the induced anchor $v^*(x,\tau)$ (the head of $e^*(x,\tau)$) and the admissible suffix family $\mathsf{Suf}(x,\tau)$ are measurable objects.*

**Lemma B.25** (Suffix family non-emptiness). *For $\mu$-a.e. $x$ and every $\tau \in \mathcal{T}_{\mathrm{act}}(x)$, the admissible suffix family $\mathsf{Suf}(x,\tau)$ is nonempty.*

**Diagnostic (non-certifying) conditions.** In empirical audits and debugging workflows, one often has fast proxies for E-conditions that are informative but do not constitute formal certificates.

**(G1$'$) Diagnostic tail estimate.** A computable proxy $\widehat{\Gamma}(x,\tau)$ (e.g. via realized-branch Jacobians, randomized projections, or layerwise heuristics) that *estimates* the tail gain factor but is not guaranteed to be a lower bound.

**(G2$'$) Realized-branch active graph.** A tractable proxy active graph $\widehat{\mathcal{G}^{\mathrm{act}}}_x \subseteq \mathcal{G}_x^{\mathrm{act}}$ obtained from a realized execution trace. Violations detected on $\widehat{\mathcal{G}^{\mathrm{act}}}_x$ are treated as diagnostic signals rather than formal refutations.

We use these diagnostics only for empirical validation and engineering feedback; the theorems and certificates are stated in terms of (E1)–(E4).

## B.5.5. Merge Certificate (Assumption M)

The tail conditions (E4-⋆) certify that *along a chosen downstream route* a perturbation generated at a bottleneck is transmitted to an active sink with nontrivial gain. However, in graphs with merges (residual additions, summations of parallel paths, multi-branch wiring), a perturbation can be *canceled* by other concurrently active routes. Assumption (M) is a certificate-level condition that rules out such cancellation on a declared, locally constant domain (typically a refined-cell ray segment; §B.3).

*Remark* B.26 (Notation collision: suffix paths vs. projectors). In §B.5.4, the symbol $Q$ is used for a *suffix path*. In this subsection, $Q$ denotes an *isolating output projector*. To avoid ambiguity, we denote suffix paths by $R$ here.

*Remark* B.27 ($\mathcal{P}$ vs. $P_S$). We reserve $\mathcal{P}$ for the *parameter distributor* (§2.1). For a linear subspace $S$ of a Euclidean space, $P_S$ denotes the *orthogonal projector* onto $S$.

**Setup: a cut interface and a downstream merge operator.** Fix a pack $\Pi$ and consider an input $x$ and an active sink $\tau \in \mathcal{T}_{\text{act}}(x)$. Let $\mathcal{G}_x^{\text{act}} \subseteq \mathcal{G}^{\text{sat}}$ be the active graph (§B.4). Let $C(x, \tau)$ be the (nonempty) set of minimum $S^{\text{sat}}$–$\tau$ edge cuts in $\mathcal{G}_x^{\text{act}}$.

By Lemma B.24 (from §B.5.4) we may select measurably: (i) a minimum cut $C^*(x, \tau) \in C(x, \tau)$ and (ii) a witness edge $e^*(x, \tau) \in C^*(x, \tau)$ that is nonaffine-incident and lies on at least one active $S^{\text{sat}}$–$\tau$ path. Write

$$C^*(x, \tau) = \{e_1, \ldots, e_k\}, \qquad k = |C^*(x, \tau)| = \text{mincut}(S^{\text{sat}}, \tau; \mathcal{G}_x^{\text{act}}).$$

For each cut edge $e_i = (u_i \to v_i)$, let $E_{v_i}$ denote the Euclidean slot (representation space) carried by the head vertex $v_i$.

**Cut-slot space.** Define the direct-sum interface space

$$E_{C^*}(x, \tau) := \bigoplus_{i=1}^{k} E_{v_i},$$

with canonical coordinate projections $\pi_i : E_{C^*}(x, \tau) \to E_{v_i}$ and injections $\iota_i : E_{v_i} \to E_{C^*}(x, \tau)$ (so $\pi_i \iota_i = \text{Id}$ and $\sum_i \iota_i \pi_i = \text{Id}$).

**Suffix families and block tail transfers.** For each cut head $v_i$, let $\mathsf{Suf}(x, \tau; v_i)$ be the family of suffix paths

$$\mathsf{Suf}(x, \tau; v_i) := \left\{ P_{\succeq v_i} : P \in \mathsf{Paths}(x, \tau) \text{ and } e_i \in P \right\},$$

where $\mathsf{Paths}(x, \tau)$ is the set of directed $S^{\text{sat}}$–$\tau$ paths in $\mathcal{G}_x^{\text{act}}$ (§B.4). For each suffix $R \in \mathsf{Suf}(x, \tau; v_i)$, the suffix transfer operator $T_R(x)$ is defined in §B.4.

Define the (linear) *merged tail block* from $v_i$ to $\tau$ by the finite superposition

$$M_i(x, \tau) := \sum_{R \in \mathsf{Suf}(x, \tau; v_i)} T_R(x) \quad : \quad E_{v_i} \to E_\tau.$$

Finally define the *cut-interface merge operator*

$$M(x, \tau) := \sum_{i=1}^{k} M_i(x, \tau)\, \pi_i \quad : \quad E_{C^*}(x, \tau) \to E_\tau.$$

Intuitively, $M(x, \tau)$ is the linearized downstream map that takes perturbations injected at the *cut interface* (the heads of cut edges) and produces the resulting perturbation at the sink, including all active downstream recombinations.

**Merge certificate data.** A merge certificate selects an *isolated* coordinate subspace of the cut interface and an *output projection* that ignores all other cut-interface directions.

**Definition B.28** (Coordinate subspace and isolating projector). A *coordinate subspace* $S_0 \subseteq E_{C^*}(x, \tau)$ is any subspace spanned by a subset of the standard coordinate basis on the direct sum $E_{C^*}(x, \tau)$ (equivalently: it is obtained by choosing a subset of coordinates across the cut slots). Let $P_{S_0}$ and $P_{S_0^\perp}$ denote the orthogonal projectors onto $S_0$ and its orthogonal complement.

An *isolating projector* is an orthogonal projector

$$Q \ : \ E_\tau \to E_\tau$$

(often a coordinate projection onto a declared subset of output coordinates, but not required).

**Merge floor.** Fix a domain $U \subseteq \mathcal{X}$ on which the relevant objects are intended to be constant (typically $U$ is a refined-cell ray segment; §B.3). For fixed $(x, \tau)$, define the *merge floor on $U$* as the largest $\underline{s}_0(U; x, \tau) \geq 0$ such that

$$\|Q\, M(x', \tau)\, P_{S_0} z\| \geq \underline{s}_0(U; x, \tau)\, \|z\| \quad \text{for all } x' \in U \text{ and all } z \in E_{C^*}(x, \tau).$$

Equivalently, $\underline{s}_0(U; x, \tau)$ is the infimum (over $x' \in U$) of the smallest singular value of $Q\, M(x', \tau)\, P_{S_0}$ restricted to $S_0$.

**(M) Merge isolation / non-cancellation.** Assumption (M) requires that there exists a projected output component in which *only* the selected cut-subspace can influence the sink, with a positive gain floor.

**(M)** (Merge certificate) For $\mu$-a.e. basepoint $x$ and every active sink $\tau \in \mathcal{T}_{\mathrm{act}}(x)$, there exists a measurable choice of:

- a minimum cut $C^*(x, \tau)$ (as above),
- a coordinate subspace $S_0(x, \tau) \subseteq E_{C^*}(x, \tau)$,
- an orthogonal projector $Q(x, \tau) : E_\tau \to E_\tau$,

and a declared domain $U = U_{x,\tau} \subseteq \mathcal{X}$ containing $x$ (typically contained in one refined cell), such that the following two conditions hold for all $x' \in U$:

(M.1) **Isolation (no leakage from the complement):**

$$Q(x, \tau)\, M(x', \tau)\, P_{S_0(x,\tau)^\perp} \;=\; 0.$$

(M.2) **Merge floor (nontrivial transmission on $S_0$):**

$$\underline{s}_0(U; x, \tau) \;:=\; \inf_{x' \in U} \sigma_{\min}\big(Q(x, \tau)\, M(x', \tau)\, P_{S_0(x,\tau)}\big) \;>\; 0.$$

Condition (M.1) is the exact non-cancellation statement: after projecting the sink by $Q(x, \tau)$, *all* cut-interface directions outside $S_0(x, \tau)$ are provably irrelevant. Condition (M.2) ensures that directions inside $S_0(x, \tau)$ survive with a uniform positive gain floor on the declared domain.

*Remark* B.29 (How (M) interacts with $\lambda_x$). The cut size $k = |C^*(x, \tau)|$ is a structural bottleneck count (a min-cut). Assumption (M) does not assert additivity across all $k$ cut edges; instead it guarantees that *at least one certifiably isolated cut-subspace exists* whose contribution cannot be canceled by the others (in the projected output). Downstream theorems package this into the structural factor $\underline{s}_0(\cdot)$.

*Remark* B.30 (When (M) is easy). (M) holds with $\underline{s}_0 = 1$ in many "wiring-separated" cases, e.g. when the post-cut computation preserves a coordinate split (concatenation, block-diagonal routing, or any architecture where different branches occupy disjoint coordinate subsets), because one can choose $Q$ as a coordinate projection onto the isolated branch output and $S_0$ as the corresponding coordinate subspace of the cut interface.

*Remark* B.31 (When (M) can fail (and why we isolate it)). In residual-addition architectures where multiple branches share the same coordinate channels, exact isolation (M.1) may fail generically: other cut-interface directions can influence every output coordinate. This is not a defect of the framework: it is precisely the point where "no-cancellation" must be supplied as an explicit certificate rather than assumed implicitly. A robust relaxation allowing $\|QMP_{S_0^\perp}\|_{\mathrm{op}} \leq \varepsilon$ (approximate isolation) is treated as whitespace.

## Verification notes (M).

- **Static checks (structure-driven).** If the merge operator $M(x, \tau)$ is induced by explicit linear wiring with known sparsity (e.g. concatenation or block-diagonal projections), then (M.1) can be certified by inspecting the corresponding sparsity pattern, and (M.2) reduces to a smallest-singular-value bound on the isolated block.
- **Cellwise constancy.** Under TAME and on domains $U$ contained in a single refined cell, the active graph and the families $\mathsf{Suf}(x', \tau; v_i)$ are constant for $x' \in U$ (§B.3); the bound in (M.2) is then a genuine infimum over a continuous family and can be lower-bounded by standard means.
- **Diagnostics.** In experiments one can test an empirical proxy of (M.1) by estimating $\|Q\, M(x, \tau)\, P_{S_0^\perp}\|_{\mathrm{op}}$ via randomized projections; failures are diagnostic, not formal. Exact certification is reserved for settings where $M$ admits a symbolic or sparsity-based proof.

### B.5.6. FEASIBLE REGION

Axioms in the Atlas play two roles. Some axioms are *semantic/regularity* assumptions that make the objects of interest well-defined (network map, graphs, measurability, stratifications). Others are *epistemic/contract* assumptions that supply quantitative constants or certificates needed to instantiate nonvacuous bounds. To make this separation operational, we package axiom satisfaction as a family of *satisfaction sets* whose intersection is the region on which the full guarantee applies.

**Axiom label universe.** Let

$$\mathcal{A} := \{S1, S2, C2, P1, P2, P3, P4, P5, \mathrm{TAME}, E1, E2, E3, E4, M\}$$

denote the full set of axiom/postulate/assumption labels used in this paper. (Throughout, "axiom" is used informally to include postulates such as TAME and assumptions such as M.)

**Satisfaction sets.** Fix a pack $\Pi$ (§2.1), an architecture template $\mathcal{G}^{\mathrm{str}}$, and parameters $\theta \in \Theta$. Let $\kappa$ denote the bundle of all *certificate data* and *declared constants* required by the axioms under $\Pi$ (e.g., directional-nonlinearity certificates for primitives, tail-gain constants, merge floors, and any declared world modifiers).

For each label $i \in \mathcal{A}$, let $a_i(x; \kappa)$ denote the predicate "axiom $i$ holds at input $x$ under pack $\Pi$ for network $N_\theta$, with certificates $\kappa$." This predicate may include the axiom's internal quantifiers. For example, $a_{S2}(x; \kappa)$ quantifies over active sinks $\tau \in \mathcal{T}_{\text{act}}(x)$ and over minimum cuts in $\mathcal{G}_x^{\text{act}}$; $a_{E4}(x; \kappa)$ quantifies over the pack-selected witness edge and suffix family (as defined in §B.4 and §B.5.4).

The *satisfaction set* for label $i$ is

$$S_i(\kappa) := \{x \in \mathcal{X} : a_i(x; \kappa) \text{ holds}\}.$$

When an axiom is global (e.g. (C2) as a realizability statement, or a fixed arithmetic world (E1)), $a_i(x; \kappa)$ is constant in $x$ and $S_i(\kappa)$ is either $\mathcal{X}$ (if the axiom holds) or $\varnothing$ (if it fails).

**Feasible region for an axiom set.**  For any axiom set $A \subseteq \mathcal{A}$, define the *feasible region*

$$F(A; \kappa) := \bigcap_{i \in A} S_i(\kappa) \subseteq \mathcal{X}.$$

This is the subset of inputs on which *all* axioms in $A$ hold simultaneously under the declared certificates. In theorems stated "$\mu$-almost everywhere," the intended success criterion is $\mu(F(A; \kappa)) = 1$, where $\mu$ is the reference measure on $\mathcal{X}$ (§B.3).

*Remark* B.32 (Extended feasible regions for rays).  When a guarantee is stated for triples $(x, v, r)$ (basepoint, direction, radius), the same construction applies on the product space $\mathcal{X} \times \mathbb{S}^{d-1} \times \mathbb{R}_{>0}$, by replacing $S_i(\kappa) \subseteq \mathcal{X}$ with $S_i(\kappa) \subseteq \mathcal{X} \times \mathbb{S}^{d-1} \times \mathbb{R}_{>0}$. For readability we present the basepoint form here.

**Semantic/regularity vs. epistemic/contract axioms.**  We use the feasible-region lens to separate axiom roles:

- **Semantic/regularity layer:**
$$\{S1, S2, C2, P1, P2, P3, P4, \text{TAME}\}.$$

  These axioms fix the meaning of $N_\theta$ and of the graph objects ($\mathcal{G}^{\text{str}}, \mathcal{G}^{\text{sat}}, \mathcal{G}_x^{\text{act}}$) and ensure the measurability/stratification properties required to state the later bounds.
- **Epistemic/contract layer:**
$$\{P5, E1, E2, E3, E4, M\}.$$

  These labels introduce pack-declared execution worlds (E1–E3) and the *certificates/quantitative contracts* (P5, E4, M) that turn structural bottlenecks into nonvacuous numerical lower bounds. In practice, this is the layer where constants are certified, estimated, or audited.

**Tame networks as feasible regions.**  Let $\mathcal{A}_{\text{full}} := \mathcal{A}$. For fixed $(\Pi, \mathcal{G}^{\text{str}}, \theta)$ and certificates $\kappa$, define the *tame region* of the network as

$$\text{TN}_\theta(\kappa) := F(\mathcal{A}_{\text{full}}; \kappa).$$

When $\mu(\text{TN}_\theta(\kappa)) = 1$ we say (informally) that $N_\theta$ is *tame* under pack $\Pi$ with certificates $\kappa$: all structural objects are well-defined almost everywhere, and the quantitative contracts needed for the main variation bound hold on a full-measure set.

**Redundancies recorded in the ledger.**  Two implications are used repeatedly and motivate keeping certain labels explicit in the proof ledger:

- Under standard closure/measurability conditions on primitives (P1–P4), (S1) is discharged automatically for acyclic graphs by composition.
- Under the pack-declared tameness postulate (TAME), the measurability properties in (E2) are automatic.

We nonetheless retain (S1) and (E2) as named waypoints so that world enumeration and axiom closures can reference them directly.

Certificates aggregate the structural audit outputs and calibrated quantitative constants into a single, audit-ready record. We reuse the canonical certificate schema verbatim.

## B.6. Certificates and Verification

This section defines the certificate objects that witness axiom satisfaction and presents verification procedures for the *structural* layer (notably S2 and $\lambda_x$). The goal is *audit-ready* evidence: a certificate should be (i) explicit, (ii) checkable by a third party, and (iii) clear about what was verified exactly versus what was only proxied.

Throughout, all semantics are relative to a fixed pack $\Pi$ (§2.1), and all graph/active-graph notation (affine saturation, $\mathcal{G}^{\text{sat}}, \mathcal{G}_x^{\text{act}}$, sinks, suffix conventions) is as defined in §B.4. We introduce no new pathwise/Jacobian notation here.

### B.6.1. CERTIFICATE SCHEMA

A *certificate* records (i) the *query* being certified, (ii) the *axiom closure* being claimed, (iii) the *verification status* for each axiom, and (iv) any *witness objects* needed to independently check the claim.

**Certificate format.** A certificate instance has the form

$$\mathsf{Cert} \;=\; (x, \theta, \kappa, A_{\mathrm{req}}, \mathbf{s}, \mathsf{W}),$$

where:

- $x \in \mathcal{X}$ is the input at which the certificate is evaluated (extended queries may include $(v, r)$; see below),
- $\theta \in \Theta$ are the network parameters (distributed via $\mathcal{P}$),
- $\kappa$ is a vector of *declared verification hyperparameters* and context modifiers (e.g. Convention A/B, arithmetic world E1, E4 variant, operator-norm thresholds, router top-$k$, etc.),
- $A_{\mathrm{req}} \subseteq \mathcal{A}$ is the requested axiom set (closure) the user wants to certify,
- $\mathbf{s}$ is a *status vector* indexed by axiom labels in $A_{\mathrm{req}}$,
- $\mathsf{W}$ is a container of checkable witnesses (cuts, partitions, constants, hashes, logs).

**Status vector.** For each axiom label $i \in A_{\mathrm{req}}$, the status $\mathbf{s}_i$ takes values in:

$$\mathbf{s}_i \in \{\texttt{evaluated-pass}, \texttt{evaluated-fail}, \texttt{proxied}, \texttt{not-evaluated}\}.$$

This is intentionally *epistemic*: it distinguishes what is known from what is assumed.

**Feasibility and supporting closures.** Given Cert, define the set of *supported axioms*

$$A_{\mathrm{sup}} \;:=\; \{\, i \in A_{\mathrm{req}} : \mathbf{s}_i = \texttt{evaluated-pass} \,\}.$$

The certificate output is:

- **Feasible** for $A_{\mathrm{req}}$ if $A_{\mathrm{sup}} = A_{\mathrm{req}}$;
- otherwise **Partially feasible**, together with the **supporting closure** $A_{\mathrm{sup}}$ (often still scientifically meaningful: e.g. "structural world verified; quantitative world not certified").

This matches the feasible-region language from §B.5.6: certifying $A_{\mathrm{req}}$ is certifying membership in $F(A_{\mathrm{req}}; \kappa)$.

**Extended query types.** Many quantitative statements are naturally indexed by a ray $(x, v, r)$ rather than just $x$. The schema extends verbatim by replacing $x$ with a query record $q \in \{x\} \cup \{(x, v, r)\}$, and interpreting axiom satisfaction sets as subsets of the relevant product space.

*Remark* B.33 (Audit metadata (recommended)). For reproducibility, $\mathsf{W}$ should include: pack identifier $\Pi$, model identifier/checkpoint hash, library versions, device type, and (under IEEE-754 worlds) the declared floating-point format and rounding mode. These are metadata, not mathematical axioms, but they are essential for third-party checking.

### B.6.2. EXACT VERIFICATION PROCEDURE (STRUCTURAL)

This subsection describes an *exact* (i.e. sound, pack-relative) verification procedure for the structural claims: construction of $\mathcal{G}^{\mathrm{sat}}$, evaluation of $\mathcal{G}_x^{\mathrm{act}}$, computation of $\lambda_x$, and checking (S2). Exactness here means: the procedure's inputs/outputs are defined in terms of the pack-declared semantics, and a failure output includes a checkable witness.

**Inputs.** A pack $\Pi$, an architecture template $\mathcal{G}^{\mathrm{str}}$, parameters $\theta$, and an input $x$.

**Outputs.** A certificate Cert that records:

- $\mathcal{G}^{\mathrm{sat}}$ (or a hash plus a replay recipe),
- $\mathcal{G}_x^{\mathrm{act}} \subseteq \mathcal{G}^{\mathrm{sat}}$ and active sinks $\mathcal{T}_{\mathrm{act}}(x)$,
- $\lambda_x$ and per-sink values $\lambda_{x,\tau} := \mathrm{mincut}(S^{\mathrm{sat}}, \tau; \mathcal{G}_x^{\mathrm{act}})$,
- the (S2) verdict together with a witness: either (i) a certified inequality ruling out affine-only minimum cuts (pass), or (ii) an affine-only minimum cut (fail).

**Procedure.**

**Step 1:** **Build $\mathcal{G}^{\mathrm{sat}}$.** Construct the affine-saturated graph $\mathcal{G}^{\mathrm{sat}}$ from $\mathcal{G}^{\mathrm{str}}$ using the pack's affine/nonaffine tags. Record the node typing and the set of *nonaffine-tagged* nodes (and hence nonaffine-incident edges).

**Step 2:** **Construct $\mathcal{G}_x^{\mathrm{act}}$ (exact, pack-relative).** Evaluate the pack-defined influence/activity predicate to obtain the path-closed active subgraph $\mathcal{G}_x^{\mathrm{act}} \subseteq \mathcal{G}^{\mathrm{sat}}$ and its active sink set $\mathcal{T}_{\mathrm{act}}(x)$. (Under TAME, $x \mapsto \mathcal{G}_x^{\mathrm{act}}$ is Borel and locally constant a.e. by Lemmas 4.1.2–4.1.3.)

**Step 3: Compute min-cuts and $\lambda_x$.** For each $\tau \in \mathcal{T}_{\mathrm{act}}(x)$, compute

$$\lambda_{x,\tau} := \mathrm{mincut}(S^{\mathrm{sat}}, \tau; \mathcal{G}_x^{\mathrm{act}}), \qquad \lambda_x := \min_{\tau \in \mathcal{T}_{\mathrm{act}}(x)} \lambda_{x,\tau}.$$

Store the returned cut partitions (reachable sets) needed to replay the check.

**Step 4: Verify (S2) without enumerating all minimum cuts.** Fix $\tau \in \mathcal{T}_{\mathrm{act}}(x)$ and write $m_x := |E(\mathcal{G}_x^{\mathrm{act}})|$. Define a modified capacity function $c^{\mathrm{aff}}$ on edges of $\mathcal{G}_x^{\mathrm{act}}$ by

$$c^{\mathrm{aff}}(e) := \begin{cases} 1, & e \text{ has both endpoints affine-tagged in } \mathcal{G}^{\mathrm{sat}}, \\ m_x + 1, & e \text{ is nonaffine-incident.} \end{cases}$$

Let

$$\lambda_{x,\tau}^{\mathrm{aff}} := \mathrm{mincut}_{c^{\mathrm{aff}}}(S^{\mathrm{sat}}, \tau; \mathcal{G}_x^{\mathrm{act}}).$$

Then:

- If $\lambda_{x,\tau}^{\mathrm{aff}} = \lambda_{x,\tau}$, there exists an *affine-only* minimum cut, yielding a **formal (S2) violation witness** for $(x, \tau)$.
- If $\lambda_{x,\tau}^{\mathrm{aff}} > \lambda_{x,\tau}$, then *no* minimum cut can be affine-only, hence (S2) holds for $(x, \tau)$.

This check is polynomial-time and avoids exponential enumeration of all minimum cuts. Repeat for all $\tau \in \mathcal{T}_{\mathrm{act}}(x)$.

**Soundness (bidirectional).** With exact $\mathcal{G}_x^{\mathrm{act}}$ construction and exact min-cut computations:

- **Pass soundness:** If the procedure returns (S2)-pass, then the pack-relative axiom (S2) holds at $x$ for every active sink $\tau$.
- **Fail soundness:** If the procedure returns (S2)-fail, the certificate includes an affine-only minimum cut, which is a checkable witness that (S2) fails at $(x, \tau)$.

*Remark* B.34 (Exactness vs. practicality). Exact $\mathcal{G}_x^{\mathrm{act}}$ construction may require symbolic reasoning or certified influence tests, depending on the pack.

### B.6.3. COMPLEXITY BOUNDS

We summarize worst-case complexity in terms of $n := |V(\mathcal{G}^{\mathrm{sat}})|$, $m := |E(\mathcal{G}^{\mathrm{sat}})|$, and $m_x := |E(\mathcal{G}_x^{\mathrm{act}})|$.

**Proposition B.35** (Structural verification complexity). *Fix a pack $\Pi$ and an input $x$.*

(i) *Saturation. Constructing $\mathcal{G}^{\mathrm{sat}}$ from $\mathcal{G}^{\mathrm{str}}$ by contracting affine-only components takes $O(n + m)$ time and $O(n + m)$ memory.*

(ii) *Active-graph construction. The cost of constructing $\mathcal{G}_x^{\mathrm{act}}$ is pack-dependent. For a purely realized-branch proxy (one forward/backward pass), the cost is $O(\mathrm{cost}_{\mathrm{fwd}}(x) + \mathrm{cost}_{\mathrm{bwd}}(x))$. For an exact influence predicate, the cost is whatever is required by the pack's certified semantics.*

(iii) *Min-cut computation. Computing $\lambda_{x,\tau}$ is a single-source/single-sink max-flow. With unit capacities and a standard polynomial-time max-flow algorithm (e.g. Dinic/push-relabel), worst-case time is polynomial in $(n, m_x)$; in common unit-capacity regimes, Dinic admits bounds of order $O\left(m_x \cdot \min\{n^{2/3}, m_x^{1/2}\}\right)$. The (S2) check uses at most two such min-cut computations per active sink $\tau$ (baseline and modified capacities).*

(iv) *Enumerating all minimum cuts is unnecessary. Although the family of minimum cuts can be exponential in $\lambda_{x,\tau}$, the capacity-modification test in §B.6.2 verifies (S2) without enumerating them.*

*Remark* B.36 (Why it is fast in transformer-style graphs). In practice, $\mathcal{G}^{\mathrm{sat}}$ is sparse and $\lambda_{x,\tau}$ is typically small (often 1), so max-flow instances are small and cuts are shallow. The dominant cost is usually the forward/backward pass used to build the (proxy) active graph.

### B.6.4. PROXY DIAGNOSTICS (NON-CERTIFYING)

Exact $\mathcal{G}_x^{\mathrm{act}}$ is often intractable to compute in large models under realistic packs. We therefore use a proxy active graph $\widehat{\mathcal{G}^{\mathrm{act}}}_x$ derived from realized execution.

**Proxy construction (typical).** A standard proxy uses:

(a) one forward pass to record realized branches, routing decisions, and activation states;
(b) one backward (or local linearization) pass to estimate edge influence;
(c) a threshold rule (declared in $\kappa$) to retain edges deemed active/influential.

This yields a realized-branch under-approximation

$$\widehat{\mathcal{G}^{\mathrm{act}}}_x \subseteq \mathcal{G}_x^{\mathrm{act}},$$

and a corresponding proxy sink set $\widehat{\mathcal{T}}_{\mathrm{act}}(x)$.

**How to interpret proxy outputs.** Because $\widehat{\mathcal{G}^{\mathrm{act}}}_x$ is an under-approximation, proxy outcomes are *diagnostic*, not certifying:

- A **proxy S2 violation** is a strong signal of a potential bypass or convention mismatch, but it is *not* a formal refutation of (S2) without additional assumptions linking proxy and oracle.
- A **proxy S2 pass** is evidence of structural health on realized execution paths, but it is *not* a proof that all oracle-influential paths satisfy (S2).

**Valid uses.** Proxy diagnostics are still valuable for:

- **Model audits at scale** (large model surveys),
- **bottleneck census** (which node types repeatedly appear in proxy min-cuts),
- **regression testing** (detecting architectural changes that introduce new bypass routes),
- **proxy–oracle calibration** on small models where exact $\mathcal{G}_x^{\mathrm{act}}$ is computable.

### B.6.5. STRUCTURAL S2 CHECK (HEURISTIC, NON-CERTIFYING)

We define a lightweight check that ignores input-dependent activity and operates purely on saturated structure.

**Definition (heuristic).** Let $\mathcal{G}^{\mathrm{sat}}$ be the affine-saturated graph for the architecture under the pack's tagging conventions. Define **S2$_{\mathrm{struct}}$** as the statement:

> For every declared sink $\tau$ in $\mathcal{G}^{\mathrm{sat}}$ (or a declared sink family), every minimum $S^{\mathrm{sat}}$–$\tau$ cut in $\mathcal{G}^{\mathrm{sat}}$ contains a nonaffine-incident edge.

Equivalently, apply the same two-cut capacity test as in §B.6.2, but with $\mathcal{G}_x^{\mathrm{act}}$ replaced by $\mathcal{G}^{\mathrm{sat}}$.

**Status and limitations.** S2$_{\mathrm{struct}}$ is *not* the same as (S2), because (S2) is formulated on $\mathcal{G}_x^{\mathrm{act}}$ and is therefore input-dependent. In particular:

- S2$_{\mathrm{struct}}$ can pass while (S2) fails on rare inputs that deactivate all nonaffine routes (e.g. gates saturating to constants).
- S2$_{\mathrm{struct}}$ can fail while (S2) holds on a large measure subset of inputs (e.g. bypass routes exist structurally but are inactive for typical data).

Therefore S2$_{\mathrm{struct}}$ should be recorded as `proxied` unless the user explicitly adopts it as a replacement axiom in a modified world.

**When S2$_{\mathrm{struct}}$ is sufficient.** S2$_{\mathrm{struct}}$ is often an adequate proxy when:

- activity is *input-invariant* at the structural level (i.e. $\mathcal{G}_x^{\mathrm{act}} = \mathcal{G}^{\mathrm{sat}}$ for all $x$ in the declared domain), or
- the application only needs a *design-time wiring guarantee* ("the architecture forbids affine-only bypasses") rather than a per-input execution guarantee.

*Remark* B.37 (Recommended reporting practice). When reporting S2 checks empirically, we recommend always reporting three items: (i) S2$_{\mathrm{struct}}$ on $\mathcal{G}^{\mathrm{sat}}$, (ii) proxy S2 on $\widehat{\mathcal{G}^{\mathrm{act}}}_x$ over a dataset, and (iii) proxy–oracle calibration on at least one small model. This makes the proxy/oracle gap explicit and prevents overstating what was certified.

This subsection collects all lemma/theorem statements used in the proofs appendix, each annotated with its axiom closure.

## B.7. Theorems

**Reading guide.** This section collects all lemma/theorem statements used in the paper, each annotated with its *axiom closure* (the minimal label set assumed). Proofs are deferred to Appendix C.1 (structural), Appendix C.2 (quantitative), and Appendix C.3 (consistency/independence). We fix a pack $\Pi$ (§2.1), an architecture template $\mathcal{G}^{\mathrm{str}}$, and parameters $\theta \in \Theta$. Ambient spaces, reference measures, and "a.e." conventions are as declared in §B.3. Graph and active-graph semantics (including $\mathcal{G}^{\mathrm{str}}$, $\mathcal{G}^{\mathrm{sat}}$, $\mathcal{G}_x^{\mathrm{act}}$, active sinks, suffix conventions, and pathwise operators) are as defined in §B.4.

### B.7.1. STRUCTURAL THEOREMS

**Measurability (discharges S1)**

**Lemma B.38** (Joint measurability of evaluation; axiom closure $\{P1, P2, P3, P4\}$ (discharges (S1))). *Assume the pack $\Pi$ satisfies primitive postulates (P1)–(P4) for every node operation that appears in $\mathcal{G}^{\mathrm{str}}$ (and that the parameter distributor $\mathcal{P}$ is Borel, as part of the pack). Then the induced evaluation map*

$$(\theta, x) \longmapsto N_\theta(x)$$

*is Borel measurable as a map $\Theta \times \mathcal{X} \to \mathcal{Y}$. In particular, for every fixed $\theta \in \Theta$, the function $N_\theta : \mathcal{X} \to \mathcal{Y}$ is Borel measurable, and axiom (S1) holds for the ideal semantics induced by $\Pi$.*

*Proof.* Appendix C.1.1.

*Remark* B.39 (How this lemma is used). Subsequent theorems may cite (S1) as shorthand for "Lemma B.38 applies," to keep axiom closures readable.

## Active Graph Borel Selection (discharges E2)

**Lemma B.40** (Borel measurability of the active graph; axiom closure $\{P1, P2, P3, P4, \mathrm{TAME}\}$ (discharges (E2))). *Assume the pack* $\Pi$ *satisfies (P1)–(P4) and* TAME. *Then the active-structure map*

$$x \longmapsto \mathcal{G}_x^{\mathrm{act}} \subseteq \mathcal{G}^{\mathrm{sat}}$$

*is Borel measurable in the following concrete sense: if* $E(\mathcal{G}^{\mathrm{sat}})$ *is the finite edge set of* $\mathcal{G}^{\mathrm{sat}}$, *then the indicator vector*

$$x \longmapsto \left(\mathbf{1}_{\{e \in \mathcal{G}_x^{\mathrm{act}}\}}\right)_{e \in E(\mathcal{G}^{\mathrm{sat}})} \in \{0, 1\}^{E(\mathcal{G}^{\mathrm{sat}})}$$

*is a Borel function. Moreover, the active sink set* $x \mapsto \mathcal{T}_{\mathrm{act}}(x) \subseteq V(\mathcal{G}^{\mathrm{sat}})$ *is Borel measurable as a set-valued map (equivalently: each sink indicator* $x \mapsto \mathbf{1}_{\{\tau \in \mathcal{T}_{\mathrm{act}}(x)\}}$ *is Borel).*

*Proof.* Appendix C.1.2.

*Remark* B.41 (E2 as bookkeeping). Axiom (E2) is stated as a pack-level declaration to make measurability dependencies explicit in world enumeration. Lemma B.40 shows that, under TAME, the declaration holds automatically.

## Local Constancy of Active Structure

**Lemma B.42** (Local constancy on refined cells; axiom closure $\{P1, P2, P3, P4, \mathrm{TAME}\}$). *Assume the pack* $\Pi$ *satisfies (P1)–(P4) and* TAME. *Then there exists a refined cell family* $\mathcal{C}$ *(as in §B.3) such that:*

(i) $\mu\left(\mathcal{X} \setminus \bigcup_{C \in \mathcal{C}} C\right) = 0$,
(ii) *for each refined cell* $C \in \mathcal{C}$, *the active graph* $\mathcal{G}_x^{\mathrm{act}}$ *is constant for all* $x \in C$,
(iii) *likewise, the active sink set* $\mathcal{T}_{\mathrm{act}}(x)$ *is constant for all* $x \in C$.

*Proof.* Appendix C.1.3.

**Lemma B.43** (Local constancy of min-cuts; axiom closure $\{P1, P2, P3, P4, \mathrm{TAME}\}$). *Assume the hypotheses of Lemma B.42. Fix any sink* $\tau \in V(\mathcal{G}^{\mathrm{sat}})$. *On every refined cell* $C \in \mathcal{C}$, *the min-cut value*

$$x \longmapsto \mathrm{mincut}(S^{\mathrm{sat}}, \tau; \mathcal{G}_x^{\mathrm{act}})$$

*is constant on* $C$ *(with the convention* $\mathrm{mincut} = 0$ *if* $\tau$ *is not reachable in* $\mathcal{G}_x^{\mathrm{act}}$). *Consequently, the connectivity statistic*

$$\lambda_x := \min_{\tau \in \mathcal{T}_{\mathrm{act}}(x)} \mathrm{mincut}(S^{\mathrm{sat}}, \tau; \mathcal{G}_x^{\mathrm{act}})$$

*is locally constant on refined cells (on the full-measure set where* $\mathcal{T}_{\mathrm{act}}(x) \neq \varnothing$).

*Proof.* Appendix C.1.3.

**Lemma B.44** (Measurable constancy radius; axiom closure $\{P1, P2, P3, P4, \mathrm{TAME}\}$). *Assume the hypotheses of Lemma B.42. There exists a Borel function* $\varepsilon : \mathcal{X} \to [0, \infty)$ *such that for* $\mu$-*almost every* $x$, $\varepsilon(x) > 0$ *and for every* $x' \in \mathcal{X}$ *with* $\|x' - x\| < \varepsilon(x)$ *we have*

$$\mathcal{G}_{x'}^{\mathrm{act}} = \mathcal{G}_x^{\mathrm{act}} \qquad \text{and} \qquad \mathcal{T}_{\mathrm{act}}(x') = \mathcal{T}_{\mathrm{act}}(x).$$

*One valid choice is the distance-to-boundary radius of the refined cell containing* $x$.

*Proof.* Appendix C.1.3.

## Nonlinear Bottleneck

**Lemma B.45** (Nonaffine witness in minimum cuts; axiom closure $\{S2\}$). *Assume axiom (S2) holds at input* $x$. *Let* $\tau \in \mathcal{T}_{\mathrm{act}}(x)$ *be any active sink, and let* $C$ *be any minimum* $S^{\mathrm{sat}}$–$\tau$ *edge cut in* $\mathcal{G}_x^{\mathrm{act}}$. *Then* $C$ *contains at least one nonaffine-incident edge. Equivalently, if* $E_{\mathrm{na}} \subseteq E(\mathcal{G}^{\mathrm{sat}})$ *denotes the set of nonaffine-incident edges, then*

$$|C \cap E_{\mathrm{na}}| \geq 1.$$

*Proof.* Appendix C.1.4.

**Lemma B.46** (Bounded path length in finite templates; axiom closure $\varnothing$). *For any finite architecture template* $\mathcal{G}^{\mathrm{str}}$ *(hence finite* $\mathcal{G}^{\mathrm{sat}}$), *every directed path in* $\mathcal{G}^{\mathrm{sat}}$ *has length at most* $|V(\mathcal{G}^{\mathrm{sat}})| - 1$.

*Proof.* Appendix C.1.4.

**Lemma B.47** (S2 is independent of realizability; axiom closure $\{S2\}$). *The truth of* (S2) *depends only on* $\mathcal{G}^{\mathrm{sat}}$, *the active subgraph* $\mathcal{G}_x^{\mathrm{act}}$, *and the affine/nonaffine tags declared by the pack. In particular,* (S2) *can hold even in packs where* (C2) *fails (restricted primitive libraries). An explicit witness is given in Theorem B.62.*

*Proof.* Appendix C.1.4.

### Kink Creation (qualitative)

**Definition B.48** (Essential vs. inessential boundary)**.** Assume TAME, so that $\mathcal{X}$ admits a refined stratification into $C^1$ cells on which $N_\theta$ is $C^1$. Let $B$ be a codimension-1 boundary separating adjacent full-dimensional refined cells $C^-$ and $C^+$. We call $B$ *inessential* if $N_\theta$ extends $C^1$ across $B$ (one-sided Jacobians match), and *essential* otherwise.

**Theorem B.49** (Smoothness on constant-activity cells; axiom closure $\{$TAME$\}$)**.** *Assume* TAME. *On every full-dimensional refined cell* $C$, *the restriction* $N_\theta|_C$ *is* $C^1$. *Consequently, along any ray segment that remains inside a single refined cell, total variation coincides with the line integral of the speed:*

$$\mathrm{TV}[N_\theta; x, v, r] \;=\; \int_0^r \left\| \tfrac{d}{dt} N_\theta(x + tv) \right\| dt \quad \text{whenever } x + tv \in C \; \forall t \in [0, r].$$

*Proof.* Appendix C.1.5.

**Theorem B.50** (Kinks at essential boundaries; axiom closure $\{$TAME$\}$)**.** *Assume* TAME. *Let $B$ be an essential boundary (Definition B.48). Then for $\mu$-almost every basepoint $x \in B$ and for almost every direction $v$ transverse to $B$ at $x$, the one-sided directional derivatives differ:*

$$\lim_{t \downarrow 0} \tfrac{d}{dt} N_\theta(x + tv) \;\neq\; \lim_{t \uparrow 0} \tfrac{d}{dt} N_\theta(x + tv).$$

*Equivalently, a kink occurs at the boundary crossing for almost every transverse ray.*

*Proof.* Appendix C.1.5.

*Remark* B.51 (Connection to quantitative bounds)**.** Kink creation is qualitative evidence that variation accumulates at cell transitions. The quantitative lower bounds do not require counting kinks explicitly, but the IEEE-754 event bound can be interpreted as a discretized boundary-crossing lower bound in an execution world where the function is step-like.

### B.7.2. QUANTITATIVE THEOREMS

### Real-RAM Variation Lower Bound

**Theorem B.52** (Real-RAM variation lower bound; axiom closure $\{S2, P1\text{–}P4, P5_{\mathrm{dir}}, \text{TAME}, E1, E4, M\}$)**.** *Assume the arithmetic world is Real-RAM (E1). Fix $x \in \mathcal{X}$, $v \in \mathbb{S}^{d-1}$, and $r > 0$ such that all pack-supplied contracts invoked below are valid on the ray segment $\{x + tv : t \in [0, r]\}$. Assume:*

- *structural nonlinearity at cuts (S2),*
- *primitive postulates (P1)–(P4),*
- *a directional nonlinearity certificate (P5$_{\mathrm{dir}}$) for the designated bottleneck primitive on this ray,*
- *TAME,*
- *a tail-gain contract (E4) in one of the declared variants (E4-$\sigma$-cell, E4-$\sigma$-ray, or E4-dir),*
- *merge isolation (M), yielding a merge floor $\underline{s}_0$ on the relevant suffix family.*

*Then the total variation along the ray satisfies*

$$\mathrm{TV}[N_\theta; x, v, r] \;\geq\; c \cdot \underline{s}_0(x, v, r) \cdot \Gamma_{E4}(x, v, r) \cdot \nu^{\mathrm{dir}}(x, v, r) \cdot \rho(x, v, r) \cdot r,$$

*where $c > 0$ is a universal constant (depending only on the norm conventions fixed in the pack), $\underline{s}_0$ is the merge floor from (M), $\Gamma_{E4}$ is the tail gain from the chosen E4 variant, and $(\nu^{\mathrm{dir}}, \rho)$ come from the (P5$_{\mathrm{dir}}$) certificate for the bottleneck primitive.*

*Proof.* Appendix C.2.2.

**Corollary B.53** (Multi-suffix amplification; axiom closure $\{S2, P1\text{–}P4, P5_{\mathrm{dir}}, \text{TAME}, E1, E4, M\}$)**.** *Under the hypotheses of Theorem B.52, suppose the E4 suffix family provides $k \geq 1$ pairwise-disjoint valid suffixes whose certified contributions can be aggregated without cancellation under (M). Then the bound improves by a $\sqrt{k}$ factor:*

$$\mathrm{TV}[N_\theta; x, v, r] \;\geq\; c \cdot \sqrt{k} \cdot \underline{s}_0 \cdot \Gamma_{E4} \cdot \nu^{\mathrm{dir}} \cdot \rho \cdot r,$$

*after substituting the $k$-suffix aggregate gain for $\Gamma_{E4}$.*

*Proof.* Appendix C.2.2.

*Remark* B.54 (Why (S1) is omitted from the closure)**.** By Lemma B.38, (S1) is implied by $\{P1, P2, P3, P4\}$ under standard packs. We list the closure in compressed form to emphasize which assumptions are structurally necessary (S2) and which are quantitative contracts (P5$_{\mathrm{dir}}$, E4, M).

### IEEE-754 Variation Lower Bound

**Definition B.55** (Event certificate along a ray (E4-event))**.** Assume the arithmetic world is IEEE-754 (E1). Fix $(x, v, r)$ and define the ray-restricted output trace

$$g(t) := N_\theta(x + tv), \qquad t \in [0, r].$$

We say that *E4-event holds on* $(x, v, r)$ if $g$ is a step function on $[0, r]$ with a finite event set $\mathsf{Ev}(x, v, r) \subset (0, r]$ of jump times such that:

(i) (**finite events**) $|\mathsf{Ev}(x, v, r)| < \infty$ and $g$ is constant on each connected component of $[0, r] \setminus \mathsf{Ev}(x, v, r)$;

(ii) (**left limits**) for each $t \in \mathsf{Ev}(x, v, r)$ the left limit $g(t^-) := \lim_{s \uparrow t} g(s)$ exists;

(iii) (**positive jump floor**) the minimum jump size

$$\Delta_{\min}(x, v, r) := \min_{t \in \mathsf{Ev}(x,v,r)} \|g(t) - g(t^-)\|$$

is well-defined (in particular, $\mathsf{Ev}(x, v, r) \neq \varnothing$) and strictly positive.

This is the IEEE-754 event-world analogue of the Real-RAM tail-gain condition (E4-$\sigma$/E4-dir).

**Theorem B.56** (IEEE-754 event-based lower bound; axiom closure $\{S2, P1\text{–}P4, \text{TAME}, E1, E4\text{-event}, M\}$)**.** *Assume the arithmetic world is IEEE-754 (E1). Fix $(x, v, r)$ and assume (S2), (P1)–(P4), TAME, merge isolation (M), and that the event certificate E4-event (Definition B.55) holds on $(x, v, r)$. Let $\kappa_{x,v,r} := |\mathsf{Ev}(x, v, r)|/r$ denote the event density. Then the total variation along the ray satisfies*

$$\mathrm{TV}[N_\theta; x, v, r] \geq \sum_{t \in \mathsf{Ev}(x,v,r)} \|g(t) - g(t^-)\| \geq |\mathsf{Ev}(x, v, r)| \cdot \Delta_{\min}(x, v, r) = \kappa_{x,v,r} \cdot r \cdot \Delta_{\min}(x, v, r).$$

*Proof.* Appendix C.2.3.

*Remark* B.57 (Certification gap (made explicit)). The inequality is formally clean; the practical bottleneck is certifying or reliably estimating $\kappa_{x,v,r}$ and $\Delta_{\min}(x, v, r)$ for deployed kernels with mixed precision and fused implementations. These issues are treated as whitespace in §B.8.3.

### B.7.3. Consistency and Independence

#### Generic Influence for LReLU Blocks

**Lemma B.58** (Degeneracy set is Lebesgue-null (LReLU block))**.** *Consider the two-layer LReLU running example from §B.1, with fixed $\alpha \in (0, 1)$. There exists an explicitly constructible algebraic set $\mathcal{E} \subset \Theta$ (depending on $(d, h, m)$ and the architecture wiring) such that $\mathcal{E}$ has Lebesgue measure zero in $\mathbb{R}^p$.*

*Proof.* Appendix C.3.1.

**Lemma B.59** (Generic influence outside the degeneracy set)**.** *Under the hypotheses of Lemma B.58, for every $\theta \in \Theta \setminus \mathcal{E}$ the running-example network exhibits generic influence in the sense required by the active-graph definitions: for $\mu$-almost every input $x$, the activation pattern is nondegenerate (no pre-activation lies exactly on a boundary), and all structurally present edges in $\mathcal{G}^{\mathrm{sat}}$ are active along a neighborhood of $x$.*

*Proof.* Appendix C.3.1.

#### Consistency

**Theorem B.60** (Consistency of the core stack)**.** *There exist a pack $\Pi$ and a network family $N_\theta$ such that (S1), (S2), and (C2) hold simultaneously. In particular, under the matching-library default pack $\mathcal{O} = \mathcal{T}$ with Real-RAM arithmetic, the two-layer LReLU running example satisfies the core semantics and is exactly realizable as a straight-line program.*

*Proof.* Appendix C.3.2.

#### Independence

**Lemma B.61** (Irrational-slope witness)**.** *Fix $\alpha \in (0, 1)$ and consider the scalar map*

$$N(x) := \mathrm{LReLU}_\alpha(\pi x).$$

*Under Real-RAM, $N$ is a well-defined Borel function and satisfies structural nonlinearity: there is no affine-only bypass, since the unique path passes through the LReLU nonlinearity.*

*Proof.* Appendix C.3.3.

**Theorem B.62** ((S1)+(S2) do not imply (C2) (relative to $\mathcal{O}$))**.** *There exists a pack $\Pi$ with a restricted primitive library $\mathcal{O}$ (e.g. rational-coefficient affine primitives) such that a network can satisfy (S1) and (S2) under the pack's ideal semantics, but fail (C2) (no exact finite straight-line realization over $\mathcal{O}$). In particular, the irrational-slope witness of Lemma B.61 is not realizable over rational-coefficient affine primitives.*

*Proof.* Appendix C.3.3.

*Remark* B.63 (Scope of independence). The independence statement is *relative* to the choice of primitive library $\mathcal{O}$. Under the matching-library default pack $\mathcal{O} = \mathcal{T}$, (C2) holds trivially. Independence matters precisely in worlds where $\mathcal{O}$ is restricted by deployment arithmetic, compiler constraints, or permitted kernel sets.

### B.7.4. CASE STUDIES

This subsection instantiates the structural lens on two modern architecture families: pre-norm transformers and mixture-of-experts (MoE) models with conditional routing. The goal is to show how $\lambda_x$ and the "min-cut bottleneck" picture can be read off from the active structural graph $\mathcal{G}_x^{\mathrm{act}} \subseteq \mathcal{G}^{\mathrm{sat}}$ (defined in §B.4). All statements below are *structural*: they are graph consequences under explicit hypotheses about wiring and activity, and they do not require quantitative contracts (P5/E4/M).

**Transformers: Pre-Norm Bottleneck**

**Terminology.** A *transformer block* consists of two residual sublayers: multi-head attention (MHA) and a positionwise feedforward network (FFN), each wrapped with residual addition and (in common implementations) a normalization. We use the standard conventions:

- **Pre-norm:** normalization is applied *before* each residual branch computation.
- **Post-norm:** normalization is applied *after* the residual addition.
- **Norm types:** LayerNorm (LN) and RMSNorm (RMS).

**Convention B (normalization is nonaffine).** Throughout this subsection, we adopt *Convention B*: data-dependent normalizations (LN/RMS) are tagged `nonaffine` in the type library. This is the convention under which normalization is an explicit nonlinearity/bottleneck in $\mathcal{G}^{\mathrm{sat}}$.

**Schematic (pre-norm).** At the level of functions (ignoring dropout), a pre-norm residual sublayer has the form

$$x \longmapsto x + \mathsf{Block}(\mathrm{Norm}(x)),$$

where $\mathsf{Block}$ is either MHA or FFN, and $\mathrm{Norm}$ is LN/RMS. In $\mathcal{G}^{\mathrm{sat}}$, affine-only regions inside $\mathsf{Block}$ are contracted, while `nonaffine` types (e.g., Norm, Softmax, GELU) remain as vertices.

**A structural sufficient criterion for $\lambda_x = 1$.** The next proposition is deliberately stated as a theorem about $\mathcal{G}_x^{\mathrm{act}}$: it is true whenever the active graph satisfies simple, checkable wiring hypotheses. This avoids arguing about *which* implementation choices place the block into this regime; that engineering variability is exactly what the experiments in §5 audit.

**Proposition B.64** (Pre-norm transformer bottleneck; $\lambda_x = 1$ on $\mathcal{G}_x^{\mathrm{act}}$). *Fix a pack $\Pi$ satisfying* TAME *and adopt Convention B. Let $x \in \mathcal{X}$ be an input, let $\mathcal{G}_x^{\mathrm{act}} \subseteq \mathcal{G}^{\mathrm{sat}}$ be the active saturated graph, and let $\mathcal{T}_{\mathrm{act}}(x)$ be its active sink set. Assume there exists a designated normalization vertex $v_{\mathrm{norm}} \in V(\mathcal{G}^{\mathrm{sat}})$ (tagged `nonaffine`) such that for every active sink $\tau \in \mathcal{T}_{\mathrm{act}}(x)$ the following hold in $\mathcal{G}_x^{\mathrm{act}}$:*

*(i)* ***Unique entry from the source block.*** *There is a unique edge*

$$e_{\mathrm{norm}} = (S^{\mathrm{sat}} \to v_{\mathrm{norm}})$$

*and every directed $S^{\mathrm{sat}}$–$\tau$ path in $\mathcal{G}_x^{\mathrm{act}}$ contains $e_{\mathrm{norm}}$.*

*(ii)* ***Fan-out after normalization.*** *The vertex $v_{\mathrm{norm}}$ reaches $\tau$ in $\mathcal{G}_x^{\mathrm{act}}$ (i.e., there exists at least one directed $v_{\mathrm{norm}}$–$\tau$ path).*

*(iii)* ***No bypass around normalization.*** *There is no directed $S^{\mathrm{sat}}$–$\tau$ path in $\mathcal{G}_x^{\mathrm{act}}$ that avoids $v_{\mathrm{norm}}$.*

*(iv)* ***Nontrivial activity.*** $\mathcal{T}_{\mathrm{act}}(x) \neq \varnothing$.

*Then for every active sink $\tau \in \mathcal{T}_{\mathrm{act}}(x)$,*

$$\mathrm{mincut}(S^{\mathrm{sat}}, \tau; \mathcal{G}_x^{\mathrm{act}}) = 1, \qquad \textit{and hence} \qquad \lambda_x := \min_{\tau \in \mathcal{T}_{\mathrm{act}}(x)} \mathrm{mincut}(S^{\mathrm{sat}}, \tau; \mathcal{G}_x^{\mathrm{act}}) = 1.$$

*Moreover, the cut $\{e_{\mathrm{norm}}\}$ is a minimum cut and is nonaffine-incident (since it is incident to $v_{\mathrm{norm}}$), so* (S2) *holds for $(x, \tau)$ with witness edge $e_{\mathrm{norm}}$.*

*Axiom closure.*
$$\{\mathrm{TAME}\} \ + \ \text{Convention B} \ + \ \text{hypotheses (i)–(iv) on } \mathcal{G}_x^{\mathrm{act}}.$$

(Neither (C2) nor quantitative axioms are used.)

*Proof.* Hypothesis (i) implies $\{e_{\mathrm{norm}}\}$ is an $S^{\mathrm{sat}}$–$\tau$ cut, so $\mathrm{mincut}(S^{\mathrm{sat}}, \tau; \mathcal{G}_x^{\mathrm{act}}) \leq 1$. Hypotheses (ii)–(iv) ensure $\tau$ is reachable from $S^{\mathrm{sat}}$ in $\mathcal{G}_x^{\mathrm{act}}$, so every $S^{\mathrm{sat}}$–$\tau$ cut has size at least 1. Thus the min-cut value is exactly 1, and the witness cut is nonaffine-incident by Convention B. $\square$

*Remark* B.65 (Implications for variation bounds). When Theorem B.52 (Real-RAM lower bound) applies, the structural factor derived from $\lambda_x$ does *not* grow with depth in this regime: $\lambda_x = 1$ for every block input where the proposition applies. Hence depth enters the bound only through quantitative contracts (tail gain, directional nonlinearity, and merge isolation), not via an accumulating structural bottleneck count. This is one precise sense in which pre-norm wiring "stabilizes" the structural side of variation propagation.

*Remark* B.66 (Post-norm blocks). In post-norm wiring, the normalization appears *after* the residual merge, and the structural hypotheses of Proposition B.64 may fail: there need not be a unique normalization entry edge that every active path must traverse. In that regime, the min-cut bottleneck in $\mathcal{G}_x^{\mathrm{act}}$ can shift to other nonaffine types (e.g., attention Softmax or FFN activation), and the bottleneck census can differ materially. This is precisely why §5 audits both pre-norm and post-norm implementations.

*Remark* B.67 (Convention A and "affine" normalization). If LN/RMS is tagged `affine` (Convention A), then $e_{\mathrm{norm}}$ is no longer nonaffine-incident, and the cut witness above ceases to certify (S2). In that world, (S2) must be enforced by other nonaffine operations on *all* active routes (e.g., explicit nonlinear gates on skip routes), or else affine bypasses can appear. The framework is designed so that this modeling choice is explicit and empirically auditable.

*Proof reference.* A fuller discussion (including template conditions ensuring (i)–(iii) in standard pre-norm implementations) appears in Appendix C.4.

## MoE and Conditional Routing

**Setup.** Mixture-of-experts (MoE) layers augment a transformer block by replacing a dense FFN with $n$ experts $\{\mathsf{E}_1, \ldots, \mathsf{E}_n\}$ and a learned router. A typical tokenwise form (suppressing token indices) is

$$x \longmapsto x + \sum_{i \in \mathrm{TopK}(x)} g_i(x)\, \mathsf{E}_i(\mathrm{Norm}(x)),$$

where $g(x)$ is produced by a router (often a Softmax followed by Top-$k$ selection), $\mathrm{TopK}(x)$ is the set of selected experts, and only selected experts are evaluated. Thus the realized computation graph is explicitly input-dependent.

**Router is nonaffine; inactive experts are inactive.** In packs aligned with standard implementations, the router contains nonaffine primitives (Softmax, Top-$k$ selection, or equivalent conditional routing). Consequently:

- router edges are nonaffine-incident by type tagging;
- experts not selected by $\mathrm{TopK}(x)$ have *zero local influence* on the output for that input, and are excluded from $\mathcal{G}_x^{\mathrm{act}}$ by the influence predicate (equivalently: their outgoing contribution is identically zero).

**Cut-size upper bound from Top-$k$ routing.** The next proposition is the clean structural statement you want in the atlas: the number of active experts per routing decision upper-bounds the source-to-sink connectivity.

**Proposition B.68** (Top-$k$ conditional routing implies $\lambda_x \leq k$). *Fix a pack $\Pi$ satisfying* TAME *and whose type tags mark the router as* `nonaffine` *(e.g., Softmax and Top-$k$ selection are* `nonaffine` *types). Consider an MoE layer with $n$ experts and tokenwise Top-$k$ routing.*

*Fix an input $x$ and an active sink $\tau \in \mathcal{T}_{\mathrm{act}}(x)$ whose value depends on a single routing decision (e.g., a single token output in a block, or a single sequence-level head after pooling). Assume that, for this $(x, \tau)$, the router activates a set $A(x, \tau) \subseteq \{1, \ldots, n\}$ of experts with $|A(x, \tau)| \leq k$, and that every $S^{\mathrm{sat}}$–$\tau$ path in $\mathcal{G}_x^{\mathrm{act}}$ that enters the expert bank must pass through exactly one router-to-expert entry edge*

$$e_i:\ v_{\mathrm{router}} \to v_{\mathsf{E}_i}, \qquad i \in A(x, \tau),$$

*with no alternative entry into expert $i$.*

*Then*

$$\mathrm{mincut}(S^{\mathrm{sat}}, \tau; \mathcal{G}_x^{\mathrm{act}}) \ \leq\ |A(x, \tau)| \ \leq\ k, \qquad \textit{and hence} \qquad \lambda_x \ \leq\ k.$$

*Axiom closure.*

$$\{\text{TAME}\} \ + \ (\text{router tagged nonaffine}) \ + \ \text{Top-}k \text{ routing hypothesis above}.$$

*Proof.* Remove the set of router-to-expert entry edges $\{e_i : i \in A(x, \tau)\}$. By hypothesis, this severs all expert-bank routes from $S^{\mathrm{sat}}$ to $\tau$, so it is an $S^{\mathrm{sat}}$–$\tau$ cut. Therefore the min-cut value is at most the cut size $|A(x, \tau)| \leq k$. □

*Remark* B.69 (When can $\lambda_x = k$ hold?). The inequality $\lambda_x \leq k$ is robust: it follows from exhibiting a cut of size $k$. Equality $\lambda_x = k$ is *not* automatic. It requires that no smaller cut exists, which in turn needs a kind of *cut-independence* or *path diversification* assumption: informally, the $k$ active expert branches must provide $k$ "independent" source-to-sink routes that cannot be simultaneously blocked by cutting fewer than $k$ edges (e.g., no earlier shared chokepoint, and no alternative bypass route that makes the expert bank irrelevant to $\tau$). In practice, shared upstream bottlenecks (such as normalization conventions) or downstream merges can reduce $\lambda_x$ below $k$ even when $k$ experts are active.

*Remark* B.70 (What this does *not* prove). Proposition B.68 is a *structural* statement about connectivity. To turn it into a quantitative variation guarantee, one still needs: (i) a directional nonlinearity certificate for at least one nonaffine bottleneck primitive (P5$_{\mathrm{dir}}$), (ii) an E4 tail-gain contract (or event certificate in IEEE-754 worlds), and (iii) merge isolation (M), which is especially delicate in MoE because outputs are sums of routed expert contributions. A full quantitative MoE treatment is deferred to the whitespace program in §B.8.3.

*Proof reference.* A fuller discussion (including multi-token routing and expert-bank sharing patterns that affect equality) appears in Appendix C.5.

The proof ledger records (i) theorem-to-axiom closures, (ii) dependency structure among lemmas, and (iii) the location of each proof artifact, so the paper can be audited end-to-end.

*Table 37.* Proof ledger: axiom–result dependency matrix $D$ for §4. **X** = assumed; **thm** = discharged by the row.

| Result (from §4) | S1 | S2 | C2 | P1 | P2 | P3 | P4 | P5$_{\mathrm{dir}}$ | TAME | E1 | E2 | E3 | E4 | M |
|---|---|---|---|---|---|---|---|---|---|---|---|---|---|---|
| Lemma 4.1.1 (Measurability of $N_\theta$; discharges S1) | thm | | | X | X | X | X | | | | | | | |
| Lemma 4.1.2 (Borel selection of $x \mapsto \mathcal{G}_x^{\mathrm{act}}$; discharges E2 under TAME) | | | | X | X | X | X | | X | | thm | | | |
| Lemma 4.1.3 (Local constancy of active structure; min-cuts; measurable radii) | | | | X | X | X | X | | X | | | | | |
| Lemma 4.1.4 (Nonlinear bottleneck consequences of S2; graph facts) | | | X | | | | | | | | | | | |
| Theorem 4.1.5 (Kink creation / smoothness-on-cells; qualitative) | | | | | | | | | X | | | | | |
| Theorem 4.2.1 (Real-RAM variation lower bound) | | X | | X | X | X | X | X | X | X | | | E4-$\sigma$/dir | X |
| Corollary 4.2.1 (Multi-suffix amplification) | | X | | X | X | X | X | X | X | X | | | E4-$\sigma$/dir | X |
| Theorem 4.2.2 (IEEE-754 variation lower bound; event form) | | X | | X | X | X | X | | X | X | | | E4-quant | X |
| Lemma 4.3.1 (Generic influence for LReLU blocks; pack-specific witness) | | | | | | | | | | | | | | |
| Theorem 4.3.2 (Consistency: joint satisfiability witness) | | | | | | | | | | | | | | |
| Theorem 4.3.3 (Independence: S1+S2 $\not\Rightarrow$ C2; irrational witness) | | | | | | | | | | | | | | |
| Proposition 4.4.1 (Transformer pre-norm: $\lambda_x = 1$ on $\mathcal{G}_x^{\mathrm{act}}$ under explicit wiring hypotheses) | | | | | | | | | X | | | | | |
| Proposition 4.4.2 (MoE Top-$k$: $\lambda_x \le k$ under routing-entry hypothesis) | | | | | | | | | X | | | | | |

## B.8. Proof Ledger and World Enumeration

**Numbering convention.** The ledger preserves the original source-outline labels 4.1.x–4.4.x for row names. The rendered appendix theorem statements use Appendix B counters, while the corresponding main-text quantitative bounds are Theorems 3.2 and 3.3.

**Guardrail (organizational device, not exhaustiveness).** We include the proof ledger and world enumeration as *organizational devices* for tracking axiom dependencies and identifying unexplored regimes. We do *not* claim exhaustive enumeration of axiom worlds, automated proof discovery, or completeness of the ledger beyond the results stated in §B.7–§B.7.4. The purpose is to make the dependency structure explicit and to surface high-leverage "whitespace" where small axiom relaxations qualitatively change what can be certified.

**Where the extended analysis belongs.** This section is intentionally compact. A fully expanded ledger (per-proof micro-dependencies, alternative closures, and a larger world catalog) is best placed in a dedicated appendix or supplement to avoid interrupting the main narrative.

### B.8.1. PROOF LEDGER

The *proof ledger* is a dependency matrix that records which axioms are invoked by each result in §4. It is intended as a reading aid and as a way to identify *Pareto-minimal* axiom closures.

**How to read the ledger.** Table 37 has: (i) one row per named lemma/theorem/proposition in §4, and (ii) one column per axiom label in the core universe

$$\mathcal{A} := \{S1, S2, C2, P1, P2, P3, P4, P5_{\mathrm{dir}}, TAME, E1, E2, E3, E4, M\}.$$

An entry **X** indicates the axiom is assumed in the stated closure of that result. Special entries:

- **thm** indicates the row *discharges* that axiom from other assumptions (e.g., Lemma 4.1.1 discharges S1 from P1–P4).
- A structured entry in the E4 column (e.g. "E4-$\sigma$/dir" or "E4-quant") indicates the *variant* of the E4 contract used by that quantitative result.

**Derived-axiom convention (S1 and E2).** In the main text we sometimes cite (S1) or (E2) as shorthand hypotheses. Formally, in the default development these are *derived*:

- Lemma 4.1.1 establishes (S1) from {P1,P2,P3,P4}.
- Lemma 4.1.2 establishes (E2) from {P1,P2,P3,P4,TAME}.

Thus, later results that require measurability may list S1/E2 for readability even though their true closure is expressed in the primitive/tameness columns.

*Remark* B.71 (Pareto-minimal closures). For a result $R$, a closure $A \subseteq \mathcal{A}$ is *Pareto-minimal (within this paper)* if $R$ is proved assuming $A$ and there is no strict subset $A' \subsetneq A$ for which $R$ is proved here. The ledger makes Pareto comparisons visually immediate; e.g., Theorem 4.2.2 drops P5$_{\mathrm{dir}}$ but requires the E4-quant world, while Theorem 4.2.1 requires P5$_{\mathrm{dir}}$ and an E4-$\sigma$/dir contract.

### B.8.2. WORLD ENUMERATION

A *world* is an axiom subset $W \subseteq \mathcal{A}$ together with declared *modifiers* (e.g., Real-RAM vs IEEE-754, Convention A vs B, and the E4 variant). In principle there are $2^{|\mathcal{A}|} - 1$ nonempty subsets, so enumeration is not the goal. Instead we list representative worlds that align with common analysis regimes and with the main results of §4.

**Modifier axes emphasized here.** The most consequential world modifiers in this paper are:

- **Arithmetic axis (E1):** Real-RAM vs IEEE-754.
- **Tail axis (E4):** E4-$\sigma$-cell vs E4-$\sigma$-ray vs E4-dir vs E4-quant.

*Table 38.* Representative explored worlds (not exhaustive). Modifiers are part of the world declaration.

| World name | Axioms/modifiers | Primary yield |
|---|---|---|
| **Full Real-RAM world** | $\{S2, P1, P2, P3, P4, P5_{\mathrm{dir}}, \mathrm{TAME}, E1, E4, M\}$ with E1=Real-RAM and E4=($\sigma$-cell / $\sigma$-ray / dir) | Structural + quantitative bounds (Theorem 4.2.1) |
| **Structural world** | $\{P1, P2, P3, P4, \mathrm{TAME}\}$ plus (optional) S2 as a design constraint | Measurability, active-graph selection, local constancy (Lemmas 4.1.1–4.1.3) |
| **S2-only diagnostic world** | $\{S2\}$ (graph-theoretic consequences once $\mathcal{G}_x^{\mathrm{act}}$ is defined) | Nonlinear bottleneck facts (Lemma 4.1.4) |
| **IEEE-754 event world** | $\{S2, P1, P2, P3, P4, \mathrm{TAME}, E1, E4\text{-quant}, M\}$ with E1=IEEE-754 | Event-form lower bound (Theorem 4.2.2) |
| **Convention comparison world** | Convention A vs B modifier for tagging normalization (LN/RMS) | Transformer bottleneck visibility / failure modes (§4.4.1) |
| **Cycle/implicit world (optional)** | Activate E3 (bounded unrolling or fixed-point semantics) | Enables DEQ/implicit-layer semantics (used in Appendix theory; not required in §4.2) |

- **Tagging axis (conventions):** Convention A vs B (notably for normalization layers).
- **Cycle axis (E3):** bounded unrolling vs fixed-point semantics when $\mathcal{G}^{\mathrm{sat}}$ is not acyclic.

### B.8.3. WHITESPACE: RESEARCH FRONTIERS

**Guardrail (examples, not claims).** We list these as examples of Atlas worlds not covered by the present axiom set, not as claims proved or contributions made in this paper.

**High-leverage unexplored regimes.** The ledger and world table highlight several frontiers where modest axiom additions/relaxations could qualitatively expand what can be certified:

- **Relaxing TAME.** Replace definable piecewise-$C^1$ structure by weaker regularity (e.g. Lipschitz a.e. without definable stratifications), or admit primitives whose cell structure is not o-minimal. This would broaden applicability but requires new proof infrastructure.
- **Stochastic execution worlds.** Extend S1 from deterministic maps to Markov kernels (dropout, stochastic depth, randomized routing), and define probability-qualified analogs of S2/E4/M.
- **Implicit layers and fixed points.** Strengthen E3 to cover DEQs and solver-based primitives with certifiable contraction or convergence budgets, and connect those contracts to variation bounds.
- **MoE full quantitative analysis.** Beyond the $\lambda_x \leq k$ structural statement (§B.7.4), a full bound likely needs additional structural hypotheses (e.g. cut-independence) and merge/leakage variants of M to control cancellation when multiple experts contribute.
- **Global variation certificates.** Move beyond cellwise/raywise bounds to global paths crossing many cell boundaries, which likely requires axioms controlling boundary crossing counts and transition geometry.

**Takeaway.** The Atlas viewpoint converts an informal space of "assumptions people usually make" into an explicit, composable set of world declarations. The proof ledger makes it clear which axioms are doing work for which theorems, and the whitespace list identifies the smallest changes most likely to yield qualitatively new guarantees.

For completeness, we include the appendix scope statement as the canonical reference for what the Atlas does and does not claim.

## B.9. Scope and Limitations

This section clarifies (i) what is *covered* by the current axiom stack and what can be certified in practice, (ii) which architecture families the framework applies to without changing the core semantics, (iii) the most common ways the structural bottleneck axiom (S2) fails in real code and how such failures present diagnostically, and (iv) fundamental limitations and non-goals.

Unless explicitly stated otherwise, all claims are interpreted relative to a fixed pack $\Pi$ (§2.1) and use the graph/active-graph semantics from §B.4. Empirical checks based on realized execution use the proxy active graph $\widehat{\mathcal{G}^{\mathrm{act}}}_x$ and are recorded as `proxied` in certificates (§B.6).

### B.9.1. COVERAGE SUMMARY

The framework separates what is *structurally* checkable from what is *quantitatively* lower-bounded with certifiable constants.

**Structural guarantees (no quantitative constants).** The following results are structural in the sense that they do not require P5, E4, or Assumption M:

- **Well-posedness / measurability.** Under the primitive closure stack $\{P1–P4\}$, the induced map $N_\theta$ is Borel measurable (Lemma 4.1.1), discharging (S1).
- **Active-graph measurability and local constancy (Real-RAM worlds).** Under $\{P1–P4, \text{TAME}\}$, the map $x \mapsto \mathcal{G}_x^{\text{act}}$ is Borel measurable (Lemma 4.1.2), discharging (E2) under TAME, and is locally constant on refined cells almost everywhere (Lemma 4.1.3).
- **Bottleneck existence and graph-cut diagnostics.** Given $\mathcal{G}_x^{\text{act}}$, one can compute $\lambda_{x,\tau} = \text{mincut}(S^{\text{sat}}, \tau; \mathcal{G}_x^{\text{act}})$ and $\lambda_x = \min_\tau \lambda_{x,\tau}$, and check (S2) exactly on $\mathcal{G}_x^{\text{act}}$ (via max-flow/min-cut; §B.6.2).
- **Qualitative kink structure (Real-RAM worlds).** Under TAME, the network is piecewise-$C^1$ on a definable stratification, and kinks occur at essential boundaries with appropriate directionality hypotheses (Theorem 4.1.5).

**Quantitative variation lower bounds (certifiable constants).**    The *numerical* lower bounds on total variation require additional axioms that supply constants:

- **Real-RAM lower bound.** Theorem 4.2.1 requires $\{S2, P1–P5_{\text{dir}}, \text{TAME}, E1, \text{an E4 tail variant, and Assumption M}\}$. It yields $\text{TV}[N_\theta; x, v, r] \geq c \cdot \underline{s}_0 \cdot \Gamma \cdot \nu \cdot \rho \cdot r$.
- **IEEE-754 (event) lower bound.** Theorem 4.2.2 requires $\{S2, P1–P4, \text{TAME}, E1, \text{the IEEE-754 event tail variant (E4-event), and Assumption M}\}$. It yields $\text{TV} \geq \kappa_x \cdot r \cdot \Delta_{\min}$ in the declared finite-precision world.

**Bounds without certificates (diagnostic only).**    One can compute the *structural* quantities ($\hat{\lambda}_x$, bottleneck census, proxy (S2) rates) at scale using $\widehat{\mathcal{G}^{\text{act}}}_x$ even when P5/E4/M constants are not certified. In that regime, the paper's claims are *organizational and diagnostic*: they identify which axioms/constants are the bottleneck for moving from structure to a numerically meaningful bound.

*Remark* B.72 (What is actually "auditable" depends on status). A certificate should always expose which axioms were `evaluated` versus `proxied` (§B.6.1). Structural audits can be exact given exact $\mathcal{G}_x^{\text{act}}$; large-model audits are typically proxy-based due to the oracle/proxy gap.

### B.9.2. ARCHITECTURE APPLICABILITY

The framework is designed to apply to any architecture whose evaluation admits a typed DAG semantics (or a declared reduction to one, e.g. via bounded unrolling). Table 39 summarizes how the current axiom stack applies across common families.

**Generic under nondegeneracy.**    We say a statement holds *generically under nondegeneracy* if it holds outside a declared exceptional set: either a Lebesgue-null set of inputs (e.g. cell boundaries) and/or a measure-zero set of parameters (e.g. degenerate weights that collapse distinct paths). Under TAME, such exceptional sets are definable and are the natural locus where activity patterns change.

### B.9.3. COMMON (S2) FAILURE MODES

Axiom (S2) prohibits affine-only bypass routes on the *active* graph. When it fails in practice, it almost always fails for one of the following reasons.

**(i) All-affine skip path (architectural bypass).**    A residual or parallel branch can create a path from $S^{\text{sat}}$ to $\tau$ that traverses only affine-tagged nodes after saturation (e.g. an identity skip around all nonlinearities due to wiring or a misplaced normalization convention). This is a *true structural* violation and is typically detectable by S2$_{\text{struct}}$ on $\mathcal{G}^{\text{sat}}$.

**(ii) Deterministic affine routing (conditional computation without nonlinearity).**    If routing/gating is implemented as a fixed affine selection for the evaluated input domain (e.g. router logits saturate, a hard-coded expert is always chosen, or routing is compiled to a constant), then the effective active graph may contain a purely affine route for those inputs.

**(iii) Misclassified normalization or reduction primitives (convention mismatch).**    Treating data-dependent normalizations as affine (Convention A) can introduce affine-only bypasses in pre-norm transformers. Conversely, tagging information-reducing but affine maps (e.g. AvgPool) as nonaffine can mask bypasses. This is not a "bug" in the framework: it is precisely why conventions are made explicit as pack modifiers.

**(iv) Degenerate inputs or parameters (rare, but real).**    Even with correct wiring and conventions, activity can degenerate: uniform attention, zero-variance normalization inputs, or parameter choices that collapse a nonaffine primitive to an affine map. These failures are typically *non-generic* (measure-zero or rare under natural data) but must be acknowledged.

**Detection and reporting.**    We recommend reporting three complementary signals: (i) S2$_{\text{struct}}$ on $\mathcal{G}^{\text{sat}}$ (design-time wiring), (ii) proxy (S2) rates on $\widehat{\mathcal{G}^{\text{act}}}_x$ over natural and synthetic data (execution-time diagnostics), and (iii) calibrated proxy–oracle comparisons on a small model where exact $\mathcal{G}_x^{\text{act}}$ is tractable. This triad makes it hard to overclaim.

### B.9.4. FUNDAMENTAL LIMITATIONS

Some limitations are intrinsic to the present semantic and axiom choices.

*Table 39.* Architecture applicability summary (pack-relative). "Exact" refers to oracle $\mathcal{G}_x^{\text{act}}$; "Proxy" refers to realized-branch $\widehat{\mathcal{G}^{\text{act}}}_x$.

| Family | Examples | Structural lens (S2, $\lambda_x$) | Quantitative bounds (P5/E4/M) |
|---|---|---|---|
| MLP / feedforward | ReLU/GeLU MLPs | Applies directly; $\mathcal{G}^{\text{sat}}$ is a layered DAG. $\lambda_x$ often small (serial bottlenecks). | Requires activation P5$_{\text{dir}}$ and an E4 tail certificate; M can be nontrivial in residual/merge-heavy designs. |
| Transformers | Pre-norm, post-norm, ViT/DiT blocks | Applies directly. Pre-norm + Convention B yields a canonical bottleneck (Proposition 4.4.1). Post-norm differs structurally. | Real-RAM bounds feasible if P5 (for the bottleneck primitive) and E4/M are certified/estimated; IEEE-754 requires event-tail variant. |
| CNN / ResNets | ResNet, ConvNeXt | Applies with a clear $\mathcal{G}^{\text{sat}}$; conventions for pooling/BN affect tagging and thus (S2) interpretation. | Quantitative constants are feasible but depend on conventions (e.g. BN as affine in eval). Merge behavior may complicate M. |
| GNNs | GraphSAGE, GAT | Applies but $\mathcal{G}^{\text{sat}}$ depends on input topology; $\lambda_x$ is topology-dependent. Active sinks are node-level. | Quantitative bounds are possible but constants depend on topology and message-passing operators; often treated structurally/diagnostically (Appendix A). |
| SSMs / recurrence | Mamba, S4/S5 (bounded unroll) | Applies via declared unrolling to a DAG; per-step $\lambda_{x,t}$ is meaningful. True feedback semantics is a whitespace extension. | Quantitative constants feasible per unroll length; costs scale with unroll $T$. Full autoregressive semantics is future work. |
| MoE / routing | Mixtral-like top-$k$ | Structural lens applies: inactive experts drop out of $\mathcal{G}_x^{\text{act}}$; $\lambda_x \leq k$ under router-as-nonlinearity. | Full quantitative theory may require an additional router independence axiom; we treat this as an extension (see §B.8.3). |

**Infinite-width and limiting objects.** The framework is formulated for finite typed graphs and finite-dimensional Euclidean slots (P3). Infinite-width limits (NNGP/NTK regimes) and continuum architectures require separate semantic choices. One can approximate such limits by finite truncations, but the resulting guarantees are about the truncation, not the limit.

**Stochastic networks and randomized execution.** The current stack treats $N_\theta$ as deterministic (S1). Dropout, stochastic depth, sampling-based inference, and randomized routing require replacing S1 by a kernel semantics and re-qualifying S2/E4/M probabilistically. This is explicitly treated as whitespace in §B.8.3.

**Implicit layers and unbounded internal iteration.** DEQs, root-finding layers, and kernels with internal loops violate atomicity unless a bounded unrolling budget is declared. E3 provides optional cycle semantics, but full implicit-layer coverage requires additional convergence contracts and is treated as an extension axis.

**Non-tame primitives and weakened regularity.** TAME is a *postulate* about definable piecewise-$C^1$ stratifications. If the pack includes primitives that are merely Lipschitz-a.e. without definable stratifications, then the measurability/local-constancy machinery must be rebuilt (whitespace item U6).

**Global optimization and training dynamics.** The paper studies *execution-time function variation* along rays, not the dynamics of training. It does not explain why SGD finds good solutions, nor does it characterize the global loss landscape. While certificate quantities can be used as diagnostics during training, that use is downstream and not analyzed here.

**Lower bounds are not "robustness guarantees."** Our main quantitative results are *lower bounds* on variation: they assert that the function changes by at least a certain amount under stated conditions. This is complementary to robustness work (upper bounds on change). Neither direction subsumes the other; together they characterize a feasible variation envelope.

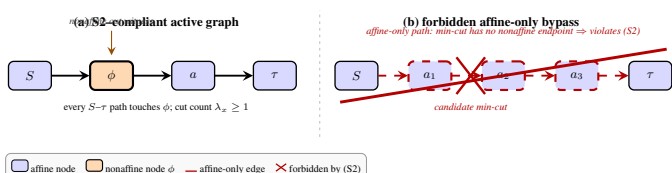

*Figure 3*. Intuition for (S2): every active source–sink path must touch a nonaffine node. Panel (a) shows the compliant orange cut witness $\phi$ and cut count $\lambda_x \geq 1$; panel (b) marks the forbidden affine-only bypass and its invalid minimum cut in red. This is the obstruction formalised by Lemma B.45.

### B.9.5. NON-GOALS

To avoid category errors, we state explicitly what this paper is not trying to do.

**Not an optimization or architecture-search paper.** The atlas/world framing and whitespace analysis identify regimes where certification machinery is missing. They do not, by themselves, specify an objective function, training procedure, or search algorithm that yields state-of-the-art performance. Coupling certificates to optimization is promising, but is a separate layer of contribution (and is intentionally deferred unless one can make end-to-end, reproducible performance claims).

**Not a solver for verification queries.** We provide certificate schemas and checkable structural procedures (max-flow/min-cut based), but we do not claim a complete decision procedure for all properties or all networks. Where exact oracle quantities are intractable, we provide explicit proxy semantics and insist on transparent status reporting.

**Not complete coverage of all neural architectures.** The framework is parametric in the pack and is intended to be extensible. Architectures with dynamic control flow, adaptive computation time, complex compilation rewrites, or hardware-specific kernels may require additional pack declarations and new axioms (as cataloged in the whitespace).

**Not an exhaustive world enumeration.** World enumeration and the proof ledger (§B.8) are organizational devices: they track dependencies and highlight unexplored regimes. We do not claim exhaustive enumeration, nor automated proof discovery across all possible axiom subsets.

# C. Proofs

We open the structural proofs with a visual intuition for the nonlinear-cut axiom; see Figure 3.

## C.1. Proofs of Structural Theorems

**Scope.** All statements and proofs in this appendix are interpreted relative to a fixed pack $\Pi$ (§2.1) and its ambient spaces/conventions (§B.3, §B.4). We write $\mathcal{X}$ for the input space, $\mathcal{Y}$ for the output space, and equip all finite-dimensional Euclidean spaces with their Borel $\sigma$-algebras. All references to "a.e." in this appendix mean *Lebesgue-a.e.* on the relevant Euclidean space unless stated otherwise.

### C.1.1. PROOF OF MEASURABILITY (LEMMA B.38)

*Proof of Lemma B.38.* Fix a pack $\Pi$ and an architecture template $\mathcal{G}^{\mathrm{str}} = (V, E)$ as in §B.4. Assume postulates (P1)–(P4).

**Step 1: Nodewise evaluation maps are Borel.** For each node $v \in V$, let $\tau_v := \mathrm{type}(v) \in \mathcal{T}$ denote its type and let $\Xi_{\tau_v}$ denote its parameter space. Write $E_{\mathrm{in},v}$ and $E_{\mathrm{out},v}$ for the corresponding input/output Euclidean slots. By (P3), these are finite-dimensional Euclidean spaces.

Let $f_{\tau_v}^{\mathrm{ideal}} : E_{\mathrm{in},v} \times \Xi_{\tau_v} \to E_{\mathrm{out},v}$ denote the pack-declared node function (ideal semantics) for type $\tau_v$. By (P4) (joint Borel measurability for the relevant atomic maps in the pack), the map $(x, \xi) \mapsto f_{\tau_v}^{\mathrm{ideal}}(x; \xi)$ is Borel measurable. By (P1), parameters are *frozen*: they are supplied by the parameter distributor

$$\mathcal{P} : \Theta \to \prod_{v \in V} \Xi_{\tau_v}, \qquad \theta \mapsto \big(\mathcal{P}(\theta)_v\big)_{v \in V},$$

which is Borel measurable by definition of a pack (§2.1). Thus $\theta \mapsto \mathcal{P}(\theta)_v$ is Borel for each fixed $v$.

**Step 2: Inductive construction of the network map.** Because $\mathcal{G}^{\mathrm{str}}$ is a finite DAG, choose a topological ordering $v_1, \ldots, v_{|V|}$. For each $i$, let $\mathrm{Pred}(v_i) \subseteq \{v_1, \ldots, v_{i-1}\}$ be the set of predecessors of $v_i$. Let $\mathrm{Agg}_i$ denote the (fixed) wiring map that collects predecessor outputs into the input slot of $v_i$:

$$\mathrm{Agg}_i : \prod_{u \in \mathrm{Pred}(v_i)} E_{\mathrm{out},u} \longrightarrow E_{\mathrm{in},v_i}.$$

Such $\mathrm{Agg}_i$ is continuous (indeed linear/projection) because it is just tuple formation and coordinate selection, so it is Borel.

Define the partial evaluation map $H_i : \Theta \times \mathcal{X} \to \prod_{j \leq i} E_{\mathrm{out}, v_j}$ by induction:

- For $i = 1$, define

$$H_1(\theta, x) := f^{\mathrm{ideal}}_{\tau_{v_1}}\big(\mathrm{Agg}_1(x); \mathcal{P}(\theta)_{v_1}\big),$$

  where $\mathrm{Agg}_1$ extracts the required input coordinates (often $\mathrm{Agg}_1(x) = x$ for the source block). This is Borel as a composition of Borel maps.
- Assume $H_{i-1}$ is Borel and write $H_{i-1}(\theta, x) = (y_{v_1}, \ldots, y_{v_{i-1}})$. Define

$$y_{v_i}(\theta, x) := f^{\mathrm{ideal}}_{\tau_{v_i}}\Big(\mathrm{Agg}_i\big((y_u)_{u \in \mathrm{Pred}(v_i)}\big) ; \mathcal{P}(\theta)_{v_i}\Big),$$

  and set $H_i(\theta, x) := (H_{i-1}(\theta, x), y_{v_i}(\theta, x))$. Each operation is Borel (projection, $\mathrm{Agg}_i$, $\mathcal{P}$, and $f^{\mathrm{ideal}}_{\tau_{v_i}}$), hence $H_i$ is Borel.

Since $|V| < \infty$, we obtain a Borel map $H_{|V|}(\theta, x)$ giving all node outputs.

**Step 3: Extract the network output.**  Let $\tau$ denote the designated sink/output node(s) (or output readout) in $\mathcal{G}^{\mathrm{str}}$. The network output map $N : \Theta \times \mathcal{X} \to \mathcal{Y}$ is obtained from $H_{|V|}$ by a coordinate projection and/or a continuous readout map (as specified by the architecture template), hence is Borel.

**Step 4: Discharge (S1).**  Fix any $\theta \in \Theta$. The section map $x \mapsto N(\theta, x)$ is Borel measurable as a section of a Borel map. Thus (S1) holds: for every $\theta$, $N_\theta : \mathcal{X} \to \mathcal{Y}$ is a well-defined Borel function. □

### C.1.2. PROOF OF ACTIVE GRAPH BOREL SELECTION (LEMMA B.40)

*Proof of Lemma B.40.*  Assume (P1)–(P4) and (TAME). Fix $\theta \in \Theta$ and consider the saturated graph $\mathcal{G}^{\mathrm{sat}}$ for the architecture (§B.4).

By definition of the active graph $\mathcal{G}^{\mathrm{act}}_x \subseteq \mathcal{G}^{\mathrm{sat}}$ (§B.4), each edge $e \in E(\mathcal{G}^{\mathrm{sat}})$ has an *activity indicator* of the form

$$\mathbf{1}_{\{e \in \mathcal{G}^{\mathrm{act}}_x\}} = \mathbf{1}_{A_e}(x),$$

where $A_e \subseteq \mathcal{X}$ is the set of inputs $x$ for which $e$ is active (according to the pack-declared influence predicate; equivalently, according to the path-closed activity rule).

Thus, it suffices to show that each $A_e$ is Borel, since then $x \mapsto \mathbf{1}_{A_e}(x)$ is Borel for every edge $e$, and the map $x \mapsto \mathcal{G}^{\mathrm{act}}_x$ (viewed as a finite vector of edge indicators) is Borel.

**TAME reduces activity predicates to definable sets.**  Under (TAME), the network map $N_\theta$ and all pack-declared primitives appearing in the influence predicate admit a definable piecewise-$C^1$ stratification of $\mathcal{X}$ into refined cells (§B.3). In particular, every regime predicate used to define edge activity (e.g. sign tests for piecewise primitives, argmax stability for discrete selections, non-vanishing of a Jacobian block, or other pack-declared influence tests) is definable on $\mathcal{X}$ and hence Borel.

Concretely, by (TAME) there exists a locally finite definable stratification $\mathcal{X} = \bigsqcup_{i \in I} C_i \sqcup B$ where: (i) each $C_i$ is a $C^1$ manifold (a refined cell) of full dimension, (ii) $B$ is a definable set of lower dimension (hence Lebesgue-null), and (iii) on each refined cell $C_i$, the combinatorial regime of every pack primitive is constant.

Since $\mathcal{G}^{\mathrm{act}}_x$ is defined by finitely many such regime predicates and finitely many Boolean operations (union/intersection/negation) across a finite graph, each set $A_e$ is definable as a union of a subcollection of refined cells plus (optionally) a subset of $B$. Therefore $A_e$ is definable, hence Borel.

**Conclusion.**  For each edge $e$, $A_e$ is Borel, so $x \mapsto \mathbf{1}_{\{e \in \mathcal{G}^{\mathrm{act}}_x\}}$ is Borel. Because $E(\mathcal{G}^{\mathrm{sat}})$ is finite, the map $x \mapsto \mathcal{G}^{\mathrm{act}}_x$ is Borel as a map into $\{0, 1\}^{|E(\mathcal{G}^{\mathrm{sat}})|}$. This is exactly the measurability requirement recorded as (E2). □

### C.1.3. PROOFS OF LOCAL CONSTANCY (LEMMAS B.42–B.44)

*Proof of Lemma B.42 (Local constancy of $\mathcal{G}^{\mathrm{act}}_x$ on refined cells).*  Assume (P1)–(P4) and (TAME). Let $\{C_i\}_{i \in I}$ be the refined $C^1$ cells from (TAME) (as used in the proof of Lemma B.40).

By construction of refined cells, *every* regime predicate used by the pack to define activity is constant on each fixed cell $C_i$. Since $\mathcal{G}^{\mathrm{act}}_x$ is computed from these predicates by a finite set of Boolean operations over a finite graph, the entire edge-indicator vector of $\mathcal{G}^{\mathrm{act}}_x$ is constant on each $C_i$. Equivalently: for any $x, x' \in C_i$, $\mathcal{G}^{\mathrm{act}}_x = \mathcal{G}^{\mathrm{act}}_{x'}$. □

*Proof of Lemma B.43 (Local constancy of min-cut values).*  Fix a refined cell $C_i$ and assume $\mathcal{G}^{\mathrm{act}}_x$ is constant on $C_i$ (Lemma B.42). For any active sink $\tau$, the value $\mathrm{mincut}(S^{\mathrm{sat}}, \tau; \mathcal{G}^{\mathrm{act}}_x)$ depends only on the graph $\mathcal{G}^{\mathrm{act}}_x$ (with unit capacities), hence is constant on $C_i$. Taking the minimum over active sinks preserves constancy, so $\lambda_x$ is constant on $C_i$ as well. □

*Proof of Lemma B.44 (Measurable local constancy radius).* Define the *local constancy radius* by

$$r(x) := \sup\left\{\epsilon > 0 : \mathcal{G}_{x'}^{\mathrm{act}} = \mathcal{G}_x^{\mathrm{act}} \text{ for all } x' \in B_\epsilon(x)\right\}.$$

If $x$ lies in the interior of a refined cell $C_i$, then by Lemma B.42 we have $\mathcal{G}_{x'}^{\mathrm{act}} = \mathcal{G}_x^{\mathrm{act}}$ for all $x'$ sufficiently close to $x$, hence $r(x) > 0$. If $x$ lies on the lower-dimensional boundary set $B$ of the stratification, it is possible that $r(x) = 0$.

To prove measurability, note that on the interior of a cell $C_i$, the maximal $\epsilon$ such that $B_\epsilon(x) \subseteq C_i$ is exactly the distance from $x$ to the complement of $C_i$:

$$\mathrm{dist}(x, \mathcal{X} \setminus C_i) := \inf\{\|x - y\| : y \in \mathcal{X} \setminus C_i\}.$$

Because each $C_i$ is definable and open in its induced topology and the stratification is locally finite, the set $\mathcal{X} \setminus C_i$ is closed in a neighborhood of any interior point of $C_i$. The distance-to-a-closed-set function is continuous, hence Borel. On the boundary set $B$ we may define $r(x) = 0$.

Therefore $r$ is Borel measurable as a piecewise-defined function: on each $C_i$ it coincides with a continuous distance function, and on $B$ it is 0. $\square$

## C.1.4. PROOFS OF NONLINEAR BOTTLENECK LEMMAS (LEMMAS B.45–B.47)

*Proof of Lemma B.45 (Nonlinear bottleneck: no affine-only active route).* Fix $x$ such that $\mathcal{T}_{\mathrm{act}}(x) \neq \varnothing$ and fix an active sink $\tau \in \mathcal{T}_{\mathrm{act}}(x)$. Assume (S2) holds at $x$.

We show that there is no directed $S^{\mathrm{sat}}$–$\tau$ path in $\mathcal{G}_x^{\mathrm{act}}$ whose vertices are all affine-tagged (equivalently, whose edges all have both endpoints affine-tagged).

Suppose for contradiction that there exists such an affine-only path $P$ from $S^{\mathrm{sat}}$ to $\tau$ in $\mathcal{G}_x^{\mathrm{act}}$. Because $\tau$ is active, $\mathcal{G}_x^{\mathrm{act}}$ contains at least one $S^{\mathrm{sat}}$–$\tau$ path, hence $\lambda_x(\tau) := \mathrm{mincut}(S^{\mathrm{sat}}, \tau; \mathcal{G}_x^{\mathrm{act}}) \geq 1$.

Pick any edge $e$ on the path $P$. Removing $\{e\}$ disconnects $S^{\mathrm{sat}}$ from $\tau$ (since $e$ lies on a directed path from source to sink), so $\{e\}$ is an $S^{\mathrm{sat}}$–$\tau$ edge cut of size 1. Thus $\lambda_x(\tau) \leq 1$. Combining with $\lambda_x(\tau) \geq 1$ yields $\lambda_x(\tau) = 1$, so $\{e\}$ is a *minimum* cut.

But $e$ has both endpoints affine-tagged by assumption (affine-only path), so $\{e\}$ is a minimum cut containing no nonaffine-incident edge, contradicting (S2). Therefore no affine-only $S^{\mathrm{sat}}$–$\tau$ path exists in $\mathcal{G}_x^{\mathrm{act}}$. $\square$

*Proof of Lemma B.46 (Bounded depth of active paths).* Because $\mathcal{G}^{\mathrm{sat}}$ (and hence $\mathcal{G}_x^{\mathrm{act}} \subseteq \mathcal{G}^{\mathrm{sat}}$) is a finite directed acyclic graph, every directed path contains no repeated vertex. Therefore the length (number of edges) of any directed path is at most $|V(\mathcal{G}^{\mathrm{sat}})| - 1$. Since $|V(\mathcal{G}^{\mathrm{sat}})| \leq |V(\mathcal{G}^{\mathrm{str}})|$ under affine saturation (contraction cannot increase vertex count), the claimed depth bound follows. $\square$

*Proof of Lemma B.47 (Independence of (S2) from (C2)).* We show that (S2) is a purely structural property and can hold even when (C2) fails, by exhibiting a witness pack.

Fix the two-layer LReLU architecture from §B.1 with $\mathcal{G}^{\mathrm{str}}$ containing a single nonaffine node (LReLU) separating affine blocks, so (S2) holds under the natural tagging (LReLU tagged nonaffine).

Now consider a *restricted* primitive library $\mathcal{O}$ in which affine primitives are only allowed to use *rational* coefficients (e.g., parameter space restricted to $\mathbb{Q}^{h \times d} \times \mathbb{Q}^h$ under an encoding), and let the arithmetic model be Real-RAM. Choose parameters $\theta$ that include at least one *irrational* coefficient in $W_1$ or $W_2$. Then the ideal network map $N_\theta$ is well-defined and (by Lemma B.38) Borel; moreover the saturated structure still contains the nonaffine LReLU bottleneck, so (S2) holds.

However, there cannot exist an exact finite SLP over the restricted primitive library $\mathcal{O}$ that computes the irrational-coefficient affine map exactly on $\mathbb{R}^d$, so (C2) fails for this $\theta$. Thus (S2) can hold while (C2) fails, proving independence (relative to the primitive library choice). $\square$

## C.1.5. PROOFS OF KINK CREATION (THEOREMS B.49 AND B.50)

*Proof of Theorem B.49 (Smoothness on constant-activity cells).* Assume (TAME). Then $\mathcal{X}$ admits a locally finite definable stratification into $C^1$ strata such that $N_\theta$ is $C^1$ on each full-dimensional stratum (refined cell).

Let $x$ lie in the interior of a refined cell $C$. Since $N_\theta$ is $C^1$ on $C$, the restriction $N_\theta|_C$ is $C^1$ by definition.

For the total variation formula, consider a ray segment $\{x + tv : t \in [0, r]\}$ that remains entirely within $C$. Define $g(t) := N_\theta(x + tv)$. Since $N_\theta$ is $C^1$ on $C$, the map $g$ is $C^1$ on $[0, r]$ with

$$g'(t) = DN_\theta(x + tv) \cdot v.$$

The total variation of $g$ on $[0, r]$ is

$$\mathrm{TV}[g; 0, r] = \int_0^r \|g'(t)\| \, dt = \int_0^r \|DN_\theta(x + tv) \cdot v\| \, dt = \int_0^r \left\|\frac{d}{dt} N_\theta(x + tv)\right\| dt,$$

where the first equality uses the standard fact that total variation of a $C^1$ curve equals the integral of its speed. $\square$

*Proof of Theorem B.50 (Kinks at essential boundaries).* Assume (TAME). Let $B$ be a codimension-1 boundary stratum separating two adjacent full-dimensional refined cells $C^+$ and $C^-$. Call the boundary *essential* at $x_0 \in B$ if the one-sided Jacobians differ:

$$\mathrm{D}N_\theta(x_0; C^+) \;\neq\; \mathrm{D}N_\theta(x_0; C^-),$$

where $\mathrm{D}N_\theta(x_0; C^\pm)$ denotes the limit of $\mathrm{D}N_\theta(x)$ as $x \to x_0$ within $C^\pm$.

Fix such an $x_0 \in B$ and consider a direction $v$ that is *transverse* to $B$ at $x_0$, i.e., $v \notin T_{x_0}B$. For sufficiently small $\varepsilon > 0$, the points $x_0 + tv$ lie in $C^+$ for $t \in (0, \varepsilon)$ and in $C^-$ for $t \in (-\varepsilon, 0)$ (by transversality and the implicit function theorem for $C^1$ hypersurfaces). Define $g(t) = N_\theta(x_0 + tv)$. Then $g$ is $C^1$ on $(-\varepsilon, 0)$ and on $(0, \varepsilon)$, with one-sided derivatives

$$g'(0^-) \;=\; \mathrm{D}N_\theta(x_0; C^-)\, v, \qquad g'(0^+) \;=\; \mathrm{D}N_\theta(x_0; C^+)\, v.$$

If $g'(0^-) \neq g'(0^+)$, then $g$ has a kink at $t = 0$ (a discontinuity in the derivative).

Since the boundary is essential, $\Delta := \mathrm{D}N_\theta(x_0; C^+) - \mathrm{D}N_\theta(x_0; C^-) \neq 0$. Therefore the set of directions $v$ satisfying $\Delta v = 0$ is a proper linear subspace of $\mathbb{R}^d$, whose intersection with the unit sphere has surface measure zero. Intersecting with the transversality condition $v \notin T_{x_0}B$ (which also excludes a measure-zero set on the sphere), we conclude: for surface-a.e. direction $v$ transverse to $B$ at $x_0$, we have $g'(0^-) \neq g'(0^+)$ and thus a kink. $\square$

**Exceptional directions have measure zero.** The stratification is locally finite, so in any bounded region only finitely many codimension-1 strata contribute. For each boundary stratum, the set of tangent directions (and the set of directions annihilated by $\Delta$) has sphere-measure zero. A finite union of measure-zero subsets is measure-zero, yielding the stated "generic direction" conclusion. $\square$

## C.1.6. SUPPORTING LEMMAS

### Affine saturation preserves nonaffine incidence

**Lemma C.1** (Nonaffine incidence is preserved under affine saturation). *Let $\pi_V : V(\mathcal{G}^{\mathrm{str}}) \to V(\mathcal{G}^{\mathrm{sat}})$ and $\pi_E : E(\mathcal{G}^{\mathrm{str}}) \to E(\mathcal{G}^{\mathrm{sat}})$ denote the quotient maps induced by affine saturation. Then for any edge $e = (u \to v) \in E(\mathcal{G}^{\mathrm{str}})$, if $e$ is nonaffine-incident in $\mathcal{G}^{\mathrm{str}}$ (i.e., $u$ or $v$ is tagged nonaffine), then $\pi_E(e)$ is nonaffine-incident in $\mathcal{G}^{\mathrm{sat}}$. Conversely, if $\bar{e} \in E(\mathcal{G}^{\mathrm{sat}})$ is nonaffine-incident, then every $e \in \pi_E^{-1}(\bar{e})$ is nonaffine-incident in $\mathcal{G}^{\mathrm{str}}$.*

*Proof.* By construction, affine saturation contracts connected affine-only regions into block nodes and never contracts a nonaffine node into an affine block. Thus, if a vertex $w$ is tagged nonaffine in $\mathcal{G}^{\mathrm{str}}$, then $\pi_V(w)$ is a nonaffine node in $\mathcal{G}^{\mathrm{sat}}$. If $e = (u \to v)$ is incident to a nonaffine endpoint in $\mathcal{G}^{\mathrm{str}}$, then $\pi_E(e) = (\pi_V(u) \to \pi_V(v))$ is incident to the corresponding nonaffine endpoint in $\mathcal{G}^{\mathrm{sat}}$.

Conversely, if $\bar{e} = (\bar{u} \to \bar{v})$ in $\mathcal{G}^{\mathrm{sat}}$ is nonaffine-incident, then either $\bar{u}$ or $\bar{v}$ is the image of a nonaffine node in $\mathcal{G}^{\mathrm{str}}$. Any preimage edge $e = (u \to v)$ of $\bar{e}$ must have $u \in \pi_V^{-1}(\bar{u})$ and $v \in \pi_V^{-1}(\bar{v})$. If $\bar{u}$ is nonaffine, then $\pi_V^{-1}(\bar{u})$ contains (at least) that nonaffine node, hence $u$ can be chosen nonaffine for any edge mapping to $\bar{e}$; similarly if $\bar{v}$ is nonaffine. Therefore every $e$ mapping to $\bar{e}$ is incident to a nonaffine node. $\square$

### Series composition min-cut formula

**Lemma C.2** (Min-cut for series composition). *Let $G$ be a directed graph formed by series composition of two subgraphs $G_1$ and $G_2$ with a single interface vertex $u$, in the sense that every $S$–$T$ path in $G$ must pass through $u$, and there are no edges from $G_1 \setminus \{u\}$ to $G_2 \setminus \{u\}$ except through $u$. Then*

$$\mathrm{mincut}(S, T; G) \;=\; \min\big\{\mathrm{mincut}(S, u; G_1),\ \mathrm{mincut}(u, T; G_2)\big\}.$$

*Proof.* Let $c_1 := \mathrm{mincut}(S, u; G_1)$ and $c_2 := \mathrm{mincut}(u, T; G_2)$.

*Upper bound.* Removing a minimum $S$–$u$ cut in $G_1$ disconnects $S$ from $u$ and hence disconnects $S$ from $T$ in $G$. Similarly, removing a minimum $u$–$T$ cut in $G_2$ disconnects $u$ from $T$ and hence disconnects $S$ from $T$. Thus $\mathrm{mincut}(S, T; G) \leq \min\{c_1, c_2\}$.

*Lower bound.* Let $C$ be any $S$–$T$ cut in $G$. Since every $S$–$T$ path must pass through $u$, the cut $C$ must either: (i) disconnect $S$ from $u$ within $G_1$, or (ii) disconnect $u$ from $T$ within $G_2$ (or both). In case (i), $C \cap E(G_1)$ is an $S$–$u$ cut in $G_1$, hence $|C| \geq |C \cap E(G_1)| \geq c_1$. In case (ii), similarly $|C| \geq c_2$. Therefore $|C| \geq \min\{c_1, c_2\}$ for every cut $C$, proving the lower bound. $\square$

### Max-flow/min-cut

**Lemma C.3** (Max-flow/min-cut (unit capacities)). *For any finite directed graph with unit edge capacities, the value of the maximum $S$–$T$ flow equals $\mathrm{mincut}(S, T; \cdot)$.*

*Proof.* This is the standard max-flow/min-cut theorem specialized to unit capacities. $\square$

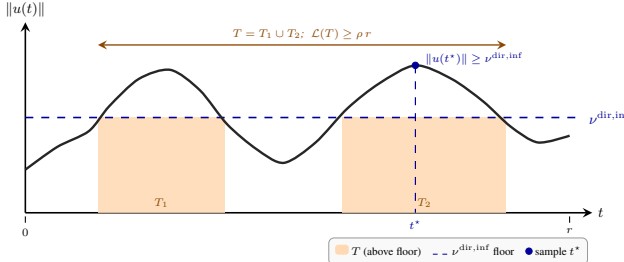

*Figure 4.* Intuition for (P5$_{\text{dir}}$): the black curve $\|u(t)\|$ clears the blue floor $\nu^{\text{dir,inf}}$ on the orange set $T = T_1 \cup T_2$ with $\mathcal{L}(T) \geq \rho r$. The marked point $t^\star$ shows one certified direction sample. The resulting $\nu^{\text{dir,inf}}\rho r$ budget is Step 1 of Theorem B.52.

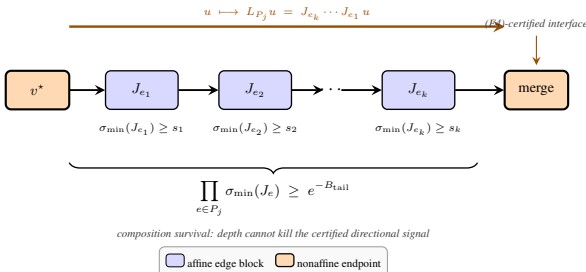

*Figure 5.* Intuition for (E4): the active suffix $P_j$ carries the directional signal from $v^\star$ to the merge through affine edge blocks. The per-edge floors $\sigma_{\min}(J_{e_i}) \geq s_i$ multiply to the certified survival bound $e^{-B_{\text{tail}}}$. This is the Step 2 composition budget in Theorem B.52.

### Measurable suffix selection and non-emptiness

**Lemma C.4** (Measurable suffix selection (finite family)). *Let $\{P_1, \ldots, P_K\}$ be a finite family of candidate suffix objects (paths/operators) indexed in advance. Assume for each $j$ the validity predicate $x \mapsto \mathbf{1}_{\{P_j \in \mathcal{F}(x)\}}$ is Borel. Define the selector*

$$j^*(x) := \min\{j \in \{1, \ldots, K\} : P_j \in \mathcal{F}(x)\},$$

*with the convention $j^*(x) = 1$ if $\mathcal{F}(x) = \varnothing$. Then $j^*(x)$ is Borel measurable.*

*Proof.* For each $j$, the event $\{j^*(x) = j\}$ can be written as the Borel set

$$\{P_j \in \mathcal{F}(x)\} \cap \bigcap_{i<j} \{P_i \notin \mathcal{F}(x)\}.$$

Since $\{1, \ldots, K\}$ is finite, this gives a Borel partition of $\mathcal{X}$ and hence a Borel selector. $\square$

**Lemma C.5** (Suffix family non-emptiness under path-closed activity). *Fix $x$ and an active sink $\tau \in \mathcal{T}_{\text{act}}(x)$. If $\mathcal{G}_x^{\text{act}}$ is path-closed and contains at least one directed $S^{\text{sat}}$–$\tau$ path, then any suffix family defined by "take suffixes of an active $S^{\text{sat}}$–$\tau$ path from a designated anchor" is non-empty.*

*Proof.* By assumption, there exists a directed path $P$ from $S^{\text{sat}}$ to $\tau$ in $\mathcal{G}_x^{\text{act}}$. Any designated anchor vertex lying on $P$ (for example the head of a designated cut edge) has a directed suffix subpath of $P$ to $\tau$. By path-closedness, this suffix is contained in $\mathcal{G}_x^{\text{act}}$. Hence the suffix family is non-empty. $\square$

Before the quantitative proofs we anchor the four contributing budgets visually. The primitive-axiom budget is illustrated in Figure 4.

The composition budget along an active suffix is illustrated in Figure 5.

The merge-axiom budget is illustrated in Figure 6.

These four budgets combine multiplicatively in Theorem 3.2; the composite intuition is shown in Figure 7.

### C.2. Proofs of Quantitative Theorems

**Standing quantifier/cell standard.** Fix a parameter vector $\theta$ outside the parameter-null discriminant set (from the structural layer), and fix an input $x$ outside the input-null set. Whenever we analyze a ray $x_v(t) = x + tv$, we implicitly restrict to radii $0 < r \leq r_0(x, \theta, v)$ small enough that: (i) $x_v([0, r])$ lies inside a single refined $C^1$ cell $U_{x,v}$; (ii) the active structure and all certificate data (suffix family, tail bounds, merge envelope) remain valid on $U_{x,v}$; and (iii) all Jacobians used below are well-defined and continuous on $U_{x,v}$. All constants with a superscript "inf" are interpreted as *cellwise safe infima* over $U_{x,v}$.

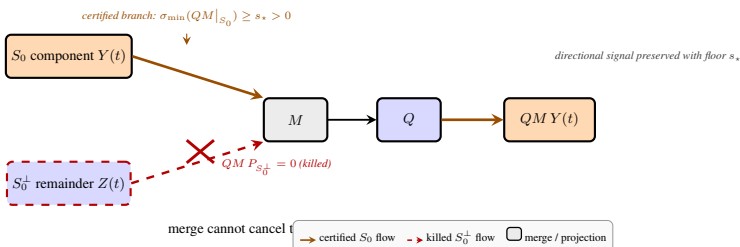

**Figure 6.** Intuition for axiom (M): the orange $S_0$ component passes through the merge–projection envelope with floor $\sigma_{\min}(QM|_{S_0}) \geq s_\star$, while the dashed red $S_0^\perp$ remainder is killed by $QMP_{S_0^\perp} = 0$. Thus the merge cannot cancel the certified contribution; the solid arrow explicitly marks the surviving signal. See Lemmas C.10 and C.11.

*Real-RAM variation bound (Theorem 3.2)*

$$\mathrm{TV}_x(\tau) \;\geq\; \lambda_x \cdot \gamma_{P5}^{\min} \cdot \Gamma_{E4} \cdot \mu_M \cdot \tau$$

**Figure 7.** Intuition for the Real-RAM variation bound (Theorem 3.2): the structural reach $\lambda_x$, primitive floor $\gamma_{P5}^{\min}$, composition factor $\Gamma_{E4}$, merge floor $\mu_M$, and ray length $\tau$ multiply from left to right. The final box is the lower bound on $\mathrm{TV}_x(\tau)$. The chain summarises Steps 1–4 of Theorem B.52.

### C.2.1. TOOLBOX LEMMAS

We use three standard lemmas repeatedly: absolute-continuity variation identity, projection monotonicity, and submultiplicativity of the least singular value.

**Lemma C.6** (Absolute continuity and total variation). *Let $f : [0, r] \to \mathbb{R}^m$ be absolutely continuous. Then $f$ is differentiable a.e., $f' \in L^1([0, r])$, and*

$$\mathrm{TV}\big[f\big]_{[0,r]} \;=\; \int_0^r \|f'(t)\| \, dt.$$

*Proof.* Because $f$ is absolutely continuous, $f(t) = f(0) + \int_0^t f'(s)\,ds$ holds for a.e. defined $f' \in L^1$. For any partition $0 = t_0 < \cdots < t_K = r$,

$$\sum_{k=0}^{K-1} \|f(t_{k+1}) - f(t_k)\| \;\leq\; \sum_{k=0}^{K-1} \int_{t_k}^{t_{k+1}} \|f'(s)\| \, ds \;=\; \int_0^r \|f'(s)\| \, ds,$$

so $\mathrm{TV}[f]_{[0,r]} \leq \int_0^r \|f'(s)\| ds$.

For the reverse inequality, define $g(t) := \|f'(t)\|$ (a.e.). By the Lebesgue differentiation theorem, for a.e. $t$ we have $\lim_{\varepsilon \downarrow 0} \frac{1}{\varepsilon} \int_t^{t+\varepsilon} g(s)\,ds = g(t)$. Choose a partition fine enough so that on each interval $[t_k, t_{k+1}]$ the increment $\|f(t_{k+1}) - f(t_k)\|$ approximates $\int_{t_k}^{t_{k+1}} g(s)\,ds$ up to an arbitrarily small relative error (using $f(t_{k+1}) - f(t_k) = \int_{t_k}^{t_{k+1}} f'(s)\,ds$ and the differentiation property). Taking the supremum over partitions yields $\mathrm{TV}[f]_{[0,r]} \geq \int_0^r g(s)\,ds$. $\square$

**Lemma C.7** (Projection monotonicity for total variation). *If $Q : \mathbb{R}^m \to \mathbb{R}^m$ is linear with $\|Q\|_{\mathrm{op}} \leq 1$, then for any $f : [0, r] \to \mathbb{R}^m$,*

$$\mathrm{TV}[Q \circ f]_{[0,r]} \;\leq\; \mathrm{TV}[f]_{[0,r]}.$$

*Equivalently, $\mathrm{TV}[f]_{[0,r]} \geq \mathrm{TV}[Q \circ f]_{[0,r]}$.*

*Proof.* For any partition $0 = t_0 < \cdots < t_K = r$,

$$\sum_k \|Q(f(t_{k+1}) - f(t_k))\| \;\leq\; \|Q\|_{\mathrm{op}} \sum_k \|f(t_{k+1}) - f(t_k)\| \;\leq\; \sum_k \|f(t_{k+1}) - f(t_k)\|.$$

Taking the supremum over partitions yields the claim. $\square$

**Lemma C.8** (Submultiplicativity of $\sigma_{\min}$). *For compatible matrices $A, B$, $\sigma_{\min}(AB) \geq \sigma_{\min}(A)\sigma_{\min}(B)$. Consequently, for any vector $w$,*

$$\|ABw\| \;\geq\; \sigma_{\min}(A)\sigma_{\min}(B) \, \|w\|.$$

*In particular, for a path operator $L_P = \prod_{e \in P} J_e$ (product ordered along the path),*

$$\|L_P w\| \geq \left( \prod_{e \in P} \sigma_{\min}(J_e) \right) \|w\|.$$

*Proof.* For any unit vector $x$,

$$\|ABx\| \geq \sigma_{\min}(A)\|Bx\| \geq \sigma_{\min}(A)\sigma_{\min}(B)\|x\|.$$

Taking the infimum over $\|x\| = 1$ yields $\sigma_{\min}(AB) \geq \sigma_{\min}(A)\sigma_{\min}(B)$. The vector inequality follows immediately, and the path bound follows by induction. $\qquad\square$

### C.2.2. PROOF OF THE REAL-RAM LOWER BOUND (THEOREM B.52)

**Setup.** Assume the hypotheses of Theorem B.52:

- $x$ has at least one active sink, and $\tau$ is an active sink attaining $\lambda_x$.
- $C$ is a minimum $S^{\mathrm{sat}}$–$\tau$ cut in $\mathcal{G}_x^{\mathrm{act}}$.
- By (S2), $C$ contains an edge incident to a nonaffine node; fix such a node and call it $v^\star$.
- (P5$_{\mathrm{dir}}$) holds at $v^\star$ on the refined cell $U_{x,v}$, producing a realized influential direction $v \in \mathbb{S}^{d-1}$ and constants $\nu_{v^\star}^{\mathrm{dir,inf}}(U_{x,v}) > 0$ and $\rho_{x,v^\star}^{\mathrm{inf}}(U_{x,v}) \in (0,1]$.
- A merge certificate (Assumption M) supplies a fixed interface subspace $S_0$, a constant linear map $Q$ with $\|Q\|_{\mathrm{op}} \leq 1$, and a certified envelope $s_0(U_{x,v}) > 0$ such that

$$\inf_{y \in U_{x,v}} \sigma_{\min}\Big(QM(y,\theta)\big|_{S_0}\Big) \geq s_0(U_{x,v}), \qquad QM(y,\theta)P_{S_0^\perp} = 0 \ \ \forall y \in U_{x,v}.$$

- A tail condition (E4-$\sigma$-cell, or E4-$\sigma$-ray, or E4-dir) supplies a finite family of active suffixes from $v^\star$ to the merge interface within $S_0$, together with a tail constant $B_{\mathrm{tail}}$ in the chosen mode.

**Goal.** We prove that for all $0 < r \leq r_0(x,\theta,v)$ with $x_v([0,r]) \subseteq U_{x,v}$,

$$\mathrm{TV}\big[N_\theta(x_v(t))\big]_{[0,r]} \geq s_0(U_{x,v}) \cdot e^{-B_{\mathrm{tail}}} \cdot \nu_{v^\star}^{\mathrm{dir,inf}}(U_{x,v}) \cdot \rho_{x,v^\star}^{\mathrm{inf}}(U_{x,v}) \cdot r.$$

*Proof of Theorem B.52.* **Step 1: Directional nonlinearity produces a coercive upstream vector.** By (P5$_{\mathrm{dir}}$), there exists a measurable set $T \subseteq [0,r]$ with

$$\mathrm{Leb}(T) \geq \rho_{x,v^\star}^{\mathrm{inf}}(U_{x,v}) r$$

and a measurable choice of generalized Jacobian elements $G_{v^\star}(x_v(t)) \in \partial^\circ \omega_{v^\star}(\zeta_{v^\star}(x_v(t)))$ such that the vector

$$u(t) := G_{v^\star}(x_v(t)) D\zeta_{v^\star}(x_v(t)) v$$

satisfies

$$\|u(t)\| \geq \nu_{v^\star}^{\mathrm{dir,inf}}(U_{x,v}) \qquad \forall t \in T.$$

**Step 2: Tail conditions yield at least one suffix with controlled gain.** Let $\{P_1, \ldots, P_m\}$ be the finite family of active suffixes from $v^\star$ to the merge interface whose coordinates lie in $S_0$ on the refined cell $U_{x,v}$. For each suffix $P_j$, let

$$L_{P_j}(y,\theta) := \prod_{e \in P_j} J_e(y,\theta)$$

denote the pathwise Jacobian product along the suffix (as defined in §B.4), evaluated on $U_{x,v}$.

Under any of the E4 variants:

- In the $\sigma$-variants, the hypothesis implies that (in the relevant mode) there exists at least one suffix $P_j$ whose product of least singular values is $\geq e^{-B_{\mathrm{tail}}}$. Then Lemma C.8 gives $\|L_{P_j}(x_v(t),\theta)u(t)\| \geq e^{-B_{\mathrm{tail}}}\|u(t)\|$.
- In the directional variant (E4-dir), the hypothesis gives the inequality $\|L_{P_j}(x_v(t),\theta)u(t)\| \geq e^{-B_{\mathrm{tail}}}\|u(t)\|$ directly (uniformly on $t \in T$) for at least one suffix $P_j$.

In all cases, for each $t \in T$ we conclude

$$\max_{1 \leq j \leq m} \|L_{P_j}(x_v(t),\theta)u(t)\| \geq e^{-B_{\mathrm{tail}}}\|u(t)\|. \tag{64}$$

**Step 3: Factorization at the merge interface and merge isolation.** Define the interface vector (living in $S_0$ by construction)

$$Y(t) := \big[\, L_{P_1}(x_v(t),\theta)u(t), \ \ldots, \ L_{P_m}(x_v(t),\theta)u(t)\,\big].$$

Then $\|Y(t)\| = \big(\sum_{j=1}^m \|L_{P_j}u(t)\|^2\big)^{1/2}$ and therefore $\|Y(t)\| \geq \max_j \|L_{P_j}u(t)\|$. Combining with (64) yields

$$\|Y(t)\| \geq e^{-B_{\mathrm{tail}}}\|u(t)\| \qquad \forall t \in T.$$

On the refined cell $U_{x,v}$ the realized map is $C^1$ (Real-RAM), hence $t \mapsto N_\theta(x_v(t))$ is absolutely continuous and differentiable a.e. The merge factorization (by the definition of the merge interface and the grouping of path contributions) has the form

$$\frac{d}{dt}N_\theta(x_v(t)) \ = \ M(x_v(t),\theta)\big(P_{S_0}Y(t) + P_{S_0^\perp}Z(t)\big),$$

for some (irrelevant) remainder term $Z(t)$ collecting other interface contributions. Applying $Q$ and using $QM(\cdot,\theta)P_{S_0^\perp} = 0$ on $U_{x,v}$ gives

$$\frac{d}{dt}\big(QN_\theta(x_v(t))\big) \ = \ QM(x_v(t),\theta)\,Y(t).$$

Therefore, for a.e. $t \in T$,

$$\Big\|\frac{d}{dt}\big(QN_\theta(x_v(t))\big)\Big\| \ \geq \ \sigma_{\min}\Big(QM(x_v(t),\theta)\big|_{S_0}\Big)\cdot\|Y(t)\| \ \geq \ s_0(U_{x,v})\cdot e^{-B_{\mathrm{tail}}}\cdot\|u(t)\|.$$

Using $\|u(t)\| \geq \nu_{v^\star}^{\mathrm{dir,inf}}(U_{x,v})$ on $T$ yields

$$\Big\|\frac{d}{dt}\big(QN_\theta(x_v(t))\big)\Big\| \ \geq \ s_0(U_{x,v})\cdot e^{-B_{\mathrm{tail}}}\cdot\nu_{v^\star}^{\mathrm{dir,inf}}(U_{x,v}) \qquad \text{for a.e. } t \in T. \tag{65}$$

**Step 4: Integrate and lift from $Q \circ N_\theta$ to $N_\theta$.** By Lemma C.6 applied to $t \mapsto QN_\theta(x_v(t))$,

$$\mathrm{TV}\big[QN_\theta(x_v(t))\big]_{[0,r]} \ = \ \int_0^r \Big\|\frac{d}{dt}\big(QN_\theta(x_v(t))\big)\Big\|\,dt \ \geq \ \int_T \Big\|\frac{d}{dt}\big(QN_\theta(x_v(t))\big)\Big\|\,dt.$$

Using (65) and $\mathrm{Leb}(T) \geq \rho_{x,v^\star}^{\mathrm{inf}}(U_{x,v})r$ gives

$$\mathrm{TV}\big[QN_\theta(x_v(t))\big]_{[0,r]} \ \geq \ s_0(U_{x,v})\cdot e^{-B_{\mathrm{tail}}}\cdot\nu_{v^\star}^{\mathrm{dir,inf}}(U_{x,v})\cdot\rho_{x,v^\star}^{\mathrm{inf}}(U_{x,v})\cdot r.$$

Finally, Lemma C.7 implies $\mathrm{TV}[N_\theta(x_v(t))]_{[0,r]} \geq \mathrm{TV}[QN_\theta(x_v(t))]_{[0,r]}$, which proves the claimed inequality. $\qquad\square$

**Corollary C.9** (Multi-suffix $\sqrt{k}$ amplification (Gram form))**.** *Assume the hypotheses of Theorem B.52 and suppose there exist $k \geq 1$ active suffixes $P_{j_1}, \ldots, P_{j_k}$ within $S_0$ such that: (i) each satisfies the same tail constant $B_{\mathrm{tail}}$ (in the chosen E4 mode), and (ii) for $t \in T$ the Gram matrix of the vectors $\{L_{P_{j_\ell}}(x_v(t),\theta)u(t)\}_{\ell=1}^k$ has $\lambda_{\min} \geq \gamma^2$ uniformly. Then*

$$\mathrm{TV}\big[N_\theta(x_v(t))\big]_{[0,r]} \ \geq \ \gamma\cdot s_0(U_{x,v})\cdot e^{-B_{\mathrm{tail}}}\cdot\nu_{v^\star}^{\mathrm{dir,inf}}(U_{x,v})\cdot\rho_{x,v^\star}^{\mathrm{inf}}(U_{x,v})\cdot r.$$

*In particular, if the $k$ suffix vectors are orthogonal with equal norm floors, then $\gamma = \sqrt{k}$.*

*Proof.* Repeat the proof of Theorem B.52, but redefine $Y(t)$ using only the selected $k$ suffixes. Then $\|Y(t)\| = \|z(t)\|$ where $z(t)$ stacks the $k$ vectors. The Gram lower bound implies $\|z(t)\| \geq \gamma\|u(t)\|$ on $T$. The remainder is identical, multiplying the final bound by $\gamma$. $\qquad\square$

### C.2.3. PROOF OF THE IEEE-754 LOWER BOUND (THEOREM B.56)

*Proof of Theorem B.56.* Under IEEE-754 arithmetic, the realized straight-line program (with fixed $\theta$) computes a map $t \mapsto N_\theta(x_v(t))$ that is piecewise constant on $[0,r]$, with jumps only at rounding-threshold crossings (after excluding a null set of degenerate tie cases if necessary).

Assume (E4-event) holds on $U_{x,v}$ with certified constants $\kappa_x > 0$, $\Delta_{\min}(x) > 0$, and separation $\delta > 0$. Then for every $0 < r \leq r_0(x,\theta,v)$, the segment triggers at least $\kappa_x r$ distinct threshold crossings, and successive crossings are separated by at least $\delta$ in $t$. At each crossing, the output changes in norm (after the certified downstream envelope) by at least $\Delta_{\min}(x)$.

Let $0 < t_1 < \cdots < t_N < r$ enumerate these crossing times, where $N \geq \kappa_x r$. Because crossings are separated, we can choose $\varepsilon > 0$ small enough that the intervals $(t_i - \varepsilon, t_i + \varepsilon)$ are disjoint and contained in $(0,r)$. Pick partition points $s_i^- \in (t_i - \varepsilon, t_i)$ and $s_i^+ \in (t_i, t_i + \varepsilon)$. On each $(s_i^-, s_i^+)$ the function experiences the $i$th jump, hence

$$\|N_\theta(x_v(s_i^+)) - N_\theta(x_v(s_i^-))\| \ \geq \ \Delta_{\min}(x).$$

Now consider the partition

$$0 < s_1^- < s_1^+ < s_2^- < s_2^+ < \cdots < s_N^- < s_N^+ < r.$$

The corresponding partition sum in the definition of total variation is at least $\sum_{i=1}^N \Delta_{\min}(x) = N\Delta_{\min}(x) \geq \kappa_x r\,\Delta_{\min}(x)$. Taking the supremum over partitions yields

$$\mathrm{TV}\big[N_\theta(x_v(t))\big]_{[0,r]} \ \geq \ \kappa_x r\,\Delta_{\min}(x),$$

which is exactly the claimed bound. $\qquad\square$

C.2.4. SUPPORTING QUANTITATIVE LEMMAS

**Lemma C.10** (Directional factorization at the merge interface). *Work on a refined $C^1$ cell $U_{x,v}$ and fix a nonaffine cut witness node $v^\star$. Let $\{P_1, \ldots, P_m\}$ be the suffix family from $v^\star$ to the merge interface whose coordinates lie in $S_0$. Let $Y(t)$ stack the vectors $L_{P_j}(x_v(t), \theta) u(t)$. Then along the ray $x_v(t)$ on $U_{x,v}$, the directional derivative can be written as*

$$\frac{d}{dt} N_\theta(x_v(t)) = M(x_v(t), \theta)\big(P_{S_0} Y(t) + P_{S_0^\perp} Z(t)\big),$$

*for some (measurable) remainder term $Z(t)$ collecting the interface contributions not in $S_0$. If additionally $QM(\cdot, \theta)P_{S_0^\perp} = 0$ on $U_{x,v}$, then*

$$\frac{d}{dt}\big(QN_\theta(x_v(t))\big) = QM(x_v(t), \theta)\, Y(t) \qquad \text{a.e. on } [0, r].$$

*Proof.* On a refined $C^1$ cell, the realized program is a composition of $C^1$ maps. Its derivative at $x_v(t)$ is a sum of pathwise Jacobian products from the input tangent to the output, grouped by the interface coordinates immediately upstream of the merge mixer. Collect the contributions corresponding to the certified suffix family (which by definition live in $S_0$) into $Y(t)$, and put the remaining interface contributions into $Z(t) \in S_0^\perp$. This yields the stated decomposition. Applying $Q$ and using $QMP_{S_0^\perp} = 0$ eliminates the $Z(t)$ term. $\square$

**Lemma C.11** (Merge lower envelope from Assumption M). *Assume the merge certificate (Assumption M) on a refined cell $U_{x,v}$ supplies $(Q, S_0)$ and one of the local certificate modes (structured block isolation or a local numerical certificate). Then there exists a certified constant $s_0(U_{x,v}) > 0$ such that*

$$\inf_{y \in U_{x,v}} \sigma_{\min}\Big(QM(y, \theta)\big|_{S_0}\Big) \geq s_0(U_{x,v}).$$

*Proof.* In the structured mode, the lower bound is part of the supplied certificate; take $s_0$ to be that bound. In the local numerical mode, suppose we have points $y_1, \ldots, y_q \in U_{x,v}$ with $\min_j \sigma_{\min}(QM(y_j, \theta)|_{S_0}) \geq \beta$ and that $M(\cdot, \theta)$ is $L$-Lipschitz in operator norm on $U_{x,v}$ with $L\,\mathrm{diam}(U_{x,v}) \leq \beta/2$. Then for any $y \in U_{x,v}$ pick $j$ with $\|y - y_j\| \leq \mathrm{diam}(U_{x,v})$ and use Weyl-type stability:

$$\sigma_{\min}(QM(y, \theta)|_{S_0}) \geq \sigma_{\min}(QM(y_j, \theta)|_{S_0}) - \|QM(y, \theta) - QM(y_j, \theta)\|_{\mathrm{op}} \geq \beta - L\mathrm{diam}(U_{x,v}) \geq \beta/2.$$

Thus $s_0(U_{x,v}) = \beta/2$ works. $\square$

**Lemma C.12** (Softmax singular value bound on the tangent hyperplane). *Let $p = \mathrm{softmax}(z) \in \mathbb{R}^n$ and let $J = \nabla\mathrm{softmax}(z) = \mathrm{diag}(p) - pp^\top$. Then $J \succeq 0$ with $\ker(J) = \mathrm{span}\{\mathbf{1}\}$. On the $(n-1)$-dimensional subspace*

$$H_p := \{x \in \mathbb{R}^n : p^\top x = 0\},$$

*the restriction $J|_{H_p}$ is positive definite and satisfies*

$$\sigma_{\min}\big(J|_{H_p}\big) \geq \min_i p_i.$$

*Proof.* A direct computation gives $J = \mathrm{diag}(p) - pp^\top$, hence $J\mathbf{1} = 0$ and $J \succeq 0$. For $x \in H_p$, we have $p^\top x = 0$, so

$$x^\top Jx = x^\top \mathrm{diag}(p)\, x - (p^\top x)^2 = \sum_{i=1}^n p_i x_i^2 \geq \Big(\min_i p_i\Big)\sum_{i=1}^n x_i^2 = \Big(\min_i p_i\Big)\|x\|^2.$$

Thus the Rayleigh quotient of $J|_{H_p}$ is at least $\min_i p_i$, yielding the stated bound on $\sigma_{\min}$. $\square$

## C.3. Proofs of Consistency, Independence, and Case Studies

This appendix contains the proofs deferred from §B.7, covering: (i) generic influence for the running LReLU block, (ii) joint satisfiability (consistency) of (S1)+(S2)+(C2), (iii) independence of (C2) from (S1)+(S2) relative to a restricted primitive library, and (iv) the transformer case-study proposition establishing $\lambda_x = 1$ under Convention B.

Throughout, $\mu$ denotes the reference input measure from §B.3 (assumed absolutely continuous with respect to Lebesgue on $\mathcal{X}$), and $\mathcal{L}$ denotes Lebesgue measure on Euclidean parameter spaces.

C.3.1. PROOFS FOR GENERIC INFLUENCE (LEMMAS B.58 AND B.59)

We work with the running example from §B.1:

$$N_\theta(x) = W_2\,\mathrm{LReLU}_\alpha(W_1 x + b_1) + b_2, \qquad \alpha \in (0, 1).$$

Write $\theta = (W_1, b_1, W_2, b_2)$ with $W_1 \in \mathbb{R}^{h \times d}$ and $W_2 \in \mathbb{R}^{m \times h}$. Let

$$\Sigma := \{1, \alpha\}^h, \qquad D_s := \mathrm{diag}(s_1, \ldots, s_h) \text{ for } s \in \Sigma.$$

For $x$ such that $(W_1 x + b_1)_i \neq 0$ for all $i \in [h]$, the activation pattern $s(x) \in \Sigma$ is defined by $s_i(x) = 1$ if $(W_1 x + b_1)_i > 0$ and $s_i(x) = \alpha$ otherwise. On such $x$, $N_\theta$ is differentiable and

$$DN_\theta(x) = W_2\, D_{s(x)}\, W_1. \tag{66}$$

**Exceptional parameter sets.** For each pattern $s \in \Sigma$, define

$$E_s := \{\theta : W_2 D_s W_1 = 0_{m \times d}\}, \qquad E := \bigcup_{s \in \Sigma} E_s.$$

We show: (i) each $E_s$ is algebraic (hence Borel), (ii) $E$ is $\mathcal{L}$-null, and (iii) for $\theta \notin E$, the unique nonaffine node in the saturated running-example graph is influential for $\mu$-a.e. input.

*Proof of Lemma B.58.* **Step 1: $E_s$ is algebraic.** Fix $s \in \Sigma$. Each entry of the matrix product $W_2 D_s W_1$ is a polynomial in the entries of $W_1$ and $W_2$ (the diagonal $D_s$ is a fixed constant matrix once $s$ is fixed). Thus $E_s$ is the simultaneous zero set of finitely many polynomials, hence an algebraic subset of the parameter space and therefore Borel.

**Step 2: $E_s$ is $\mathcal{L}$-null (hence $E$ is $\mathcal{L}$-null).** Define the polynomial

$$g_s(\theta) := \|W_2 D_s W_1\|_F^2 = \sum_{i=1}^{m} \sum_{j=1}^{d} (W_2 D_s W_1)_{ij}^2.$$

Then $E_s = \{\theta : g_s(\theta) = 0\}$. To see $g_s$ is not the zero polynomial, choose (for example) parameters with $W_1 = \begin{bmatrix} I_d \\ 0 \end{bmatrix}$ and $W_2 = \begin{bmatrix} I_d & 0 \end{bmatrix}$ (when $h \geq d$; if $h < d$, choose any $W_1, W_2$ with a nonzero entry so that $W_2 D_s W_1 \neq 0$). Then $W_2 D_s W_1 \neq 0$, so $g_s(\theta) > 0$ at that point. Hence $g_s$ is a nontrivial polynomial.

The zero set of a nontrivial polynomial in $\mathbb{R}^p$ has Lebesgue measure 0. Therefore $\mathcal{L}(E_s) = 0$ for each $s \in \Sigma$, and since $\Sigma$ is finite,

$$\mathcal{L}(E) = \mathcal{L}\Big(\bigcup_{s \in \Sigma} E_s\Big) = 0.$$

This completes the proof of Lemma B.58. $\qquad \square$

*Proof of Lemma B.59.* Fix $\theta \notin E$. Let

$$B_\theta := \bigcup_{i=1}^{h} \{x \in \mathbb{R}^d : (W_1 x + b_1)_i = 0\}.$$

Each set $\{(W_1 x + b_1)_i = 0\}$ is an affine hyperplane (or empty), hence $\mu$-null because $\mu \ll \mathcal{L}$ on $\mathcal{X}$. Thus $\mu(B_\theta) = 0$.

For $x \notin B_\theta$, the activation pattern $s(x) \in \Sigma$ is well-defined and (66) holds. Since $\theta \notin E$, in particular $\theta \notin E_{s(x)}$, so $W_2 D_{s(x)} W_1 \neq 0$. Therefore there exists a direction $v \in \mathbb{R}^d$ with $DN_\theta(x) v \neq 0$. In the saturated running-example graph, the unique nonaffine node is the LReLU block, and the unique suffix from that node to the sink is the affine map induced by $W_2$. Thus a nonzero directional derivative of $N_\theta$ implies that the LReLU node is influential at $x$ under the influence predicate used to define $\mathcal{G}_x^{\mathrm{act}}$ (it has a witness direction whose pushforward to the sink is nonzero). Consequently, the unique nonaffine node is influential for $\mu$-a.e. $x$.

This completes the proof of Lemma B.59. $\qquad \square$

## C.3.2. PROOF OF CONSISTENCY (THEOREM B.60)

*Proof of Theorem B.60.* We exhibit a concrete pack $\Pi$ and a parameter choice $\theta$ for which (S1), (S2), and (C2) hold simultaneously.

**Pack.** Take Real-RAM arithmetic. Let the type library $\mathcal{T}$ contain the types needed for the running example (Affine and $\mathrm{LReLU}_\alpha$), and take the matching primitive library $\mathcal{O} = \mathcal{T}$ (with the identity type-realization map). Use the standard parameter distributor that assigns $(W_1, b_1)$ to the first affine node, $\varnothing$ to $\mathrm{LReLU}_\alpha$, and $(W_2, b_2)$ to the second affine node.

**Choose parameters.** Pick any $\theta$ such that $\theta \notin E$ from §C.3.1 (equivalently: $W_2 D_s W_1 \neq 0$ for every $s \in \Sigma$).

**(C2).** Because $\mathcal{O} = \mathcal{T}$ and the network graph is finite and acyclic, the network is realizable by the evident SLP that evaluates the graph. Thus (C2) holds.

**(S1).** Under Real-RAM, Affine and $\mathrm{LReLU}_\alpha$ are continuous maps, hence Borel measurable, and the parameter distributor is Borel. Therefore the induced network map $N_\theta$ is Borel, either directly by closure of Borel maps under finite composition or by invoking Lemma B.38. Hence (S1) holds.

**(S2).** The affine-saturated graph for the running example has the form $S \to \mathrm{LReLU}_\alpha \to \tau$ with a unique $S$–$\tau$ path, so every $S$–$\tau$ cut has size 1 and every minimum cut contains an edge incident to the nonaffine node $\mathrm{LReLU}_\alpha$. Moreover, since $\theta \notin E$, Lemma B.59 implies that the $\mathrm{LReLU}_\alpha$ node is influential for $\mu$-a.e. $x$, so for those $x$ the unique path is active and $\tau \in \mathcal{T}_{\mathrm{act}}(x)$. Thus (S2) holds on the required domain of quantification (all $x$ with active sinks).

Therefore (S1)+(S2)+(C2) are jointly satisfiable under this explicit model. $\qquad \square$

C.3.3. PROOFS OF INDEPENDENCE (LEMMA B.61 AND THEOREM B.62)

*Proof of Lemma B.61.* Fix $\alpha \in (0, 1)$ and consider the scalar map $N(x) = \text{LReLU}_\alpha(\pi x)$.

**Well-defined Borel function.** The map $x \mapsto \pi x$ is continuous (hence Borel), and $\text{LReLU}_\alpha$ is continuous (hence Borel). The composition $N = \text{LReLU}_\alpha \circ (\pi \cdot)$ is therefore Borel measurable.

**Structural nonlinearity.** The affine-saturated graph consists of an affine source block (multiplication by $\pi$), followed by the nonaffine $\text{LReLU}_\alpha$ node, followed by an identity sink. The unique $S$–$\tau$ path passes through the $\text{LReLU}_\alpha$ node. Since there is only one path and it contains the nonaffine node, there is no affine-only bypass: every minimum cut (which has size 1) is incident to the nonaffine node. □

*Proof of Theorem B.62.* We construct a pack $\Pi$ and a network $N_\theta$ such that (S1) and (S2) hold but (C2) fails. The key is that independence is *relative to the primitive library*.

**Fix $\alpha \in \mathbb{Q} \cap (0, 1)$.** Consider the one-dimensional network from Lemma B.61:

$$N_\theta(x) = \text{LReLU}_\alpha(\pi x), \qquad x \in \mathbb{R}.$$

At the level of ideal semantics, this is an Affine node (multiply by $\pi$) followed by $\text{LReLU}_\alpha$.

**Pack.** Work under Real-RAM arithmetic. Let the *type* library $\mathcal{T}$ include an unconstrained scalar Affine type (with real parameter space $\Xi_{\text{Aff}} = \mathbb{R}^2$ for $(a, b)$) and $\text{LReLU}_\alpha$. Let the *primitive* library be the restricted set

$$\mathcal{O}_{\mathbb{Q}, \alpha} := \{\text{Aff}_\mathbb{Q}, \ \text{LReLU}_\alpha\},$$

where $\text{Aff}_\mathbb{Q}$ is the affine primitive with *rational* coefficients (i.e., its parameter space is $\Xi_{\text{Aff}_\mathbb{Q}} = \mathbb{Q}^2$ and it realizes $x \mapsto ax + b$ with $(a, b) \in \mathbb{Q}^2$).

**(S1).** By Lemma B.61, $N_\theta$ is a well-defined Borel function. Hence (S1) holds.

**(S2).** By Lemma B.61, the affine-saturated graph has a single nonaffine node on the unique $S$–$\tau$ route. For any $x$ with an active sink, the minimum $S$–$\tau$ cut has size 1 and is incident to the nonaffine node, so (S2) holds (structurally).

**(C2) fails.** Assume for contradiction that (C2) holds: there exists a finite acyclic SLP over $\mathcal{O}_{\mathbb{Q}, \alpha}$ that computes $N_\theta$ exactly as a real function.

Any such SLP computes a piecewise-affine function on $\mathbb{R}$ whose slopes on each linear region lie in the field $\mathbb{Q}(\alpha)$, by a straightforward induction over the program:

- An $\text{Aff}_\mathbb{Q}$ node maps $x \mapsto ax + b$ with $a, b \in \mathbb{Q} \subseteq \mathbb{Q}(\alpha)$.
- On any region where the branch of $\text{LReLU}_\alpha$ is fixed, $\text{LReLU}_\alpha(u)$ equals either $u$ or $\alpha u$, so it multiplies local slopes by 1 or by $\alpha$, preserving membership in $\mathbb{Q}(\alpha)$.
- Finite composition preserves the property that regionwise slopes lie in $\mathbb{Q}(\alpha)$.

Since $\alpha \in \mathbb{Q}$, we have $\mathbb{Q}(\alpha) = \mathbb{Q}$.

But $N_\theta(x) = \text{LReLU}_\alpha(\pi x)$ has slope $\pi$ on $(0, \infty)$ and slope $\alpha\pi$ on $(-\infty, 0)$, and $\pi \notin \mathbb{Q}$. Therefore $N_\theta$ cannot agree with any SLP over $\mathcal{O}_{\mathbb{Q}, \alpha}$ on any open interval, a contradiction. Hence (C2) fails.

This establishes that (S1)+(S2) do not imply (C2), i.e. $(S1) + (S2) \nvdash (C2)$ (relative to the chosen primitive library). □

C.3.4. PROOF OF TRANSFORMER $\lambda_x = 1$ (PROPOSITION B.64)

*Proof of Proposition B.64.* We prove that for a pre-norm transformer block satisfying the structural hypotheses (i)–(iv) in the proposition statement, and under Convention B (data-dependent normalizations tagged nonaffine), the active-graph edge connectivity satisfies $\lambda_x = 1$ whenever $\mathcal{T}_{\text{act}}(x) \neq \varnothing$.

Let $\mathcal{G}^{\text{sat}}$ be the affine-saturated graph of the block and $\mathcal{G}_x^{\text{act}} \subseteq \mathcal{G}^{\text{sat}}$ the active subgraph. Let Norm denote the (shared) normalization node (e.g. LayerNorm/RMSNorm) in $\mathcal{G}^{\text{sat}}$. By hypothesis (i), Norm has a unique incoming edge $e_{\text{in}} = (u \to \text{Norm})$ on the block input route.

**Step 1: $e_{\text{in}}$ is on every active $S$–$\tau$ path.** Fix an active sink $\tau \in \mathcal{T}_{\text{act}}(x)$. By hypothesis (iii) (no bypasses around Norm), every $S$–$\tau$ path in $\mathcal{G}^{\text{sat}}$ must pass through Norm. In particular, every $S$–$\tau$ path must traverse the unique in-edge $e_{\text{in}}$ to enter Norm. Since $\mathcal{G}_x^{\text{act}}$ is a path-closed subgraph of $\mathcal{G}^{\text{sat}}$ (§B.4), every active $S$–$\tau$ path in $\mathcal{G}_x^{\text{act}}$ is also an $S$–$\tau$ path in $\mathcal{G}^{\text{sat}}$, hence must traverse $e_{\text{in}}$.

**Step 2: Upper bound $\text{mincut}(S, \tau; \mathcal{G}_x^{\text{act}}) \leq 1$.** Removing $e_{\text{in}}$ disconnects $S$ from $\tau$ in $\mathcal{G}_x^{\text{act}}$ by Step 1. Thus $\{e_{\text{in}}\}$ is an $S$–$\tau$ edge cut in $\mathcal{G}_x^{\text{act}}$, so $\text{mincut}(S, \tau; \mathcal{G}_x^{\text{act}}) \leq 1$.

**Step 3: Lower bound $\text{mincut}(S, \tau; \mathcal{G}_x^{\text{act}}) \geq 1$.** Hypothesis (iv) states $\mathcal{T}_{\text{act}}(x) \neq \varnothing$, so $\tau$ is reachable from $S$ in $\mathcal{G}_x^{\text{act}}$. Hence at least one edge must be removed to disconnect $S$ from $\tau$, giving $\text{mincut}(S, \tau; \mathcal{G}_x^{\text{act}}) \geq 1$.

Combining Steps 2–3 yields $\text{mincut}(S, \tau; \mathcal{G}_x^{\text{act}}) = 1$ for each active sink $\tau$, and therefore $\lambda_x = 1$ by the definition of $\lambda_x$ as the minimum over active sinks.

Finally, under Convention B, Norm is tagged `nonaffine`, so $e_{\text{in}}$ is nonaffine-incident. Thus the size-1 minimum cut is witnessed by a nonaffine-incident edge, aligning with (S2) and enabling the transformer specialization of the quantitative bounds. $\qquad\square$

### C.3.5. PROOF OF MoE $\lambda_x \leq k$ (PROPOSITION B.68)

*Proof of Proposition B.68.* We prove that for an MoE layer with Top-$k$ routing satisfying the structural hypotheses in the proposition statement, the active-graph edge connectivity satisfies $\lambda_x \leq k$.

Let $\mathcal{G}^{\text{sat}}$ be the affine-saturated graph of the MoE layer and $\mathcal{G}_x^{\text{act}} \subseteq \mathcal{G}^{\text{sat}}$ the active subgraph. Fix an active sink $\tau \in \mathcal{T}_{\text{act}}(x)$. Let $A(x, \tau) \subseteq \{1, \ldots, n\}$ denote the set of experts activated by the router for this $(x, \tau)$, with $|A(x, \tau)| \leq k$ by the Top-$k$ routing assumption.

**Constructing a cut of size $|A(x, \tau)|$.** By hypothesis, every $S^{\text{sat}}$–$\tau$ path in $\mathcal{G}_x^{\text{act}}$ that enters the expert bank must pass through exactly one router-to-expert entry edge $e_i = (v_{\text{router}} \to v_{\text{E}_i})$ for some $i \in A(x, \tau)$. Consider the edge set

$$C := \{e_i : i \in A(x, \tau)\}.$$

Removing $C$ severs all routes from $S^{\text{sat}}$ to $\tau$ that pass through any active expert, since each such route must use exactly one of the edges in $C$. By the hypothesis that all $S^{\text{sat}}$–$\tau$ paths enter the expert bank through these edges, the set $C$ is an $S^{\text{sat}}$–$\tau$ edge cut in $\mathcal{G}_x^{\text{act}}$.

**Upper bound on min-cut.** Since $C$ is a cut and $|C| = |A(x, \tau)| \leq k$, we have

$$\text{mincut}(S^{\text{sat}}, \tau; \mathcal{G}_x^{\text{act}}) \leq |C| \leq k.$$

Taking the minimum over all active sinks yields $\lambda_x \leq k$. $\qquad\square$

# D. Calibration of Certificate Constants

This appendix explains how the quantitative constants appearing in the lower bounds are obtained in practice. The framework distinguishes two modes:

- **Certificates (provable).** A finite object whose validity can be checked mechanically and which implies the axiom/postulate in the stated world (pack) without appealing to sampling.
- **Estimates (empirical).** A numerically measured surrogate used in experiments to assess typical magnitude. Estimates may suggest that a certificate is *plausible*, but do not imply the axiom.

Our main-text experiments use estimates for large models; formal certification is feasible for small models and for pack components with closed-form lower bounds.

**Guardrail.** All pathwise and suffix notation (e.g. suffix families, $T_{P_{\succeq v^\star}}$, pathwise Jacobian blocks, and merge operators) is exactly as defined in §B.4. This appendix adds *only* calibration guidance for the constants required by (P5$_{\text{dir}}$), (E4 variants), and Assumption (M), as introduced in §B.5.3, §B.5.4, and §B.5.5.

## D.1. General Methodology

Calibration follows a common three-step workflow.

**Step 1: Choose the world and the validity region.** Fix a pack $\Pi$ and a world choice (Real-RAM or IEEE-754; and an (E4) variant). For each input instance $(x, v, r)$ used in certification or evaluation, identify: (i) the active sink $\tau \in \mathcal{T}_{\text{act}}(x)$, (ii) the designated bottleneck location (e.g. a nonaffine node or cut edge), and (iii) a *validity region* $U$ on which the primitive-level inequalities will be claimed. Typical choices include:

- a refined cell (Real-RAM, piecewise-$C^1$ behavior),
- an interval along a ray $\{x + tv : t \in [0, r]\}$,
- a neighborhood $B_\varepsilon(x)$ when local constancy is invoked.

**Step 2: Calibrate primitive-level constants.** For each primitive $\omega$ used as a bottleneck witness, calibrate a (P5$_{\text{dir}}$)-style tuple on $U_\omega$: direction selector $\text{Sel}_\omega$ and constants such as $\nu_\omega^{\text{dir}}$ (a directional gain floor) and $\rho_\omega$ (a one-dimensional variation floor), as required by the exact statement of (P5$_{\text{dir}}$) in §B.5.3. For (E4), calibrate a *tail gain* constant (spectral or directional) for suffix operators.

**Step 3: Compose into network-level constants.** Combine: (i) structural floors derived from $\hat{\lambda}_x$ / cut structure, (ii) tail gains from (E4), and (iii) merge survival from (M), yielding a single numerical lower bound on total variation (or an IEEE-754 event bound, depending on the arithmetic world). Composition is purely algebraic once all constants are supplied.

**Certificates vs estimates.** A *certificate* records: the world choice, the validity region $U$, the witness directions (if any), and *provable* bounds. An *estimate* records the same objects but with bounds obtained by numerical probes (sampling directions, finite differences, power iteration, etc.). Estimates are appropriate for the experimental questions in §5 (tightness, scaling, typical magnitudes), but are not used to claim axiom satisfaction.

## D.2. Directional Nonlinearity Calibration (P5$_{\text{dir}}$)

This subsection describes how to calibrate the constants demanded by (P5$_{\text{dir}}$) for common bottleneck primitives in modern architectures.

**General pattern.** Most primitives of interest have one or more *invariance directions* along which the Jacobian must vanish (e.g. softmax is invariant to adding a constant logit). Calibration therefore proceeds by:

1. identifying the appropriate *effective subspace* $H_\omega(z) \subseteq E_{\text{in}, \omega}$ on which the primitive is non-degenerate,
2. ensuring the selector $\text{Sel}_\omega$ returns a direction $u \in H_\omega(z)$,
3. lower-bounding the gain of the primitive on $H_\omega(z)$ on the validity region.

## SOFTMAX FLOORS ON THE TANGENT SUBSPACE

Let $\omega = \text{Softmax}_n$. For logits $z \in \mathbb{R}^n$, write $p = \text{softmax}(z) \in \Delta^{n-1}$ and recall

$$J_{\text{softmax}}(z) = \text{diag}(p) - pp^\top, \qquad J_{\text{softmax}}(z)\,\mathbf{1} = 0.$$

Thus the effective subspace is the tangent space

$$H_{\text{softmax}} := \{u \in \mathbb{R}^n : \langle u, \mathbf{1}\rangle = 0\}.$$

A convenient certified floor (proved as Lemma C.12 in Appendix C.2) is:

$$\sigma_{\min}\left(J_{\text{softmax}}(z)\big|_{H_{\text{softmax}}}\right) \geq \min_i p_i. \tag{67}$$

**How to certify** $\min_i p_i$ **on a region.** A simple sufficient condition uses a logit range bound. If one can certify on $U$ that $\max_i z_i - \min_i z_i \leq G$, then for all $z \in U$,

$$\min_i p_i \geq \frac{e^{\min z}}{\sum_j e^{z_j}} \geq \frac{e^{\min z}}{n\, e^{\max z}} = \frac{e^{-G}}{n}.$$

Thus a valid (certificate) choice is

$$\nu_{\text{softmax}}^{\text{dir}}(U) := \frac{e^{-G}}{n}, \qquad \text{with selector } \text{Sel}_{\text{softmax}}(z) \in H_{\text{softmax}} \text{ (e.g. } u = e_i - e_j).$$

In experiments, $G$ is estimated from observed logits; for formal certification, $G$ can be obtained from interval bounds or other sound range-bounding procedures on the pre-softmax logit computation.

## LAYERNORM FLOORS ON THE NON-INVARIANT SUBSPACE

Let $\omega = \text{LayerNorm}_h$ in Convention B semantics (data-dependent normalization tagged nonaffine). Ignoring $\beta$ and treating $\gamma$ as a fixed diagonal scaling, the core normalization map is

$$\text{LN}(z) = \frac{z - \mu(z)\mathbf{1}}{\sigma(z)}, \qquad \mu(z) = \frac{1}{h}\langle z, \mathbf{1}\rangle, \quad \sigma(z) = \sqrt{\frac{1}{h}\|z - \mu(z)\mathbf{1}\|_2^2 + \varepsilon}.$$

LN is invariant to adding constants in the $\mathbf{1}$ direction and (locally, for positive scalings) to scaling of the centered vector. Accordingly, the Jacobian has (at least) the two-dimensional null space spanned by $\mathbf{1}$ and $z - \mu(z)\mathbf{1}$ when $\varepsilon$ is treated as negligible and $z$ is nondegenerate. A stable effective choice is the codimension-2 subspace

$$H_{\text{LN}}(z) := \{u \in \mathbb{R}^h : \langle u, \mathbf{1}\rangle = 0 \text{ and } \langle u, z - \mu(z)\mathbf{1}\rangle = 0\}.$$

**Practical certified floor.** On $H_{\text{LN}}(z)$ the Jacobian simplifies (the rank-1 correction term vanishes), and one obtains a gain of order $1/\sigma(z)$ before the $\gamma$ scaling. A conservative certificate usable in the framework is:

$$\sigma_{\min}\left(J_{\text{LayerNorm}}(z)\big|_{H_{\text{LN}}(z)}\right) \geq \frac{\gamma_{\min}}{\sigma_{\max}(U)}, \qquad \gamma_{\min} := \min_i |\gamma_i|,$$

provided one can certify an upper bound $\sigma(z) \leq \sigma_{\max}(U)$ for all $z$ in the validity region. (Upper-bounding $\sigma$ is typically much easier than lower-bounding it, and it yields a safe *lower* gain floor.) As above, $\sigma_{\max}(U)$ may be certified by range bounds on the pre-norm activations.

## RMSNORM FLOORS

For RMSNorm (again tagged nonaffine under Convention B analogs),

$$\text{RMS}(z) = \frac{z}{\text{rms}(z)}, \qquad \text{rms}(z) = \sqrt{\frac{1}{h}\|z\|_2^2 + \varepsilon}.$$

RMSNorm is invariant to positive scaling of $z$, so a natural effective subspace is

$$H_{\text{RMS}}(z) := \{u \in \mathbb{R}^h : \langle u, z\rangle = 0\}.$$

On this subspace, the Jacobian admits a gain floor of order $1/\text{rms}(z)$, hence a conservative certificate:

$$\sigma_{\min}\left(J_{\text{RMSNorm}}(z)\big|_{H_{\text{RMS}}(z)}\right) \geq \frac{\gamma_{\min}}{\text{rms}_{\max}(U)}, \qquad \text{rms}_{\max}(U) := \sup_{z \in U} \text{rms}(z).$$

**Remark (pointwise activations).** For pointwise primitives like ReLU/LReLU/GELU, calibration is usually simpler: the effective subspace is a coordinate axis or a low-dimensional span on which the primitive is known to have a nontrivial derivative change (or slope gap). In experiments, one can often take $\text{Sel}_\omega$ to pick the coordinate with largest preactivation margin to avoid boundary degeneracy; formal certificates require a margin bound to exclude measure-zero boundary cases.

## D.3. Coercive Tail Calibration (E4 Variants)

The (E4) conditions supply a quantitative *tail gain* ensuring that variation injected at the bottleneck propagates to the chosen sink through the suffix computation, without being annihilated by a nearly singular tail. The variants differ only in what notion of gain is bounded:

- **(E4-$\sigma$-cell):** a cellwise lower bound on a spectral quantity (e.g. a minimum singular value) of the relevant suffix operator.
- **(E4-$\sigma$-ray):** a raywise uniform spectral lower bound along $t \in [0, r]$.
- **(E4-dir):** a directional gain lower bound for a *selected* witness direction propagated through the suffix.

**How to certify (E4-$\sigma$) in small models.** When the suffix operator is an explicit matrix (or composition of small matrices) on a fixed refined cell, one may compute its minimum singular value exactly and record it. When the suffix includes known primitives (e.g. softmax, LayerNorm), one combines certified floors such as (67) with submultiplicativity of $\sigma_{\min}$ (Lemma C.8) to obtain a conservative product bound.

**How to estimate (E4-$\sigma$) in large models.** For large models, exact $\sigma_{\min}$ is expensive. We therefore report estimates based on:

- power iteration / Lanczos for extremal singular values of linearized blocks,

- randomized probing of $\|Ju\|$ over random $u$ in the effective subspace (gives directional surrogates),
- layerwise lower bounds with explicit tracking of the slack contributed by each layer.

These are *not* certificates, but they diagnose whether the spectral tail is the dominant looseness source.

**How to certify (E4-dir).** (E4-dir) is designed to be easier to validate: given a witness direction $u$ at the bottleneck output, one needs a lower bound on $\|Tu\|$, where $T$ is the suffix operator (as defined in §B.4). A certificate may include:

- the direction $u$ (or a finite set of candidate directions),
- a proof that $u$ lies in the effective subspace of the bottleneck primitive,
- a lower bound on $\|Tu\|$ obtained by composing primitive-level floors and affine matrix gains.

Experimentally, (E4-dir) is estimated by computing Jacobian-vector products (JVPs) for the suffix.

## D.4. Merge Calibration (Assumption M)

Assumption (M) ensures that variation that survives the bottleneck and tail is not destroyed at merge nodes (residual adds, concatenations, gated merges, expert aggregation, etc.). The assumption is stated in §B.5.5 in terms of:

- a **coordinate subspace** $S_0$ (where the certificate tracks a protected component),
- an **isolating projector** $Q$ applied downstream of the merge,
- a **merge floor** $\underline{s}_0(U)$ bounding how much of the $S_0$ component survives on a validity set $U$,
- and an **isolation condition** excluding leakage from $S_0^\perp$ into the protected channel (typically written as $Q\,M\,P_{S_0^\perp} = 0$ in the idealized exact-isolation regime).

### M.1 STRUCTURED MERGES (EXACT ISOLATION BY DESIGN)

Some merges are isolating by architecture:

- **Concatenation.** If $M(a,b) = (a,b)$ and the sink reads only coordinates of $a$, take $S_0$ as the $a$ coordinates and $Q$ as projection onto those coordinates. Then $QMP_{S_0^\perp} = 0$ holds.
- **Head/channel partitioning.** If the merge is a block-diagonal linear map across disjoint channel groups, take $S_0$ as a block, and $Q$ as its projection.

In these cases, $\underline{s}_0$ is often 1 (up to known diagonal scalings).

### M.2 UNIFORM SURVIVAL BOUND (BOUNDED CANCELLATION)

For additive merges $M(a,b) = a + b$, exact isolation rarely holds. A practical certifiable route is to choose $Q$ so that, on $U$, the protected component of $a$ dominates:
$$\|Qa\| \geq \eta\|Qb\| \quad \text{for all points in } U,$$
for some $\eta > 0$. Then one obtains a survival floor
$$\|Q(a+b)\| \;\geq\; \|Qa\| - \|Qb\| \;\geq\; \left(1 - \eta^{-1}\right)\|Qa\|.$$
Thus $\underline{s}_0(U)$ can be taken as $1 - \eta^{-1}$. Formally, a merge certificate records: the choice of $(S_0, Q)$ and a proof of the dominance inequality on $U$ (e.g. from sound bounds on each branch, or from structural disjointness).

### M.3 LIPSCHITZ INTERPOLATION (ROBUSTNESS-STYLE MERGE BOUNDS)

When exact isolation does not hold and dominance is not uniform, one may certify a weaker form: the protected projection is *Lipschitz* in the nuisance branch and the nuisance magnitude is small on $U$. Concretely, if for all $z$ in a validity region one has
$$\|Q\,M(z_a, z_b) - Q\,M(z_a, 0)\| \leq L_M\,\|z_b\|,$$
and one can lower bound $\|Q\,M(z_a, 0)\|$ on $U$, then a nontrivial survival floor follows. This is the natural "graceful degradation" posture (exact constants are pack- and merge-dependent); it is also the bridge to the $\varepsilon$-leakage relaxation discussed in the whitespace (§B.8.3).

**Projector selection heuristics.** In experiments, we choose $Q$ to be: (i) a coordinate projection onto a head/channel group, (ii) a random low-dimensional projection (for robustness of detection), or (iii) the top left singular vectors of a downstream linear map (to align with sensitive output directions). For formal certification, (i) is preferred because it is exactly checkable and stable under implementation details.

## D.5. IEEE-754 Event Constants

In the IEEE-754 world, the realized network is (typically) piecewise constant as a function of real inputs, and total variation along a ray is the sum of discrete jumps at event points. The IEEE-754 lower bound in §B.7.2 depends on two kinds of constants:

- an **event density** (or event count) $\kappa_x$ on an interval of length $r$,
- a **jump floor** $\Delta_{\min}$, a lower bound on the output jump magnitude per event after tail+merge propagation.

**Certification burden and scope.** Full certification of IEEE-754 constants is substantially harder than Real-RAM because: (i) primitives are discontinuous at rounding thresholds, and (ii) fused kernels and mixed precision can deviate from "pure IEEE-754 per-op" semantics. Accordingly, this paper treats IEEE-754 constants as *certifiable in principle* under a declared arithmetic pack, and *estimated in practice* in §5.

### EVENT DENSITY $\kappa_x$ AND SPACING $\delta$

A common route is to track a scalar monitored quantity $g(t)$ along the ray (e.g. a preactivation coordinate, a logit difference that triggers an argmax switch, or an INT8 bucket index). If events correspond to crossing regularly spaced thresholds with spacing at most $\delta_{\max}$ on $U$, and if one has a lower bound $|\dot{g}(t)| \geq \dot{g}_{\min}$ on $U$, then
$$\kappa_x \;\gtrsim\; \frac{\dot{g}_{\min}}{\delta_{\max}} \quad \text{(events per unit length)}.$$

For quantization buckets, $\delta_{\max}$ is the bucket width; for floating-point rounding, $\delta_{\max}$ is controlled by local ulp spacing (which depends on exponent range on $U$).

JUMP FLOOR $\Delta_{\min}$

At the triggering primitive output, a single rounding/bucket event produces a change of at least one representable step:

$$\Delta_{\text{prim}} \geq \text{ulp}_{\min}(U) \quad \text{(FP formats)} \qquad \text{or} \qquad \Delta_{\text{prim}} \geq \delta_{\min} \quad \text{(uniform quantizers)}.$$

To translate this into an *output* jump, one composes:

- a tail gain bound (an IEEE-754 analog of (E4), often directional in practice),
- a merge survival floor $\underline{s}_0(U)$ from Assumption (M),
- and any fixed output projection $Q$ used by the certificate.

Thus a conservative jump floor takes the schematic form

$$\Delta_{\min}(U) \geq \underline{s}_0(U) \cdot \Gamma_{\text{event}}(U) \cdot \Delta_{\text{prim}}(U),$$

where $\Gamma_{\text{event}}$ is the certified/estimated propagation gain from the event site to the sink under the declared IEEE-754 semantics.

**Remark (estimation in experiments).** In §5, we estimate $\kappa_x$ by direct event counting along rays and estimate $\Delta_{\min}$ by measuring the smallest observed nonzero output jump, stratified by precision and quantization scheme. These estimates validate whether the event bound is *informative*, but they do not certify it.

## D.6. Empirical Estimation vs. Certification

The framework is intentionally explicit about what is *proved* versus what is *measured*:

- **To publish theory:** it is sufficient to define the constants and state the axioms/postulates they satisfy, together with theorems that use them (the present paper).
- **To publish audits/verification:** one must supply either certificates or carefully scoped proxies. Our main-text experiments are framed as audits and calibrations, not as formal proofs of axiom satisfaction, except in settings where certificates are tractable (small models, closed-form floors).
- **To deploy certificates:** one must commit to a concrete arithmetic pack, including kernel semantics (especially for mixed precision and fused operations), and use sound range-bounding/verification tools to produce machine-checkable bounds. This deployment-grade regime motivates the whitespace items in §B.8.3.

# E. Limitations and Future Work

**Limitations.** The Atlas has several limitations that scope its applicability. First, bounds are conditional on pack declarations; if operators violate their stated contracts (a claimed Lipschitz constant that is too optimistic, or a derivative floor that does not hold in saturation), certificates may be invalid. Users must verify operator properties independently. Second, calibration via Jacobian sampling provides statistical estimates, not exact values; the factors $\gamma_{P5}^{\min}$, $\Gamma_{E4}$, and $\mu_M$ may be miscalibrated outside the sampled input distribution. Third, we treat explicit computation graphs; implicit layers (deep equilibrium models, neural ODEs) and stochastic execution (dropout, stochastic depth) require extensions discussed in the appendix. Fourth, the Atlas certifies necessary conditions for variation floors but does not guarantee downstream task performance; an architecture may satisfy all axioms yet fail for reasons outside the framework's scope. Finally, our experiments use controlled settings to isolate mechanisms; scaling to production systems requires additional engineering.

