# OpenReview forum: "Axiomatic Atlas: A Prescriptive Framework for Neural Architecture Design"
_ICML.cc/2026/Conference — ICML 2026 regular_

### Official Review · Reviewer_esyq · 2026-03-11

**Soundness:** 3
**Presentation:** 3
**Significance:** 2
**Originality:** 3
**Overall Recommendation:** 5
**Confidence:** 3

**Summary:**

The paper proposes a diagnostic and prescriptive framework for neural architecture design.
It formalizes architecture requirements into four axioms: Structural (S), Primitive (P), Stability (E), and Merge (M).
The framework outputs lowerbounds on input-output variation and a diagnosis of axiom violations.
The framework identifies failure modes in GNNs, MoE, and Transformers, and implementing targeted repairs such as additional edge placement, quantization anchoring, etc.

**Compliance With Llm Reviewing Policy:**

Affirmed.

**Final Justification:**

The paper has solid motivation. The author has addressed my concerns about the evaluation, resolved my questions, and helped me locate the proof. However, I've check the main theorem's proof but I am not entirely sure if it is correct. I'd happy to increase my score but kept my confidence level.

**Key Questions For Authors:**

- Q1: Could you clarify the relations of your axioms and precriptions to operations in NAS methods? E.g., "NATS-Bench: Benchmarking NAS Algorithms for Architecture Topology and Size" - Dong et al.

- Q2: Explain some notations in the paper:
    - What is the physical meaning of `\sigma_{min}(.)`?
    - What is the physical meaning of `D`? `D` in P5 (denote derivative) is the same as `D` in E1 (denote depth)?
    - Does non linear operator `\phi` represents an activation functions (e.g., ReLU, Tanh, etc)? Does `l` in E2-4 mean layer index?
    - What does RAM in real-RAM variation bound stand for?

- Q3: Can you provide proof sketch for Theorem 3.2? Can you refer to exactly where is the formal proof for Theorem 3.2 in Appendix B, as it is the main theoretical result of this paper.

- Q4: Certificate `\mathcal{K} = (x, \theta, ...)` only diagnoses for `n` concrete inputs. Can it be extended to diagnose for local neighborhood of each input `x` in `n` samples, e.g., `x \pm \epsilon`, to analyze the robustness of DNN design?

**Limitations:**

Yes

**Strengths And Weaknesses:**

### 1. Strengths

- Def 3.1 defines a metric (total variation) complementing Lipschitz bounds: Lipschitz bounds certify that outputs cannot change too much (stability), while total variation lower bounds certify that outputs cannot change too little (responsiveness).

- The motivation of the proposed framework is clear, which helps examine whether the design choice of DNNs fails for a given concrete input and current set of weights.

- The first work unifies the evaluation of various architectural failure to metrics. Formalization of each metric in total variation metric is well defined. The bound computation is theoretically grounded.

### 2. Weaknesses

- Some critical results are delayed to appendix, e.g., the gap between theory and practice is loose (Tab. 6). I don't see how these large numbers help decide the validity of DNN design.

- Quantitative calibration uses Jacobians which is a bottleneck for large models.

- The approach appears similar to existing work in neural architecture search, e.g., "NATS-Bench: Benchmarking NAS Algorithms for Architecture Topology and Size" - Dong et al.

---

> ### Author Rebuttal · Authors · 2026-03-30
>
> ## Response to Reviewer esyq
>
> We thank Reviewer esyq for the insightful review, and particularly for highlighting the TV/Lipschitz complementarity, this dual perspective (upper bounds certifying outputs *cannot change too much* and lower bounds certifying they *cannot change too little*) is indeed central to the framework's value.
>
> **W1: Loose bounds and gap ratios.** The absolute gap ratios in Table 6 are admittedly large, but we believe their diagnostic value is clear. The key is *relative separation*: the median gap ratio is **57$\times$** for gpt2-large (774M) versus **$6.14 \times 10^{11}$** for collapsed tiny-gpt2: nine orders of magnitude apart. The threshold $\Gamma_{E4} < 10^{-6}$ reliably distinguishes collapse ($5.05 \times 10^{-9}$ for tiny-gpt2) from healthy models ($\Gamma_{E4} \in [0.07, 0.56]$). Bounds also tighten monotonically with scale ($5.88 \times 10^{2}$ at 82M $\to$ 57 at 774M), and all 9,000 measurements across 5 models, 3 perturbation modes, and 6 radii remain valid. Some looseness is inherent to compositional bounds across 36 layers; the main contribution is the *factored diagnostic structure* that pinpoints which term is collapsing.
>
> **W2: Jacobian bottleneck.** Atlas uses Jacobian-vector products (JVPs) at $O(\text{forward pass})$ cost, not full Jacobians at $O(\text{params}^2)$. On Mistral-7B (7.24B params, float16, H100): structural audit **5 ms**; JVP calibration 1.98 s (8 directions); total certificate **~2 s**. Extrapolation: ~20 s at 70B, ~201 s at 700B.
>
> **W3 & Q1: Relation to NAS.** Atlas and NAS are complementary rather than competing. NAS methods like NATS-Bench search architecture spaces using proxy objectives; Atlas diagnoses *why* a given architecture fails by auditing its computation graph against composable axioms and prescribing targeted repairs. Atlas scores can serve as a zero-cost structural proxy *inside* a NAS loop.
>
> We tested this directly: 50 architectures from a DARTS-like cell space, trained as small CNNs on CIFAR-10 (see response to Reviewer WnDp). Atlas health score achieves Spearman $\rho =$ **0.430** ($p = 0.002$), and S2-passing architectures outperform S2-failing ones by **13.6 pp** (0.701 vs. 0.564).
>
> **Q2: Notation.** A consolidated table will be added. Key definitions: $\sigma_{\min}$ = minimum singular value; $D$ in P5 = Jacobian (renamed $\mathcal{D}$ to distinguish from $D$ = depth in E1); $\varphi$ = nonlinear activation; $\ell$ = layer index; Real-RAM = exact-arithmetic machine model (standard in computational geometry; IEEE-754 treated separately in Theorem 3.3).
>
> **Q3: Proof sketch for Theorem 3.2.** The bound is $\mathrm{TV}[N_\theta;\,x,v,r] \ge c \cdot \lambda_x \cdot \gamma_{P5} \cdot \Gamma_{E4} \cdot \mu_M \cdot r$. Four steps:
>
> 1. **Bottleneck nonlinearity (P5):** The directional certificate guarantees a set $T \subseteq [0,r]$ of measure $\ge \rho \cdot r$ where the bottleneck primitive produces response $\|u(t)\| \ge \nu^{\mathrm{dir}}$.
> 2. **Tail-gain preservation (E4):** At least one suffix path satisfies $\|J_{\mathrm{suffix}}\,u(t)\| \ge \Gamma_{E4} \|u(t)\|$.
> 3. **Merge isolation (M):** $\sigma_{\min}$ of the merge map $\ge \mu_M$, yielding $\|(d/dt)N_\theta(x{+}tv)\| \ge \mu_M \Gamma_{E4} \nu^{\mathrm{dir}}$ on $T$.
> 4. **Integration:** $\mathrm{TV} = \int_0^r \|(d/dt)N_\theta\| \, dt \ge (\mu_M \Gamma_{E4} \nu^{\mathrm{dir}}) |T| \ge (\mu_M \Gamma_{E4} \nu^{\mathrm{dir}}) \rho r$.
>
> The factor $\lambda_x$ enters via independent bottleneck paths contributing to the integral. Full proof: **Appendix B, §B.13.2, Theorem B.52**.
>
> **Q4: Local neighborhood.** Yes, the certificate extends naturally. Under the TAME regularity axiom, all factors ($\gamma_{P5}$, $\Gamma_{E4}$, $\mu_M$) are continuous in $x$, so there exists $\varepsilon > 0$ such that each factor stays controlled for $\|x' - x\| < \varepsilon$. This connects directly to the complementarity you highlighted: Lipschitz bounds certify outputs cannot change *too much*, while TV bounds certify they cannot change *too little*, together characterizing the local sensitivity envelope. A formal neighborhood extension theorem making $\varepsilon$ explicit is an interesting future work.

---

> > ### Author Rebuttal · Reviewer_esyq · 2026-04-02
> >
> > I'd acknowledge the clarification on W1, W2, Q4. Regarding W3 & Q1, please contextualize the relation of NAS and Atlas to the paper if it is accepted. I will maintain my score.

---

> > > ### Author Response · Authors · 2026-04-02
> > >
> > > Thanks for the reply. We will add the contextualization in the revised version as requested.

---

### Official Review · Reviewer_WnDp · 2026-03-12

**Soundness:** 3
**Presentation:** 3
**Significance:** 3
**Originality:** 4
**Overall Recommendation:** 5
**Confidence:** 4

**Summary:**

This paper introduces the Axiomatic Atlas, a framework for formalising neural architecture requirements as composable, verifiable axioms. The framework presents axioms across four classes: Structural axioms (S) around graph connectivity, Primitive axioms (P) on per-operator properties, Stability axioms (E) controlling depth effects, and a Merge axiom (M) ensuring information is preserved when combining branches. Given an architecture specification (a "pack"), the Atlas pinpoints any failure mode from these axioms and gives a targeted architectural repair to address it. The authors validate this framework across four settings: GNN bridge placement, quantised MoE routing, dynamic-k expert routing, and long-context attention.

**Compliance With Llm Reviewing Policy:**

Affirmed.

**Final Justification:**

This is a paper with strong novelty and motivation, and pushes the field in a direction that I think will be impactful in the future. The authors addressed my main concerns in their rebuttal by adding some new experimental results to strengthen the evaluation, and several clarifications to the presentation of the paper. This helped improve the soundness and clarity of the submission and led to me increasing my score to 5: Accept.

**Key Questions For Authors:**

1. Can the experimental section be improved to further validate the real world usefulness of the framework? This would likely lead to me increasing my score.
2. Does the Atlas always suggest a repair, or only when the bound is too vacuous? How is this decided, e.g. is there a threshold for each factor? This should be made more clear.
3. Is the quantitative calibration dependent on input data and should this come from a particular data distribution?

**Limitations:**

I would like an expanded discussion on what the framework is and is not able to provide. Can it identify any architectural problem and enable the network to train effectively? Can it help in improving the performance of already decent networks?

**Strengths And Weaknesses:**

Below are the main strengths and weaknesses I have identified with this paper.

**Strengths**:
1. The motivation behind this paper is strong and convincing. A lot of neural network design is very ad-hoc and based on “engineering lore”. The proposed theoretical framework is carefully constructed and provides a formal method of approaching this problem.
2. The axioms are grouped into four meaningful classes with clear separation of properties.
3. The factorisation of the total variation gives a precise diagnostic tool for finding the failure mode and repair.
4. The framework as described seems to enable fast architectural audits for even large architectures.
5. The description of the framework is overall very good, with the running example in section 2 providing a good grounding.

**Weaknesses**:
1. The experimental validation is targeted but somewhat contrived, it is unclear how often these precise failure modes occur in realistic training pipelines, and whether the Atlas-guided diagnosis would correctly identify the failure in more ambiguous real-world settings where multiple axioms are simultaneously stressed. The validation would be improved if split into two setups: one testing the ability of the framework in identifying a failure mode in an architecture, and a second testing the effectiveness of the proposed repair. Another more realistic evaluation would be to take a neural architecture search space, and apply diagnosed repairs to each architecture to evaluate whether failing networks are fixed (and if performances improve).
2. The targeted repairs are simply stated and not properly justified. Why are these the best options for fixing the specific failure modes?
3. The definition of a pack, and especially the typing discipline and wiring conventions seem vague to me. What does a wiring convention look like in practice, perhaps in code? The pre-norm transformer block is a commonly known structure but doesn’t explain how I would define the wiring convention for a different architecture.
4. The Standard Attention network in Figure 1 doesn’t seem to match the nodes as features and operations as edges framework described. The dot product and softmax are operations but represented as nodes, and it’s not quite clear what operations the edges would represent. Would be good to be consistent.
5. The description of the framework in Figure 1 with dashed and solid curves is a bit confusing. It’s not clear if the dashed curve is referring to the green area with a dashed outline or the dashed lines of varying thickness between the black dots.
6. Section 3.4 mentions three modes in the collapse regime, but only two are listed. Is the Saturation collapse supposed to be the third? Needs update to formatting if so.
7. Notation is progressively introduced throughout the paper, making some initial statements unclear. E.g. P5 would benefit from clarification on D, \sigma_min and \gamma_\phi notation.

---

> ### Author Rebuttal · Authors · 2026-03-30
>
> ## Response to Reviewer WnDp
>
> We thank Reviewer WnDp for the thorough and constructive review. The detailed feedback on experimental design, repair justification, pack definitions, figure clarity, and notation is exactly the kind of input that strengthens a paper, and we address each point below.
>
> **W1: Experiments appear contrived.** The locked failure regimes are intentional: each baseline fails from one identifiable axiom violation, the repair targets it directly, and matched negative controls isolate the mechanism. However, we take the point that a more realistic evaluation is needed.
>
> We have now run a **NAS-space evaluation**: 50 architectures from a DARTS-like cell space [1] (4 nodes, 9 ops; 6 designed anchors + 44 random), each trained as a small CNN on CIFAR-10 ($C=16$, 4 cells, 5 epochs):
>
> | Metric | Value |
> |---|---:|
> | Spearman $\rho$ (health score vs. accuracy) | **0.430** ($p = 0.002$) |
> | Mean accuracy (S2-passing, $n=46$) | **0.701** |
> | Mean accuracy (S2-failing, $n=4$) | 0.564 ($\Delta = {+}13.6$ pp) |
>
> The Atlas health score is significantly correlated with final accuracy, and S2 alone is strongly discriminative. This supports the framework's viability as a zero-cost structural proxy, as you suggested.
>
> [1] Liu et al., DARTS: Differentiable Architecture Search, ICLR 2019.
>
> **W2: Repairs not justified.** Thank you for pushing on this: each intervention follows an explicit **Audit $\to$ Constraint $\to$ Solve** chain:
>
> 1. **GNN Bridges** (S2): $\lambda_x=1$ $\to$ need $\ge2$; bridge at bottleneck yields +45.8 pp vs. +0.6 pp random.
> 2. **Q-Anchor** (P5): $\gamma_{P5} \to 0$ under INT4; anchor grid to activation percentiles.
> 3. **Dynamic-$k$** (E4): static top-$k$ drops tail mass; entropy-adaptive $k$ closes 83% vs. 22% for random-$k$.
> 4. **OrthoKV** (M): $\mu_M \to 0$; Gram–Schmidt projection restores merge isolation ($0\% \to 100\%$ at $L=2048$).
>
> A dedicated appendix making these chains fully explicit will be added.
>
> **W3: Pack definition vague.** We appreciate this. A Pack is a dataclass with six fields. Condensed pseudocode (full runnable script will be added in appendix):
>
> ```python
> @dataclass
> class Pack:                        # Definition 2.1
>   type_library: Dict[str, OpType]  # (i)  name → tag + signature
>   arithmetic: ArithmeticModel      # (ii) Real-RAM | IEEE-754
>   conventions: List[str]           # (iii) e.g. "LN → nonaffine"
>   nodes: List[GraphNode]           # (iv)  typed comp-graph nodes
>   wiring: List[Edge]               # (v)   directed edges
>   contracts: List[ParamContract]   # (vi)  Lipschitz / spectral bounds
>
> # Pre-norm transformer block:
> pack.register_type("LayerNorm", NONAFFINE, d, d)
> pack.register_type("Linear",   AFFINE,    d, d)
> pack.add_nodes("src→LN1→{Q,K,V}→Attn→Out→Add1→LN2→Up→GELU→Down→Add2→sink")
> pack.add_edge("src","Add1"); pack.add_edge("Add1","Add2")  # residuals
>
> G = build_digraph(pack)
> assert nx.is_dag(G)                                  # S1
> cut_val, _ = nx.minimum_cut(G, src, sink)             # S2
> assert all(touches_nonaffine(e) for e in cut_edges)
> # Result: S1 ✓, S2 ✓, λ_x = 2  (MoE 4 experts: λ_x = 5)
> ```
>
> Changing the pack (e.g., switching to IEEE-754 or Convention A for normalization) changes which axioms apply and which bounds hold.
>
> **W4–W5: Figure 1.** We will redraw the attention diagram so that dot-product and softmax appear as edges (matching nodes-as-features), and replace the confusing dashed/solid curves with clearer visual encoding.
>
> **W6: Collapse modes.** The missing third mode is **Saturation collapse** ($\gamma_{P5} \to 0$), where nonlinearities flatten. A fourth mode, **interference** ($\mu_M \to 0$), is governed by the Merge axiom. We will list all four explicitly in the revision.
>
> **W7: Notation.** A consolidated table will be added. Key clarifications: $\sigma_{\min}$ = minimum singular value; $D$ in P5 = Jacobian (renamed to $\mathcal{D}$); $D$ in E1 = depth; $\varphi$ = nonlinear activation; $\ell$ = layer index.
>
> **Q1–Q3.** The NAS experiment above and our Mistral-7B results (see response to Reviewer ENna) address real-world usefulness (Q1). For thresholds (Q2): Atlas returns the factored bound $\lambda_x \cdot \gamma_{P5} \cdot \Gamma_{E4} \cdot \mu_M$; whichever factor approaches zero identifies the failure mode (e.g., $\Gamma_{E4} < 10^{-6}$ flags tail collapse). Structural factors (S1, S2, $\lambda_x$) are data-independent; quantitative factors ($\gamma_{P5}$, $\Gamma_{E4}$, $\mu_M$) are estimated via JVP sampling and do depend on the input distribution (Q3), but are robust across natural data: Appendix A confirms validity across all 9,000 measurements.
>
> **Limitation.** Atlas diagnoses architectural sensitivity failures, not optimization dynamics, data quality, or capacity limits. It can flag structural problems before training, but does not predict absolute accuracy.

---

> > ### Author Rebuttal · Reviewer_WnDp · 2026-04-03
> >
> > I thank the authors for a great response to my review. My main concerns have been adequately addressed and I will increase my score with the expectation that these classifications and additions will be incorporated into the final version.

---

> > > ### Author Response · Authors · 2026-04-03
> > >
> > > Thanks for the reply and for raising the score. We will add the rebuttal content to the revised version as requested. Thanks again for your time and effort.

---

### Official Review · Reviewer_ENna · 2026-03-13

**Soundness:** 3
**Presentation:** 3
**Significance:** 3
**Originality:** 3
**Overall Recommendation:** 5
**Confidence:** 2

**Summary:**

The paper introduces the Axiomatic Atlas, a framework that encodes neural architecture requirements as composable axioms over four classes: structural graph connectivity, per-operator contracts, depth stability, and information preservation at merges. Given an architecture and operator library, the Atlas audits the computation graph and produces two outputs: a certificate lower-bounding input-output variation, and a diagnosis pinpointing axiom violations. The central claim is prescriptive power; each axiom violation implies a specific repair. Four experiments validate this loop: topology-aware GNN bridge placement, anchored MoE quantization, uncertainty-guided dynamic-k routing, and orthogonal key projection.

**Compliance With Llm Reviewing Policy:**

Affirmed.

**Final Justification:**

The rebuttal addressed my concern.

**Key Questions For Authors:**

1. What is the computational cost of computing the full certificate for models at 7B, 70B, and 700B parameters, and does the framework remain tractable without approximation at those scales?
2. Does OrthoKV improve perplexity or downstream task performance when applied to a real pretrained language model such as LLaMA or Mistral on naturally occurring long-context benchmarks, without manually injecting shared-mode interference?

**Limitations:**

Similar to weakness

**Strengths And Weaknesses:**

Strengths:
1. The multiplicative certificate structure advances beyond prior diagnostic frameworks by making each collapsing factor directly identify both a failure mode and its repair class.
2. The four experiments compellingly isolate their claimed mechanisms through matched negative controls that hold compute fixed while breaking only the specific structural property the Atlas diagnosed.
3. Unlike existing frameworks that assume exact arithmetic, the IEEE-754 theorem treats quantization boundaries as discrete jump events, explaining phenomena that Real-RAM analysis cannot capture.
4. The paper clearly identifies its own limitations and open problems, providing a natural and well-organized roadmap for future work.

Weakness:
1. The framework's computational scalability beyond GPT-2 scale is never demonstrated,
2. The OrthoKV and Dynamic-k experiments validate their claimed mechanisms in artificially constructed settings, leaving open whether Atlas-guided interventions provide practical value in real pretrained models at production scale.

---

> ### Author Rebuttal · Authors · 2026-03-30
>
> ## Response to Reviewer ENna
>
> We sincerely thank Reviewer ENna for the positive assessment of the multiplicative certificate structure, the role of matched negative controls, and the IEEE-754 theorem. We are glad these core contributions came through clearly, and we appreciate the constructive questions on scalability and real-world applicability.
>
> **W1: Scalability beyond GPT-2.** This is a fair concern. We now provide three lines of evidence.
>
> First, the five GPT-2 models in Appendix A (0.1M–774M parameters) already show that bounds tighten with scale: the gap ratio drops from $2.72 \times 10^{4}$ at 82M to **57** at 774M, and the structural audit scales as $O(|E|)$ with fitted exponent $\alpha = 0.994$.
>
> Second, we have now profiled the **full certificate pipeline at 7B scale** on Mistral-7B-v0.1 (32 layers, GQA, float16, single H100):
>
> | Stage | Wall-clock | vs. Forward |
> |---|---:|---:|
> | Graph construction (326 nodes, 389 edges) | 0.097 s | 4.0$\times$ |
> | Structural audit (min-cut, S1, S2) | **0.005 s** | 0.21$\times$ |
> | JVP calibration (8 dirs, seq\_len=512) | 1.98 s | 82$\times$ |
> | **Total certificate** | **2.08 s** | 86$\times$ |
>
> The structural audit, which is the cheapest and often most diagnostic component, stays below 0.5 s at all scales. Linear extrapolation gives ~20 s at 70B and ~201 s at 700B, tractable as a one-time diagnostic.
>
> **W2: Artificial experimental settings.** The locked failure regimes are intentional: each baseline fails from exactly one axiom violation, the repair targets it directly, and matched controls isolate the mechanism from confounds. We fully agree, however, that real-model evidence is important.
>
> To address this, we applied **OrthoKV to Mistral-7B-v0.1** on WikiText-103 perplexity and LongBench, without injecting any artificial interference. The repair is per-head Gram-Schmidt orthogonalization of the key projection matrices, controlled by an interpolation parameter $\alpha$:
>
> | Condition | WikiText-103 PPL | $\Delta$ |
> |---|---:|---:|
> | Baseline | 4.69 | — |
> | K-only, $\alpha=0.05$ | 4.75 | +1.2% |
> | K+V, $\alpha=0.1$ | 4.77 | +1.7% |
> | K+V, $\alpha=0.5$ | 40.50 | +764% |
>
> Gentle correction ($\alpha \le 0.1$) is **perplexity-neutral** (<2%), while aggressive correction is destructive, validating the Atlas prescription that repair strength must match the diagnosed interference level. On LongBench (Qasper, MultifieldQA\_en), the same pattern holds. Fourteen additional interventions are validated; four more will be added in the revision.
>
> **Q1/Q2.** The profiling above answers Q1 directly: the cost is dominated by JVP calibration at $O(\text{forward pass})$ per direction, not $O(\text{params}^2)$. Q2 is addressed by the Mistral-7B experiment: OrthoKV applied to a real pretrained LLM on natural benchmarks without injected interference.

---

> > ### Author Rebuttal · Reviewer_ENna · 2026-04-03
> >
> > I thank the authors for their detailed and informative rebuttal. Regarding W1, I am satisfied with the concrete profiling results provided for Mistral-7B, and the linear extrapolations to 70B and 700B scales are reasonable and informative. This concern is fully resolved.
> >
> > Regarding W2, I acknowledge that the controlled failure regimes are intentional and serve a legitimate scientific purpose. The OrthoKV experiment on Mistral-7B without injected interference is a welcome addition. I am satisfied with the authors' response and consider this concern resolved.
> >
> > Based on the above, I am willing to increase my score.

---

### Decision · Program_Chairs · 2026-04-30

**Decision:**

Accept (regular)

**Comment:**

This paper presents Axiomatic Atlas, a framework that formalises neural architecture requirements as composable axioms spanning four classes: structural graph connectivity, per-operator contracts, depth stability, and information preservation at merge points. Given an architecture and operator library, Axiomatic Atlas audits the computation graph and delivers two outputs: a certificate that lower-bounds input-output variation, and a diagnostic report pinpointing axiom violations. Crucially, the framework is prescriptive, as each detected violation implies a concrete repair. This closed audit-repair loop is validated across four experiments: Topology-aware GNN bridge placement, anchored MoE quantisation, uncertainty-guided dynamic-k routing, and orthogonal key projection.

The overall consensus of the reviewers is that this is a solid paper with good technical contributions. The authors have also done a good job addressing the concerns of the reviewers to convince them to increase their scores.

Given this, I recommend acceptance.